# Gradient Descent on Two-layer Nets:
# Margin Maximization and Simplicity Bias

**Kaifeng Lyu**[*][†]
Princeton University
klyu@cs.princeton.edu

**Zhiyuan Li**[*]
Princeton University
zhiyuanli@cs.princeton.edu

**Runzhe Wang**[*][†]
Princeton University
runzhew@princeton.edu

**Sanjeev Arora**
Princeton University
arora@cs.princeton.edu

## Abstract

The generalization mystery of overparametrized deep nets has motivated efforts to understand how gradient descent (GD) converges to low-loss solutions that generalize well. Real-life neural networks are initialized from small random values and trained with cross-entropy loss for classification (unlike the "lazy" or "NTK" regime of training where analysis was more successful), and a recent sequence of results (Lyu and Li, 2020; Chizat and Bach, 2020; Ji and Telgarsky, 2020a) provide theoretical evidence that GD may converge to the "max-margin" solution with zero loss, which presumably generalizes well. However, the global optimality of margin is proved only in some settings where neural nets are infinitely or exponentially wide. The current paper is able to establish this global optimality for two-layer Leaky ReLU nets trained with gradient flow on linearly separable and symmetric data, regardless of the width. The analysis also gives some theoretical justification for recent empirical findings (Kalimeris et al., 2019) on the so-called simplicity bias of GD towards linear or other "simple" classes of solutions, especially early in training. On the pessimistic side, the paper suggests that such results are fragile. A simple data manipulation can make gradient flow converge to a linear classifier with suboptimal margin.

## 1 Introduction

One major mystery in deep learning is why deep neural networks generalize despite overparameterization (Zhang et al., 2017). To tackle this issue, many recent works turn to study the *implicit bias* of gradient descent (GD) — what kind of theoretical characterization can we give for the low-loss solution found by GD?

The seminal works by Soudry et al. (2018a,b) revealed an interesting connection between GD and margin maximization: for linear logistic regression on linearly separable data, there can be multiple linear classifiers that perfectly fit the data, but GD with any initialization always converges to the max-margin (hard-margin SVM) solution, even when there is no explicit regularization. Thus the solution found by GD has the same margin-based generalization bounds as hard-margin SVM. Subsequent works on linear models have extended this theoretical understanding of GD to SGD (Nacson et al., 2019b), other gradient-based methods (Gunasekar et al., 2018a), other loss functions with certain poly-exponential tails (Nacson et al., 2019a), linearly non-separable data (Ji and Telgarsky, 2018, 2019b), deep linear nets (Ji and Telgarsky, 2019a; Gunasekar et al., 2018b).

---

[*]Equal contribution
[†]Most of the work is done when Kaifeng Lyu and Runzhe Wang were at Tsinghua University.

35th Conference on Neural Information Processing Systems (NeurIPS 2021).

Given the above results, a natural question to ask is whether GD has the same implicit bias towards max-margin solutions for machine learning models in general. Lyu and Li (2020) studied the relationship between GD and margin maximization on *deep homogeneous neural network*, i.e., neural network whose output function is (positively) homogeneous with respect to its parameters. For homogeneous neural networks, only the direction of parameter matters for classification tasks. For logistic and exponential loss, Lyu and Li (2020) assumed that GD decreases the loss to a small value and achieves full training accuracy at some time point, and then provided an analysis for the training dynamics after this time point (Theorem 3.1), which we refer to as *late phase analysis*. It is shown that GD decreases the loss to 0 in the end and converges to a direction satisfying the Karush-Kuhn-Tucker (KKT) conditions of a constrained optimization problem (P) on margin maximization.

However, given the non-convex nature of neural networks, KKT conditions do not imply global optimality for margins. Several attempts are made to prove the global optimality specifically for two-layer nets. Chizat and Bach (2020) provided a mean-field analysis for infinitely wide two-layer Squared ReLU nets showing that gradient flow converges to the solution with global max margin, which also corresponds to the max-margin classifier in some non-Hilbertian space of functions. Ji and Telgarsky (2020a) extended the proof to finite-width neural nets, but the width needs to be exponential in the input dimension (due to the use of a covering condition). Both works build upon late phase analyses. Under a restrictive assumption that the data is orthogonally separable, i.e., any data point $x_i$ can serve as a perfect linear separator, Phuong and Lampert (2021) analyzed the full trajectory of gradient flow on two-layer ReLU nets with small initialization, and established the convergence to a piecewise linear classifier that maximizes the margin, irrespective of network width.

In this paper, we study the implicit bias of gradient flow on two-layer neural nets with Leaky ReLU activation (Maas et al., 2013) and logistic loss. To avoid the *lazy* or *Neural Tangent Kernel (NTK)* regime where the weights are initialized to large random values and do not change much during training (Jacot et al., 2018; Chizat et al., 2019; Du et al., 2019b,a; Allen-Zhu et al., 2018, 2019; Zou et al., 2018; Arora et al., 2019b), we use small initialization to encourage the model to learn features actively, which is closer to real-life neural network training.

When analyzing convergence behavior of training on neural networks, one can simplify the problem and gain insights by assuming that the data distribution has a simple structure. Many works particularly study the case where the labels are generated by an unknown teacher network that is much smaller/simpler than the (student) neural network to be trained. Following Brutzkus et al. (2018); Sarussi et al. (2021) and many other works, we consider the case where the dataset is linearly separable, namely the labels are generated by a linear teacher, and study the training dynamics of two-layer Leaky ReLU nets on such dataset.

### 1.1 Our Contribution

Among all the classifiers that can be represented by the two-layer Leaky ReLU nets, we show **any global-max-margin classifier is exactly linear** under one more data assumption: the dataset is *symmetric*, i.e., if $x$ is in the training set, then so is $-x$. Note that such symmetry can be ensured by simple data augmentation.

Still, little is known about what kind of classifiers neural network trained by GD learns. Though Lyu and Li (2020) showed that gradient flow converges to a classifier along KKT-margin direction, we note that this result is not sufficient to guarantee the global optimality since such classifier can have nonlinear decision boundaries. See Figure 1 (left) for an example.

In this paper, we provide a multi-phase analysis for the full trajectory of gradient flow, in contrast with previous late phase analyses which only analyzes the trajectory after achieving $100\%$ training accuracy. We show that **gradient flow with small initialization converges to a global-max-margin linear classifier** (Theorem 4.2). The proof leverages power iteration to show that neuron weights align in two directions in an early phase of training, inspired by Li et al. (2021). We further show the alignment at any constant training time by associating the dynamics of wide neural net with that of two-neuron neural net, and finally, extend the alignment to the infinite time limit by applying Kurdyka-Łojasiewicz (KL) inquality in a similar way as Ji and Telgarsky (2020a). The alignment at convergence implies that the convergent classifier is linear.

The above results also justify a recent line of works studying the so-called *simplicity bias*: GD first learns linear functions in the early phase of training, and the complexity of the solution increases

as training goes on (Kalimeris et al., 2019; Hu et al., 2020; Shah et al., 2020). Indeed, our result establishes a form of *extreme simplicity bias* of GD: *if the dataset can be fitted by a linear classifier, then GD learns a linear classifier not only in the beginning but also at convergence.*

On the pessimistic side, this paper suggests that such global margin maximization result could be fragile. Even for linearly separable data, global-max-margin classifiers may be nonlinear without the symmetry assumption. In particular, we show that for any linearly separable dataset, **gradient flow can be led to converge to a linear classifier with suboptimal margin by adding only** 3 **extra data points** (Theorem 6.2). See Figure 1 (right) for an example.

## 2    Related Works

**Generalization Aspect of Margin Maxmization.**    Margin often appears in the generalization bounds for neural networks (Bartlett et al., 2017; Neyshabur et al., 2018), and larger margin leads to smaller bounds. Jiang et al. (2020) conducted an empirical study for the causal relationships between complexity measures and generalization errors, and showed positive results for normalized margin, which is defined by the output margin divided by the product (or powers of the sum) of Frobenius norms of weight matrices from each layer. On the pessimistic side, negative results are also shown if Frobenius norm is replaced by spectral norm. In this paper, we do use the normalized margin with Frobenius norm (see Section 3).

**Learning on Linearly Separable Data.**    Some works studied the training dynamics of (nonlinear) neural networks on linearly separable data (labels are generated by a linear teacher). Brutzkus et al. (2018) showed that SGD on two-layer Leaky ReLU nets with hinge loss fits the training set in finite steps and generalizes well. Frei et al. (2021) studied online SGD (taking a fresh sample from the population in each step) on the two-layer Leaky ReLU nets with logistic loss. For any data distribution, they proved that there exists a time step in the early phase such that the net has a test error competitive with that of the best linear classifier over the distribution, and hence generalizes well on linearly separable data. Both two papers reveal that the weight vectors in the first layer have positive correlations with the weight of the linear teacher, but their analyses do not imply that the learned classifier is linear. In the NTK regime, Ji and Telgarsky (2020b); Chen et al. (2021) showed that GD on shallow/deep neural nets learns a kernel predictor with good generalization on linearly separable data, and it suffices to have width polylogarithmic in the number of training samples. Still, they do not imply that the learned classifier is linear. Pellegrini and Biroli (2020) provided a mean-field analysis for two-layer ReLU net showing that training with hinge loss and infinite data leads to a linear classifier, but their analysis requires the data distribution to be spherically symmetric (i.e., the probability density only depends on the distance to origin), which is a more restrictive assumption than ours. Sarussi et al. (2021) provided a late phase analysis for gradient flow on two-layer Leaky ReLU nets with logistic loss, which establishes the convergence to linear classifier based on an assumption called *Neural Agreement Regime* (NAR): starting from some time point, for any training sample, the outputs of all the neurons have the same sign. However, it is unclear why this can happen a priori. Comparing with our work, we analyze the full trajectory of gradient flow and establish the convergence to linear classifier without assuming NAR. Phuong and Lampert (2021) analyzed the full trajectory for gradient flow on orthogonally separable data, but every KKT-margin direction attains the global max margin (see Appendix H) in their setting, which it is not necessarily true in general. In our setting, KKT-margin direction with suboptimal margin does exist.

**Simplicity Bias.**    Kalimeris et al. (2019) empirically observed that neural networks in the early phase of training are learning linear classifiers, and provided evidence that SGD learns functions of increasing complexity. Hu et al. (2020) justified this view by proving that the learning dynamics of two-layer neural nets and simple linear classifiers are close to each other in the early phase, for dataset drawn from a data distribution where input coordinates are independent after some linear transformation. The aforementioned work by Frei et al. (2021) can be seen as another theoretical justification for online SGD on aribitrary data distribution. Shah et al. (2020) pointed out that extreme simplicity bias can lead to suboptimal generalization and negative effects on adversarial robustness.

**Small Initialization.**    Several theoretical works studying neural network training with small initialization can be connected to simplicity bias. Maennel et al. (2018) uncovered a weight quantization effect in training two-layer nets with small initialization: gradient flow biases the weight vectors to a certain number of directions determined by the input data (independent of neural network width). It

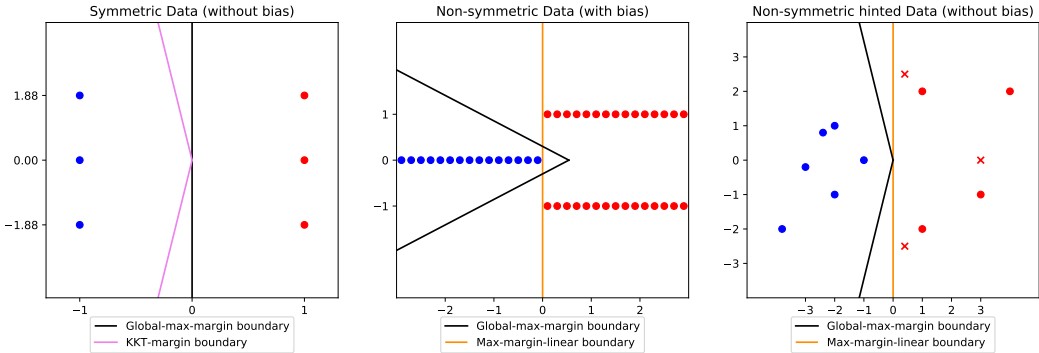

Figure 1: Two-layer Leaky ReLU nets ($\alpha_{\text{leaky}} = 1/2$) with KKT margin and global max margin on linearly separable data. See Appendix I.1 for detailed discussions. **Left**: Theorem 4.3 is not vacuous: a symmetric dataset can have KKT directions with suboptimal margin, but our theory shows that gradient flow from small initialization goes to global max margin. **Middle**: The linear classifier (orange) is along a KKT-margin direction with a much smaller margin comparing to the (nonlinear) global-max-margin classifier (black), but our theory suggests that gradient flow converges to the linear classifier. **Right**: Adding three extra data points (marked as "x"; see Definition 6.1) to a linearly separable dataset makes the linear classifier (orange) has suboptimal margin but causes the neural net to be biased to it.

is hence argued that gradient flow has a bias towards "simple" functions, but their proof is not entirely rigorous and no clear definition of simplicity is given. This weight quantization effect has also been studied under the names of weight clustering (Brutzkus and Globerson, 2019), condensation (Luo et al., 2021; Xu et al., 2021). Williams et al. (2019) studied univariate regression and showed that two-layer ReLU nets with small initialization tend to learn linear splines. For the matrix factorization problem, which can be related to training neural networks with linear or quadratic activations, we can measure the complexity of the learned solution by rank. A line of works showed that gradient descent learns solutions with gradually increasing rank (Li et al., 2018; Arora et al., 2019a; Gidel et al., 2019; Gissin et al., 2020; Li et al., 2021). Such results have been generalized to tensor factorization where the complexity measure is replaced by tensor rank (Razin et al., 2021). Beyond small initialization of our interest and large initialization in the lazy or NTK regime, Woodworth et al. (2020); Moroshko et al. (2020); Mehta et al. (2021) studied feature learning when the initialization scale transitions from small to large scale.

## 3 Preliminaries

We denote the set $\{1, \ldots, n\}$ by $[n]$ and the unit sphere $\{\boldsymbol{x} \in \mathbb{R}^d : \|\boldsymbol{x}\|_2 = 1\}$ by $\mathbb{S}^{d-1}$. We call a function $h : \mathbb{R}^D \to \mathbb{R}$ *L-homogeneous* if $h(c\boldsymbol{\theta}) = c^L h(\boldsymbol{\theta})$ for all $\boldsymbol{\theta} \in \mathbb{R}^D$ and $c > 0$. For $S \subseteq \mathbb{R}^D$, $\text{conv}(S)$ denotes the convex hull of $S$. For locally Lipschitz function $f : \mathbb{R}^D \to \mathbb{R}$, we define Clarke's subdifferential (Clarke, 1975; Clarke et al., 2008; Davis et al., 2020) to be $\partial^\circ f(\boldsymbol{\theta}) := \text{conv}\left\{\lim_{n \to \infty} \nabla f(\boldsymbol{\theta}_n) : f \text{ differentiable at } \boldsymbol{\theta}_n, \lim_{n \to \infty} \boldsymbol{\theta}_n = \boldsymbol{\theta}\right\}$ (see also Appendix B.1).

### 3.1 Logistic Loss Minimization and Margin Maximization

For a neural net, we use $f_{\boldsymbol{\theta}}(\boldsymbol{x}) \in \mathbb{R}$ to denote the output logit on input $\boldsymbol{x} \in \mathbb{R}^d$ when the parameter is $\boldsymbol{\theta} \in \mathbb{R}^D$. We say that the neural net is *L-homogeneous* if $f_{\boldsymbol{\theta}}(\boldsymbol{x})$ is $L$-homogeneous with respect to $\boldsymbol{\theta}$, i.e., $f_{c\boldsymbol{\theta}}(\boldsymbol{x}) = c^L f_{\boldsymbol{\theta}}(\boldsymbol{x})$ for all $\boldsymbol{\theta} \in \mathbb{R}^D$ and $c > 0$. VGG-like CNNs can be made homogeneous if we remove all the bias terms expect those in the first layer (Lyu and Li, 2020).

Throughout this paper, we restrict our attention to $L$-homogeneous neural nets with $f_{\boldsymbol{\theta}}(\boldsymbol{x})$ definable with respect to $\boldsymbol{\theta}$ in an o-minimal structure for all $\boldsymbol{x}$. (See Coste 2000 for reference for o-minimal structures.) This is a technical condition needed by Theorem 3.1, and it is a mild regularity condition as almost all modern neural networks satisfy this condition, including the two-layer Leaky ReLU networks studied in this paper.

For a dataset $\mathcal{S} = \{(\boldsymbol{x}_1, y_1), \ldots, (\boldsymbol{x}_n, y_n)\}$, we define $q_i(\boldsymbol{\theta}) := y_i f_{\boldsymbol{\theta}}(\boldsymbol{x}_i)$ to be the *output margin on the data point* $(\boldsymbol{x}_i, y_i)$, and $q_{\min}(\boldsymbol{\theta}) := \min_{i \in [n]} q_i(\boldsymbol{\theta})$ to be the *output margin on the dataset* $\mathcal{S}$ (or

*margin* for short). It is easy to see that $q_1(\boldsymbol{\theta}), \dots, q_n(\boldsymbol{\theta})$ are $L$-homogeneous functions, and so is $q_{\min}(\boldsymbol{\theta})$. We define the *normalized margin* $\gamma(\boldsymbol{\theta}) := q_{\min}\left(\frac{\boldsymbol{\theta}}{\|\boldsymbol{\theta}\|_2}\right) = \frac{q_{\min}(\boldsymbol{\theta})}{\|\boldsymbol{\theta}\|_2^L}$ to be the output margin (on the dataset) for the normalized parameter $\frac{\boldsymbol{\theta}}{\|\boldsymbol{\theta}\|_2}$.

We refer the problem of finding $\boldsymbol{\theta}$ that maximizes $\gamma(\boldsymbol{\theta})$ as *margin maximization*. Note that once we have found an optimal solution $\boldsymbol{\theta}^* \in \mathbb{R}^D$, $c\boldsymbol{\theta}^*$ is also optimal for all $c > 0$. We can put the norm constraint on $\boldsymbol{\theta}$ to eliminate this freedom on rescaling:

$$\max_{\boldsymbol{\theta} \in \mathbb{S}^{D-1}} \quad \gamma(\boldsymbol{\theta}). \tag{M}$$

Alternatively, we can also constrain the margin to have $q_{\min} \geq 1$ and minimize the norm:

$$\min \quad \frac{1}{2}\|\boldsymbol{\theta}\|_2^2 \quad \text{s.t.} \quad q_i(\boldsymbol{\theta}) \geq 1, \quad \forall i \in [n]. \tag{P}$$

One can easily show that $\boldsymbol{\theta}^*$ is a global maximizer of (M) if and only if $\frac{\boldsymbol{\theta}^*}{(q_{\min}(\boldsymbol{\theta}^*))^{1/L}}$ is a global minimizer of (P). For convenience, we make the following convention: if $\frac{\boldsymbol{\theta}}{\|\boldsymbol{\theta}\|_2}$ is a local/global maximizer of (M), then we say $\boldsymbol{\theta}$ is along a *local-max-margin direction*/*global-max-margin direction*; if $\frac{\boldsymbol{\theta}}{(q_{\min}(\boldsymbol{\theta}))^{1/L}}$ satisfies the KKT conditions of (P), then we say $\boldsymbol{\theta}$ is along a *KKT-margin direction*.

Gradient flow with logistic loss is defined by the following differential inclusion,

$$\frac{\mathrm{d}\boldsymbol{\theta}}{\mathrm{d}t} \in -\partial^\circ \mathcal{L}(\boldsymbol{\theta}), \quad \text{with } \mathcal{L}(\boldsymbol{\theta}) := \frac{1}{n}\sum_{i=1}^{n} \ell(q_i(\boldsymbol{\theta})), \tag{1}$$

where $\ell(q) := \ln(1 + e^{-q})$ is the logistic loss. Lyu and Li (2020); Ji and Telgarsky (2020a) showed that $\boldsymbol{\theta}(t)/\|\boldsymbol{\theta}(t)\|_2$ always converges to a KKT-margin direction. We restate the results below.

**Theorem 3.1** (Lyu and Li 2020; Ji and Telgarsky 2020a)**.** *For homogeneous neural networks, if* $\mathcal{L}(\boldsymbol{\theta}(0)) < \frac{\ln 2}{n}$*, then* $\mathcal{L}(\boldsymbol{\theta}(t)) \to 0$*,* $\|\boldsymbol{\theta}(t)\|_2 \to +\infty$*, and* $\frac{\boldsymbol{\theta}(t)}{\|\boldsymbol{\theta}(t)\|_2}$ *converges to a KKT-margin direction as* $t \to +\infty$*.*

### 3.2 Two-Layer Leaky ReLU Networks on Linearly Separable Data

Let $\phi(x) = \max\{x, \alpha_{\text{leaky}}x\}$ be Leaky ReLU, where $\alpha_{\text{leaky}} \in (0, 1)$. Throughout the following sections, we consider a two-layer neural net defined as below,

$$f_{\boldsymbol{\theta}}(\boldsymbol{x}) = \sum_{k=1}^{m} a_k \phi(\boldsymbol{w}_k^\top \boldsymbol{x}).$$

where $\boldsymbol{w}_1, \dots, \boldsymbol{w}_m \in \mathbb{R}^d$ are the weights in the first layer, $a_1, \dots, a_m \in \mathbb{R}$ are the weights in the second layer, and $\boldsymbol{\theta} = (\boldsymbol{w}_1, \dots, \boldsymbol{w}_m, a_1, \dots, a_m) \in \mathbb{R}^D$ is the concatenation of all trainable parameters, where $D = md + m$. We can verify that $f_{\boldsymbol{\theta}}(\boldsymbol{x})$ is 2-homogeneous with respect to $\boldsymbol{\theta}$.

Let $\mathcal{S} := \{(\boldsymbol{x}_1, y_1), \dots, (\boldsymbol{x}_n, y_n)\}$ be the training set. For simplicity, we assume that $\|\boldsymbol{x}_i\|_2 \leq 1$. We focus on linearly separable data, thus we assume that $\mathcal{S}$ is linearly separable throughout the paper.

**Assumption 3.2** (Linear Separable)**.** There exists a $\boldsymbol{w} \in \mathbb{R}^d$ such that $y_i \langle \boldsymbol{w}, \boldsymbol{x}_i \rangle \geq 1$ for all $i \in [n]$.

**Definition 3.3** (Max-margin Linear Separator)**.** For the linearly separable dataset $\mathcal{S}$, we say that $\boldsymbol{w}^* \in \mathbb{S}^{d-1}$ is the max-margin linear separator if $\boldsymbol{w}^*$ maximizes $\min_{i \in [n]} y_i \langle \boldsymbol{w}, \boldsymbol{x}_i \rangle$ over $\boldsymbol{w} \in \mathbb{S}^{d-1}$.

## 4 Training on Linearly Separable and Symmetric Data

In this section, we study the implicit bias of gradient flow assuming the training data is linearly separable and *symmetric*. We say a dataset is symmetric if whenever $\boldsymbol{x}$ is present in the training set, the input $-\boldsymbol{x}$ is also present. By linear separability, $\boldsymbol{x}$ and $-\boldsymbol{x}$ must have different labels because $\langle \boldsymbol{w}^*, \boldsymbol{x} \rangle = -\langle \boldsymbol{w}^*, -\boldsymbol{x} \rangle$, where $\boldsymbol{w}^*$ is the max-margin linear separator. The formal statement for this assumption is given below.

**Assumption 4.1** (Symmetric)**.** $n$ is even and $\boldsymbol{x}_i = -\boldsymbol{x}_{i+n/2}, y_i = 1, y_{i+n/2} = -1$ for $1 \leq i \leq n/2$.

This symmetry can be ensured via data augmentation. Given a dataset, if it is known that the ground-truth labels are produced by an unknown linear classifier, then one can augment each data point $(\boldsymbol{x}, y)$ by flipping the sign, i.e., replace it with two data points $(\boldsymbol{x}, y), (-\boldsymbol{x}, -y)$ (and thus the dataset size is doubled).

Our results show that gradient flow directionally converges to a global-max-margin direction for two-layer Leaky ReLU networks, when the dataset is linearly separable and symmetric. To achieve such result, the key insight is that any global-max-margin direction represents a linear classifier, which we will see in Section 4.1. Then we will present our main convergence results in Section 4.2.

## 4.1 Global-Max-Margin Classifiers are Linear

Theorem 4.2 below characterizes the global-max-margin direction in our case by showing that margin maximization and simplicity bias coincide with each other: a network that representing the *max-margin linear classifier* (i.e., $f_{\boldsymbol{\theta}}(\boldsymbol{x}) = c\langle \boldsymbol{w}^*, \boldsymbol{x}\rangle$ for some $c > 0$) can simultaneously achieve the goals of being simple and maximizing the margin.

**Theorem 4.2.** *Under Assumptions 3.2 and 4.1, for the two-layer Leaky ReLU network with width $m \geq 2$, any global-max-margin direction $\boldsymbol{\theta}^* \in \mathbb{S}^{D-1}$, $f_{\boldsymbol{\theta}^*}$ represents a linear classifier. Moreover, we have $f_{\boldsymbol{\theta}^*}(\boldsymbol{x}) = \frac{1+\alpha_{\text{leaky}}}{4}\langle \boldsymbol{w}^*, \boldsymbol{x}\rangle$ for all $\boldsymbol{x} \in \mathbb{R}^d$, where $\boldsymbol{w}^*$ is the max-margin linear separator.*

The result of Theorem 4.2 is based on the observation that replacing each neuron $(a_k, \boldsymbol{w}_k)$ in a network with two neurons of oppositing parameters $(a_k, \boldsymbol{w}_k)$ and $(-a_k, -\boldsymbol{w}_k)$ does not decrease the normalized margin on the symmetric dataset, while making the classifier linear in function space. Thus if any direction attains the global max margin, we can construct a new global-max-margin direction which corresponds to a linear classifier. We can show that every weight vector $\boldsymbol{w}_k$ of this linear classifier must be in the direction of $\boldsymbol{w}^*$ or $-\boldsymbol{w}^*$. Then the original classifier must also be linear in the same direction.

## 4.2 Convergence to Global-Max-Margin Directions

Though Theorem 3.1 guarantees that gradient flow directionally converges to a KKT-margin direction if the loss is optimized successfully, we note that KKT-margin directions can be non-linear and have complicated decision boundaries. See Figure 1 (left) for an example. Therefore, to establish the convergence to linear classifiers, Theorem 3.1 is not enough and we need a new analysis for the trajectory of gradient flow.

We use initialization $\boldsymbol{w}_k \overset{\text{i.i.d.}}{\sim} \mathcal{N}(\boldsymbol{0}, \sigma_{\text{init}}^2 \boldsymbol{I})$, $a_k \overset{\text{i.i.d.}}{\sim} \mathcal{N}(0, c_{\text{ainit}}^2 \sigma_{\text{init}}^2)$, where $c_{\text{ainit}}$ is a fixed constant throughout this paper and $\sigma_{\text{init}}$ controls the initialization scale. We call this distribution as $\boldsymbol{\theta}_0 \sim \mathcal{D}_{\text{init}}(\sigma_{\text{init}})$. An alternative way to generate this distribution is to first draw $\bar{\boldsymbol{\theta}}_0 \sim \mathcal{D}_{\text{init}}(1)$, and then set $\boldsymbol{\theta}_0 = \sigma_{\text{init}}\bar{\boldsymbol{\theta}}_0$. With small initialization, we can establish the following convergence result.

**Theorem 4.3.** *Under Assumptions 3.2 and 4.1 and certain regularity conditions (see Assumptions 4.5 and 4.6 below), consider gradient flow on a Leaky ReLU network with width $m \geq 2$ and initialization $\boldsymbol{\theta}_0 = \sigma_{\text{init}}\bar{\boldsymbol{\theta}}_0$ where $\bar{\boldsymbol{\theta}}_0 \sim \mathcal{D}_{\text{init}}(1)$. With probability $1 - 2^{-(m-1)}$ over the random draw of $\bar{\boldsymbol{\theta}}_0$, if the initialization scale is sufficiently small, then gradient flow directionally converges and $f^\infty(\boldsymbol{x}) := \lim_{t \to +\infty} f_{\boldsymbol{\theta}(t)/\|\boldsymbol{\theta}(t)\|_2}(\boldsymbol{x})$ represents the max-margin linear classifier. That is,*

$$\Pr_{\bar{\boldsymbol{\theta}}_0 \sim \mathcal{D}_{\text{init}}(1)}\left[\exists \sigma_{\text{init}}^{\max} > 0 \text{ s.t. } \forall \sigma_{\text{init}} < \sigma_{\text{init}}^{\max}, \forall \boldsymbol{x} \in \mathbb{R}^d, f^\infty(\boldsymbol{x}) = C\langle \boldsymbol{w}^*, \boldsymbol{x}\rangle\right] \geq 1 - 2^{-(m-1)},$$

*where $C := \frac{1+\alpha_{\text{leaky}}}{4}$ is a scaling factor.*

Combining Theorem 4.2 and Theorem 4.3, we can conclude that gradient flow achieves the global max margin in our case.

**Corollary 4.4.** *In the settings of Theorem 4.3, gradient flow on linearly separable and symmetric data directionally converges to the global-max-margin direction with probability $1 - 2^{-(m-1)}$.*

## 4.3 Additional Notations and Assumptions

Let $\boldsymbol{\mu} := \frac{1}{n}\sum_{i=1}^n y_i \boldsymbol{x}_i$, which is non-zero since $\langle \boldsymbol{\mu}, \boldsymbol{w}_*\rangle = \frac{1}{n}\sum_{i\in[n]} y_i \boldsymbol{w}_*^\top \boldsymbol{x}_i \geq 1$. Let $\bar{\boldsymbol{\mu}} := \frac{\boldsymbol{\mu}}{\|\boldsymbol{\mu}\|_2}$. We use $\varphi(\boldsymbol{\theta}_0, t) \in \mathbb{R}^d$ to the value of $\boldsymbol{\theta}$ at time $t$ for $\boldsymbol{\theta}(0) = \boldsymbol{\theta}_0$.

We make the following technical assumption, which holds if we are allowed to add a slight perturbation to the training set.

**Assumption 4.5.** For all $i \in [n]$, $\langle \boldsymbol{\mu}, \boldsymbol{x}_i \rangle \neq 0$.

Another technical issue we face is that the gradient flow may not be unique due to non-smoothness. It is possible that $\varphi(\boldsymbol{\theta}_0, t)$ is not well-defined as the solution of (1) may not be unique. See Appendix I.2 for more discussions. In this case, we assign $\varphi(\boldsymbol{\theta}_0, \cdot)$ to be an arbitrary gradient flow trajectory starting from $\boldsymbol{\theta}_0$. In the case where $\varphi(\boldsymbol{\theta}_0, t)$ has only one possible value for all $t \geq 0$, we say that $\boldsymbol{\theta}_0$ is a *non-branching starting point*. We assume the following technical assumption.

**Assumption 4.6.** For any $m \geq 2$, there exist $r, \epsilon > 0$ such that $\boldsymbol{\theta}$ is a non-branching starting point if its neurons can be partitioned into two groups: in the first group, $a_k = \|\boldsymbol{w}_k\|_2 \in (0, r)$ and all $\boldsymbol{w}_k$ point to the same direction $\boldsymbol{w}^+ \in \mathbb{S}^{d-1}$ with $\|\boldsymbol{w}^+ - \bar{\boldsymbol{\mu}}\|_2 \leq \epsilon$; in the second group, $-a_k = \|\boldsymbol{w}_k\|_2 \in (0, r)$ and all $\boldsymbol{w}_k$ point to the same direction $\boldsymbol{w}^- \in \mathbb{S}^{d-1}$ with $\|\boldsymbol{w}^- + \bar{\boldsymbol{\mu}}\|_2 \leq \epsilon$.

## 5 Proof Sketch for the Symmetric Case

In this section, we provide a proof sketch for Theorem 4.3. Our proof uses a multi-phase analysis, which divides the training process into 3 phases, from small initialization to the final convergence. We will now elaborate the analyses for them one by one.

### 5.1 Phase I: Dynamics Near Zero

Gradient flow starts with small initialization. In Phase I, we analyze the dynamics when gradient flow does not go far away from zero. Inspired by Li et al. (2021), we relate such dynamics to power iterations and show that every weight vector $\boldsymbol{w}_k$ in the first layer moves towards the directions of either $\bar{\boldsymbol{\mu}}$ or $-\bar{\boldsymbol{\mu}}$. To see this, the first step is to note that $f_{\boldsymbol{\theta}}(\boldsymbol{x}_i) \approx 0$ when $\boldsymbol{\theta}$ is close to $\mathbf{0}$. Applying Taylor expansion on $\ell(y_i f_{\boldsymbol{\theta}}(\boldsymbol{x}_i))$,

$$\mathcal{L}(\boldsymbol{\theta}) = \frac{1}{n} \sum_{i \in [n]} \ell(y_i f_{\boldsymbol{\theta}}(\boldsymbol{x}_i)) \approx \frac{1}{n} \sum_{i \in [n]} \left( \ell(0) + \ell'(0) y_i f_{\boldsymbol{\theta}}(\boldsymbol{x}_i) \right). \tag{2}$$

Expanding $f_{\boldsymbol{\theta}}(\boldsymbol{x}_i)$ and reorganizing the terms, we have

$$\mathcal{L}(\boldsymbol{\theta}) \approx \frac{1}{n} \sum_{i \in [n]} \ell(0) + \frac{1}{n} \sum_{i \in [n]} \ell'(0) \sum_{k \in [m]} y_i a_k \phi(\boldsymbol{w}_k^\top \boldsymbol{x}_i) = \ell(0) + \frac{\ell'(0)}{n} \sum_{k \in [m]} \sum_{i \in [n]} y_i a_k \phi(\boldsymbol{w}_k^\top \boldsymbol{x}_i)$$

$$= \ell(0) - \sum_{k \in [m]} a_k G(\boldsymbol{w}_k),$$

where $G$-function (Maennel et al., 2018) is defined below:

$$G(\boldsymbol{w}) := \frac{-\ell'(0)}{n} \sum_{i \in [n]} y_i \phi(\boldsymbol{w}^\top \boldsymbol{x}_i) = \frac{1}{2n} \sum_{i \in [n]} y_i \phi(\boldsymbol{w}^\top \boldsymbol{x}_i).$$

This means gradient flow optimizes each $-a_k G(\boldsymbol{w}_k)$ separately near origin.

$$\frac{\mathrm{d}\boldsymbol{w}_k}{\mathrm{d}t} \approx a_k \partial^\circ G(\boldsymbol{w}_k), \qquad \frac{\mathrm{d}a_k}{\mathrm{d}t} \approx G(\boldsymbol{w}_k). \tag{3}$$

In the case where Assumption 4.1 holds, we can pair each $\boldsymbol{x}_i$ with $-\boldsymbol{x}_i$ and use the identity $\phi(z) - \phi(-z) = \max\{z, \alpha_{\text{leaky}} z\} - \max\{-z, -\alpha_{\text{leaky}} z\} = (1 + \alpha_{\text{leaky}})z$ to show that $G(\boldsymbol{w})$ is linear:

$$G(\boldsymbol{w}) = \frac{1}{2n} \sum_{i \in [n/2]} \left( \phi(\boldsymbol{w}^\top \boldsymbol{x}_i) - \phi(-\boldsymbol{w}^\top \boldsymbol{x}_i) \right) = \frac{1}{2n} \sum_{i \in [n/2]} (1 + \alpha_{\text{leaky}}) \boldsymbol{w}^\top \boldsymbol{x}_i = \langle \boldsymbol{w}, \tilde{\boldsymbol{\mu}} \rangle,$$

where $\tilde{\boldsymbol{\mu}} := \frac{1+\alpha_{\text{leaky}}}{2} \boldsymbol{\mu} = \frac{1+\alpha_{\text{leaky}}}{2n} \sum_{i \in [n]} y_i \boldsymbol{x}_i$. Substituting this formula for $G$ into (3) reveals that the dynamics of two-layer neural nets near zero has a close relationship to power iteration (or matrix exponentiation) of a matrix $\boldsymbol{M}_{\tilde{\boldsymbol{\mu}}} \in \mathbb{R}^{(d+1) \times (d+1)}$ that only depends on data.

$$\frac{\mathrm{d}}{\mathrm{d}t} \begin{bmatrix} \boldsymbol{w}_k \\ a_k \end{bmatrix} \approx \boldsymbol{M}_{\tilde{\boldsymbol{\mu}}} \begin{bmatrix} \boldsymbol{w}_k \\ a_k \end{bmatrix}, \qquad \text{where} \qquad \boldsymbol{M}_{\tilde{\boldsymbol{\mu}}} := \begin{bmatrix} \mathbf{0} & \tilde{\boldsymbol{\mu}} \\ \tilde{\boldsymbol{\mu}}^\top & 0 \end{bmatrix}.$$

Simple linear algebra shows that $\lambda_0 := \|\tilde{\boldsymbol{\mu}}\|_2, \frac{1}{\sqrt{2}}(\bar{\boldsymbol{\mu}}, 1) \in \mathbb{R}^{d+1}$ are the unique top eigenvalue and eigenvector of $\boldsymbol{M}_{\tilde{\boldsymbol{\mu}}}$, which suggests that $(\boldsymbol{w}_k(t), a_k(t)) \in \mathbb{R}^{d+1}$ aligns to this top eigenvector direction if the approximation (3) holds for a sufficiently long time. With small initialization, this can indeed be true and we obtain the following lemma.

**Definition 5.1** (M-norm). For parameter vector $\boldsymbol{\theta} = (\boldsymbol{w}_1, \ldots, \boldsymbol{w}_m, a_1, \ldots, a_m)$, we define the M-norm to be $\|\boldsymbol{\theta}\|_{\mathrm{M}} = \max_{k \in [m]} \{\max\{\|\boldsymbol{w}_k\|_2, |a_k|\}\}$.

**Lemma 5.2.** *Let $r > 0$ be a small value. With probability $1$ over the random draw of $\bar{\boldsymbol{\theta}}_0 = (\bar{\boldsymbol{w}}_1, \ldots, \bar{\boldsymbol{w}}_m, \bar{a}_1, \ldots, \bar{a}_m) \sim \mathcal{D}_{\mathrm{init}}(1)$, if we take $\sigma_{\mathrm{init}} \leq \frac{r^3}{\sqrt{m}\|\bar{\boldsymbol{\theta}}_0\|_{\mathrm{M}}}$, then any neuron $(\boldsymbol{w}_k, a_k)$ at time $T_1(r) := \frac{1}{\lambda_0} \ln \frac{r}{\sqrt{m}\sigma_{\mathrm{init}}\|\bar{\boldsymbol{\theta}}_0\|_{\mathrm{M}}}$ can be decomposed into*

$$\boldsymbol{w}_k(T_1(r)) = r\bar{b}_k\bar{\boldsymbol{\mu}} + \Delta\boldsymbol{w}_k, \qquad a_k(T_1(r)) = r\bar{b}_k + \Delta a_k,$$

*where $\bar{b}_k := \frac{\langle\bar{\boldsymbol{w}}_k, \bar{\boldsymbol{\mu}}\rangle + \bar{a}_k}{2\sqrt{m}\|\bar{\boldsymbol{\theta}}_0\|_{\mathrm{M}}}$ and the error term $\Delta\boldsymbol{\theta} := (\Delta\boldsymbol{w}_1, \ldots, \Delta\boldsymbol{w}_m, \Delta a_1, \ldots, \Delta a_m)$ is bounded by $\|\Delta\boldsymbol{\theta}\|_{\mathrm{M}} \leq \frac{Cr^3}{\sqrt{m}}$ for some universal constant C.*

## 5.2 Phase II: Near-Two-Neuron Dynamics

By Lemma 5.2, we know that at time $T_1(r)$ we have $\boldsymbol{w}_k(T_1(r)) \approx r\bar{b}_k\bar{\boldsymbol{\mu}}$ and $a_k(T_1(r)) \approx r\bar{b}_k$, where $\bar{\boldsymbol{b}} \in \mathbb{R}^d$ is some fixed vector. This motivates us to couple the training dynamics of $\boldsymbol{\theta}(t) = (\boldsymbol{w}_1(t), \ldots, \boldsymbol{w}_m(t), a_1(t), \ldots, a_m(t))$ after the time $T_1(r)$ with another gradient flow starting from the point $(r\bar{b}_1\bar{\boldsymbol{\mu}}, \ldots, r\bar{b}_m\bar{\boldsymbol{\mu}}, r\bar{b}_1, \ldots, r\bar{b}_m)$. Interestingly, the latter dynamic can be seen as a dynamic of two neurons "embedded" into the $m$-neuron neural net, and we will show that $\boldsymbol{\theta}(t)$ is close to this "embedded" two-neuron dynamic for a long time. Now we first introduce our idea of embedding a two-neuron network into an $m$-neuron network.

**Embedding.** For any $\boldsymbol{b} \in \mathbb{R}^m$, we say that $\boldsymbol{b}$ is a *good embedding vector* if it has at least one positive entry and one negative entry, and all the entries are non-zero. For a good embedding vector $\boldsymbol{b}$, we use $b_+ := \sqrt{\sum_{j \in [m]} \mathbb{1}_{[b_j > 0]} b_j^2}$ and $b_- := -\sqrt{\sum_{j \in [m]} \mathbb{1}_{[b_j < 0]} b_j^2}$ to denote the root-sum-squared of the positive entries and the negative root-sum-squared of the negative entries. For parameter $\hat{\boldsymbol{\theta}} := (\hat{\boldsymbol{w}}_1, \hat{\boldsymbol{w}}_2, \hat{a}_1, \hat{a}_2)$ of a two-neuron neural net with $\hat{a}_1 > 0$ and $\hat{a}_2 < 0$, we define the *embedding* from two-neuron into $m$-neuron neural nets as $\pi_{\boldsymbol{b}}(\hat{\boldsymbol{w}}_1, \hat{\boldsymbol{w}}_2, \hat{a}_1, \hat{a}_2) = (\boldsymbol{w}_1, \ldots, \boldsymbol{w}_m, a_1, \ldots, a_m)$, where

$$a_k = \begin{cases} \frac{b_k}{b_+}\hat{a}_1, & \text{if } b_k > 0 \\ \frac{b_k}{b_-}\hat{a}_2, & \text{if } b_k < 0 \end{cases}, \qquad \boldsymbol{w}_k = \begin{cases} \frac{b_k}{b_+}\hat{\boldsymbol{w}}_1, & \text{if } b_k > 0 \\ \frac{b_k}{b_-}\hat{\boldsymbol{w}}_2, & \text{if } b_k < 0 \end{cases}.$$

It is easy to check that $f_{\hat{\boldsymbol{\theta}}}(\boldsymbol{x}) = f_{\pi_{\boldsymbol{b}}(\hat{\boldsymbol{\theta}})}(\boldsymbol{x})$ by the homogeneity of the activation ($\phi(cz) = c\phi(z)$ for $c > 0$):

$$f_{\pi_{\boldsymbol{b}}(\hat{\boldsymbol{\theta}})}(\boldsymbol{x}) = \sum_{b_k > 0} a_k\phi(\boldsymbol{w}_k^\top\boldsymbol{x}) + \sum_{b_k < 0} a_k\phi(\boldsymbol{w}_k^\top\boldsymbol{x})$$

$$= \sum_{b_k > 0} \frac{b_k^2}{b_+^2}\hat{a}_1\phi(\hat{\boldsymbol{w}}_1^\top\boldsymbol{x}) + \sum_{b_k < 0} \frac{b_k^2}{b_-^2}\hat{a}_2\phi(\hat{\boldsymbol{w}}_2^\top\boldsymbol{x}) = \hat{a}_1\phi(\hat{\boldsymbol{w}}_1^\top\boldsymbol{x}) + \hat{a}_2\phi(\hat{\boldsymbol{w}}_2^\top\boldsymbol{x}) = f_{\hat{\boldsymbol{\theta}}}(\boldsymbol{x}).$$

Moreover, by taking the chain rule, we can obtain the following lemma showing that the trajectories starting from $\hat{\boldsymbol{\theta}}$ and $\pi_{\boldsymbol{b}}(\hat{\boldsymbol{\theta}})$ are essentially the same.

**Lemma 5.3.** *Given $\hat{\boldsymbol{\theta}} := (\hat{\boldsymbol{w}}_1, \hat{\boldsymbol{w}}_2, \hat{a}_1, \hat{a}_2)$ with $\hat{a}_1 > 0$ and $\hat{a}_2 < 0$, if both $\hat{\boldsymbol{\theta}}$ and $\pi_{\boldsymbol{b}}(\hat{\boldsymbol{\theta}})$ are non-branching starting points, then $\varphi(\pi_{\boldsymbol{b}}(\hat{\boldsymbol{\theta}}), t) = \pi_{\boldsymbol{b}}(\varphi(\hat{\boldsymbol{\theta}}, t))$ for all $t \geq 0$.*

**Approximate Embedding.** Back to our analysis for Phase II, $\bar{\boldsymbol{b}}$ is a good embedding vector with high probability (see lemma below). Let $\hat{\boldsymbol{\theta}} := (\bar{b}_+, \bar{b}_+\bar{\boldsymbol{\mu}}, \bar{b}_-, \bar{b}_-\bar{\boldsymbol{\mu}})$. By Lemma 5.2, $\pi_{\bar{\boldsymbol{b}}}(r\hat{\boldsymbol{\theta}}) = (r\bar{b}_1\bar{\boldsymbol{\mu}}, \ldots, r\bar{b}_m\bar{\boldsymbol{\mu}}, r\bar{b}_1, \ldots, r\bar{b}_m) \approx \boldsymbol{\theta}(T_1(r))$, which means $r\hat{\boldsymbol{\theta}} \to \boldsymbol{\theta}(T_1(r))$ is approximately an embedding. Suppose that the approximation happens to be exact, namely $\pi_{\bar{\boldsymbol{b}}}(r\hat{\boldsymbol{\theta}}) = \boldsymbol{\theta}(T_1(r))$, then $\boldsymbol{\theta}(T_1(r) + t) = \pi_{\bar{\boldsymbol{b}}}(\varphi(r\hat{\boldsymbol{\theta}}, t))$ by Lemma 5.3. Inspired by this, we consider the case where $\sigma_{\mathrm{init}} \to 0, r \to 0$ so that the approximate embedding is infinitely close to the exact one, and prove the following lemma. We shift the training time by $T_2(r)$ to avoid trivial limits (such as $\boldsymbol{0}$).

**Lemma 5.4.** *Follow the notations in Lemma 5.2 and take $\sigma_{\text{init}} \leq \frac{r^3}{\sqrt{m}\|\bar{\theta}_0\|_{\text{M}}}$. Let $T_2(r) := \frac{1}{\lambda_0} \ln \frac{1}{r}$, then $T_{12} := T_1(r) + T_2(r) = \frac{1}{\lambda_0} \ln \frac{1}{\sqrt{m}\sigma_{\text{init}}\|\bar{\theta}_0\|_{\text{M}}}$ regardless the choice of $r$. For width $m \geq 2$, with probability $1 - 2^{-(m-1)}$ over the random draw of $\bar{\theta}_0 \sim \mathcal{D}_{\text{init}}(1)$, the vector $\bar{b} \in \mathbb{R}^m$ is a good embedding vector, and for the two-neuron dynamics starting with rescaled initialization in the direction of $\hat{\theta} := (\bar{b}_+, \bar{b}_+\bar{\mu}, \bar{b}_-, \bar{b}_-\bar{\mu})$, the following limit exists for all $t$,*

$$\tilde{\theta}(t) := \lim_{r \to 0} \varphi\left(r\hat{\theta}, T_2(r) + t\right) \neq \mathbf{0}, \tag{4}$$

*and moreover, for the $m$-neuron dynamics of $\theta(t)$, the following holds for all $t$,*

$$\lim_{\sigma_{\text{init}} \to 0} \theta\left(T_{12} + t\right) = \pi_{\bar{b}}(\tilde{\theta}(t)). \tag{5}$$

### 5.3 Phase III: Dynamics near Global-Max-Margin Direction

With some efforts, we have the following characterization for the two-neuron dynamics.

**Theorem 5.5.** *For $m = 2$, if initially $a_1 = \|w_1\|_2$, $a_2 = -\|w_2\|_2$, $\langle w_1, w^* \rangle > 0$ and $\langle w_2, w^* \rangle < 0$, then $\theta(t)$ directionally converges to the following global-max-margin direction,*

$$\lim_{t \to +\infty} \frac{\theta(t)}{\|\theta(t)\|_2} = \frac{1}{4}(w^*, -w^*, 1, -1),$$

*where $w^*$ is the max-margin linear separator.*

It is not hard to verify that $\tilde{\theta}(t)$ satisfies the conditions required by Theorem 5.5. Given this result, a first attempt to establish the convergence of $\theta(t)$ to global-max-margin direction is to take $t \to +\infty$ on both sides of (5). However, this only proves that $\theta(T_{12} + t)$ directionally converges to the global-max-margin direction if we take the limit $\sigma_{\text{init}} \to 0$ first then take $t \to +\infty$, while we are interested in the convergent solution when $t \to +\infty$ first then $\sigma_{\text{init}} \to 0$ (i.e., solution gradient flow converges to with infinite training time, if it starts from sufficiently small initialization). These two double limits are not equivalent because the order of limits cannot be exchanged without extra conditions.

To overcome this issue, we follow a similar proof strategy as Ji and Telgarsky (2020a) to prove local convergence near a local-max-margin direction, as formally stated below. Theorem 5.6 holds for $L$-homogeneous neural networks in general and we believe is of independent interest.

**Theorem 5.6.** *Consider any $L$-homogeneous neural networks with logistic loss. Given a local-max-margin direction $\bar{\theta}^* \in \mathbb{S}^{D-1}$ and any $\delta > 0$, there exists $\epsilon_0 > 0$ and $\rho_0 \geq 1$ such that for any $\theta_0$ with norm $\|\theta_0\|_2 \geq \rho_0$ and direction $\left\|\frac{\theta_0}{\|\theta_0\|_2} - \bar{\theta}^*\right\|_2 \leq \epsilon_0$, gradient flow starting with $\theta_0$ directionally converges to some direction $\bar{\theta}$ with the same normalized margin $\gamma$ as $\bar{\theta}^*$, and $\|\bar{\theta} - \bar{\theta}^*\|_2 \leq \delta$.*

Using Theorem 5.6, we can finish the proof for Theorem 4.3 as follows. First we note that the two-neuron global-max-margin direction $\frac{1}{4}(w^*, -w^*, 1, -1)$ after embedding is a global-max-margin direction for $m$-neurons, and we can prove that any direction with distance no more than a small constant $\delta > 0$ is still a global-max-margin direction. Then we can take $t$ to be large enough so that $\pi_{\bar{b}}(\tilde{\theta}(t))$ satisfies the conditions in Theorem 5.6. According to (5), we can also make the conditions hold for $\theta(T_{12} + t)$ by taking $\sigma_{\text{init}}$ and $r$ to be sufficiently small. Finally, applying Theorem 5.6 finishes the proof.

## 6 Non-symmetric Data Complicates the Picture

Now we turn to study the case without assuming symmetry and the question is whether the implicit bias to global-max-margin solution still holds. Unfortunately, it turns out the convergence to global-max-margin classifier is very fragile — for any linearly separable dataset, we can add 3 extra data points so that every linear classifier has suboptimal margin but still gradient flow with small initialization converges to a linear classifier.[3] See Definition 6.1 for the construction and Figure 1 (right) for an example.

---

[3]Here linear classifier refers to a classifier whose decision boundary is linear.

Unlike the symmetric case, we use balanced Gaussian initialization instead of purely random Gaussian initialization: $\boldsymbol{w}_k \sim \mathcal{N}(\boldsymbol{0}, \sigma_{\text{init}}^2 \boldsymbol{I})$, $a_k = s_k \|\boldsymbol{w}_k\|_2$, where $s_k \sim \text{unif}\{\pm 1\}$. We call this distribution as $\boldsymbol{\theta}_0 \sim \tilde{\mathcal{D}}_{\text{init}}(\sigma_{\text{init}})$. This adaptation can greatly simplify our analysis since it ensures that $a_k(t) = s_k \|\boldsymbol{w}_k(t)\|_2$ for all $t \geq 0$ (Corollary B.18). Similar as the symmetric case, an alternative way to generate this distribution is to first draw $\bar{\boldsymbol{\theta}}_0 \sim \tilde{\mathcal{D}}_{\text{init}}(1)$, and then set $\boldsymbol{\theta}_0 = \sigma_{\text{init}} \bar{\boldsymbol{\theta}}_0$.

**Definition 6.1** $((H, K \epsilon, \boldsymbol{w}_\perp)$-Hinted Dataset)**.** Given a linearly separable dataset $\mathcal{S}$ with max-margin linear separator $\boldsymbol{w}^*$, for constants $H, K, \epsilon > 0$ and unit vector $\boldsymbol{w}_\perp \in \mathbb{S}^{d-1}$ perpendicular to $\boldsymbol{w}^*$, we define the $(H, K, \epsilon, \boldsymbol{w}_\perp)$-hinted dataset $\mathcal{S}'$ by the dataset containing all the data points in $\mathcal{S}$ and the following 3 data points (numbered by $1, 2, 3$) that can serve as hints to the max-margin linear separator $\boldsymbol{w}^*$:

$$(\boldsymbol{x}_1, y_1) = (H\boldsymbol{w}^*, 1), \qquad (\boldsymbol{x}_2, y_2) = (\epsilon\boldsymbol{w}^* + K\boldsymbol{w}_\perp, 1), \qquad (\boldsymbol{x}_3, y_3) = (\epsilon\boldsymbol{w}^* - K\boldsymbol{w}_\perp, 1).$$

**Theorem 6.2.** *Given a linearly separable dataset $\mathcal{S}$ and a unit vector $\boldsymbol{w}_\perp \in \mathbb{S}^{d-1}$ perpendicular to the max-margin linear separator $\boldsymbol{w}^*$, for any sufficiently large $H > 0, K > 0$ and sufficiently small $\epsilon > 0$, the following statement holds for the $(H, K, \epsilon, \boldsymbol{w}_\perp)$-Hinted Dataset $\mathcal{S}'$. Under a regularity assumption for gradient flow (see Assumption A.6), consider gradient flow on a Leaky ReLU network with width $m \geq 1$ and initialization $\boldsymbol{\theta}_0 = \sigma_{\text{init}} \bar{\boldsymbol{\theta}}_0$ where $\bar{\boldsymbol{\theta}}_0 \sim \tilde{\mathcal{D}}_{\text{init}}(1)$. With probability $1 - 2^{-m}$ over the draw of $\bar{\boldsymbol{\theta}}_0$, if the initialization scale is sufficiently small, then gradient flow directionally converges and $f^\infty(\boldsymbol{x}) := \lim_{t \to +\infty} f_{\boldsymbol{\theta}(t)/\|\boldsymbol{\theta}(t)\|_2}(\boldsymbol{x})$ represents the one-Leaky-ReLU classifier $\frac{1}{2}\phi(\langle \boldsymbol{w}^*, \boldsymbol{x}\rangle)$ with linear decision boundary. That is,*

$$\Pr_{\bar{\boldsymbol{\theta}}_0 \sim \mathcal{D}_{\text{init}}(1)} \left[ \exists \sigma_{\text{init}}^{\max} > 0 \text{ s.t. } \forall \sigma_{\text{init}} < \sigma_{\text{init}}^{\max}, \forall \boldsymbol{x} \in \mathbb{R}^d, f^\infty(\boldsymbol{x}) = \frac{1}{2}\phi(\langle \boldsymbol{w}^*, \boldsymbol{x}\rangle) \right] \geq 1 - \delta.$$

*Moreover, the convergent classifier only attains a suboptimal margin.*

Theorem 6.2 is actually a simple corollary general theorem under data assumptions that hold for a broader class of linearly separable data. From a high-level perspective, we only require two assumptions: (1). There is a direction such that data points have large inner products with this direction on average; (2). The support vectors for the max-margin linear separator $\boldsymbol{w}^*$ have nearly the same labels. The first hint data point is for the first condition and the second and third data point is for the second condition. We defer formal statements of the assumptions and theorems to Appendix A.

## 7 Conclusions and Future Works

We study the implicit bias of gradient flow in training two-layer Leaky ReLU networks on linearly separable datasets. When the dataset is symmetric, we show any global-max-margin classifier is exactly linear and gradient flow converges to a global-max-margin direction. On the pessimistic side, we show such margin maximization result is fragile — for any linearly separable dataset, we can lead gradient flow to converge to a linear classifier with suboptimal margin by adding only 3 extra data points. A critical assumption for our convergence analysis is the linear separability of data. We left it as a future work to study simplicity bias and global margin maximization without assuming linear separability.

## Acknowledgments and Disclosure of Funding

The authors acknowledge support from NSF, ONR, Simons Foundation, DARPA and SRC. ZL is also supported by Microsoft Research PhD Fellowship.

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
