# A Theorem Statements for the Non-symmetric Case

## A.1 Assumptions and Main Theorems

For every $\boldsymbol{x}_i$, define $\boldsymbol{x}_i^+ := \boldsymbol{x}_i$ if $y_i = 1$ and $\boldsymbol{x}_i^+ := \alpha_{\text{leaky}}\boldsymbol{x}_i$ if $y_i = -1$. Similarly, we define $\boldsymbol{x}_i^- := \alpha_{\text{leaky}}\boldsymbol{x}_i$ if $y_i = 1$ and $\boldsymbol{x}_i^+ := \boldsymbol{x}_i$ if $y_i = -1$. Then we define $\boldsymbol{\mu}^+$ to be the mean vector of $y_i\boldsymbol{x}_i^+$, and $\boldsymbol{\mu}^-$ to be the mean vector of $y_i\boldsymbol{x}_i^-$, that is,

$$\boldsymbol{\mu}^+ := \frac{1}{n}\sum_{i\in[n]}y_i\boldsymbol{x}_i^+, \qquad \boldsymbol{\mu}^- := \frac{1}{n}\sum_{i\in[n]}y_i\boldsymbol{x}_i^-. \tag{6}$$

Theorem 6.2 is indeed a simple corollary of Theorem A.7 below which holds for a broader class of datasets. Now we illustrate the assumptions one by one.

We first make the following assumption saying that there is a principal direction $\boldsymbol{w}^\diamond \in \mathbb{S}^{d-1}$ such that data points on average have much larger inner products with $\boldsymbol{w}^\diamond$ than any other direction perpendicular to $\boldsymbol{w}^\diamond$. This ensures at small initialization, the moving direction of each neurons lies in a small cone around the the direction of $\pm\boldsymbol{w}^\diamond$, and thus will converge to that cone eventually. The opening angle of this small cone is $2\arcsin\frac{\gamma^\diamond}{\max_{i\in[n]}\|\boldsymbol{x}_i\|_2}$, which ensures the sign pattern inside the cone $\{\langle\boldsymbol{w},\boldsymbol{x}_i\rangle\}_{i=1}^n$ is unique and indeed equal to $\{y_i\}_{i=1}^n$, and thus all neurons converge to two directions, $\boldsymbol{\mu}^+$ and $\boldsymbol{\mu}^-$ (defined in (6)).

**Assumption A.1** (Existence of Principal Direction). There exists a unit-norm vector $\boldsymbol{w}^\diamond$ such that $\gamma^\diamond := \min_{i\in[n]} y_i\langle\boldsymbol{w}^\diamond,\boldsymbol{x}_i\rangle > 0$ and

$$\frac{\frac{1}{n}\sum_{i\in[n]}\|\boldsymbol{P}^\diamond\boldsymbol{x}_i\|_2}{\alpha_{\text{leaky}}\langle\boldsymbol{\mu},\boldsymbol{w}^\diamond\rangle} < \frac{\gamma^\diamond}{\max_{i\in[n]}\|\boldsymbol{P}^\diamond\boldsymbol{x}_i\|_2},$$

where $\boldsymbol{P}^\diamond := \boldsymbol{I} - \boldsymbol{w}^\diamond\boldsymbol{w}^{\diamond\top}$ is the projection matrix onto the space perpendicular to $\boldsymbol{w}^\diamond$, and $\boldsymbol{\mu} := \frac{1}{n}\sum_{i\in[n]}y_i\boldsymbol{x}_i$ is the mean vector of $y_i\boldsymbol{x}_i$.

Indeed, our main theorem is based on a weaker assumption than Assumption A.1, which is Assumption A.2 below, but the geometric meaning of Assumption A.2 is not as clear as Assumption A.1. We will show in Lemma G.1 that Assumption A.1 implies Assumption A.2.

**Assumption A.2.** For all $i \in [n]$, we have

$$\langle\boldsymbol{\mu},y_i\boldsymbol{x}_i\rangle > \frac{1-\alpha_{\text{leaky}}}{n\cdot\alpha_{\text{leaky}}}\sum_{j\in[n]}\max\{-\langle y_i\boldsymbol{x}_i,y_j\boldsymbol{x}_j\rangle,0\}.$$

In general, the norms $\|\boldsymbol{\mu}^+\|_2$ and $\|\boldsymbol{\mu}^-\|_2$ should not be equal: for any given dataset $\mathcal{S}$, we can make $\|\boldsymbol{\mu}^+\|_2 \neq \|\boldsymbol{\mu}^-\|_2$ by adding arbitrarily small perturbations to the data points. This motivates us to assume that $\|\boldsymbol{\mu}^+\|_2 \neq \|\boldsymbol{\mu}^-\|_2$. Without loss of generality, we can assume that $\|\boldsymbol{\mu}^+\|_2 > \|\boldsymbol{\mu}^-\|_2$ for convenience (Assumption A.3). When the reverse is true, i.e., $\|\boldsymbol{\mu}^+\|_2 < \|\boldsymbol{\mu}^-\|_2$, we can change the direction of the inequality by flipping all the labels in the dataset so that our theorems can apply. We include the theorem statements for this reversed case in Appendix A.3.

**Assumption A.3.** The norm of $\boldsymbol{\mu}^+$ is strictly larger than $\boldsymbol{\mu}^-$, i.e., $\|\boldsymbol{\mu}^+\|_2 > \|\boldsymbol{\mu}^-\|_2$.

Now we define $\boldsymbol{w}^+$ to be the max-margin linear separator of the dataset consisting of $(\boldsymbol{x}_i^+, y_i)$, where $i \in [n]$, and define $\gamma^+$ to be this max margin. That is,

$$\boldsymbol{w}^+ := \arg\max_{\boldsymbol{w}\in\mathbb{S}^{d-1}}\left\{\min_{i\in[n]}y_i\langle\boldsymbol{w},\boldsymbol{x}_i^+\rangle\right\}, \qquad \gamma^+ := \max_{\boldsymbol{w}\in\mathbb{S}^{d-1}}\left\{\min_{i\in[n]}y_i\langle\boldsymbol{w},\boldsymbol{x}_i^+\rangle\right\}.$$

The reason that we care about $\boldsymbol{w}^+$ and $\gamma^+$ is because that it can be related to margin maximization on one-neuron Leaky ReLU nets. The following lemma is easy to prove.

**Lemma A.4.** For $m = 1$, if $\boldsymbol{\theta} = (\boldsymbol{w}_1, a_1) \in \mathbb{S}^{D-1}$ is a KKT-margin direction and $a_1 \geq 0$, then $\boldsymbol{\theta} = (\frac{1}{\sqrt{2}}\boldsymbol{w}^+, \frac{1}{\sqrt{2}})$, and it attains the global max margin $\frac{1}{2}\gamma^+$.

The third assumption we made is that this margin cannot be obtained when all $a_i$ are negative, regardless of the width. This assumption holds when all the support vectors $\boldsymbol{x}_i^+$ have positive labels, i.e., $y_i = 1$. Conceptually, this assumption is about whether nearly all the support vectors have positive labels (or negative labels in the reversed case where $\|\boldsymbol{\mu}^+\|_2 < \|\boldsymbol{\mu}^-\|_2$).

**Assumption A.5.** For any $m \geq 1$ and any $\boldsymbol{\theta} = (\boldsymbol{w}_1, \ldots, \boldsymbol{w}_m, a_1, \ldots, a_m) \in \mathbb{R}^D$, if $a_k \leq 0$ for all $k \in [m]$, then the normalized margin $\gamma(\boldsymbol{\theta})$ on the dataset $\{(\boldsymbol{x}_i, y_i) : i \in [n], y_i \langle \boldsymbol{w}^+, \boldsymbol{x}_i^+ \rangle = \gamma^+\}$ is less than $\frac{1}{2}\gamma^+$.

Similar to Assumption 4.6 in the symmetric case, we need Assumption A.6 on non-branching starting point due to the technical difficulty for the potential non-uniqueness of gradient flow trajectory.

**Assumption A.6.** For any $m \geq 1$, there exist $r, \epsilon > 0$ such that $\boldsymbol{\theta}$ is a non-branching starting point if $a_k = \|\boldsymbol{w}_k\|_2 \in (0, r)$ holds for all $k \in [m]$, and all $\boldsymbol{w}_k$ point to the same direction $\boldsymbol{v} \in \mathbb{S}^{d-1}$ with $\left\| \boldsymbol{v} - \frac{\boldsymbol{\mu}^+}{\|\boldsymbol{\mu}^+\|_2} \right\|_2 \leq \epsilon$.

Now we are ready to state our theorem, and we defer the proofs to Appendix G.

**Theorem A.7.** *Under Assumptions 3.2, A.2, A.3, A.5 and A.6, consider gradient flow on a Leaky ReLU network with width $m \geq 1$ and initialization $\boldsymbol{\theta}_0 = \sigma_{\mathrm{init}}\bar{\boldsymbol{\theta}}_0$ where $\bar{\boldsymbol{\theta}}_0 \sim \tilde{\mathcal{D}}_{\mathrm{init}}(1)$. With probability $1 - 2^{-m}$ over the draw of $\bar{\boldsymbol{\theta}}_0$, if the initialization scale is sufficiently small, then gradient flow directionally converges and $f^\infty(\boldsymbol{x}) := \lim_{t \to +\infty} f_{\boldsymbol{\theta}(t)/\|\boldsymbol{\theta}(t)\|_2}(\boldsymbol{x})$ represents the one-Leaky-ReLU classifier $\frac{1}{2}\phi(\langle \boldsymbol{w}^+, \boldsymbol{x} \rangle)$ with linear decision boundary. That is,*

$$\Pr_{\bar{\boldsymbol{\theta}}_0 \sim \mathcal{D}_{\mathrm{init}}(1)} \left[ \exists \sigma_{\mathrm{init}}^{\max} > 0 \text{ s.t. } \forall \sigma_{\mathrm{init}} < \sigma_{\mathrm{init}}^{\max}, \forall \boldsymbol{x} \in \mathbb{R}^d, f^\infty(\boldsymbol{x}) = \frac{1}{2}\phi(\langle \boldsymbol{w}^+, \boldsymbol{x} \rangle) \right] \geq 1 - 2^{-m}.$$

## A.2 Applying Theorem A.7 to prove Theorem 6.2

We give a proof of Theorem 6.2 here given the result of Theorem A.7.

*Proof.* With a $(H, K \epsilon, \boldsymbol{w}_\perp)$-Hinted Dataset (Definition 6.1) with proper $H, K, \epsilon$, we only need to show that Assumptions A.2, A.3 and A.5 hold for Theorem 6.2. Specifically, we choose the parameters such that

- $K > 0$;

- $\epsilon < \alpha_{\mathrm{leaky}} \min_{i>3} y_i \langle \boldsymbol{w}^*, \boldsymbol{x}_i \rangle$;

- $H > \max\{\epsilon, H_0, n \|\boldsymbol{\mu}^-\|_2 + \|\sum_{j>1} y_j \boldsymbol{x}_j^+\|_2\}$, where
  $H_0 = \frac{\max_{i \in [n]} \|\boldsymbol{P}^* \boldsymbol{x}_i\|_2 \sum_{i \in [n]} \|\boldsymbol{P}^* \boldsymbol{x}_i\|_2}{\alpha_{\mathrm{leaky}} \min_{i>1} \langle y_i \boldsymbol{x}_i, \boldsymbol{w}^* \rangle} - \sum_{i>1} \langle \boldsymbol{w}^*, y_i \boldsymbol{x}_i \rangle$ and $\boldsymbol{P}^* = \boldsymbol{I} - \boldsymbol{w}^* \boldsymbol{w}^{*\top}$ is the projection matrix onto the orthogonal space of $\boldsymbol{w}^*$.

Notice that $H_0$ is indepenent of $H$ as the data point $\boldsymbol{x}_1$ has projection $\|\boldsymbol{P}^* x_1\|_2 = 0$. For Assumption A.1, $\boldsymbol{w}^\diamond = \boldsymbol{w}^*$ is a valid principal direction in this case, as

$$\max_{i \in [n]} \|\boldsymbol{P}^\diamond \boldsymbol{x}_i\|_2 \frac{\frac{1}{n} \sum_{i \in [n]} \|\boldsymbol{P}^\diamond \boldsymbol{x}_i\|_2}{\alpha_{\mathrm{leaky}} \gamma^\diamond} = \frac{1}{n}(H_0 + \sum_{i>1} \langle w^*, y_i \boldsymbol{x}_i \rangle) < \langle \boldsymbol{\mu}, \boldsymbol{w}^\diamond \rangle.$$

Then Assumption A.2 follows from Assumption A.1 by Lemma G.1. Since $H > n \|\boldsymbol{\mu}^-\|_2 + \|\sum_{j>1} y_j \boldsymbol{x}_j^+\|_2$,

$$\|\boldsymbol{\mu}^+\|_2 \geq \frac{1}{n} H - \left\| \frac{1}{n} \sum_{j>1} y_j \boldsymbol{x}_j^+ \right\|_2 > \|\boldsymbol{\mu}^-\|_2,$$

and thus Assumption A.3 holds. Furthermore, with $\epsilon < \alpha_{\mathrm{leaky}} \min_{i>3} y_i \langle \boldsymbol{w}^*, \boldsymbol{x}_i \rangle$ and $H > \epsilon$, $(\boldsymbol{x}_2, y_2) = (\epsilon \boldsymbol{w}^* + K \boldsymbol{w}_\perp, 1)$ and $(\boldsymbol{x}_3, y_3) = (\epsilon \boldsymbol{w}^* - K \boldsymbol{w}_\perp, 1)$ are the only support vectors for the linear margin problem on $\{(\boldsymbol{x}_i, y_i)\}$ and that on $\{(\boldsymbol{x}_i^+, y_i)\}$ as well. Then $\boldsymbol{w}^+ = \boldsymbol{w}^*$ and $\gamma^+ = \epsilon$. For a neuron with $a_k < 0$, the total output margin on the hints $(\boldsymbol{x}_2, y_2)$ and $(\boldsymbol{x}_3, y_3)$ is $a_k \phi(\boldsymbol{w}_k^\top \boldsymbol{x}_2) + a_k \phi(\boldsymbol{w}_k^\top \boldsymbol{x}_3) \leq 2\alpha_{\mathrm{leaky}} \epsilon |a_k| \|\boldsymbol{w}_k\|_2 \leq \alpha_{\mathrm{leaky}} \epsilon (a_k^2 + \|\boldsymbol{w}_k\|_2^2)$. Thus the normalized margin for multiple such neurons is at most $\frac{\alpha_{\mathrm{leaky}} \epsilon}{2} < \frac{\epsilon}{2}$, so Assumption A.5 will also be true. $\square$

## A.3 Results in the Reversed Case

In a reversed case where $\|\boldsymbol{\mu}^+\|_2 < \|\boldsymbol{\mu}^-\|_2$, we can apply Theorem A.7 by flipping the labels in the dataset. Below we state the assumptions and the theorem in the reversed case.

**Assumption A.8.** $\|\boldsymbol{\mu}^+\|_2 < \|\boldsymbol{\mu}^-\|_2$.

Now similarly we define $\boldsymbol{w}^-$ and $\gamma^-$.

$$\boldsymbol{w}^- := \underset{\boldsymbol{w} \in \mathbb{S}^{d-1}}{\arg\max} \left\{ \min_{i \in [n]} y_i \left\langle \boldsymbol{w}, \boldsymbol{x}^- \right\rangle \right\}, \qquad \gamma^- := \max_{\boldsymbol{w} \in \mathbb{S}^{d-1}} \left\{ \min_{i \in [n]} y_i \left\langle \boldsymbol{w}, \boldsymbol{x}_i^- \right\rangle \right\}.$$

**Assumption A.9.** For any $m \geq 1$ and any $\boldsymbol{\theta} = (\boldsymbol{w}_1, \ldots, \boldsymbol{w}_m, a_1, \ldots, a_m) \in \mathbb{R}^D$, if $a_k \leq 0$ for all $k \in [m]$, then the normalized margin $\gamma(\boldsymbol{\theta})$ on the dataset $\{(\boldsymbol{x}_i, y_i) : i \in [n], y_i \langle \boldsymbol{w}^-, \boldsymbol{x}_i^- \rangle = \gamma^-\}$ is less than $\frac{1}{2}\gamma^-$.

**Theorem A.10.** *Under Assumptions 3.2, A.2, A.6, A.8 and A.9, consider gradient flow on a Leaky ReLU network with width $m \geq 1$ and initialization $\boldsymbol{\theta}_0 = \sigma_{\mathrm{init}}\bar{\boldsymbol{\theta}}_0$ where $\bar{\boldsymbol{\theta}}_0 \sim \tilde{\mathcal{D}}_{\mathrm{init}}(1)$. With probability $1 - 2^{-m}$ over the draw of $\bar{\boldsymbol{\theta}}_0$, there is an sufficiently small initialization scale, such that gradient flow directionally converges and $f^\infty(\boldsymbol{x}) := \lim_{t \to +\infty} f_{\boldsymbol{\theta}(t)/\|\boldsymbol{\theta}(t)\|_2}(\boldsymbol{x})$ represents the one-Leaky-ReLU classifier $-\frac{1}{2}\phi(-\langle \boldsymbol{w}^-, \boldsymbol{x}\rangle)$ with linear decision boundary. That is,*

$$\Pr_{\bar{\boldsymbol{\theta}}_0 \sim \tilde{\mathcal{D}}_{\mathrm{init}}(1)} \left[ \exists \sigma_{\mathrm{init}}^{\max} > 0 \ s.t. \ \forall \sigma_{\mathrm{init}} < \sigma_{\mathrm{init}}^{\max}, \forall \boldsymbol{x} \in \mathbb{R}^d, f^\infty(\boldsymbol{x}) = -\frac{1}{2}\phi(-\langle \boldsymbol{w}^-, \boldsymbol{x}\rangle) \right] \geq 1 - 2^{-m}.$$

# B  Additional Preliminaries and Lemmas

In this section, we will introduce additional notations and give some preliminary results for the dynamics of the two-layer Leaky ReLU network. The only assumption we will use for the results in the section is that the input norm is bounded $\max_{i \in [n]} \|\boldsymbol{x}_i\|_2 \leq 1$ and we do not assume other properties of the dataset (such as symmetry) except we assume it explicitly.

## B.1  Additional Notations

For notational convenience for calculation with subgradients, we generalize the following notations for vectors to vector sets. More specifically, we define

- $\forall A, B \subseteq \mathbb{R}^d$, $A + B := \{\boldsymbol{x} + \boldsymbol{y} : \boldsymbol{x} \in A, \boldsymbol{y} \in B\}$ and $A - B := A + (-B)$;
- $\forall A \subseteq \mathbb{R}^d, \lambda \in \mathbb{R}, \lambda A := \{\lambda \boldsymbol{x} : \boldsymbol{x} \in A\}$;
- Let $\|\cdot\|$ be any norm on $\mathbb{R}^d, \forall A \subseteq \mathbb{R}^d, \|A\| := \{\|x\| : x \in A\} \subseteq \mathbb{R}$;
- $\forall A \subseteq \mathbb{R}^d$ and $\boldsymbol{y} \in \mathbb{R}^d, \langle \boldsymbol{y}, A \rangle \equiv \langle A, \boldsymbol{y} \rangle := \{\langle \boldsymbol{x}, \boldsymbol{y} \rangle : \boldsymbol{x} \in A\}$;
- We use $\mathrm{dist}(\boldsymbol{x}, \boldsymbol{y}) := \|\boldsymbol{x} - \boldsymbol{y}\|_2$ to denote the $L^2$-distance between $\boldsymbol{x} \in \mathbb{R}^d$ and $\boldsymbol{y} \in \mathbb{R}^d$, $\mathrm{dist}(A, \boldsymbol{y}) := \inf_{\boldsymbol{x} \in A} \|\boldsymbol{x} - \boldsymbol{y}\|_2$ to denote the minimum $L^2$-distance between any $\boldsymbol{x} \in A$ and $\boldsymbol{y} \in \mathbb{R}^d$, and $\mathrm{dist}(A, B) := \inf_{\boldsymbol{x} \in A, \boldsymbol{y} \in B} \|\boldsymbol{x} - \boldsymbol{y}\|_2$ to denote the minimum $L^2$-distance between any $\boldsymbol{x} \in A$ and any $\boldsymbol{y} \in B$.

By Rademacher theorem, any real-valued locally Lipschitz function on $\mathbb{R}^D$ is differentiable almost everywhere (a.e.) in the sense of Lebesgue measure. For a locally Lipschitz function $\mathcal{L} : \mathbb{R}^D \to \mathbb{R}$, we use $\nabla\mathcal{L}(\boldsymbol{\theta}) \in \mathbb{R}^D$ to denote the usual gradient (if $\mathcal{L}$ is differentiable at $\boldsymbol{\theta}$) and $\partial^\circ\mathcal{L}(\boldsymbol{\theta}) \subseteq \mathbb{R}^D$ to denote Clarke's subdifferential. The definition of Clarke's subdifferential is given by (7): for any sequence of differentiable points converging to $\boldsymbol{\theta}$, we collect convergent gradients from such sequences and take the convex hull as the Clarke's subdifferential at $\boldsymbol{\theta}$.

$$\partial^\circ\mathcal{L}(\boldsymbol{\theta}) := \mathrm{conv}\left\{ \lim_{n \to \infty} \nabla\mathcal{L}(\boldsymbol{\theta}_n) : \mathcal{L} \text{ differentiable at } \boldsymbol{\theta}_n, \lim_{n \to \infty} \boldsymbol{\theta}_n = \boldsymbol{\theta} \right\}. \tag{7}$$

For any full measure set $\Omega \subseteq \mathbb{R}^D$ that does not contain any non-differentiable points, (7) also has the following equivalent form:

$$\partial^\circ\mathcal{L}(\boldsymbol{\theta}) = \mathrm{conv}\left\{ \lim_{n \to \infty} \nabla\mathcal{L}(\boldsymbol{\theta}_n) : \boldsymbol{\theta}_n \in \Omega \text{ for all } n \text{ and } \lim_{n \to \infty} \boldsymbol{\theta}_n = \boldsymbol{\theta} \right\}. \tag{8}$$

The Clarke's subdifferential $\partial^\circ \mathcal{L}(\boldsymbol{\theta})$ is convex compact if $\mathcal{L}$ is locally Lipschitz, and it is upper-semicontinuous with respect to $\boldsymbol{\theta}$ (or equivalently it has closed graph) if $\mathcal{L}$ is definable. We use $\bar{\partial}^\circ \mathcal{L}(\boldsymbol{\theta}) \in \mathbb{R}^D$ to denote the min-norm gradient vector in the Clarke's subdifferential at $\boldsymbol{\theta}$, i.e., $\bar{\partial}^\circ \mathcal{L}(\boldsymbol{\theta}) := \arg\min_{\boldsymbol{g} \in \partial^\circ \mathcal{L}(\boldsymbol{\theta})} \|\boldsymbol{g}\|_2$. If $\mathcal{L}$ is continuously differentiable at $\boldsymbol{\theta}$, then $\partial^\circ \mathcal{L}(\boldsymbol{\theta}) = \{\nabla \mathcal{L}(\boldsymbol{\theta})\}$ and $\bar{\partial}^\circ \mathcal{L}(\boldsymbol{\theta}) = \nabla \mathcal{L}(\boldsymbol{\theta})$.

If $\boldsymbol{\theta}$ can be written as $\boldsymbol{\theta} = (\boldsymbol{\theta}_1, \boldsymbol{\theta}_2) \in \mathbb{R}^{D_1} \times \mathbb{R}^{D_2}$, then we use $\frac{\partial \mathcal{L}(\boldsymbol{\theta})}{\partial \boldsymbol{\theta}_1} \in \mathbb{R}^{D_1}$ to denote the usual partial derivatives (partial gradient) and $\frac{\partial^\circ \mathcal{L}(\boldsymbol{\theta})}{\partial \boldsymbol{\theta}_1} \subseteq \mathbb{R}^{D_1}$ to denote the partial subderivatives (partial subgradient) in the sense of Clarke.

Furthermore, we use the following notations to denote the radial and spherical components of $\bar{\partial}^\circ \mathcal{L}(\boldsymbol{\theta})$ (which will be used in analyzing Phase III):

$$\bar{\partial}_{\mathrm{r}}^\circ \mathcal{L}(\boldsymbol{\theta}) := \frac{\boldsymbol{\theta}\boldsymbol{\theta}^\top}{\|\boldsymbol{\theta}\|_2^2} \bar{\partial}^\circ \mathcal{L}(\boldsymbol{\theta}), \qquad \bar{\partial}_{\perp}^\circ \mathcal{L}(\boldsymbol{\theta}) := \left( \boldsymbol{I} - \frac{\boldsymbol{\theta}\boldsymbol{\theta}^\top}{\|\boldsymbol{\theta}\|_2^2} \right) \bar{\partial}^\circ \mathcal{L}(\boldsymbol{\theta}).$$

For univariate function $f : \mathbb{R} \to \mathbb{R}$, we use $f'(z) \in \mathbb{R}$ to denote the usual derivative (if $f$ is differentiable at $z$) and $f^\circ(z) \subseteq \mathbb{R}$ to denote the Clarke's subdifferential.

The logistic loss is defined by $\ell(q) = \ln(1 + e^{-q})$, which satisfies $\ell(0) = \ln 2$, $\ell'(0) = -1/2$, $|\ell'(q)| \leq 1$, $|\ell''(q)| \leq 1$. Given a dataset $\mathcal{S} = \{(\boldsymbol{x}_1, y_1), \ldots, (\boldsymbol{x}_n, y_n)\}$, we consider gradient flow on two-layer Leaky ReLU network with output function $f_{\boldsymbol{\theta}}(\boldsymbol{x}_i)$ and logistic loss $\mathcal{L}(\boldsymbol{\theta}) := \frac{1}{n} \sum_{i \in [n]} \ell(q_i(\boldsymbol{\theta}))$, where $q_i(\boldsymbol{\theta}) := y_i f_{\boldsymbol{\theta}}(\boldsymbol{x}_i)$. Following Davis et al. (2020); Lyu and Li (2020), we say that a function $\boldsymbol{z}(t) \in \mathbb{R}^D$ on an interval $I$ is an *arc* if $\boldsymbol{z}$ is absolutely continuous on any compact subinterval of $I$. An arc $\boldsymbol{\theta}(t)$ is a trajectory of gradient flow on $\mathcal{L}$ if $\boldsymbol{\theta}(t)$ satisfies the following gradient inclusion for a.e. $t \geq 0$:

$$\frac{\mathrm{d}\boldsymbol{\theta}(t)}{\mathrm{d}t} \in -\partial^\circ \mathcal{L}(\boldsymbol{\theta}(t)).$$

Let $\Omega_{\mathcal{S}}$ be the set of parameter vectors $\boldsymbol{\theta} = (\boldsymbol{w}_1, \ldots, \boldsymbol{w}_m, a_1, \ldots, a_m)$ so that $\langle \boldsymbol{w}_k, \boldsymbol{x}_i \rangle \neq 0$ for all $i \in [n], k \in [m]$, i.e., no activation function has zero input. For any $\boldsymbol{\theta} \in \Omega_{\mathcal{S}}$, $f_{\boldsymbol{\theta}}(\boldsymbol{x}_i)$ and $\mathcal{L}(\boldsymbol{\theta})$ are continuously differentiable at $\boldsymbol{\theta}$, and the gradients are given by

$$\frac{\partial f_{\boldsymbol{\theta}}(\boldsymbol{x})}{\partial \boldsymbol{w}_k} = a_k \phi'(\boldsymbol{w}_k^\top \boldsymbol{x}_i) \boldsymbol{x}_i, \qquad\qquad \frac{\partial f_{\boldsymbol{\theta}}(\boldsymbol{x})}{\partial a_k} = \phi(\boldsymbol{w}_k^\top \boldsymbol{x}_i). \qquad (9)$$

$$\frac{\partial \mathcal{L}(\boldsymbol{\theta})}{\partial \boldsymbol{w}_k} = \frac{1}{n} \sum_{i \in [n]} \ell'(q_i(\boldsymbol{\theta})) y_i a_k \phi'(\boldsymbol{w}_k^\top \boldsymbol{x}_i) \boldsymbol{x}_i, \qquad \frac{\partial \mathcal{L}(\boldsymbol{\theta})}{\partial a_k} = \frac{1}{n} \sum_{i \in [n]} \ell'(q_i(\boldsymbol{\theta})) y_i \phi(\boldsymbol{w}_k^\top \boldsymbol{x}_i). \qquad (10)$$

Then the Clarke's subdifferential for any $\boldsymbol{\theta}$ can be computed from (8) with $\Omega = \Omega_{\mathcal{S}}$ if needed.

Recall that $G$-function (Section 5.1) is defined by

$$G(\boldsymbol{w}) := \frac{-\ell'(0)}{n} \sum_{i \in [n]} y_i \phi(\boldsymbol{w}^\top \boldsymbol{x}_i) = \frac{1}{2n} \sum_{i \in [n]} y_i \phi(\boldsymbol{w}^\top \boldsymbol{x}_i).$$

Define $\tilde{\mathcal{L}}(\boldsymbol{\theta})$ to the linear approximation of $\mathcal{L}(\boldsymbol{\theta})$:

$$\tilde{\mathcal{L}}(\boldsymbol{\theta}) := \ell(0) - \sum_{k \in [m]} a_k G(\boldsymbol{w}_k).$$

For every $\boldsymbol{\theta}_0 \in \mathbb{R}^D$, we define $\varphi(\boldsymbol{\theta}_0, t)$ to be the value of $\boldsymbol{\theta}(t)$ for gradient flow on $\mathcal{L}(\boldsymbol{\theta})$ starting with $\boldsymbol{\theta}(0) = \theta_0$. For every $\tilde{\boldsymbol{\theta}}_0 \in \mathbb{R}^D$, we define $\tilde{\varphi}(\tilde{\boldsymbol{\theta}}_0, t)$ to be the value of $\tilde{\boldsymbol{\theta}}(t)$ for gradient flow on $\tilde{\mathcal{L}}(\tilde{\boldsymbol{\theta}})$ starting with $\tilde{\boldsymbol{\theta}}(0) = \tilde{\boldsymbol{\theta}}_0$. In the case where the gradient flow trajectory may not be unique, we assign $\varphi(\boldsymbol{\theta}_0, \cdot)$ (or $\tilde{\varphi}(\tilde{\boldsymbol{\theta}}_0, \cdot)$) by an arbitrary trajectory of gradient flow on $\mathcal{L}$ (or $\tilde{\mathcal{L}}$) starting from $\boldsymbol{\theta}_0$ (or $\tilde{\boldsymbol{\theta}}_0$).

### B.2 Grönwall's Inequality

We frequently use Grönwall's inequality in our analysis.

**Lemma B.1** (Grönwall's Inequality). *Let $\alpha, \beta, u$ be real-valued functions defined on $[a, b]$. Suppose that $\beta, u$ are continuous and $\min\{\alpha, 0\}$ is integrable on every compact subinterval of $[a, b]$. If $\beta \geq 0$ and $u$ satisfies the following inequality for all $t \in [a, b]$:*

$$u(t) \leq \alpha(t) + \int_a^t \beta(\tau) u(\tau) \mathrm{d}\tau,$$

*then for all $t \in [a, b]$,*

$$u(t) \leq \alpha(t) + \int_a^t \alpha(\tau) \beta(\tau) \exp\left(\int_\tau^t \beta(\tau') \mathrm{d}\tau'\right) \mathrm{d}\tau. \tag{11}$$

*Furthermore, if $\alpha$ is non-decreasing, then for all $t \in [a, b]$,*

$$u(t) \leq \alpha(t) \exp\left(\int_a^t \beta(\tau) \mathrm{d}\tau\right). \tag{12}$$

### B.3 Homogeneous Functions

For $L \geq 0$, we say that a function $f : \mathbb{R}^d \to \mathbb{R}$ is (positively) $L$-homogeneous if $f(c\boldsymbol{\theta}) = c^L f(\boldsymbol{\theta})$ for all $c > 0$ and $\boldsymbol{\theta} \in \mathbb{R}^d$. The proof for the following two theorems can be found in Lyu and Li (2020, Theorem B.2) and Ji and Telgarsky (2020a, Lemma C.1) respectively.

**Theorem B.2.** *For locally Lipschitz and $L$-homogeneous function $f : \mathbb{R}^d \to \mathbb{R}$, we have*

$$\partial^\circ f(c\boldsymbol{\theta}) = c^{L-1} \partial^\circ f(\boldsymbol{\theta}).$$

*for all $\boldsymbol{\theta} \in \mathbb{R}^d$.*

**Theorem B.3** (Euler's homogeneous function theorem). *For locally Lipschitz and $L$-homogeneous function $f : \mathbb{R}^d \to \mathbb{R}$, we have*

$$\forall \boldsymbol{g} \in \partial^\circ f(\boldsymbol{\theta}) : \quad \langle \boldsymbol{g}, \boldsymbol{\theta} \rangle = L f(\boldsymbol{\theta}),$$

*for all $\boldsymbol{\theta} \in \mathbb{R}^d$.*

For the maximizer of a homogeneous function on $\mathbb{S}^{d-1}$, we have the following useful lemma.

**Lemma B.4.** *For locally Lipschitz and $L$-homogeneous function $f : \mathbb{R}^d \to \mathbb{R}$, if $\boldsymbol{\theta} \in \mathbb{S}^{d-1}$ is a local/global maximizer of $f(\boldsymbol{\theta})$ on $\mathbb{S}^{d-1}$ and $f$ is differentiable at $\boldsymbol{\theta}$, then $\nabla f(\boldsymbol{\theta}) = L f(\boldsymbol{\theta}) \boldsymbol{\theta}$.*

*Proof.* Since $\boldsymbol{\theta}$ is a local/global maximizer of $f(\boldsymbol{\theta})$ on $\mathbb{S}^{d-1}$ and $f$ is differentiable at $\boldsymbol{\theta}$, $\nabla f(\boldsymbol{\theta})$ is parallel to $\boldsymbol{\theta}$, i.e., $\nabla f(\boldsymbol{\theta}) = c\boldsymbol{\theta}$ for some $c \in \mathbb{R}$. By Theorem B.3 we know that $\langle \nabla f(\boldsymbol{\theta}), \boldsymbol{\theta} \rangle = L f(\boldsymbol{\theta})$. So $c = L f(\boldsymbol{\theta})$. $\square$

The following is a direct corollary of Lemma B.4.

**Lemma B.5.** *If $\boldsymbol{w} \in \mathbb{S}^{d-1}$ attains the maximum of $|G(\boldsymbol{w})|$ on $\mathbb{S}^{d-1}$ and $G(\boldsymbol{w})$ is differentiable at $\boldsymbol{w}$, then $\nabla G(\boldsymbol{w}) = G(\boldsymbol{w}) \boldsymbol{w}$.*

*Proof.* Note that $G(\boldsymbol{w})$ is 1-homogeneous. If $\boldsymbol{w}$ attains the maximum of $|G(\boldsymbol{w})|$ on $\mathbb{S}^{d-1}$, then $\boldsymbol{w}$ is either a maximizer of $G(\boldsymbol{w})$ or $-G(\boldsymbol{w})$. Applying Lemma B.4 gives $\nabla G(\boldsymbol{w}) = G(\boldsymbol{w}) \boldsymbol{w}$. $\square$

### B.4 Karush-Kuhn-Tucker Conditions for Margin Maximization

**Definition B.6** (Feasible Point and KKT Point, Dutta et al. 2013; Lyu and Li 2020). Let $f, g_1, \ldots, g_n : \mathbb{R}^D \to \mathbb{R}$ be locally Lipschitz functions. Consider the following constrained optimization problem for $\boldsymbol{\theta} \in \mathbb{R}^D$:

$$\begin{aligned} \min \quad & f(\boldsymbol{\theta}) \\ \text{s.t.} \quad & g_i(\boldsymbol{\theta}) \leq 0, \qquad \forall i \in [n]. \end{aligned}$$

We say that $\boldsymbol{\theta}$ is a *feasible point* if $g_i(\boldsymbol{\theta}) \leq 0$ for all $i \in [n]$. A feasible point $\boldsymbol{\theta}$ is a *KKT point* if it satisfies Karush-Kuhn-Tucker Conditions: there exist $\lambda_1, \ldots, \lambda_n \geq 0$ such that

1. $\mathbf{0} \in \partial^\circ f(\boldsymbol{\theta}) + \sum_{i \in [n]} \lambda_i \partial^\circ g_i(\boldsymbol{\theta})$;

2. $\forall i \in [n] : \lambda_i g_i(\boldsymbol{\theta}) = 0$.

Recall that we say that a parameter vector $\boldsymbol{\theta} \in \mathbb{R}^D$ of a $L$-homogeneous network is along a KKT-margin direction if $\frac{\boldsymbol{\theta}}{(q_{\min}(\boldsymbol{\theta}))^{1/L}}$ is a KKT point of (P), where $f(\boldsymbol{\theta}) = \frac{1}{2}\|\boldsymbol{\theta}\|_2^2$ and $g_i(\boldsymbol{\theta}) = 1 - q_i(\boldsymbol{\theta})$. Alternatively, we can use the following equivalent definition.

**Definition B.7** (KKT-margin Direction for Homogeneous Network, Lyu and Li 2020)**.** For a parameter vector $\boldsymbol{\theta} \in \mathbb{R}^D$ of a homogeneous network, we say $\boldsymbol{\theta}$ is along a KKT-margin direction if $q_i(\boldsymbol{\theta}) > 0$ for all $i \in [n]$ and there exist $\lambda_1, \ldots, \lambda_n \geq 0$ such that

1. $\boldsymbol{\theta} \in \sum_{i \in [n]} \lambda_i \partial^\circ q_i(\boldsymbol{\theta})$;

2. For all $i \in [n]$, if $q_i(\boldsymbol{\theta}) \neq q_{\min}(\boldsymbol{\theta})$ then $\lambda_i = 0$.

For two-layer Leaky ReLU network, $q_i(\boldsymbol{\theta}) := y_i \sum_{k \in [m]} a_k \phi(\boldsymbol{w}_k^\top \boldsymbol{x}_i)$. Then the KKT-margin direction is defined as follows.

**Definition B.8** (KKT-margin Direction for Two-layer Leaky ReLU Network)**.** For a parameter vector $\boldsymbol{\theta} = (\boldsymbol{w}_1, \ldots, \boldsymbol{w}_m, a_1, \ldots, a_m) \in \mathbb{R}^D$ of a two-layer Leaky ReLU network, we say $\boldsymbol{\theta}$ is along a KKT-margin direction if $q_i(\boldsymbol{\theta}) > 0$ for all $i \in [n]$ and there exist $\lambda_1, \ldots, \lambda_n \geq 0$ such that

1. For all $k \in [m]$, $\boldsymbol{w}_k \in \sum_{i \in [n]} \lambda_i y_i a_k \phi^\circ(\boldsymbol{w}_k^\top \boldsymbol{x}_i) \boldsymbol{x}_i$;

2. For all $k \in [m]$, $a_k = \sum_{i \in [n]} \lambda_i y_i \phi(\boldsymbol{w}_k^\top \boldsymbol{x}_i)$;

3. For all $i \in [n]$, if $q_i(\boldsymbol{\theta}) \neq q_{\min}(\boldsymbol{\theta})$ then $\lambda_i = 0$.

For $\boldsymbol{\theta}$ along a KKT-margin direction of two-layer Leaky ReLU network, Lemma B.9 below shows that $|a_k| = \|\boldsymbol{w}_k\|_2$ for all $k \in [m]$.

**Lemma B.9.** *If $\boldsymbol{\theta} = (\boldsymbol{w}_1, \ldots, \boldsymbol{w}_m, a_1, \ldots, a_m) \in \mathbb{R}^D$ is along a KKT-margin direction of a two-layer Leaky ReLU network, then $|a_k| = \|\boldsymbol{w}_k\|_2$ for all $k \in [m]$.*

*Proof.* By Definition B.8 and Theorem B.3, we have

$$\|\boldsymbol{w}_k\|_2^2 \in \left\langle \boldsymbol{w}_k, \sum_{i \in [n]} \lambda_i y_i a_k \phi^\circ(\boldsymbol{w}_k^\top \boldsymbol{x}_i) \boldsymbol{x}_i \right\rangle = \left\{ \sum_{i \in [n]} \lambda_i y_i a_k \phi(\boldsymbol{w}_k^\top \boldsymbol{x}_i) \right\},$$

$$|a_k|^2 = a_k \cdot \sum_{i \in [n]} \lambda_i y_i \phi(\boldsymbol{w}_k^\top \boldsymbol{x}_i) = \sum_{i \in [n]} \lambda_i y_i a_k \phi(\boldsymbol{w}_k^\top \boldsymbol{x}_i).$$

Therefore $\|\boldsymbol{w}_k\|_2^2 = |a_k|^2$. $\qquad\qquad\square$

### B.5 Lemmas for Perturbation Bounds

Recall that $\|\boldsymbol{\theta}\|_M$ is defined in Definition 5.1.

**Lemma B.10.** *For $\|\boldsymbol{x}\|_2 \leq 1$, $|f_{\boldsymbol{\theta}}(\boldsymbol{x})| \leq m\|\boldsymbol{\theta}\|_M^2$, $|f_{\boldsymbol{\theta}}(\boldsymbol{x}) - f_{\tilde{\boldsymbol{\theta}}}(\boldsymbol{x})| \leq m\|\boldsymbol{\theta} - \tilde{\boldsymbol{\theta}}\|_M \left( \|\boldsymbol{\theta}\|_M + \|\tilde{\boldsymbol{\theta}}\|_M \right)$.*

*Proof.* The proof is straightforward by definition of $f_{\boldsymbol{\theta}}(\boldsymbol{x})$ and $\|\boldsymbol{\theta}\|_M$. For the first inequality,

$$|f_{\boldsymbol{\theta}}(\boldsymbol{x})| \leq \sum_{k=1}^m |a_k \phi(\boldsymbol{w}_k^\top \boldsymbol{x})| \leq \sum_{k=1}^m |a_k| \cdot |\boldsymbol{w}_k^\top \boldsymbol{x}| \leq \sum_{k=1}^m |a_k| \cdot \|\boldsymbol{w}_k\|_2 \leq m\|\boldsymbol{\theta}\|_M^2.$$

For the second inequality,

$$
\begin{aligned}
|f_{\boldsymbol{\theta}}(\boldsymbol{x}) - f_{\tilde{\boldsymbol{\theta}}}(\boldsymbol{x})| &\leq \sum_{k=1}^{m} |a_k \phi(\boldsymbol{w}_k^\top \boldsymbol{x}) - \tilde{a}_k \phi(\tilde{\boldsymbol{w}}_k^\top \boldsymbol{x})| \\
&\leq \sum_{k=1}^{m} |a_k \phi(\boldsymbol{w}_k^\top \boldsymbol{x}) - a_k \phi(\tilde{\boldsymbol{w}}_k^\top \boldsymbol{x})| + |a_k \phi(\tilde{\boldsymbol{w}}_k^\top \boldsymbol{x}) - \tilde{a}_k \phi(\tilde{\boldsymbol{w}}_k^\top \boldsymbol{x})| \\
&\leq \sum_{k=1}^{m} |a_k| \cdot \|\boldsymbol{w}_k - \tilde{\boldsymbol{w}}_k\|_2 + |a_k - \tilde{a}_k| \cdot \|\tilde{\boldsymbol{w}}_k\|_2 \\
&\leq m\|\boldsymbol{\theta} - \tilde{\boldsymbol{\theta}}\|_{\mathrm{M}} \left( \|\boldsymbol{\theta}\|_{\mathrm{M}} + \|\tilde{\boldsymbol{\theta}}\|_{\mathrm{M}} \right),
\end{aligned}
$$

which completes the proof. $\qquad\square$

We have the following bound for the difference between $\partial^\circ \mathcal{L}(\boldsymbol{\theta})$ and $\partial^\circ \tilde{\mathcal{L}}(\boldsymbol{\theta})$.

**Lemma B.11.** *Assume that $\|\boldsymbol{x}_i\|_2 \leq 1$ for all $i \in [n]$. For any $\boldsymbol{\theta} = (\boldsymbol{w}_1, \ldots, \boldsymbol{w}_m, a_1, \ldots, a_m) \in \mathbb{R}^D$, we have the following bounds for the partial derivatives of $\mathcal{L}(\boldsymbol{\theta}) - \tilde{\mathcal{L}}(\boldsymbol{\theta})$:*

$$
\left\| \frac{\partial^\circ (\mathcal{L}(\boldsymbol{\theta}) - \tilde{\mathcal{L}}(\boldsymbol{\theta}))}{\partial \boldsymbol{w}_k} \right\|_2 \subseteq \left( -\infty, m\|\boldsymbol{\theta}\|_{\mathrm{M}}^2 |a_k| \right], \qquad \left| \frac{\partial (\mathcal{L}(\boldsymbol{\theta}) - \tilde{\mathcal{L}}(\boldsymbol{\theta}))}{\partial a_k} \right| \leq m\|\boldsymbol{\theta}\|_{\mathrm{M}}^2 \|\boldsymbol{w}_k\|_2.
$$

*for all $k \in [m]$.*

*Proof.* We only need to prove the following bounds for gradients at any $\boldsymbol{\theta} \in \Omega_S$, i.e., $\langle \boldsymbol{w}_k, \boldsymbol{x}_i \rangle \neq 0$ for all $i \in [n], k \in [m]$. For the general case where $\boldsymbol{\theta}$ can be non-differentiable, we can prove the same bounds for Clarke's sub-differential at every point $\boldsymbol{\theta} \in \mathbb{R}^D$ by taking limits in $\Omega_S$ through (8).

$$
\left\| \frac{\partial (\mathcal{L}(\boldsymbol{\theta}) - \tilde{\mathcal{L}}(\boldsymbol{\theta}))}{\partial \boldsymbol{w}_k} \right\|_2 \leq m\|\boldsymbol{\theta}\|_{\mathrm{M}}^2 |a_k|, \qquad \left| \frac{\partial (\mathcal{L}(\boldsymbol{\theta}) - \tilde{\mathcal{L}}(\boldsymbol{\theta}))}{\partial a_k} \right| \leq m\|\boldsymbol{\theta}\|_{\mathrm{M}}^2 \|\boldsymbol{w}_k\|_2.
$$

By Taylor expansion, we have

$$
\ell(y_i f_{\boldsymbol{\theta}}(\boldsymbol{x}_i)) = \ell(0) + \ell'(0) y_i f_{\boldsymbol{\theta}}(\boldsymbol{x}_i) + \int_0^{y_i f_{\boldsymbol{\theta}}(\boldsymbol{x}_i)} \ell''(z)(y_i f_{\boldsymbol{\theta}}(\boldsymbol{x}_i) - z) \mathrm{d}z.
$$

Taking average over $i \in [n]$ gives

$$
\begin{aligned}
\mathcal{L}(\boldsymbol{\theta}) &= \ell(0) + \frac{1}{n} \sum_{i \in [n]} \ell'(0) y_i f_{\boldsymbol{\theta}}(\boldsymbol{x}_i) + \frac{1}{n} \sum_{i \in [n]} \int_0^{y_i f_{\boldsymbol{\theta}}(\boldsymbol{x}_i)} \ell''(z)(y_i f_{\boldsymbol{\theta}}(\boldsymbol{x}_i) - z) \mathrm{d}z \\
&= \tilde{\mathcal{L}}(\boldsymbol{\theta}) + \frac{1}{n} \sum_{i \in [n]} \int_0^{y_i f_{\boldsymbol{\theta}}(\boldsymbol{x}_i)} \ell''(z)(y_i f_{\boldsymbol{\theta}}(\boldsymbol{x}_i) - z) \mathrm{d}z.
\end{aligned}
$$

By Leibniz integral rule,

$$
\begin{aligned}
\nabla_{\boldsymbol{\theta}} \left( \mathcal{L}(\boldsymbol{\theta}) - \tilde{\mathcal{L}}(\boldsymbol{\theta}) \right) &= \nabla_{\boldsymbol{\theta}} \left( \frac{1}{n} \sum_{i \in [n]} \int_0^{y_i f_{\boldsymbol{\theta}}(\boldsymbol{x}_i)} \ell''(z)(y_i f_{\boldsymbol{\theta}}(\boldsymbol{x}_i) - z) \mathrm{d}z \right) \\
&= -\frac{1}{n} \sum_{i \in [n]} \int_0^{y_i f_{\boldsymbol{\theta}}(\boldsymbol{x}_i)} \ell''(z) y_i \nabla_{\boldsymbol{\theta}}(f_{\boldsymbol{\theta}}(\boldsymbol{x}_i)) \mathrm{d}z \\
&= -\frac{1}{n} \sum_{i \in [n]} \left( \int_0^{y_i f_{\boldsymbol{\theta}}(\boldsymbol{x}_i)} \ell''(z) \mathrm{d}z \right) y_i \nabla_{\boldsymbol{\theta}}(f_{\boldsymbol{\theta}}(\boldsymbol{x}_i)).
\end{aligned}
$$

Since $\ell''(z) \leq 1$, there exists $\delta_i \in [-|f_{\boldsymbol{\theta}}(\boldsymbol{x}_i)|, |f_{\boldsymbol{\theta}}(\boldsymbol{x}_i)|]$ for all $i \in [n]$ such that

$$
\nabla_{\boldsymbol{\theta}} \left( \mathcal{L}(\boldsymbol{\theta}) - \tilde{\mathcal{L}}(\boldsymbol{\theta}) \right) = \frac{1}{n} \sum_{i \in [n]} \delta_i \nabla_{\boldsymbol{\theta}}(f_{\boldsymbol{\theta}}(\boldsymbol{x}_i)). \tag{13}
$$

Writing the formula with respect to $\boldsymbol{w}_k, a_k$, we have

$$\left\|\frac{\partial(\mathcal{L}(\boldsymbol{\theta}) - \tilde{\mathcal{L}}(\boldsymbol{\theta}))}{\partial \boldsymbol{w}_k}\right\|_2 \leq \frac{1}{n} \sum_{i \in [n]} |\delta_i| \cdot \left\|\frac{\partial f_{\boldsymbol{\theta}}(\boldsymbol{x}_i)}{\partial \boldsymbol{w}_k}\right\|_2 \leq \frac{1}{n} \sum_{i \in [n]} |f_{\boldsymbol{\theta}}(\boldsymbol{x}_i)| \cdot \|a_k \phi'(\langle \boldsymbol{w}_k, \boldsymbol{x}_i \rangle) \boldsymbol{x}_i\|_2 \,.$$

$$\left|\frac{\partial(\mathcal{L}(\boldsymbol{\theta}) - \tilde{\mathcal{L}}(\boldsymbol{\theta}))}{\partial a_k}\right| \leq \frac{1}{n} \sum_{i \in [n]} |\delta_i| \cdot \left\|\frac{\partial f_{\boldsymbol{\theta}}(\boldsymbol{x}_i)}{\partial a_k}\right\|_2 \leq \frac{1}{n} \sum_{i \in [n]} |f_{\boldsymbol{\theta}}(\boldsymbol{x}_i)| \cdot |\phi(\langle \boldsymbol{w}_k, \boldsymbol{x}_i \rangle)| \,.$$

By Lemma B.10, $|f_{\boldsymbol{\theta}}(\boldsymbol{x}_i)| \leq m\|\boldsymbol{\theta}\|_{\mathrm{M}}^2$. Since Leaky ReLU is 1-Lipschitz and $\|\boldsymbol{x}_i\|_2 \leq 1$, we have $\|a_k \phi'(\langle \boldsymbol{w}_k, \boldsymbol{x}_i \rangle) \boldsymbol{x}_i\|_2 \leq |a_k|, |\phi(\langle \boldsymbol{w}_k, \boldsymbol{x}_i \rangle)| \leq \|\boldsymbol{w}_k\|_2$. Then we have

$$\left\|\frac{\partial(\mathcal{L}(\boldsymbol{\theta}) - \tilde{\mathcal{L}}(\boldsymbol{\theta}))}{\partial \boldsymbol{w}_k}\right\|_2 \leq \frac{1}{n} \sum_{i \in [n]} m\|\boldsymbol{\theta}\|_{\mathrm{M}}^2 \cdot |a_k| = m\|\boldsymbol{\theta}\|_{\mathrm{M}}^2 \cdot |a_k|,$$

$$\left|\frac{\partial(\mathcal{L}(\boldsymbol{\theta}) - \tilde{\mathcal{L}}(\boldsymbol{\theta}))}{\partial a_k}\right| \leq \frac{1}{n} \sum_{i \in [n]} m\|\boldsymbol{\theta}\|_{\mathrm{M}}^2 \cdot \|\boldsymbol{w}_k\|_2 = m\|\boldsymbol{\theta}\|_{\mathrm{M}}^2 \cdot \|\boldsymbol{w}_k\|_2,$$

which completes the proof for $\boldsymbol{\theta} \in \Omega_{\mathcal{S}}$ and thus the same bounds hold for the general case. $\qquad\square$

Lemma B.11 is a lemma for bounding the partial subderivatives. For the full subgradient, we have the following lemma.

**Lemma B.12.** *Assume that $\|\boldsymbol{x}_i\|_2 \leq 1$ for all $i \in [n]$. For any $\boldsymbol{\theta} \in \mathbb{R}^D$, we have*

$$\forall \boldsymbol{g} \in \partial^\circ \left(\mathcal{L}(\boldsymbol{\theta}) - \tilde{\mathcal{L}}(\boldsymbol{\theta})\right) : \qquad \|\boldsymbol{g}\|_{\mathrm{M}} \leq m\|\boldsymbol{\theta}\|_{\mathrm{M}}^3.$$

*Proof.* Note that $|a_k| \leq \|\boldsymbol{\theta}\|_{\mathrm{M}}$ and $\|\boldsymbol{w}_k\|_2 \leq \|\boldsymbol{\theta}\|_{\mathrm{M}}$. Combining this with Lemma B.11 gives $\|\partial^\circ \mathcal{L}(\boldsymbol{\theta})\|_{\mathrm{M}} \subseteq (-\infty, m\|\boldsymbol{\theta}\|_{\mathrm{M}}^3]$. $\qquad\square$

When $\tilde{\mathcal{L}}(\boldsymbol{\theta})$ is smooth, we have the following direct corollary.

**Corollary B.13.** *Assume that $\|\boldsymbol{x}_i\|_2 \leq 1$ for all $i \in [n]$. If $\tilde{\mathcal{L}}$ is continuously differentiable at $\boldsymbol{\theta} \in \mathbb{R}^D$, then we have*

$$\forall \boldsymbol{g} \in \left(\partial^\circ \mathcal{L}(\boldsymbol{\theta}) - \nabla \tilde{\mathcal{L}}(\boldsymbol{\theta})\right) : \qquad \|\boldsymbol{g}\|_{\mathrm{M}} \leq m\|\boldsymbol{\theta}\|_{\mathrm{M}}^3.$$

Note that $\partial^\circ(\mathcal{L}(\boldsymbol{\theta}) - \tilde{\mathcal{L}}(\boldsymbol{\theta})) \neq \partial^\circ \mathcal{L}(\boldsymbol{\theta}) - \nabla \tilde{\mathcal{L}}(\boldsymbol{\theta})$ because the exact sum rule does not hold for Clarke's subdifferential when $\tilde{\mathcal{L}}(\boldsymbol{\theta})$ is not smooth. In the non-smooth case, we have the following lemma:

**Lemma B.14.** *Assume that $\|\boldsymbol{x}_i\|_2 \leq 1$ for all $i \in [n]$. For any $\epsilon > 0$ and $\|\boldsymbol{\theta}\|_{\mathrm{M}} \leq \sqrt{\frac{\epsilon}{2m}}$, we have*

$$\forall k \in [m], \quad \frac{\partial^\circ \mathcal{L}(\boldsymbol{\theta})}{\partial \boldsymbol{w}_k} \subseteq \left\{-\frac{a_k}{2n} \sum_{i=1}^n (1 + \epsilon_i)\alpha_i y_i \boldsymbol{x}_i : \alpha_i \in \phi^\circ(\boldsymbol{w}_k^\top \boldsymbol{x}_i), \epsilon_i \in [-\epsilon, \epsilon], \forall i \in [n]\right\}.$$

*Proof.* If $\boldsymbol{\theta} \in \Omega_{\mathcal{S}}$, by (13), there exists $\delta_i \in [-|f_{\boldsymbol{\theta}}(\boldsymbol{x}_i)|, |f_{\boldsymbol{\theta}}(\boldsymbol{x}_i)|]$ for all $i \in [n]$ such that

$$\nabla \mathcal{L}(\boldsymbol{\theta}) = \nabla \tilde{\mathcal{L}}(\boldsymbol{\theta}) + \frac{1}{n} \sum_{i \in [n]} \delta_i \nabla_{\boldsymbol{\theta}} (f_{\boldsymbol{\theta}}(\boldsymbol{x}_i)).$$

Writing it with respect to $\boldsymbol{w}_k$, we have

$$\frac{\partial \mathcal{L}}{\partial \boldsymbol{w}_k} = -a_k \frac{\partial G(\boldsymbol{w}_k)}{\partial \boldsymbol{w}_k} + \frac{1}{n} \sum_{i \in [n]} \delta_i \frac{\partial f_{\boldsymbol{\theta}}(\boldsymbol{x}_i)}{\partial \boldsymbol{w}_k}$$

$$= -\frac{a_k}{2n} \sum_{i \in [n]} y_i \phi'(\boldsymbol{w}_k^\top \boldsymbol{x}_i) \boldsymbol{x}_i + \frac{1}{n} \sum_{i \in [n]} \delta_i a_k \phi'(\boldsymbol{w}_k^\top \boldsymbol{x}_i) \boldsymbol{x}_i$$

$$= -\frac{a_k}{2n} \sum_{i \in [n]} y_i (1 - 2y_i \delta_i) \phi'(\boldsymbol{w}_k^\top \boldsymbol{x}_i) \boldsymbol{x}_i.$$

Regarding Clarke's subdifferential at a point $\boldsymbol{\theta} \in \mathbb{R}^D$, we can take limits in a neighborhood of $\boldsymbol{\theta}$ in $\Omega_{\mathcal{S}}$ through (8), then

$$\frac{\partial^{\circ}\mathcal{L}}{\partial \boldsymbol{w}_k} \subseteq \left\{ -\frac{a_k}{2n} \sum_{i \in [n]} y_i(1 + \epsilon_i)\alpha_i \boldsymbol{x}_i : \alpha_i \in \phi^{\circ}(\boldsymbol{w}_k^{\top}\boldsymbol{x}_i), \epsilon_i \in [-2|f_{\boldsymbol{\theta}}(\boldsymbol{x}_i)|, 2|f_{\boldsymbol{\theta}}(\boldsymbol{x}_i)|], \forall i \in [n] \right\}.$$

We conclude the proof by noticing that $[-2|f_{\boldsymbol{\theta}}(\boldsymbol{x}_i)|, 2|f_{\boldsymbol{\theta}}(\boldsymbol{x}_i)|] \subseteq [-\epsilon, \epsilon]$ by Lemma B.10. $\qquad \square$

## B.6 Basic Properties of Gradient Flow

The following lemma is a simple corollary from Davis et al. (2020).

**Lemma B.15.** *For gradient flow $\boldsymbol{\theta}(t)$ on a two-layer Leaky ReLU network with logistic loss, we have*

$$\frac{\mathrm{d}\boldsymbol{\theta}(t)}{\mathrm{d}t} = -\bar{\partial}^{\circ}\mathcal{L}(\boldsymbol{\theta}(t)), \qquad \frac{\mathrm{d}\mathcal{L}(\boldsymbol{\theta}(t))}{\mathrm{d}t} = -\left\|\frac{\mathrm{d}\boldsymbol{\theta}(t)}{\mathrm{d}t}\right\|_2^2$$

*for a.e. $t \geq 0$.*

The following lemma is from Du et al. (2018). We provide a simple proof here for completeness.

**Lemma B.16.** *For gradient flow $\boldsymbol{\theta}(t) = (\boldsymbol{w}_1(t), \ldots, \boldsymbol{w}_m(t), a_1(t), \ldots, a_m(t))$ on a two-layer Leaky ReLU network with logistic loss, the following holds for all $t \geq 0$,*

$$\frac{1}{2}\frac{\mathrm{d}\|\boldsymbol{w}_k\|_2^2}{\mathrm{d}t} = \frac{1}{2}\frac{\mathrm{d}|a_k|^2}{\mathrm{d}t} = -\frac{1}{n}\sum_{i=1}^n \ell'(q_i(\boldsymbol{\theta}))y_i a_k \phi(\boldsymbol{w}_k^{\top}\boldsymbol{x}_i),$$

*where $q_i(\boldsymbol{\theta}) := y_i f_{\boldsymbol{\theta}}(\boldsymbol{x}_i)$. Therefore, $\frac{\mathrm{d}}{\mathrm{d}t}(\|\boldsymbol{w}_k\|_2^2 - |a_k|^2) = 0$ for all $t \geq 0$.*

*Proof.* By (9), we have the following for any $\boldsymbol{\theta} \in \Omega_{\mathcal{S}}$,

$$a_k \cdot \frac{\partial f_{\boldsymbol{\theta}}(\boldsymbol{x})}{\partial a_k} = a_k \phi(\boldsymbol{w}_k^{\top}\boldsymbol{x}_i), \qquad \left\langle \boldsymbol{w}_k, \frac{\partial f_{\boldsymbol{\theta}}(\boldsymbol{x})}{\partial \boldsymbol{w}_k} \right\rangle = a_k \phi'(\boldsymbol{w}_k^{\top}\boldsymbol{x}_i)\boldsymbol{w}_k^{\top}\boldsymbol{x}_i.$$

By 1-homogeneity of $\phi$ and Theorem B.3, we have $\phi'(\boldsymbol{w}_k^{\top}\boldsymbol{x}_i)\boldsymbol{w}_k^{\top}\boldsymbol{x}_i = \phi(\boldsymbol{w}_k^{\top}\boldsymbol{x}_i)$, which implies that $\left\langle \boldsymbol{w}_k, \frac{\partial f_{\boldsymbol{\theta}}(\boldsymbol{x})}{\partial \boldsymbol{w}_k} \right\rangle = a_k \phi(\boldsymbol{w}_k^{\top}\boldsymbol{x}_i)$.

For any $\boldsymbol{\theta} \in \mathbb{R}^D$, we can take limits in $\Omega_{\mathcal{S}}$ through (8) to show that the same equation holds in general.

$$a_k \cdot \frac{\partial^{\circ} f_{\boldsymbol{\theta}}(\boldsymbol{x})}{\partial a_k} = \left\langle \boldsymbol{w}_k, \frac{\partial^{\circ} f_{\boldsymbol{\theta}}(\boldsymbol{x})}{\partial \boldsymbol{w}_k} \right\rangle = \left\{ a_k \phi(\boldsymbol{w}_k^{\top}\boldsymbol{x}_i) \right\}.$$

By chain rule, for a.e. $t \geq 0$ we have

$$\frac{1}{2}\frac{\mathrm{d}|a_k|^2}{\mathrm{d}t} = \frac{\mathrm{d}a_k}{\mathrm{d}t} \cdot a_k \in -\frac{1}{n}\sum_{i=1}^n \ell'(q_i(\boldsymbol{\theta}))y_i \frac{\partial^{\circ} f_{\boldsymbol{\theta}}(\boldsymbol{x}_i)}{\partial a_k} \cdot a_k.$$

$$\frac{1}{2}\frac{\mathrm{d}\|\boldsymbol{w}_k\|_2^2}{\mathrm{d}t} = \left\langle \frac{\mathrm{d}\boldsymbol{w}_k}{\mathrm{d}t}, \boldsymbol{w}_k \right\rangle \in -\frac{1}{n}\sum_{i=1}^n \ell'(q_i(\boldsymbol{\theta}))y_i \left\langle \frac{\partial^{\circ} f_{\boldsymbol{\theta}}(\boldsymbol{x}_i)}{\partial \boldsymbol{w}_k}, \boldsymbol{w}_k \right\rangle.$$

Therefore we have

$$\frac{1}{2}\frac{\mathrm{d}|a_k|^2}{\mathrm{d}t} = \frac{1}{2}\frac{\mathrm{d}\|\boldsymbol{w}_k\|_2^2}{\mathrm{d}t} = -\frac{1}{n}\sum_{i=1}^n \ell'(q_i(\boldsymbol{\theta}))y_i a_k \phi(\boldsymbol{w}_k^{\top}\boldsymbol{x}_i),$$

for a.e. $t \geq 0$. Note that $-\frac{1}{n}\sum_{i=1}^n \ell'(q_i(\boldsymbol{\theta}))y_i a_k \phi(\boldsymbol{w}_k^{\top}\boldsymbol{x}_i)$ is continuous in $\boldsymbol{\theta}$ and thus continuous in time $t$. This means we can further deduce that this equation holds for all $t \geq 0$. This automatically proves that $\frac{\mathrm{d}}{\mathrm{d}t}(\|\boldsymbol{w}_k\|_2^2 - |a_k|^2) = 0$. $\qquad \square$

The following lemma shows that if a neuron has zero weights, then it stays with zero weights forever. Conversely, this also implies that the weights stay non-zero if they are initially non-zero.

**Lemma B.17.** *If $a_k(t_0) = 0$ and $\boldsymbol{w}_k(t_0) = \boldsymbol{0}$ at some time $t_0 \geq 0$, then $a_k(t) = 0$ and $\boldsymbol{w}_k(t) = \boldsymbol{0}$ for all $t \geq 0$.*

*Proof.* By Lemma B.16, we know that $\|\boldsymbol{w}_k\|_2 = |a_k|$ hold for all $t \geq 0$. Also, we have $\frac{1}{2}\left|\frac{\mathrm{d}\|\boldsymbol{w}_k\|_2^2}{\mathrm{d}t}\right| =$
$\frac{1}{2}\left|\frac{\mathrm{d}|a_k|^2}{\mathrm{d}t}\right| \leq C \cdot |a_k| \|\boldsymbol{w}_k\|_2 = C\|\boldsymbol{w}_k\|_2^2$, where $C > 0$ is some constant. Then

$$\|\boldsymbol{w}_k(t)\|_2^2 \leq \|\boldsymbol{w}_k(t_0)\|_2^2 + \int_{t_0}^t 2C\|\boldsymbol{w}_k(\tau)\|_2^2 \mathrm{d}\tau.$$

By Grönwall's inequality (12) this implies that $\|\boldsymbol{w}_k(t)\|_2 = 0$ for all $t \geq t_0$. Similarly,

$$\|\boldsymbol{w}_k(t)\|_2^2 \leq \|\boldsymbol{w}_k(t_0)\|_2^2 + \int_t^{t_0} 2C\|\boldsymbol{w}_k(\tau)\|_2^2 \mathrm{d}\tau.$$

By Grönwall's inequality (12) again, $\|\boldsymbol{w}_k(t)\|_2 = 0$ for all $t \leq t_0$, which completes the proof. □

A direct corollary of Lemma B.16 and Lemma B.17 is the following characterization in the case where the weights are initially balanced.

**Corollary B.18.** *If $|a_k| = \|\boldsymbol{w}_k\|_2$ initially for $t = 0$, then this equation holds for all $t \geq 0$. Moreover,*

1. *If $a_k(0) = \|\boldsymbol{w}_k(0)\|_2$, then $a_k(t) = \|\boldsymbol{w}_k(t)\|_2$ for all $t \geq 0$;*

2. *If $a_k(0) = -\|\boldsymbol{w}_k(0)\|_2$, then $a_k(t) = -\|\boldsymbol{w}_k(t)\|_2$ for all $t \geq 0$.*

### B.7 A Useful Theorem for Loss Convergence

In this section we prove a useful theorem for loss convergence, which will be used later in our analysis for both symmetric and non-symmetric datasets.

**Theorem B.19.** *Under Assumption 3.2, for any linear seprator $\boldsymbol{w}^*$ of the data with positive linear margin (e.g. $y_i \langle \boldsymbol{w}^*, x_i \rangle \geq \gamma^* > 0$ for all $i \in [n]$), if initially there exists $k \in [m]$ such that*

$$\mathrm{sgn}(a_k(0)) \langle \boldsymbol{w}_k(0), \boldsymbol{w}^* \rangle > 0, \qquad \langle \boldsymbol{w}_k(0), \boldsymbol{w}^* \rangle^2 > \|\boldsymbol{w}_k(0)\|_2^2 - |a_k(0)|^2,$$

*then $a_k(t) \neq 0$ for all $t > 0$, and $\mathcal{L}(\boldsymbol{\theta}(t)) \to 0$ and $\|\boldsymbol{\theta}(t)\|_2 \to +\infty$ as $t \to +\infty$.*

Before proving Theorem B.19, we first prove a lemma on gradient lower bounds.

**Lemma B.20.** *For a.e. $t \geq 0$,*

$$\left\langle \mathrm{sgn}(a_k) \frac{\mathrm{d}\boldsymbol{w}_k}{\mathrm{d}t}, \boldsymbol{w}^* \right\rangle \geq |a_k| \alpha_{\mathrm{leaky}} \gamma^* \cdot \frac{1 - \exp(-n\mathcal{L})}{n}.$$

*Proof.* By (10), there exist $h_1^{(k)}(t), \ldots, h_n^{(k)}(t) \in [\alpha_{\mathrm{leaky}}, 1]$ such that

$$\frac{\mathrm{d}\boldsymbol{w}_k}{\mathrm{d}t} = \frac{a_k}{n} \sum_{i \in [n]} g_i(\boldsymbol{\theta}(t)) h_i^{(k)}(t) y_i \boldsymbol{x}_i$$

where $g_i(\boldsymbol{\theta}(t)) = -\ell'(y_i f_{\boldsymbol{\theta}(t)}(\boldsymbol{x}_i)) > 0$. Then we have

$$\left\langle \mathrm{sgn}(a_k) \frac{\mathrm{d}\boldsymbol{w}_k}{\mathrm{d}t}, \boldsymbol{w}^* \right\rangle \geq \frac{|a_k|}{n} \sum_{i \in [n]} g_i(\boldsymbol{\theta}(t)) \alpha_{\mathrm{leaky}} \gamma^*.$$

Note that $-\ell'(q) = \frac{1}{1+e^q} = 1 - \frac{1}{1+e^{-q}} = 1 - \exp(-\ell(q))$ for all $q$. So we have the following lower bound for $\sum_{i \in [n]} g_i(\boldsymbol{\theta}(t))$:

$$\sum_{i \in [n]} g_i(\boldsymbol{\theta}(t)) = \sum_{i \in [n]} -\ell'(y_i f_{\boldsymbol{\theta}}(\boldsymbol{x}_i)) = \sum_{i \in [n]} (1 - \exp(-\ell(y_i f_{\boldsymbol{\theta}}(\boldsymbol{x}_i))))$$

$$\geq \max_{i \in [n]} (1 - \exp(-\ell(y_i f_{\boldsymbol{\theta}}(\boldsymbol{x}_i))))$$

$$\geq 1 - \exp\left(-\max_{i \in [n]} \ell(y_i f_{\boldsymbol{\theta}}(\boldsymbol{x}_i))\right)$$

$$\geq 1 - \exp(-n\mathcal{L}).$$

Therefore,

$$\left\langle \mathrm{sgn}(a_k)\frac{\mathrm{d}\boldsymbol{w}_k}{\mathrm{d}t}, \boldsymbol{w}^* \right\rangle \geq |a_k|\alpha_{\mathrm{leaky}}\gamma^* \cdot \frac{1 - \exp(-n\mathcal{L})}{n},$$

which completes the proof. $\qquad\square$

*Proof for Theorem B.19.* We only need to show that there exists $t_0$ such that $\mathcal{L}(\boldsymbol{\theta}(t_0)) < \frac{\ln 2}{n}$, then we can apply Theorem 3.1 to show that $\mathcal{L}(\boldsymbol{\theta}(t)) \to 0$. Assume to the contrary that $\mathcal{L}(\boldsymbol{\theta}(t)) \geq \frac{\ln 2}{n}$ for all $t \geq 0$. By Lemma B.20,

$$\mathrm{sgn}(a_k)\left\langle \frac{\mathrm{d}\boldsymbol{w}_k}{\mathrm{d}t}, \boldsymbol{w}^* \right\rangle \geq |a_k| \cdot \frac{\alpha_{\mathrm{leaky}}\gamma^*}{2n}.$$

Let $c := \langle \boldsymbol{w}_k(0), \boldsymbol{w}^* \rangle^2 - \|\boldsymbol{w}_k(0)\|_2^2 + |a_k(0)|^2 > 0$. First we show that $\mathrm{sgn}(a_k(t)) = \mathrm{sgn}(a_k(0))$ for all $t > 0$. Otherwise let $t_{\mathrm{s}} := \inf\{t : \mathrm{sgn}(a_k(t)) \neq \mathrm{sgn}(a_k(0))\}$, and since $a_k(t)$ is continuous, $a_k(t_{\mathrm{s}}) = 0$. We know for $t \in [0, t_{\mathrm{s}}]$, $\mathrm{sgn}(a_k(0))\frac{\mathrm{d}}{\mathrm{d}t}\langle \boldsymbol{w}_k(t), \boldsymbol{w}^* \rangle > 0$, and

$$|a_k(t_{\mathrm{s}})|^2 = |a_k(0)|^2 - \|\boldsymbol{w}_k(0)\|_2^2 + \|\boldsymbol{w}_k(t_{\mathrm{s}})\|_2^2 \geq |a_k(0)|^2 - \|\boldsymbol{w}_k(0)\|_2^2 + \langle \boldsymbol{w}_k(t_{\mathrm{s}}), \boldsymbol{w}^* \rangle^2$$
$$> |a_k(0)|^2 - \|\boldsymbol{w}_k(0)\|_2^2 + \langle \boldsymbol{w}_k(0), \boldsymbol{w}^* \rangle^2 = c > 0.$$

This contradicts to the fact that $a_k(t_{\mathrm{s}}) = 0$, and thus $\mathrm{sgn}(a_k(t))$ does not change during all time. Therefore for any $t > 0$, $a_k(t) \neq 0$. Then for all $t > 0$,

$$|a_k(t)|^2 = |a_k(0)|^2 - \|\boldsymbol{w}_k(0)\|_2^2 + \|\boldsymbol{w}_k(t)\|_2^2 \geq |a_k(0)|^2 - \|\boldsymbol{w}_k(0)\|_2^2 + \langle \boldsymbol{w}_k(t), \boldsymbol{w}^* \rangle^2$$
$$> |a_k(0)|^2 - \|\boldsymbol{w}_k(0)\|_2^2 + \langle \boldsymbol{w}_k(0), \boldsymbol{w}^* \rangle^2 = c.$$

Lemma B.15 ensures that $-\frac{\mathrm{d}\mathcal{L}}{\mathrm{d}t} = \left\|\frac{\mathrm{d}\boldsymbol{\theta}}{\mathrm{d}t}\right\|_2^2$ for a.e. $t \geq 0$. Then we have

$$-\frac{\mathrm{d}\mathcal{L}}{\mathrm{d}t} \geq \left\|\frac{\mathrm{d}\boldsymbol{w}_k}{\mathrm{d}t}\right\|_2^2 \geq \left\langle \frac{\mathrm{d}\boldsymbol{w}_k}{\mathrm{d}t}, \boldsymbol{w}^* \right\rangle^2 \geq |a_k|^2 \left(\frac{\alpha_{\mathrm{leaky}}\gamma^*}{2n}\right)^2 \geq c^2 \cdot \left(\frac{\alpha_{\mathrm{leaky}}\gamma^*}{2n}\right)^2.$$

Then we can conclude that

$$\mathcal{L}(\boldsymbol{\theta}(0)) - \mathcal{L}(\boldsymbol{\theta}(t)) \geq c^2 \left(\frac{\alpha_{\mathrm{leaky}}\gamma^*}{2n}\right)^2 t.$$

Integrating on $t$ from 0 to $+\infty$, we can see that the LHS is upper bounded by $\mathcal{L}(\boldsymbol{\theta}(0)) - \frac{\ln 2}{n}$ while the RHS is unbounded, which leads to a contradiction. Therefore, there exist time $t_0$ such that $\mathcal{L}(\boldsymbol{\theta}(t_0)) < \frac{\ln 2}{n}$, and thus $\mathcal{L}(\boldsymbol{\theta}(t)) \to 0$ as $t \to +\infty$. $\qquad\square$

## C   Proofs for Linear Maximality for the Symmetric Case

For linearly separable and symmetric data, we show that all global-max-margin directions represent linear functions in Theorem 4.2. We give a proof here.

*Proof for Theorem 4.2.* Let $\boldsymbol{\theta}^* = (\boldsymbol{w}_1, \ldots, \boldsymbol{w}_m, a_1, \ldots, a_m) \in \mathbb{S}^{D-1}$ be any global-max-margin direction with output margin $q_{\min}(\boldsymbol{\theta}^*) = \gamma(\boldsymbol{\theta}^*)$. As the dataset is symmetric,

$$\gamma(\boldsymbol{\theta}^*) = \min_{i \in [n]}\{y_i f_{\boldsymbol{\theta}^*}(\boldsymbol{x}_i), -y_i f_{\boldsymbol{\theta}^*}(-\boldsymbol{x}_i)\}.$$

Now we define $A := \sqrt{\sum_{k \in [m]} a_k^2}$ and let $\boldsymbol{\theta}' = (\boldsymbol{w}_1', \ldots, \boldsymbol{w}_m', a_1', \ldots, a_m')$ where

$$\boldsymbol{w}_1' = \frac{1}{\sqrt{2}A}\sum_{k \in [m]} a_k\boldsymbol{w}_k, \qquad \boldsymbol{w}_2' = -\boldsymbol{w}_1', \qquad a_1' = \frac{A}{\sqrt{2}}, \qquad a_2' = -a_1',$$

and $a'_k = 0, \boldsymbol{w}'_k = \boldsymbol{0}$ for $k > 2$. We claim that $\gamma(\boldsymbol{\theta}') \geq \gamma(\boldsymbol{\theta}^*)$. First we prove that $q_i(\boldsymbol{\theta}') \geq \gamma(\boldsymbol{\theta}^*)$ by repeatedly applying $\phi(z) - \phi(-z) = (1 + \alpha_{\text{leaky}})z$.

$$
\begin{aligned}
q_i(\boldsymbol{\theta}') = y_i f_{\boldsymbol{\theta}'}(\boldsymbol{x}_i) &= y_i \left( a'_1 \phi(\langle \boldsymbol{w}'_1, \boldsymbol{x}_i \rangle) + a'_2 \phi(\langle \boldsymbol{w}'_2, \boldsymbol{x}_i \rangle) \right) \\
&= y_i(1 + \alpha_{\text{leaky}})a'_1 \langle \boldsymbol{w}'_1, \boldsymbol{x}_i \rangle \\
&= \frac{y_i}{2} \sum_{k \in [m]} \langle (1 + \alpha_{\text{leaky}})a_k \boldsymbol{w}_k, \boldsymbol{x}_i \rangle \\
&= \frac{y_i}{2} \sum_{k \in [m]} \left( a_k \phi(\boldsymbol{w}_k^\top \boldsymbol{x}_i) - a_k \phi(-\boldsymbol{w}_k^\top \boldsymbol{x}_i) \right) \\
&= \frac{1}{2}(y_i f_{\boldsymbol{\theta}^*}(\boldsymbol{x}_i) - y_i f_{\boldsymbol{\theta}^*}(-\boldsymbol{x}_i)) \geq \gamma(\boldsymbol{\theta}^*).
\end{aligned}
$$

Meanwhile, by the Cauchy-Schwarz inequality,

$$
\|\boldsymbol{\theta}'\|_2^2 = A^2 + \left\| \sum_{k \in [m]} \frac{a_k}{A} \boldsymbol{w}_k \right\|_2^2 \leq A^2 + \sum_{k \in [m]} \left( \frac{a_k}{A} \right)^2 \cdot \sum_{k \in [m]} \|\boldsymbol{w}_k\|_2^2 = A^2 + \sum_{k \in [m]} \|\boldsymbol{w}_k\|_2^2 = \|\boldsymbol{\theta}^*\|_2^2.
$$

Thus $\gamma(\boldsymbol{\theta}') = \frac{q_{\min}(\boldsymbol{\theta}')}{\|\boldsymbol{\theta}'\|_2^2} \geq \gamma(\boldsymbol{\theta}^*)$. As $\boldsymbol{\theta}^*$ is already a global-max-margin direction, equalities should hold in all the inequalities above, so

$$
\min_{i \in [n]} \{y_i f_{\boldsymbol{\theta}^*}(\boldsymbol{x}_i) - y_i f_{\boldsymbol{\theta}^*}(-\boldsymbol{x}_i)\} = 2\gamma(\boldsymbol{\theta}^*), \qquad \left\| \sum_{k \in [m]} \frac{a_k}{A} \boldsymbol{w}_k \right\|_2^2 = \sum_{k \in [m]} \left( \frac{a_k}{A} \right)^2 \cdot \sum_{k \in [m]} \|\boldsymbol{w}_k\|_2^2.
$$

Then we know the following:

- There is $\boldsymbol{c} \in \mathbb{R}^d$ that $\boldsymbol{w}_k = a_k \boldsymbol{c}$ for all $k$;
- There is $j \in [n]$ that $y_j f_{\boldsymbol{\theta}^*}(\boldsymbol{x}_j) = -y_j f_{\boldsymbol{\theta}^*}(-\boldsymbol{x}_j) = \gamma(\boldsymbol{\theta}^*)$.

Note that $\phi(z) + \phi(-z) = (1 - \alpha_{\text{leaky}})|z|$. Then we have

$$
0 = f_{\boldsymbol{\theta}^*}(\boldsymbol{x}_j) + f_{\boldsymbol{\theta}^*}(-\boldsymbol{x}_j) = \sum_{k \in [m]} a_k \left( \phi(a_k \boldsymbol{c}^\top \boldsymbol{x}_j) + \phi(-a_k \boldsymbol{c}^\top \boldsymbol{x}_j) \right) = \sum_{k=1}^m (1 - \alpha_{\text{leaky}}) a_k |a_k \boldsymbol{c}^\top \boldsymbol{x}_j|.
$$

Certainly $\boldsymbol{c}^\top \boldsymbol{x}_j \neq 0$ as otherwise the margin would be zero. Then $\sum_{k \in [m]} a_k |a_k| = 0$, which means $\sum_{k: a_k \geq 0} a_k^2 = \sum_{k: a_k < 0} a_k^2 = \frac{1}{2}A^2$, and therefore

$$
\begin{aligned}
f_{\boldsymbol{\theta}^*}(\boldsymbol{x}) = \sum_{k=1}^m a_k \phi(a_k \boldsymbol{c}^\top \boldsymbol{x}) &= \sum_{k=1}^m a_k |a_k| \phi(\text{sgn}(a_k) \boldsymbol{c}^\top \boldsymbol{x}) \\
&= \frac{1}{2}A^2(\phi(\boldsymbol{c}^\top \boldsymbol{x}) - \phi(-\boldsymbol{c}^\top \boldsymbol{x})) = \frac{1}{2}A^2(1 + \alpha_{\text{leaky}})\boldsymbol{c}^\top \boldsymbol{x}
\end{aligned}
$$

is a linear function in $\boldsymbol{x}$.

Finally, let $\gamma_{\boldsymbol{w}^*} = \min_{i \in [n]} y_i \langle \boldsymbol{w}^*, \boldsymbol{x}_i \rangle$ be the maximum linear margin, where $\boldsymbol{w}^* \in \mathbb{S}^{d-1}$ is the max-margin linear separator. As $\|\boldsymbol{\theta}^*\|_2^2 = 1 = (1 + \|\boldsymbol{c}\|_2^2)A^2$,

$$
\begin{aligned}
\gamma(\boldsymbol{\theta}^*) = \frac{1}{2}A^2(1 + \alpha_{\text{leaky}}) \min_{i \in [n]} y_i \boldsymbol{c}^\top \boldsymbol{x}_i &\leq \frac{1}{2}A^2(1 + \alpha_{\text{leaky}}) \|\boldsymbol{c}\|_2 \gamma_{\boldsymbol{w}^*} \\
&= \frac{\|\boldsymbol{c}\|_2}{2(1 + \|\boldsymbol{c}\|_2^2)}(1 + \alpha_{\text{leaky}})\gamma_{\boldsymbol{w}^*} \leq \frac{1}{4}(1 + \alpha_{\text{leaky}})\gamma_{\boldsymbol{w}^*}.
\end{aligned}
$$

By choosing $\boldsymbol{c} = \boldsymbol{w}^*$ with $A = \frac{1}{\sqrt{2}}$, the network is able to attain the margin $\frac{1}{4}(1 + \alpha_{\text{leaky}})\gamma_{\boldsymbol{w}^*}$. As $\boldsymbol{\theta}^*$ is already a global-max-margin direction, we know again that the equalities must hold. Therefore we know

- $\min_{i \in [n]} y_i \boldsymbol{c}^\top \boldsymbol{x}_i = \|\boldsymbol{c}\|_2 \, \gamma_{\boldsymbol{w}^*}$;

- $\frac{\|\boldsymbol{c}\|_2}{1 + \|\boldsymbol{c}\|_2^2} = \frac{1}{2}$.

Then we know $\boldsymbol{c} = \boldsymbol{w}^*$ due to the uniqueness of the max-margin linear separator, and thus $A = \frac{1}{\sqrt{2}}$. Therefore the function is $f_{\boldsymbol{\theta}^*}(\boldsymbol{x}) = \frac{1 + \alpha_{\text{leaky}}}{4} \langle \boldsymbol{w}^*, \boldsymbol{x} \rangle$. $\hfill\square$

# D  Proofs for Phase I

In the subsequent sections we first show the proofs for the symmetric datasets under Assumption 4.1. Additional proofs for the non-symmetric counterparts are provided in Appendix G.

As we have illustrated in Section 5.1, we have $G(\boldsymbol{w}) = \langle \boldsymbol{w}, \tilde{\boldsymbol{\mu}} \rangle$ under Assumption 4.1. Then we have

$$\tilde{\mathcal{L}}(\boldsymbol{\theta}) := \ell(0) - \sum_{k \in [m]} a_k G(\boldsymbol{w}_k) = \ell(0) - \sum_{k \in [m]} a_k \langle \boldsymbol{w}_k, \tilde{\boldsymbol{\mu}} \rangle .$$

This means the dynamics of $\tilde{\boldsymbol{\theta}}(t) = (\tilde{\boldsymbol{w}}_1(t), \dots, \tilde{\boldsymbol{w}}_m(t), \tilde{a}_1(t), \dots, \tilde{a}_m(t)) = \tilde{\varphi}(\tilde{\boldsymbol{\theta}}_0, t)$ can be described by linear ODE:

$$\frac{\mathrm{d}\tilde{\boldsymbol{w}}_k}{\mathrm{d}t} = a_k \tilde{\boldsymbol{\mu}}, \qquad \frac{\mathrm{d}\tilde{a}_k}{\mathrm{d}t} = \langle \tilde{\boldsymbol{w}}_k, \tilde{\boldsymbol{\mu}} \rangle .$$

**Lemma D.1.** *Let $\tilde{\boldsymbol{\theta}}(t) = \tilde{\varphi}(\tilde{\boldsymbol{\theta}}_0, t)$. Then*

$$\|\tilde{\boldsymbol{\theta}}(t)\|_{\mathrm{M}} \leq \exp(t\lambda_0)\|\tilde{\boldsymbol{\theta}}_0\|_{\mathrm{M}}.$$

*Proof.* By definition and Cauchy-Schwartz inequality,

$$\left\| \frac{\mathrm{d}\tilde{\boldsymbol{w}}_k}{\mathrm{d}t} \right\|_2 \leq \|\tilde{a}_k \tilde{\boldsymbol{\mu}}\|_2 \leq \lambda_0 |\tilde{a}_k|, \qquad \left| \frac{\mathrm{d}\tilde{a}_k}{\mathrm{d}t} \right| \leq |\tilde{\boldsymbol{w}}_k^\top \tilde{\boldsymbol{\mu}}| \leq \lambda_0 \|\tilde{\boldsymbol{w}}_k\|_2.$$

So we have $\|\tilde{\boldsymbol{\theta}}(t)\|_{\mathrm{M}} \leq \|\tilde{\boldsymbol{\theta}}_0\|_{\mathrm{M}} + \int_0^t \lambda_0 \|\tilde{\boldsymbol{\theta}}(\tau)\|_{\mathrm{M}} \mathrm{d}\tau$. Then we can finish the proof by Grönwall's inequality (11). $\hfill\square$

**Lemma D.2.** *For initial point $\boldsymbol{\theta}_0 \neq \mathbf{0}$, we have*

$$\|\boldsymbol{\theta}(t) - \tilde{\varphi}(\boldsymbol{\theta}_0, t)\|_{\mathrm{M}} \leq \frac{4m\|\boldsymbol{\theta}_0\|_{\mathrm{M}}^3}{\lambda_0} \exp(3\lambda_0 t),$$

*for all $t \leq \frac{1}{\lambda_0} \ln \frac{\sqrt{\lambda_0/4}}{\sqrt{m}\|\boldsymbol{\theta}_0\|_{\mathrm{M}}}$.*

*Proof.* Let $\tilde{\boldsymbol{\theta}}(t) = \tilde{\varphi}(\boldsymbol{\theta}_0, t)$. By Corollary B.13, the following holds for a.e. $t \geq 0$,

$$\left\| \frac{\mathrm{d}\boldsymbol{\theta}}{\mathrm{d}t} - \frac{\mathrm{d}\tilde{\boldsymbol{\theta}}}{\mathrm{d}t} \right\|_{\mathrm{M}} \leq \sup \left\{ \|\boldsymbol{\delta} - \nabla\tilde{\mathcal{L}}(\boldsymbol{\theta})\|_{\mathrm{M}} : \boldsymbol{\delta} \in \partial^\circ \mathcal{L}(\boldsymbol{\theta}) \right\} + \|\nabla\tilde{\mathcal{L}}(\boldsymbol{\theta}) - \nabla\tilde{\mathcal{L}}(\tilde{\boldsymbol{\theta}})\|_{\mathrm{M}}$$

$$\leq m\|\boldsymbol{\theta}(t)\|_{\mathrm{M}}^3 + \lambda_0 \|\boldsymbol{\theta} - \tilde{\boldsymbol{\theta}}\|_{\mathrm{M}}.$$

Taking integral, we have

$$\|\boldsymbol{\theta}(t) - \tilde{\boldsymbol{\theta}}(t)\|_{\mathrm{M}} \leq \int_0^t \left( m\|\boldsymbol{\theta}(\tau)\|_{\mathrm{M}}^3 + \lambda_0 \|\boldsymbol{\theta}(\tau) - \tilde{\boldsymbol{\theta}}(\tau)\|_{\mathrm{M}} \right) \mathrm{d}\tau.$$

Let $t_0 := \inf\{t \geq 0 : \|\boldsymbol{\theta}(t)\|_{\mathrm{M}} \geq 2\|\boldsymbol{\theta}_0\|_{\mathrm{M}} \exp(\lambda_0 t)\}$. Then for all $0 \leq t \leq t_0$ (or for all $t \geq 0$ if $t_0 = +\infty$),

$$\|\boldsymbol{\theta}(t) - \tilde{\boldsymbol{\theta}}(t)\|_{\mathrm{M}} \leq \int_0^t \left( 8m\|\boldsymbol{\theta}_0\|_{\mathrm{M}}^3 \exp(3\lambda_0 \tau) + \lambda_0 \|\boldsymbol{\theta}(\tau) - \tilde{\boldsymbol{\theta}}(\tau)\|_{\mathrm{M}} \right) \mathrm{d}\tau$$

$$\leq \frac{8m\|\boldsymbol{\theta}_0\|_{\mathrm{M}}^3}{3\lambda_0} \exp(3\lambda_0 t) + \lambda_0 \int_0^t \|\boldsymbol{\theta}(\tau) - \tilde{\boldsymbol{\theta}}(\tau)\|_{\mathrm{M}} \mathrm{d}\tau.$$

By Grönwall's inequality (12),

$$\|\boldsymbol{\theta}(t) - \tilde{\boldsymbol{\theta}}(t)\|_{\mathrm{M}} \le \frac{8m\|\boldsymbol{\theta}_0\|_{\mathrm{M}}^3}{3\lambda_0}\left(\exp(3\lambda_0 t) + \lambda_0\int_0^t \exp(3\lambda_0\tau)\exp(\lambda_0(t-\tau))\mathrm{d}\tau\right)$$

$$= \frac{8m\|\boldsymbol{\theta}_0\|_{\mathrm{M}}^3}{3\lambda_0}\left(\exp(3\lambda_0 t) + \frac{1}{2}\exp(3\lambda_0 t)\right) = \frac{4m\|\boldsymbol{\theta}_0\|_{\mathrm{M}}^3}{\lambda_0}\exp(3\lambda_0 t),$$

If $t_0 < \frac{1}{2\lambda_0}\ln\frac{\lambda_0}{4m\|\boldsymbol{\theta}_0\|_{\mathrm{M}}^2}$, then

$$\|\boldsymbol{\theta}(t)\|_{\mathrm{M}} \le \|\tilde{\boldsymbol{\theta}}(t)\|_{\mathrm{M}} + \frac{4m\|\boldsymbol{\theta}_0\|_{\mathrm{M}}^3}{\lambda_0}\exp(3\lambda_0 t)$$

$$\le \|\tilde{\boldsymbol{\theta}}(t)\|_{\mathrm{M}} + \frac{4m\|\boldsymbol{\theta}_0\|_{\mathrm{M}}^2}{\lambda_0}\exp(2\lambda_0 t_0)\cdot\|\boldsymbol{\theta}_0\|_{\mathrm{M}}\exp(\lambda_0 t)$$

$$< \|\tilde{\boldsymbol{\theta}}(t)\|_{\mathrm{M}} + \|\boldsymbol{\theta}_0\|_{\mathrm{M}}\exp(\lambda_0 t).$$

By Lemma D.1, $\|\tilde{\boldsymbol{\theta}}(t)\|_{\mathrm{M}} \le \|\boldsymbol{\theta}_0\|_{\mathrm{M}}\exp(\lambda_0 t)$. So $\|\boldsymbol{\theta}(t)\|_{\mathrm{M}} < 2\|\boldsymbol{\theta}_0\|_{\mathrm{M}}\exp(\lambda_0 t)$ for all $0 \le t \le t_0$, which contradicts to the definition of $t_0$. Therefore, $t_0 \ge \frac{1}{2\lambda_0}\ln\frac{\lambda_0}{4m\|\boldsymbol{\theta}_0\|_{\mathrm{M}}^2} = \frac{1}{\lambda_0}\ln\frac{\sqrt{\lambda_0/4}}{\sqrt{m}\|\boldsymbol{\theta}_0\|_{\mathrm{M}}}$. $\qquad\square$

*Proof for Lemma 5.2.* Let $\tilde{\boldsymbol{\theta}}(t) = (\tilde{\boldsymbol{w}}_1(t), \ldots, \tilde{\boldsymbol{w}}_m(t), \tilde{a}_1(t), \ldots, \tilde{a}_m(t)) = \tilde{\varphi}(\boldsymbol{\theta}_0, t)$. Then

$$[\tilde{\boldsymbol{w}}_k(t), \tilde{a}_k(t)]^\top = \exp(T_1(r)\boldsymbol{M}_{\tilde{\boldsymbol{\mu}}})[\tilde{\boldsymbol{w}}_k(0), \tilde{a}_k(0)]^\top = \exp(T_1(r)\boldsymbol{M}_{\tilde{\boldsymbol{\mu}}})[\sigma_{\mathrm{init}}\bar{\boldsymbol{w}}_k, \sigma_{\mathrm{init}}\bar{a}_k]^\top,$$

where $\boldsymbol{M}_{\tilde{\boldsymbol{\mu}}}$ is defined in Section 5.1,

$$\boldsymbol{M}_{\tilde{\boldsymbol{\mu}}} := \begin{bmatrix} \boldsymbol{0} & \tilde{\boldsymbol{\mu}} \\ \tilde{\boldsymbol{\mu}}^\top & 0 \end{bmatrix}.$$

Let $\bar{\boldsymbol{\mu}}_2 := \frac{1}{\sqrt{2}}[\bar{\boldsymbol{\mu}}, 1]^\top$ be the top eigenvector of $\boldsymbol{M}_{\tilde{\boldsymbol{\mu}}}$, which is associated with eigenvalue $\lambda_0$. All the other eigenvalues of $\boldsymbol{M}_{\tilde{\boldsymbol{\mu}}}$ are no greater than 0. Note that

$$\exp(T_1(r)\lambda_0)\bar{\boldsymbol{\mu}}_2\bar{\boldsymbol{\mu}}_2^\top\begin{bmatrix}\sigma_{\mathrm{init}}\bar{\boldsymbol{w}}_k \\ \sigma_{\mathrm{init}}\bar{a}_k\end{bmatrix} = \frac{r}{\sqrt{m}\|\boldsymbol{\theta}_0\|_{\mathrm{M}}}\left(\frac{\sigma_{\mathrm{init}}}{\sqrt{2}}(\bar{\boldsymbol{\mu}}^\top\bar{\boldsymbol{w}}_k + \bar{a}_k)\right)\bar{\boldsymbol{\mu}}_2 = \sqrt{2}r\bar{b}_k\bar{\boldsymbol{\mu}}_2 = r\bar{b}_k\begin{bmatrix}\bar{\boldsymbol{\mu}} \\ 1\end{bmatrix}.$$

So we have

$$\left\|\begin{bmatrix}\tilde{\boldsymbol{w}}_k(T_1(r)) \\ \tilde{a}_k(T_1(r))\end{bmatrix} - r\bar{b}_k\begin{bmatrix}\bar{\boldsymbol{\mu}} \\ 1\end{bmatrix}\right\|_2 = \left\|(\exp(T_1(r)\boldsymbol{M}_{\tilde{\boldsymbol{\mu}}}) - \exp(T_1(r)\lambda_0)\bar{\boldsymbol{\mu}}_2\bar{\boldsymbol{\mu}}_2^\top)\begin{bmatrix}\sigma_{\mathrm{init}}\bar{\boldsymbol{w}}_k \\ \sigma_{\mathrm{init}}\bar{a}_k\end{bmatrix}\right\|_2$$

$$\le \sigma_{\mathrm{init}}\left\|\begin{bmatrix}\bar{\boldsymbol{w}}_k \\ \bar{a}_k\end{bmatrix}\right\|_2 \le \sqrt{2}\sigma_{\mathrm{init}}\|\bar{\boldsymbol{\theta}}_0\|_{\mathrm{M}}.$$

With probability 1, $\bar{\boldsymbol{\theta}}_0 \ne \boldsymbol{0}$. For $r \le \sqrt{\lambda_0/4}$, we have $T_1(r) = \frac{1}{\lambda_0}\ln\frac{r}{\sqrt{m}\|\boldsymbol{\theta}_0\|_{\mathrm{M}}} \le \frac{1}{\lambda_0}\ln\frac{\sqrt{\lambda_0/4}}{\sqrt{m}\|\boldsymbol{\theta}_0\|_{\mathrm{M}}}$. Then by Lemma D.2,

$$\|\boldsymbol{\theta}(T_1(r)) - \tilde{\boldsymbol{\theta}}(T_1(r))\|_{\mathrm{M}} \le \frac{4m\|\boldsymbol{\theta}_0\|_{\mathrm{M}}^3}{\lambda_0}\exp(3\lambda_0 T_1(r)) = \frac{4r^3}{\lambda_0\sqrt{m}}.$$

By triangle inequality, we have

$$\left\|\begin{bmatrix}\boldsymbol{w}_k(T_1(r)) \\ a_k(T_1(r))\end{bmatrix} - r\bar{b}_k\begin{bmatrix}\bar{\boldsymbol{\mu}} \\ 1\end{bmatrix}\right\|_{\mathrm{M}} \le \left\|\begin{bmatrix}\boldsymbol{w}_k(T_1(r)) \\ a_k(T_1(r))\end{bmatrix} - \begin{bmatrix}\tilde{\boldsymbol{w}}_k(T_1(r)) \\ \tilde{a}_k(T_1(r))\end{bmatrix}\right\|_{\mathrm{M}} + \left\|\begin{bmatrix}\tilde{\boldsymbol{w}}_k(T_1(r)) \\ \tilde{a}_k(T_1(r))\end{bmatrix} - r\bar{b}_k\begin{bmatrix}\bar{\boldsymbol{\mu}} \\ 1\end{bmatrix}\right\|_{\mathrm{M}}$$

$$\le \frac{4r^3}{\lambda_0\sqrt{m}} + \sqrt{2}\sigma_{\mathrm{init}}\|\bar{\boldsymbol{\theta}}_0\|_{\mathrm{M}} \le \frac{Cr^3}{\sqrt{m}},$$

for some universal constant $C$, where the last step is due to our choice of $\sigma_{\mathrm{init}} \le \frac{r^3}{\sqrt{m}\|\bar{\boldsymbol{\theta}}_0\|_{\mathrm{M}}}$. $\qquad\square$

# E  Proofs for Phase II

## E.1  Proof for Exact Embedding

To prove Lemma 5.3, we start from the following lemma.

**Lemma E.1.** *Given $\hat{\boldsymbol{\theta}}_0 := (\hat{\boldsymbol{w}}_1, \hat{\boldsymbol{w}}_2, \hat{a}_1, \hat{a}_2)$ with $\hat{a}_1 > 0$ and $\hat{a}_2 < 0$, then $\boldsymbol{\theta}(t) = \pi_{\boldsymbol{b}}(\varphi(\hat{\boldsymbol{\theta}}_0, t))$ is a gradient flow trajectory on $\mathcal{L}(\boldsymbol{\theta})$ starting from $\boldsymbol{\theta}(0) = \pi_{\boldsymbol{b}}(\hat{\boldsymbol{\theta}}_0)$.*

First we notice the following fact.

**Lemma E.2.** *For any $\hat{\boldsymbol{\theta}}$ and $\boldsymbol{g} \in \partial^\circ \mathcal{L}(\hat{\boldsymbol{\theta}})$, $\pi_{\boldsymbol{b}}(\boldsymbol{g}) \in \partial^\circ \mathcal{L}(\pi_{\boldsymbol{b}}(\hat{\boldsymbol{\theta}}))$.*

Below we use $\pi_{\boldsymbol{b}}(S) = \{\pi_{\boldsymbol{b}}(\boldsymbol{s}) : \boldsymbol{s} \in S\}$ to denote the embedding of a parameter set.

*Proof.* For every $\hat{\boldsymbol{\theta}} = (\hat{\boldsymbol{w}}_1, \hat{\boldsymbol{w}}_2, \hat{a}_1, \hat{a}_2) \in \Omega_{\mathcal{S}}$ (i.e., no activation function has zero input), let $\boldsymbol{\theta} = \pi_{\boldsymbol{b}}(\hat{\boldsymbol{\theta}}) = (\boldsymbol{w}_1, \ldots, \boldsymbol{w}_m, a_1, \ldots, a_m)$, and clearly $\boldsymbol{\theta} \in \Omega_{\mathcal{S}}$. Then $\partial^\circ \mathcal{L}(\hat{\boldsymbol{\theta}}) = \{\nabla \mathcal{L}(\hat{\boldsymbol{\theta}})\}$ and $\partial^\circ \mathcal{L}(\boldsymbol{\theta}) = \{\nabla \mathcal{L}(\boldsymbol{\theta})\}$ are the usual differentials. In this case, we can apply the chain rule as

$$\nabla \mathcal{L}(\hat{\boldsymbol{\theta}}) = \frac{1}{n} \sum_{i \in [n]} y_i \ell'(y_i f_{\hat{\boldsymbol{\theta}}}(\boldsymbol{x}_i)) \frac{\partial f_{\hat{\boldsymbol{\theta}}}(\boldsymbol{x}_i)}{\partial \hat{\boldsymbol{\theta}}},$$

$$\nabla \mathcal{L}(\boldsymbol{\theta}) = \frac{1}{n} \sum_{i \in [n]} y_i \ell'(y_i f_{\boldsymbol{\theta}}(\boldsymbol{x}_i)) \frac{\partial f_{\boldsymbol{\theta}}(\boldsymbol{x}_i)}{\partial \boldsymbol{\theta}}.$$

Notice that the embedding preserves the function value,

$$f_{\boldsymbol{\theta}}(\boldsymbol{x}_i) = \sum_{j=1}^{m} a_j \phi(\boldsymbol{w}_j^\top \boldsymbol{x}_i) = \sum_{j : b_j > 0} \frac{b_j^2}{b_+^2} \hat{a}_1 \phi(\hat{\boldsymbol{w}}_1^\top \boldsymbol{x}_i) + \sum_{j : b_j < 0} \frac{b_j^2}{b_-^2} \hat{a}_2 \phi(\hat{\boldsymbol{w}}_2^\top \boldsymbol{x}_i)$$

$$= \hat{a}_1 \phi(\hat{\boldsymbol{w}}_1^\top \boldsymbol{x}_i) + \hat{a}_2 \phi(\hat{\boldsymbol{w}}_2^\top \boldsymbol{x}_i) = f_{\hat{\boldsymbol{\theta}}}(\boldsymbol{x}_i);$$

and the also preserves the gradient

$$\frac{\partial f_{\boldsymbol{\theta}}(\boldsymbol{x}_i)}{\partial \boldsymbol{w}_k} = a_k \phi'(\boldsymbol{w}_k^\top \boldsymbol{x}_i) \boldsymbol{x}_i = \begin{cases} \frac{b_k}{b_+} \hat{a}_1 \phi'(\hat{\boldsymbol{w}}_1^\top \boldsymbol{x}_i) \boldsymbol{x}_i & \text{if } b_k > 0 \\ \frac{b_k}{b_-} \hat{a}_2 \phi'(\hat{\boldsymbol{w}}_2^\top \boldsymbol{x}_i) \boldsymbol{x}_i & \text{if } b_k < 0 \end{cases},$$

$$\frac{\partial f_{\boldsymbol{\theta}}(\boldsymbol{x}_i)}{\partial a_k} = \phi(\boldsymbol{w}_k^\top \boldsymbol{x}_i) = \begin{cases} \frac{b_k}{b_+} \phi(\hat{\boldsymbol{w}}_1^\top \boldsymbol{x}_i) & \text{if } b_k > 0 \\ \frac{b_k}{b_-} \phi(\hat{\boldsymbol{w}}_2^\top \boldsymbol{x}_i) & \text{if } b_k < 0 \end{cases},$$

so $\frac{\partial f_{\boldsymbol{\theta}}(\boldsymbol{x}_i)}{\partial \boldsymbol{\theta}} = \pi_{\boldsymbol{b}} \left( \frac{\partial f_{\hat{\boldsymbol{\theta}}}(\boldsymbol{x}_i)}{\partial \hat{\boldsymbol{\theta}}} \right)$. Then from the chain rule above we can see $\nabla \mathcal{L}(\boldsymbol{\theta}) = \pi_{\boldsymbol{b}}(\nabla \mathcal{L}(\hat{\boldsymbol{\theta}}))$, and we proved the lemma in this case.

In the general case, by the definition of Clarke's subdifferential,

$$\partial^\circ \mathcal{L}(\boldsymbol{\theta}) := \text{conv} \left\{ \lim_{n \to \infty} \nabla \mathcal{L}(\boldsymbol{\theta}_n) : \mathcal{L} \text{ differentiable at } \boldsymbol{\theta}_n, \lim_{n \to \infty} \boldsymbol{\theta}_n = \boldsymbol{\theta} \right\}.$$

For any $\hat{\boldsymbol{\theta}}_n \to \hat{\boldsymbol{\theta}}$ with $\hat{\boldsymbol{\theta}}_n \in \Omega_{\mathcal{S}}$, $\pi_{\boldsymbol{b}}(\hat{\boldsymbol{\theta}}_n) \to \pi_{\boldsymbol{b}}(\hat{\boldsymbol{\theta}})$, and

$$\lim_{n \to \infty} \nabla \mathcal{L}(\pi_{\boldsymbol{b}}(\hat{\boldsymbol{\theta}}_n)) = \lim_{n \to \infty} \pi_{\boldsymbol{b}}(\nabla \mathcal{L}(\hat{\boldsymbol{\theta}}_n)) = \pi_{\boldsymbol{b}} \left( \lim_{n \to \infty} \nabla \mathcal{L}(\hat{\boldsymbol{\theta}}_n) \right).$$

Taking the convex hull, it follows that $\pi_{\boldsymbol{b}}(\partial^\circ \mathcal{L}(\hat{\boldsymbol{\theta}})) \subseteq \partial^\circ \mathcal{L}(\pi_{\boldsymbol{b}}(\hat{\boldsymbol{\theta}}))$, and we finished the proof.  □

*Proof for Lemma E.1.* For notations we write $\hat{\boldsymbol{\theta}}(t) := \varphi(\hat{\boldsymbol{\theta}}_0, t)$ and $\boldsymbol{\theta}(t) = \pi_{\boldsymbol{b}}(\hat{\boldsymbol{\theta}}(t))$. Then $\frac{\mathrm{d}}{\mathrm{d}t} \hat{\boldsymbol{\theta}}(t) \in -\partial^\circ \mathcal{L}(\hat{\boldsymbol{\theta}}(t))$ for a.e. $t$. At these $t$, $\frac{\mathrm{d}}{\mathrm{d}t} \boldsymbol{\theta}(t) = \pi_{\boldsymbol{b}}(\frac{\mathrm{d}}{\mathrm{d}t} \hat{\boldsymbol{\theta}}(t)) \in \pi_{\boldsymbol{b}}(-\partial^\circ \mathcal{L}(\hat{\boldsymbol{\theta}}(t)))$. From Lemma E.2 we know $\pi_{\boldsymbol{b}}(\partial^\circ \mathcal{L}(\hat{\boldsymbol{\theta}}(t))) \subseteq \partial^\circ \mathcal{L}(\boldsymbol{\theta}(t))$. Then $\frac{\mathrm{d}}{\mathrm{d}t} \boldsymbol{\theta}(t) \in -\partial^\circ \mathcal{L}(\boldsymbol{\theta}(t))$ for a.e. $t$, and therefore $\boldsymbol{\theta}(t)$ is indeed a gradient flow trajectory.  □

*Proof for Lemma 5.3.* By Lemma E.1, $\pi_{\boldsymbol{b}}(\varphi(\hat{\boldsymbol{\theta}}_0, t))$ is indeed a gradient flow trajectory. Then, as $\pi_{\boldsymbol{b}}(\varphi(\hat{\boldsymbol{\theta}}_0, 0)) = \pi_{\boldsymbol{b}}(\hat{\boldsymbol{\theta}}_0)$, as well as the fact that $\hat{\boldsymbol{\theta}}_0$ and $\pi_{\boldsymbol{b}}(\hat{\boldsymbol{\theta}}_0)$ are non-branching starting points, the gradient flow trajectory is unique and therefore $\pi_{\boldsymbol{b}}(\varphi(\hat{\boldsymbol{\theta}}_0, t)) = \varphi(\pi_{\boldsymbol{b}}(\hat{\boldsymbol{\theta}}_0), t)$ for all $t \geq 0$.  □

## E.2 A General Theorem for Limiting Trajectory Near Zero

Before analyzing Phase II, we first give a characterization for gradient flow on Leaky ReLU networks with logistic loss, starting near $r\hat{\boldsymbol{\theta}}$, where $\hat{\boldsymbol{\theta}}$ is a well-aligned parameter vector defined below. We only assume that the inputs are bounded $\|\boldsymbol{x}_i\|_2 \leq 1$ and $\lambda := \max\{|G(\boldsymbol{w})| : \boldsymbol{w} \in \mathbb{S}^{d-1}\} > 0$. Theorems in the section will be used again in the non-symmetric case.

**Definition E.3** (Well-aligned Parameter Vector). We say that $\hat{\boldsymbol{\theta}} := (\hat{\boldsymbol{w}}_1, \ldots, \hat{\boldsymbol{w}}_m, \hat{a}_1, \ldots, \hat{a}_m)$ is a *well-aligned parameter vector* if it satisfies the following for some $1 \leq p \leq m$:

1. For $1 \leq k \leq p$, $\frac{\hat{\boldsymbol{w}}_k}{\|\hat{\boldsymbol{w}}_k\|_2}$ attains the maximum value of $|G(\boldsymbol{w})|$ on $\mathbb{S}^{d-1}$, i.e., $\left|G(\frac{\hat{\boldsymbol{w}}_k}{\|\hat{\boldsymbol{w}}_k\|_2})\right| = \lambda$;

2. For $1 \leq k \leq p$, $\hat{a}_k = \text{sgn}(G(\hat{\boldsymbol{w}}_k))\|\hat{\boldsymbol{w}}_k\|_2$;

3. For $1 \leq k \leq p$, $\langle \hat{\boldsymbol{w}}_k, \boldsymbol{x}_i \rangle \neq 0$ for all $i \in [n]$;

4. For $p + 1 \leq k \leq m$, $\hat{\boldsymbol{w}}_k = \mathbf{0}$, $\hat{a}_k = 0$.

Our analysis for Phase I shows that weight vectors approximately align to either of $\bar{\boldsymbol{\mu}}$ or $-\bar{\boldsymbol{\mu}}$, and both of them are maximizers of $|G(\boldsymbol{w})|$. Therefore, gradient flow goes near a well-aligned parameter vector (with $p = m$) at the end of Phase I.

The following is the main theorem of this subsection.

**Theorem E.4.** *Let $\hat{\boldsymbol{\theta}}$ be a well-aligned parameter vector. Let $\hat{\Delta} := \min_{k\in[p], i\in[n]} \frac{|\langle\hat{\boldsymbol{w}}_k, \boldsymbol{x}_i\rangle|}{\|\hat{\boldsymbol{\theta}}\|_{\mathrm{M}}} > 0$. Define $T_2(r) := \frac{1}{\lambda}\ln\frac{1}{r}$ and let $t_0$ be the following time constant*

$$t_0 := \frac{1}{2\lambda}\ln\frac{\lambda\hat{\Delta}}{16m\|\hat{\boldsymbol{\theta}}\|_{\mathrm{M}}^2}. \tag{14}$$

*Then for all $t \in (-\infty, t_0]$, the following is true:*

1. *$\lim_{r\to 0}\varphi(r\hat{\boldsymbol{\theta}}, T_2(r) + t)$ exists. This limit is independent of the choice of $\varphi$ when the gradient flow may not be unique.*

2. *$\lim_{r\to 0}\varphi(r\hat{\boldsymbol{\theta}}, T_2(r) + t)$ lies near $e^{\lambda t}\hat{\boldsymbol{\theta}}$:*

$$\left\|\lim_{r\to 0}\varphi\left(r\hat{\boldsymbol{\theta}}, T_2(r) + t\right) - e^{\lambda t}\hat{\boldsymbol{\theta}}\right\|_{\mathrm{M}} \leq \frac{4m\|\hat{\boldsymbol{\theta}}\|_{\mathrm{M}}^3}{\lambda}e^{3\lambda t}.$$

3. *Let $\boldsymbol{\theta}_1, \boldsymbol{\theta}_2, \ldots$ be a series of parameters converging to $\mathbf{0}$, $r_1, r_2, \ldots$ be a series of positive real numbers converging to 0. If $\|\boldsymbol{\theta}_s - r_s\hat{\boldsymbol{\theta}}\|_2 \leq Cr_s^{1+\kappa}$ for some $C > 0, \kappa > 0$, then*

$$\lim_{s\to\infty}\varphi\left(\boldsymbol{\theta}_s, T_2(r_s) + t\right) = \lim_{r\to 0}\varphi(r\hat{\boldsymbol{\theta}}, T_2(r) + t).$$

Now we prove Theorem E.4. Throughout this subsection, we fix a well-aligned parameter vector $\hat{\boldsymbol{\theta}} := (\hat{\boldsymbol{w}}_1, \ldots, \hat{\boldsymbol{w}}_m, \hat{a}_1, \ldots, \hat{a}_m)$ with constant $p \in [m]$. We also use $t_0$ and $T_2(r)$ to denote the same constant $t_0$ defined by (14) and the same function $T_2(r) := \frac{1}{\lambda}\ln\frac{1}{r}$ as in Theorem E.4.

For any parameter $\boldsymbol{\theta} = (\boldsymbol{w}_1, \ldots, \boldsymbol{w}_m, a_1, \ldots, a_m)$, we use $\|\boldsymbol{\theta}\|_{\mathrm{P}}, \|\boldsymbol{\theta}\|_{\mathrm{R}}$ to denote the following semi-norms respectively,

$$\|\boldsymbol{\theta}\|_{\mathrm{P}} := \max_{k\in[p]}\{\max\{\|\boldsymbol{w}_k\|_2, |a_k|\}\}, \qquad \|\boldsymbol{\theta}\|_{\mathrm{R}} := \max_{p<k\leq m}\{\max\{\|\boldsymbol{w}_k\|_2, |a_k|\}\}.$$

The M-norm can be expressed in terms of P-norm and R-norm: $\|\boldsymbol{\theta}\|_{\mathrm{M}} = \max\{\|\boldsymbol{\theta}\|_{\mathrm{P}}, \|\boldsymbol{\theta}\|_{\mathrm{R}}\}$. Also note that Condition 4 in Definition E.3 is now equivalent to $\|\hat{\boldsymbol{\theta}}\|_{\mathrm{R}} = 0$.

For $k \in [p]$, define $\widehat{\mathcal{W}}_k := \{\boldsymbol{w} \in \mathbb{R}^d : \langle\hat{\boldsymbol{w}}_k, \boldsymbol{x}_i\rangle \cdot \langle\boldsymbol{w}, \boldsymbol{x}_i\rangle > 0, \forall i \in [n]\}$ to be the set of weights that share the same activation pattern as $\hat{\boldsymbol{w}}_k$.

**Lemma E.5.** *If $r > 0$ is small enough and the initial point $\boldsymbol{\theta}_0$ of gradient flow satisfies $\|\boldsymbol{\theta}_0 - r\hat{\boldsymbol{\theta}}\|_{\mathrm{M}} \leq Cr^{1+\kappa}$ for some $C > 0, \kappa > 0$, then for any $-T_2(r) \leq t \leq t_0$, the following four properties hold:*

1. For all $k \in [p]$, $\boldsymbol{w}_k(T_2(r) + t) \in \widehat{\mathcal{W}}_k$;

2. $\|\varphi(\boldsymbol{\theta}_0, T_2(r) + t)\|_{\mathrm{M}} \leq 2e^{\lambda t}\|\hat{\boldsymbol{\theta}}\|_{\mathrm{M}}$;

3. $\|\varphi(\boldsymbol{\theta}_0, T_2(r) + t) - e^{\lambda t}\hat{\boldsymbol{\theta}}\|_{\mathrm{P}} \leq Cr^\kappa e^{\lambda t} + \frac{4m\|\hat{\boldsymbol{\theta}}\|_{\mathrm{M}}^3}{\lambda}e^{3\lambda t}$;

4. $\|\varphi(\boldsymbol{\theta}_0, T_2(r) + t) - e^{\lambda t}\hat{\boldsymbol{\theta}}\|_{\mathrm{R}} \leq 2Cr^\kappa e^{\lambda t}$.

*Proof.* Let $\boldsymbol{\theta}(t) := \varphi(\boldsymbol{\theta}_0, t)$ be gradient flow on $\mathcal{L}$ starting from $\boldsymbol{\theta}_0$, and $\tilde{\boldsymbol{\theta}}(t) := re^{\lambda t}\hat{\boldsymbol{\theta}}$. Let $t_1, t_2$ be the following time constants and define $t_{\max} := \min\{t_0, t_1, t_2\}$:

$$t_1 := \inf\{t \geq 0 : \exists k \in [p], \boldsymbol{w}_k(t) \notin \widehat{\mathcal{W}}_k\} - T_2(r),$$

$$t_2 := \inf\{t \geq 0 : \|\boldsymbol{\theta}(t)\|_{\mathrm{M}} \geq 2re^{\lambda t}\|\hat{\boldsymbol{\theta}}\|_{\mathrm{M}}\} - T_2(r).$$

We also define $r_{\max} := \left(\frac{\|\hat{\boldsymbol{\theta}}\|_{\mathrm{M}}\hat{\Delta}}{8C}\right)^{1/\kappa}$. We only consider the dynamics for $r \leq r_{\max}$, $t < T_2(r) + t_{\max}$. Our goal is to show that

$$\|\boldsymbol{\theta}(t) - \tilde{\boldsymbol{\theta}}(t)\|_{\mathrm{P}} \leq Cr^{1+\kappa}e^{\lambda t} + \frac{4m\|\hat{\boldsymbol{\theta}}\|_{\mathrm{M}}^3}{\lambda}r^3 e^{3\lambda t}, \qquad \|\boldsymbol{\theta}(t) - \tilde{\boldsymbol{\theta}}(t)\|_{\mathrm{R}} \leq 2Cr^{1+\kappa}e^{\lambda t}$$

within the time interval $[0, T_2(r) + t_{\max})$ (and thus it also holds for $[0, T_2(r) + t_{\max}]$ by continuity), and to show that $t_0$ is actually equal to $t_{\max}$, i.e., $t_0$ is the minimum among $t_0, t_1, t_2$. It is easy to see that proving these suffice to deduce the original lemma statement, given the translation of time $e^{\lambda T_2(r)} = \frac{1}{r}$.

For $k \in [m]$, by Lemma B.11 we have

$$\left\|\frac{\mathrm{d}\boldsymbol{w}_k}{\mathrm{d}t} - a_k\partial^\circ G(\boldsymbol{w}_k)\right\|_2 \subseteq (-\infty, m\|\boldsymbol{\theta}\|_{\mathrm{M}}^2|a_k|], \qquad \left|\frac{\mathrm{d}a_k}{\mathrm{d}t} - G(\boldsymbol{w}_k)\right| \leq m\|\boldsymbol{\theta}\|_{\mathrm{M}}^2\|\boldsymbol{w}_k\|_2. \quad (15)$$

For $\tilde{\boldsymbol{\theta}}(t)$, a simple calculus shows that for all $t \geq 0$,

$$\forall k \in [p]: \qquad \frac{\mathrm{d}\tilde{\boldsymbol{w}}_k}{\mathrm{d}t} = \lambda\tilde{a}_k\frac{\hat{\boldsymbol{w}}_k}{\|\hat{\boldsymbol{w}}_k\|_2}, \qquad \frac{\mathrm{d}\tilde{a}_k}{\mathrm{d}t} = \lambda\left\langle\frac{\hat{\boldsymbol{w}}_k}{\|\hat{\boldsymbol{w}}_k\|_2}, \tilde{\boldsymbol{w}}_k\right\rangle. \quad (16)$$

$$\forall p < k \leq m: \qquad |\tilde{a}_k| = \|\tilde{\boldsymbol{w}}_k\|_2 = 0. \quad (17)$$

**Bounding** $\|\boldsymbol{\theta}(t) - \tilde{\boldsymbol{\theta}}(t)\|_{\mathrm{P}}$. For $k \in [p]$, $\partial^\circ G(\boldsymbol{w}_k) = \{\nabla G(\boldsymbol{w}_k)\} = \{\nabla G(\hat{\boldsymbol{w}}_k)\}$. Also note that $\nabla G(\hat{\boldsymbol{w}}_k) = \nabla G(\frac{\hat{\boldsymbol{w}}_k}{\|\hat{\boldsymbol{w}}_k\|_2}) = \lambda\frac{\hat{\boldsymbol{w}}_k}{\|\hat{\boldsymbol{w}}_k\|_2}$ by Lemma B.5. Then $a_k\partial^\circ G(\boldsymbol{w}_k) = \{\lambda a_k\frac{\hat{\boldsymbol{w}}_k}{\|\hat{\boldsymbol{w}}_k\|_2}\}$ and $G(\boldsymbol{w}_k) = \lambda\left\langle\frac{\hat{\boldsymbol{w}}_k}{\|\hat{\boldsymbol{w}}_k\|_2}, \boldsymbol{w}_k\right\rangle$. Combining these with (15) gives

$$\max\left\{\left\|\frac{\mathrm{d}\boldsymbol{w}_k}{\mathrm{d}t} - \lambda a_k\frac{\hat{\boldsymbol{w}}_k}{\|\hat{\boldsymbol{w}}_k\|_2}\right\|_2, \left|\frac{\mathrm{d}a_k}{\mathrm{d}t} - \lambda\left\langle\frac{\hat{\boldsymbol{w}}_k}{\|\hat{\boldsymbol{w}}_k\|_2}, \boldsymbol{w}_k\right\rangle\right|\right\} \leq m\|\boldsymbol{\theta}\|_{\mathrm{M}}^3. \quad (18)$$

Then by (16) we have

$$\left\|\frac{\mathrm{d}\boldsymbol{\theta}}{\mathrm{d}t} - \frac{\mathrm{d}\tilde{\boldsymbol{\theta}}}{\mathrm{d}t}\right\|_{\mathrm{P}} \leq m\|\boldsymbol{\theta}\|_{\mathrm{M}}^3 + \max_{k \in [p]}\left\{\left\|\lambda(a_k - \tilde{a}_k)\frac{\hat{\boldsymbol{w}}_k}{\|\hat{\boldsymbol{w}}_k\|_2}\right\|_2, \left|\lambda\left\langle\frac{\hat{\boldsymbol{w}}_k}{\|\hat{\boldsymbol{w}}_k\|_2}, \boldsymbol{w}_k - \tilde{\boldsymbol{w}}_k\right\rangle\right|\right\}$$

$$\leq m\|\boldsymbol{\theta}\|_{\mathrm{M}}^3 + \lambda\|\boldsymbol{\theta} - \tilde{\boldsymbol{\theta}}\|_{\mathrm{P}}.$$

Taking the integral gives $\|\boldsymbol{\theta}(t) - \tilde{\boldsymbol{\theta}}(t)\|_{\mathrm{P}} \leq \|\boldsymbol{\theta}(0) - \tilde{\boldsymbol{\theta}}(0)\|_{\mathrm{P}} + \int_0^t (m\|\boldsymbol{\theta}(\tau)\|_{\mathrm{M}}^3 + \lambda\|\boldsymbol{\theta}(\tau) - \tilde{\boldsymbol{\theta}}(\tau)\|_{\mathrm{P}})\mathrm{d}\tau$. Note that $t_{\max} \leq t_2$. Then

$$\|\boldsymbol{\theta}(t) - \tilde{\boldsymbol{\theta}}(t)\|_{\mathrm{P}} \leq \|\boldsymbol{\theta}(0) - \tilde{\boldsymbol{\theta}}(0)\|_{\mathrm{P}} + \int_0^t \left(8mr^3 e^{3\lambda t}\|\hat{\boldsymbol{\theta}}\|_{\mathrm{M}}^3 + \lambda\|\boldsymbol{\theta}(\tau) - \tilde{\boldsymbol{\theta}}(\tau)\|_{\mathrm{P}}\right)\mathrm{d}\tau$$

$$\leq Cr^{1+\kappa} + \frac{8}{3\lambda}mr^3 e^{3\lambda t}\|\hat{\boldsymbol{\theta}}\|_{\mathrm{M}}^3 + \lambda\int_0^t \|\boldsymbol{\theta}(\tau) - \tilde{\boldsymbol{\theta}}(\tau)\|_{\mathrm{P}}\mathrm{d}\tau.$$

By Grönwall's inequality (11), we have

$$\|\boldsymbol{\theta}(t) - \tilde{\boldsymbol{\theta}}(t)\|_{\mathrm{P}} \le Cr^{1+\kappa} + \frac{8}{3\lambda}mr^3e^{3\lambda t}\|\hat{\boldsymbol{\theta}}\|_{\mathrm{M}}^3 + \int_0^t \left(Cr^{1+\kappa} + \frac{8}{3\lambda}mr^3e^{3\lambda\tau}\|\hat{\boldsymbol{\theta}}\|_{\mathrm{M}}^3\right)\lambda e^{\lambda(t-\tau)}\mathrm{d}\tau$$

$$\le Cr^{1+\kappa} + Cr^{1+\kappa}(e^{\lambda t}-1) + \frac{8}{3\lambda}mr^3e^{3\lambda t}\|\hat{\boldsymbol{\theta}}\|_{\mathrm{M}}^3 + \frac{8}{3\lambda}mr^3 \cdot \frac{e^{\lambda t}}{2}(e^{2\lambda t}-1)\|\hat{\boldsymbol{\theta}}\|_{\mathrm{M}}^3$$

$$\le Cr^{1+\kappa}e^{\lambda t} + \frac{8}{3\lambda}mr^3e^{3\lambda t}(1+1/2)\|\hat{\boldsymbol{\theta}}\|_{\mathrm{M}}^3.$$

Therefore we can conclude that

$$\|\boldsymbol{\theta}(t) - \tilde{\boldsymbol{\theta}}(t)\|_{\mathrm{P}} \le Cr^{1+\kappa}e^{\lambda t} + \frac{4m\|\hat{\boldsymbol{\theta}}\|_{\mathrm{M}}^3}{\lambda}r^3e^{3\lambda t}. \tag{19}$$

**Bounding $\|\boldsymbol{\theta}(t) - \tilde{\boldsymbol{\theta}}(t)\|_{\mathrm{R}}$.** For $p < k \le m$, we can combine Theorem B.3 and (15) to give the following bound for the norm growth:

$$\frac{1}{2}\frac{\mathrm{d}\|\boldsymbol{w}_k\|_2^2}{\mathrm{d}t} = \frac{1}{2}\frac{\mathrm{d}|a_k|^2}{\mathrm{d}t} \le a_k G(\boldsymbol{w}_k) + |a_k| \cdot m\|\boldsymbol{\theta}\|_{\mathrm{M}}^2\|\boldsymbol{w}_k\|_2.$$

This implies

$$\frac{1}{2}\frac{\mathrm{d}|a_k|^2}{\mathrm{d}t} = \frac{1}{2}\frac{\mathrm{d}\|\boldsymbol{w}_k\|_2^2}{\mathrm{d}t} \le \|\boldsymbol{\theta}\|_{\mathrm{R}}^2(\lambda + m\|\boldsymbol{\theta}\|_{\mathrm{M}}^2). \tag{20}$$

Taking the integral gives $\|\boldsymbol{\theta}(t)\|_{\mathrm{R}}^2 \le \|\boldsymbol{\theta}(0)\|_{\mathrm{R}}^2 + \int_0^t 2\|\boldsymbol{\theta}(\tau)\|_{\mathrm{R}}^2(\lambda + m\|\boldsymbol{\theta}(\tau)\|_{\mathrm{M}}^2)\mathrm{d}\tau$. Note that $t_{\max} \le t_2$ and $\|\boldsymbol{\theta}(0)\|_{\mathrm{R}} \le Cr^{1+\kappa}$. Then

$$\|\boldsymbol{\theta}(t)\|_{\mathrm{R}}^2 \le C^2r^{2(1+\kappa)} + \int_0^t 2\|\boldsymbol{\theta}(\tau)\|_{\mathrm{R}}^2(\lambda + 4mr^2e^{2\lambda\tau}\|\hat{\boldsymbol{\theta}}\|_{\mathrm{M}}^2)\mathrm{d}\tau$$

By Grönwall's inequality (12), we have

$$\|\boldsymbol{\theta}(t)\|_{\mathrm{R}}^2 \le C^2r^{2(1+\kappa)}\exp\left(\int_0^t 2(\lambda + 4mr^2e^{2\lambda\tau}\|\hat{\boldsymbol{\theta}}\|_{\mathrm{M}}^2)\mathrm{d}\tau\right)$$

$$\le C^2r^{2(1+\kappa)}\exp\left(2\lambda t + \frac{4m\|\hat{\boldsymbol{\theta}}\|_{\mathrm{M}}^2}{\lambda}r^2e^{2\lambda t}\right).$$

Taking the square root gives

$$\|\boldsymbol{\theta}(t)\|_{\mathrm{R}} \le Cr^{1+\kappa}\exp\left(\lambda t + \frac{2m\|\hat{\boldsymbol{\theta}}\|_{\mathrm{M}}^2}{\lambda}r^2e^{2\lambda t}\right).$$

For $t \le T(r) + t_{\max} \le T(r) + t_0$, we can use the the definition (14) of $t_0$ to deduce that $\frac{2m\|\hat{\boldsymbol{\theta}}\|_{\mathrm{M}}^2}{\lambda}r^2e^{2\lambda t} \le \frac{2m\|\hat{\boldsymbol{\theta}}\|_{\mathrm{M}}^2}{\lambda}e^{2\lambda t_0} = \hat{\Delta}/8 \le 1/8$. Therefore, we have

$$\|\boldsymbol{\theta}(t) - \tilde{\boldsymbol{\theta}}(t)\|_{\mathrm{R}} = \|\boldsymbol{\theta}(t)\|_{\mathrm{R}} \le Cr^{1+\kappa}e^{\lambda t+1/8} < Cr^{1+\kappa}e^{\lambda t+\ln 2} = 2Cr^{1+\kappa}e^{\lambda t}. \tag{21}$$

**Bounding $t_{\max}$.** To prove the lemma, now we only need to show that $t_{\max} = t_0$. Combining (19) and (21), we have for $t \le T_2(r) + t_{\max}$,

$$\|\boldsymbol{\theta}(t) - \tilde{\boldsymbol{\theta}}(t)\|_{\mathrm{M}} \le 2Cr^{1+\kappa}e^{\lambda t} + \frac{4m\|\hat{\boldsymbol{\theta}}\|_{\mathrm{M}}^3}{\lambda}r^3e^{3\lambda t}.$$

Since $r \le r_{\max}$, $2Cr^\kappa \le \frac{1}{4}\|\hat{\boldsymbol{\theta}}\|_{\mathrm{M}}\hat{\Delta}$. By definition (14) of $t_0$, $\frac{4m\|\hat{\boldsymbol{\theta}}\|_{\mathrm{M}}^3}{\lambda}r^2e^{2\lambda t} \le \frac{4m\|\hat{\boldsymbol{\theta}}\|_{\mathrm{M}}^3}{\lambda}e^{2\lambda t_0} \le \frac{1}{4}\|\hat{\boldsymbol{\theta}}\|_{\mathrm{M}}\hat{\Delta}$. Then we have $2Cr^\kappa + \frac{4m\|\hat{\boldsymbol{\theta}}\|_{\mathrm{M}}^3}{\lambda}r^2e^{2\lambda t} \le \frac{1}{2}\|\hat{\boldsymbol{\theta}}\|_{\mathrm{M}}\hat{\Delta}$ and thus

$$\|\boldsymbol{\theta}(t) - \tilde{\boldsymbol{\theta}}(t)\|_{\mathrm{M}} \le re^{\lambda t}\left(2Cr^\kappa + \frac{4m\|\hat{\boldsymbol{\theta}}\|_{\mathrm{M}}^3}{\lambda}r^2e^{2\lambda t}\right) \le \frac{1}{2}re^{\lambda t}\|\hat{\boldsymbol{\theta}}\|_{\mathrm{M}}\hat{\Delta}. \tag{22}$$

For all time $0 \leq t < T_2(r) + t_{\max}$, we can use (22) to deduce

$$\mathrm{sgn}(\langle \hat{\boldsymbol{w}}_k, \boldsymbol{w}_i \rangle) \langle \boldsymbol{w}(t), \boldsymbol{x}_i \rangle \geq \mathrm{sgn}(\langle \hat{\boldsymbol{w}}_k, \boldsymbol{x}_i \rangle) \langle re^{\lambda t} \hat{\boldsymbol{w}}_k, \boldsymbol{x}_i \rangle - \frac{1}{2} re^{\lambda t} \|\hat{\boldsymbol{\theta}}\|_{\mathrm{M}} \hat{\Delta}$$

$$= re^{\lambda t} \left( |\langle \hat{\boldsymbol{w}}_k, \boldsymbol{x}_i \rangle| - \frac{1}{2} \|\hat{\boldsymbol{\theta}}\|_{\mathrm{M}} \hat{\Delta} \right)$$

$$\geq re^{\lambda t} \|\hat{\boldsymbol{\theta}}\|_{\mathrm{M}} \hat{\Delta}/2 > 0,$$

which implies $t_1 > t_{\max}$.

For norm growth, we can again use (22) to deduce

$$\|\boldsymbol{\theta}(t)\|_{\mathrm{M}} \leq \|\tilde{\boldsymbol{\theta}}(t)\|_{\mathrm{M}} + \frac{1}{2} re^{\lambda t} \|\hat{\boldsymbol{\theta}}\|_{\mathrm{M}} \hat{\Delta} = re^{\lambda t} \left( \|\hat{\boldsymbol{\theta}}\|_{\mathrm{M}} + \frac{1}{2} \|\hat{\boldsymbol{\theta}}\|_{\mathrm{M}} \hat{\Delta} \right)$$

$$\leq \frac{3}{2} re^{\lambda t} \|\hat{\boldsymbol{\theta}}\|_{\mathrm{M}} < 2re^{\lambda t} \|\hat{\boldsymbol{\theta}}\|_{\mathrm{M}},$$

which implies $t_2 > t_{\max}$.

Now we have $t_1 > t_{\max}, t_2 > t_{\max}$. Recall that $t_{\max} := \min\{t_0, t_1, t_2\}$ by definition. Then $t_{\max} = t_0$ must hold, which completes the proof. $\qquad\square$

**Lemma E.6.** *If $r > 0$ is small enough and the initial point $\boldsymbol{\theta}_0$ of gradient flow satisfies $\|\boldsymbol{\theta}_0 - r\hat{\boldsymbol{\theta}}\|_{\mathrm{M}} \leq Cr^{1+\kappa}$ for some $C > 0, \kappa > 0$, then for all $t \in [-T_2(r), t_0]$,*

$$\|\varphi(\boldsymbol{\theta}_0, T_2(r) + t) - \varphi(r\hat{\boldsymbol{\theta}}, T_2(r) + t)\|_{\mathrm{M}} \leq 4Cr^{\kappa} e^{\lambda t}.$$

*Proof.* Let $\boldsymbol{\theta}(t) := \varphi(\boldsymbol{\theta}_0, t)$ and $\tilde{\boldsymbol{\theta}}(t) := \varphi(r\hat{\boldsymbol{\theta}}, t)$ be gradient flows starting from $\boldsymbol{\theta}_0$ and $r\hat{\boldsymbol{\theta}}$. For notation simplicity, let $h_{ki} = y_i \phi'(\hat{\boldsymbol{w}}_k^\top \boldsymbol{x}_i)$. Let $g_i := -\ell'(y_i f_{\boldsymbol{\theta}}(\boldsymbol{x}_i))$, $\tilde{g}_i := -\ell'(y_i f_{\tilde{\boldsymbol{\theta}}}(\boldsymbol{x}_i))$.

By Lemma E.5, we can make $r$ to be small enough so that the four properties hold for both $\boldsymbol{\theta}(T_2(r)+t)$ and $\tilde{\boldsymbol{\theta}}(T_2(r) + t)$ when $t \leq t_0$.

**Bounding the Difference for $1 \leq k \leq p$.** For all $t \leq t_0$ and $k \in [p]$, we know that $\phi'(\boldsymbol{w}_k^\top \boldsymbol{x}_i) = \phi'(\tilde{\boldsymbol{w}}_k^\top \boldsymbol{x}_i) = h_{ki}$, and thus for $\boldsymbol{w}_k, \tilde{\boldsymbol{w}}_k$ we have

$$\left\| \frac{\mathrm{d}\boldsymbol{w}_k}{\mathrm{d}t} - \frac{\mathrm{d}\tilde{\boldsymbol{w}}_k}{\mathrm{d}t} \right\|_2 = \left\| \frac{a_k}{n} \sum_{i=1}^n g_i h_{ki} \boldsymbol{x}_i - \frac{\tilde{a}_k}{n} \sum_{i=1}^n \tilde{g}_i h_{ki} \boldsymbol{x}_i \right\|_2$$

$$\leq |a_k - \tilde{a}_k| \cdot \underbrace{\left\| \frac{1}{n} \sum_{i=1}^n \tilde{g}_i h_{ki} \boldsymbol{x}_i \right\|_2}_{\Lambda(t)} + |a_k| \cdot \underbrace{\left\| \frac{1}{n} \sum_{i=1}^n (g_i - \tilde{g}_i) h_{ki} \boldsymbol{x}_i \right\|_2}_{\Delta(t)}$$

$$=: \Lambda(t) \cdot |a_k - \tilde{a}_k| + |a_k| \cdot \Delta(t).$$

and for $a_k, \tilde{a}_k$ we have

$$\left\| \frac{\mathrm{d}a_k}{\mathrm{d}t} - \frac{\mathrm{d}\tilde{a}_k}{\mathrm{d}t} \right\|_2 = \left| \frac{1}{n} \sum_{i=1}^n g_i \phi(\boldsymbol{w}_k^\top \boldsymbol{x}_i) - \frac{1}{n} \sum_{i=1}^n \tilde{g}_i \phi(\tilde{\boldsymbol{w}}_k^\top \boldsymbol{x}_i) \right|$$

$$= \left| \frac{1}{n} \sum_{i=1}^n g_i h_{ki} \boldsymbol{w}_k^\top \boldsymbol{x}_i - \frac{1}{n} \sum_{i=1}^n \tilde{g}_i h_{ki} \tilde{\boldsymbol{w}}_k^\top \boldsymbol{x}_i \right|$$

$$= \|\boldsymbol{w}_k - \tilde{\boldsymbol{w}}_k\|_2 \cdot \left\| \frac{1}{n} \sum_{i=1}^n \tilde{g}_i h_{ki} \boldsymbol{x}_i \right\|_2 + \|\boldsymbol{w}_k\|_2 \cdot \left\| \frac{1}{n} \sum_{i=1}^n (g_i - \tilde{g}_i) h_{ki} \boldsymbol{x}_i \right\|_2$$

$$= \Lambda(t) \cdot \|\boldsymbol{w}_k - \tilde{\boldsymbol{w}}_k\|_2 + \|\boldsymbol{w}_k\|_2 \cdot \Delta(t).$$

Therefore, $\left\| \frac{\mathrm{d}\boldsymbol{\theta}}{\mathrm{d}t} - \frac{\mathrm{d}\tilde{\boldsymbol{\theta}}}{\mathrm{d}t} \right\|_{\mathrm{P}} \leq \Lambda(t) \cdot \|\boldsymbol{\theta} - \tilde{\boldsymbol{\theta}}\|_{\mathrm{M}} + \|\boldsymbol{\theta}\|_{\mathrm{M}} \cdot \Delta(t)$. Now we turn to bound $\Lambda(t)$ and $\Delta(t)$.

By Lipschitzness of $\ell'$ and Lemma B.10, we have

$$|-\ell'(0) - \tilde{g}_i| \leq m\|\tilde{\boldsymbol{\theta}}\|_{\mathrm{M}}^2, \qquad |g_i - \tilde{g}_i| \leq m\|\boldsymbol{\theta} - \tilde{\boldsymbol{\theta}}\|_{\mathrm{M}} \left( \|\boldsymbol{\theta}\|_{\mathrm{M}} + \|\tilde{\boldsymbol{\theta}}\|_{\mathrm{M}} \right).$$

For $\Lambda(t)$, by triangle inequality and Lemma B.5 we have

$$\Lambda(t) \leq \left\| \frac{-\ell'(0)}{n} \sum_{i=1}^{n} h_{ki} \boldsymbol{x}_i \right\|_2 + m\|\tilde{\boldsymbol{\theta}}\|_{\mathrm{M}}^2 = \|\nabla G(\hat{\boldsymbol{w}}_k)\|_2 + m\|\tilde{\boldsymbol{\theta}}\|_{\mathrm{M}}^2 = \lambda + m\|\tilde{\boldsymbol{\theta}}\|_{\mathrm{M}}^2,$$

For $\Delta(t)$, we use triangle inequality again to give the following bound:

$$\Delta(t) \leq \frac{1}{n} \sum_{i=1}^{n} |g_i - \tilde{g}_i| \leq m\|\boldsymbol{\theta} - \tilde{\boldsymbol{\theta}}\|_{\mathrm{M}} \left( \|\boldsymbol{\theta}\|_{\mathrm{M}} + \|\tilde{\boldsymbol{\theta}}\|_{\mathrm{M}} \right).$$

Therefore, we can conclude that

$$
\begin{aligned}
\left\| \frac{\mathrm{d}\boldsymbol{\theta}}{\mathrm{d}t} - \frac{\mathrm{d}\tilde{\boldsymbol{\theta}}}{\mathrm{d}t} \right\|_{\mathrm{P}} &\leq (\lambda + m\|\tilde{\boldsymbol{\theta}}\|_{\mathrm{M}}^2) \cdot \|\boldsymbol{\theta} - \tilde{\boldsymbol{\theta}}\|_{\mathrm{M}} + \|\boldsymbol{\theta}\|_{\mathrm{M}} \cdot m\|\boldsymbol{\theta} - \tilde{\boldsymbol{\theta}}\|_{\mathrm{M}} \left( \|\boldsymbol{\theta}\|_{\mathrm{M}} + \|\tilde{\boldsymbol{\theta}}\|_{\mathrm{M}} \right) \\
&\leq \left( \lambda + 3m \max\{\|\boldsymbol{\theta}\|_{\mathrm{M}}, \|\tilde{\boldsymbol{\theta}}\|_{\mathrm{M}}\}^2 \right) \|\boldsymbol{\theta} - \tilde{\boldsymbol{\theta}}\|_{\mathrm{M}} \\
&\leq \left( \lambda + 12mr^2 e^{2\lambda t} \|\hat{\boldsymbol{\theta}}\|_{\mathrm{M}}^2 \right) \|\boldsymbol{\theta} - \tilde{\boldsymbol{\theta}}\|_{\mathrm{M}},
\end{aligned}
$$

where the last inequality uses the 2nd property in Lemma E.5. Note that $\|\boldsymbol{\theta}_0 - r\hat{\boldsymbol{\theta}}\|_{\mathrm{P}} \leq Cr^{1+\kappa}$. So we can write it into the integral form:

$$\|\boldsymbol{\theta}(t) - \tilde{\boldsymbol{\theta}}(t)\|_{\mathrm{P}} \leq Cr^{1+\kappa} + \int_0^t \left( \lambda + 12mr^2 e^{2\lambda\tau} \|\hat{\boldsymbol{\theta}}\|_{\mathrm{M}}^2 \right) \|\boldsymbol{\theta}(\tau) - \tilde{\boldsymbol{\theta}}(\tau)\|_{\mathrm{M}} \mathrm{d}\tau. \qquad (23)$$

**Bounding the Difference for $p < k \leq m$.** By Lemma B.17, $\|\tilde{\boldsymbol{\theta}}(t)\|_{\mathrm{R}} = 0$ for all $t \geq 0$, so $\|\boldsymbol{\theta} - \tilde{\boldsymbol{\theta}}\|_{\mathrm{R}} = \|\boldsymbol{\theta}\|_{\mathrm{R}}$. By the 4th property in Lemma E.5, we then have

$$\|\boldsymbol{\theta}(t) - \tilde{\boldsymbol{\theta}}(t)\|_{\mathrm{R}} = \|\boldsymbol{\theta}(t)\|_{\mathrm{R}} = \|\boldsymbol{\theta}(t) - re^{\lambda t}\hat{\boldsymbol{\theta}}\|_{\mathrm{R}} \leq 2Cr^{1+\kappa} e^{\lambda t}.$$

So we can verify that $\|\boldsymbol{\theta}(t) - \tilde{\boldsymbol{\theta}}(t)\|_{\mathrm{R}}$ satisfies the following inequality:

$$\|\boldsymbol{\theta}(t) - \tilde{\boldsymbol{\theta}}(t)\|_{\mathrm{R}} \leq 2Cr^{1+\kappa} + \int_0^t \lambda\|\boldsymbol{\theta}(\tau) - \tilde{\boldsymbol{\theta}}(\tau)\|_{\mathrm{R}} \mathrm{d}\tau. \qquad (24)$$

**Bounding the Difference for All.** Combining Lemma E.5 and Lemma E.5, we have the following inequality for $\|\boldsymbol{\theta}(t) - \tilde{\boldsymbol{\theta}}(t)\|_{\mathrm{M}}$:

$$\|\boldsymbol{\theta}(t) - \tilde{\boldsymbol{\theta}}(t)\|_{\mathrm{M}} \leq 2Cr^{1+\kappa} + \int_0^t \left( \lambda + 12mr^2 e^{2\lambda\tau} \|\hat{\boldsymbol{\theta}}\|_{\mathrm{M}}^2 \right) \|\boldsymbol{\theta}(\tau) - \tilde{\boldsymbol{\theta}}(\tau)\|_{\mathrm{M}} \mathrm{d}\tau.$$

By Grönwall's inequality (12),

$$
\begin{aligned}
\|\boldsymbol{\theta}(t) - \tilde{\boldsymbol{\theta}}(t)\|_{\mathrm{M}} &\leq 2Cr^{1+\kappa} \exp \left( \int_0^t \left( \lambda + 12mr^2 e^{2\lambda\tau} \|\hat{\boldsymbol{\theta}}\|_{\mathrm{M}}^2 \right) \mathrm{d}\tau \right) \\
&\leq 2Cr^{1+\kappa} \exp \left( \lambda t + \frac{6m\|\hat{\boldsymbol{\theta}}\|_{\mathrm{M}}^2}{\lambda} r^2 e^{2\lambda t} \right).
\end{aligned}
$$

By definition (14) of $t_0$, we have $\frac{6m\|\hat{\boldsymbol{\theta}}\|_{\mathrm{M}}^2}{\lambda} r^2 e^{2\lambda t} \leq \frac{6m\|\hat{\boldsymbol{\theta}}\|_{\mathrm{M}}^2}{\lambda} e^{2\lambda t_0} = \frac{3\hat{\Delta}}{8} \leq 3/8 < \ln 2$. Therefore we have the following bound for $\|\boldsymbol{\theta}(t) - \tilde{\boldsymbol{\theta}}(t)\|_{\mathrm{M}}$:

$$\|\boldsymbol{\theta}(t) - \tilde{\boldsymbol{\theta}}(t)\|_{\mathrm{M}} \leq 2Cr^{1+\kappa} e^{\lambda t + \ln 2} = 4Cr^{1+\kappa} e^{\lambda t}.$$

At time $T_2(r) + t \in [0, T_2(r) + t_0]$, this bound can be rewritten as

$$\|\boldsymbol{\theta}(T_2(r) + t) - \tilde{\boldsymbol{\theta}}(T_2(r) + t)\|_{\mathrm{M}} \leq 4Cr^\kappa e^{\lambda t},$$

which completes the proof. $\qquad \square$

*Proof for Theorem E.4.* First we show that $\lim_{r \to 0} \varphi(r\hat{\boldsymbol{\theta}}, T_2(r) + t)$ exists. We consider the case of $r \le r_{\max}$, where $r_{\max}$ is chosen to be small enough so that the properties in Lemma E.5 hold. For any $r' < r$, by Lemma E.5 we have

$$\left\| \varphi\left(r'\hat{\boldsymbol{\theta}}, T_2(r') + \frac{1}{\lambda}\ln r\right) - r\hat{\boldsymbol{\theta}} \right\|_{\mathrm{M}} \le \frac{4m\|\hat{\boldsymbol{\theta}}\|_{\mathrm{M}}^3}{\lambda}r^3 \le C'r^{1+\kappa'},$$

where $C' = \frac{4m\|\hat{\boldsymbol{\theta}}\|_{\mathrm{M}}^3}{\lambda}$, $\kappa' = 2$. Applying Lemma E.6, we then have

$$\left\| \varphi\left(\varphi\left(r'\hat{\boldsymbol{\theta}}, T_2(r') + \frac{1}{\lambda}\ln r\right), T_2(r) + t\right) - \varphi\left(r\hat{\boldsymbol{\theta}}, T_2(r) + t\right) \right\|_{\mathrm{M}} \le 4C'r^{\kappa'}e^{\lambda t}.$$

Note that $T_2(r') + \frac{1}{\lambda}\ln r + T_2(r) + t = T_2(r') + t$. So this proves

$$\|\varphi(r'\hat{\boldsymbol{\theta}}, T_2(r') + t) - \varphi(r\hat{\boldsymbol{\theta}}, T_2(r) + t)\|_{\mathrm{M}} \le 4C'r^{\kappa'}e^{\lambda t}.$$

For any fixed $t \le t_0$, the RHS converges to 0 as $r \to 0$, which implies Cauchy convergence of the limit $\lim_{r \to 0} \varphi(r\hat{\boldsymbol{\theta}}, T_2(r) + t)$ and thus the limit exists. By the 1st property in Lemma E.5, we know that there is no activation pattern switch in the time interval $t \in [0, T_2(r) + t_0]$ if $r$ is small enough. This means $\mathcal{L}$ is locally smooth near the trajectory of $\varphi(r\hat{\boldsymbol{\theta}}, T_2(r) + t)$ and thus the trajectory is unique. Therefore, the limit $\lim_{r \to 0} \varphi(r\hat{\boldsymbol{\theta}}, T_2(r) + t)$ is uniquely defined.

By Lemma E.5,

$$\|\varphi(r\hat{\boldsymbol{\theta}}, T_2(r) + t) - e^{\lambda t}\hat{\boldsymbol{\theta}}\|_{\mathrm{M}} \le \frac{4m\|\hat{\boldsymbol{\theta}}\|_{\mathrm{M}}^3}{\lambda}e^{3\lambda t}.$$

Taking $r \to 0$ on both sides gives the range of the limit $\lim_{r \to 0} \varphi(r\hat{\boldsymbol{\theta}}, T_2(r) + t)$:

$$\left\| \lim_{r \to 0} \varphi(r\hat{\boldsymbol{\theta}}, T_2(r) + t) - e^{\lambda t}\hat{\boldsymbol{\theta}} \right\|_{\mathrm{M}} \le \frac{4m\|\hat{\boldsymbol{\theta}}\|_{\mathrm{M}}^3}{\lambda}e^{3\lambda t}.$$

For $s \to \infty$, by Lemma E.6, we have

$$\lim_{s \to \infty} \left\| \varphi\left(\hat{\boldsymbol{\theta}}_s, T_2(r_s) + t\right) - \varphi\left(r_s\hat{\boldsymbol{\theta}}, T_2(r_s) + t\right) \right\|_{\mathrm{M}} = 0.$$

So $\lim_{s \to \infty} \varphi(\boldsymbol{\theta}_s, T_2(r_s) + t) = \lim_{r \to 0} \varphi(r\hat{\boldsymbol{\theta}}, T_2(r) + t)$ is proved. $\qquad\square$

## E.3  Proof for Approximate Embedding

To analyze Phase II, we need to deal with approximate embedding instead of the exact one. For this, we further divide Phase II into Phase II.1 and II.2 and analyze them in order. At the end of this subsection we will prove Lemma 5.4.

### E.3.1  Proofs for Phase II.1

Given the discussions in the previous sections, we are ready to present proofs for the phase II dynamics (Lemma 5.4) here.

We subdivide the dynamics of Phase II into Phase II.1 and Phase II.2. At the end of Phase I, we show that the parameters grow to norm $O(r)$ in time $T_1(r)$. In Phase II.1, we extend the dynamic to time $T_1(r) + T_2(r)$ so that the parameters grow into constant norms (irrelevant to $r$ and $\sigma_{\mathrm{init}}$). Then, when the initialization scale becomes sufficiently small, at the end of Phase II.1 the parameters become sufficiently close to the embedded parameters from two neurons at constant norms, so the subsequent dynamics is a good approximate embedding until the norm of the parameters grow sufficiently large to ensure directional convergence in Phase III. Here we show the results in Phase II.1.

**Lemma E.7.** *For $m \ge 2$, with probability $1 - 2^{-(m-1)}$ over the random draw of $\bar{\boldsymbol{\theta}}_0 \sim \mathcal{D}_{\mathrm{init}}(1)$, the vector $\bar{\boldsymbol{b}} \in \mathbb{R}^m$ with entries $\bar{b}_k := \frac{\langle \bar{\boldsymbol{w}}_k, \bar{\boldsymbol{\mu}}\rangle + \bar{a}_k}{2\sqrt{m}\|\bar{\boldsymbol{\theta}}_0\|_{\mathrm{M}}}$ defined as in Lemma 5.2 is a good embedding vector.*

*Proof.* Note that $\bar{\boldsymbol{b}}$ is a good embedding vector iff $\bar{\boldsymbol{b}}' = 2\sqrt{m}\|\bar{\boldsymbol{\theta}}_0\|_{\mathrm{M}}\bar{\boldsymbol{b}}$ is a good embedding vector. Recall that $\bar{\boldsymbol{w}}_k \overset{\mathrm{i.i.d.}}{\sim} \mathcal{N}(\mathbf{0}, \boldsymbol{I})$, $\bar{a}_k \overset{\mathrm{i.i.d.}}{\sim} \mathcal{N}(0, c_{\mathrm{ainit}}^2)$. By the property of Gaussian variables,

$$\bar{b}'_k = \langle \bar{\boldsymbol{w}}_k, \bar{\boldsymbol{\mu}}\rangle + \bar{a}_k \sim \mathcal{N}(0, 1 + c_{\mathrm{ainit}}^2).$$

Thus $\bar{b}' \sim \mathcal{N}(\mathbf{0}, (1 + c_{\text{ainit}}^2)\mathbf{I})$. Since it is a continuous probability distribution, $\bar{b}' \neq \mathbf{0}$ with probability 1. By symmetry and independence, we know that $\Pr[\forall k \in [m] : \bar{b}'_k > 0] = 2^{-m}$ and $\Pr[\forall k \in [m] : \bar{b}'_k < 0] = 2^{-m}$. So $\bar{b}'$ is a good embedding vector with probability $1 - 2^{-m} - 2^{-m} = 1 - 2^{-(m-1)}$, and so is $\bar{b}$. $\qquad\square$

**Lemma E.8.** *Let* $T_2(r) := \frac{1}{\lambda_0} \ln \frac{1}{r}$, *then* $T_{12} := T_1(r) + T_2(r) = \frac{1}{\lambda_0} \ln \frac{1}{\sqrt{m}\sigma_{\text{init}}\|\bar{\theta}_0\|_{\text{M}}}$ *regardless the choice of* $r$. *For random draw of* $\bar{\theta}_0 \sim \mathcal{D}_{\text{init}}(1)$, *if* $\bar{b} \in \mathbb{R}^m$ *defined as in Lemma 5.2 is a good embedding vector, then there exists* $t_0 \in \mathbb{R}$ *such that the following holds:*

1. *For the two-neuron dynamics starting with rescaled initialization in the direction of* $\hat{\theta} := (\bar{b}_+, \bar{b}_+\bar{\mu}, \bar{b}_-, \bar{b}_-\bar{\mu})$, *for all* $t \in (-\infty, t_0]$, *the limit* $\tilde{\theta}(t) := \lim_{r \to 0} \varphi(r\hat{\theta}, T_2(r) + t)$ *exists and lies near* $e^{\lambda_0 t}\hat{\theta}$:

$$\left\| \tilde{\theta}(t) - e^{\lambda_0 t}\hat{\theta} \right\|_{\text{M}} \leq \frac{4m\|\hat{\theta}\|_{\text{M}}^3}{\lambda_0} e^{3\lambda_0 t} = O(e^{3\lambda_0 t}).$$

2. *For the* $m$-*neuron dynamics* $\theta(t)$, *the following holds for all* $t \in (-\infty, t_0]$,

$$\lim_{\sigma_{\text{init}} \to 0} \theta(T_{12} + t) = \pi_{\bar{b}}(\tilde{\theta}(t)).$$

*Proof.* Under Assumptions 4.1 and 4.5, the maximum value of $|G(\boldsymbol{w})|$ on $\mathbb{S}^{d-1}$ is $\lambda_0$ and is attained at $\pm\bar{\mu}$. Given a good embedding vector $\bar{b}$, both $\hat{\theta}$ and $\hat{\theta}_\pi := \pi_{\bar{b}}(\hat{\theta})$ are well-aligned parameter vectors (Definition E.3) for width-2 and width-$m$ Leaky ReLU nets respectively. By Theorem E.4, there exists $t_0 \in \mathbb{R}$ such that the following two limits exist for all $t \in (-\infty, t_0]$:

$$\tilde{\theta}(t) := \lim_{r \to 0} \varphi\left(r\hat{\theta}, T_2(r) + t\right), \qquad \tilde{\theta}_\pi(t) := \lim_{r \to 0} \varphi\left(r\hat{\theta}_\pi, T_2(r) + t\right).$$

Note that by Lemma E.1, we have $\pi_{\bar{b}}(\varphi(r\hat{\theta}, T_2(r) + t))$ is a trajectory of gradient flow starting from $r\hat{\theta}_\pi$. The uniqueness of $\tilde{\theta}_\pi(t)$ (for all possible choices of $\varphi$) shows that

$$\pi_{\bar{b}}(\tilde{\theta}(t)) = \lim_{r \to 0} \pi_{\bar{b}}\left(\varphi\left(r\hat{\theta}, T_2(r) + t\right)\right) = \tilde{\theta}_\pi(t).$$

By Lemma 5.2, for $\sigma_{\text{init}}$ small enough, if we choose $r$ so that $\sigma_{\text{init}} = \frac{r^3}{\sqrt{m}\|\bar{\theta}_0\|_{\text{M}}}$, then for some universal constant $C$ we have

$$\|\theta(T_1(r)) - r\hat{\theta}_\pi\|_{\text{M}} \leq \frac{Cr^3}{\sqrt{m}}.$$

Applying Theorem E.4 proves the following for all $t \in (-\infty, t_0]$:

$$\lim_{\sigma_{\text{init}} \to 0} \varphi(\theta(T_1(r)), T_2(r) + t) = \tilde{\theta}_\pi(t).$$

Therefore $\lim_{\sigma_{\text{init}} \to 0} \theta(T_{12} + t) = \pi_{\bar{b}}(\tilde{\theta}(t))$. $\qquad\square$

### E.4 Proofs for Phase II.2

Next, at the end of Phase II.1, $\theta(T_{12} + t_0)$ has a constant norm. Then we show the trajectory convergence with respect to the initialization scale in Phase II.2.

**Lemma E.9.** *If* $\pi_{\bar{b}}(\tilde{\theta}(t_0))$ *is non-branching and* $\lim_{\sigma_{\text{init}} \to 0} \theta(T_{12} + t_0) = \pi_{\bar{b}}(\tilde{\theta}(t_0))$ *for some constant* $t_0$, *then for all* $t > t_0$, $\lim_{\sigma_{\text{init}} \to 0} \theta(T_{12} + t) = \pi_{\bar{b}}(\tilde{\theta}(t))$.

We first start with a simple lemma on gradient upper bounds, and then show that the trajectory of gradient flow is Lipschitz with time.

**Lemma E.10.** *For every* $\theta \in \mathbb{R}^D$, $\|g\|_2 \leq \|\theta\|_2$ *for all* $g \in \partial^\circ \mathcal{L}(\theta)$.

*Proof.* Note that $|\ell'(q)| \leq 1, |\phi'(z)| \leq 1 \|\boldsymbol{x}_i\|_2 \leq 1, |y_i| \leq 1$. For every $\boldsymbol{\theta} \in \Omega_{\mathcal{S}}$ (i.e., no activation function has zero input), by the chain rule (10), we have

$$\left\|\frac{\partial \mathcal{L}(\boldsymbol{\theta})}{\partial \boldsymbol{w}_k}\right\|_2 = \left\|\frac{1}{n}\sum_{i\in[n]}\ell'(q_i(\boldsymbol{\theta}))y_i a_k \phi'(\boldsymbol{w}_k^\top \boldsymbol{x}_i)\boldsymbol{x}_i\right\|_2 \leq \frac{1}{n}\sum_{i\in[n]}|a_k| = |a_k|,$$

$$\left\|\frac{\partial \mathcal{L}(\boldsymbol{\theta})}{\partial a_k}\right\|_2 = \left\|\frac{1}{n}\sum_{i\in[n]}\ell'(q_i(\boldsymbol{\theta}))y_i \phi(\boldsymbol{w}_k^\top \boldsymbol{x}_i)\right\|_2 \leq \frac{1}{n}\sum_{i\in[n]}|\boldsymbol{w}_k^\top \boldsymbol{x}_i| \leq \|\boldsymbol{w}_k\|_2.$$

So $\|\nabla \mathcal{L}(\boldsymbol{\theta})\|_2 \leq \|\boldsymbol{\theta}\|_2$. We can finish the proof for any $\boldsymbol{\theta} \in \mathbb{R}^D$ by taking limits in $\Omega_{\mathcal{S}}$. $\qquad\square$

**Lemma E.11.** *The gradient flow trajectory $\boldsymbol{\theta}(T_{12} + t)$, in the interval $t \in [t_0, t_s]$ for any $t_s > t_0$, is Lipschitz in $t$ with Lipschitz constant $\|\boldsymbol{\theta}(T_{12} + t_0)\|_2 e^{(t_s - t_0)}$.*

*Proof.* By Lemma E.10, $\left\|\frac{\mathrm{d}}{\mathrm{d}t}\boldsymbol{\theta}(T_{12} + t)\right\|_2 \leq \|\boldsymbol{\theta}(T_{12} + t)\|_2$. So $\frac{\mathrm{d}\|\boldsymbol{\theta}(T_{12}+t)\|_2}{\mathrm{d}t} \leq \|\boldsymbol{\theta}(T_{12} + t)\|_2$. By Grönwall's inequality (12), $\|\boldsymbol{\theta}(T_{12} + t)\|_2 \leq \|\boldsymbol{\theta}(T_{12} + t_0)\|_2 e^{t-t_0}$. Then $\left\|\frac{\mathrm{d}}{\mathrm{d}t}\boldsymbol{\theta}(T_{12} + t)\right\|_2 \leq \|\boldsymbol{\theta}(T_{12} + t)\|_2 \leq \|vtheta(T_{12} + t_0)\|_2 e^{t_s - t_0}$. $\qquad\square$

Now we are ready to prove Lemma E.9.

*Proof of Lemma E.9.* When $\sigma_{\text{init}} \to 0$, as $\boldsymbol{\theta}(T_{12} + t_0) \to \pi_{\bar{\boldsymbol{b}}}(\tilde{\boldsymbol{\theta}}(t_0))$, we can select a countable sequence $(\sigma_{\text{init}})_i \to 0$ and trajectories $\boldsymbol{\theta}_i(T_{12} + t)$ with initialization scale $(\sigma_{\text{init}})_i$. We show that for every $t \geq t_0$, there must be $\boldsymbol{\theta}_i(T_{12}+t) = \varphi(\boldsymbol{\theta}_i(T_{12}+t_0), t-t_0) \to \pi_{\bar{\boldsymbol{b}}}(\tilde{\boldsymbol{\theta}}(t)) = \varphi(\pi_{\bar{\boldsymbol{b}}}(\tilde{\boldsymbol{\theta}}(t_0)), t-t_0)$.

If this does not hold for some $t = T$, then there must be a limit point $\boldsymbol{q}_T$ of points in $\{\varphi(\boldsymbol{\theta}_i(T_{12} + t_0), T - t_0)\}_{i=1}^\infty$ such that $\boldsymbol{q}_T \neq \varphi(\pi_{\bar{\boldsymbol{b}}}(\tilde{\boldsymbol{\theta}}(t_0)), T - t_0)$ and a converging subsequence in $\{\varphi(\boldsymbol{\theta}_i(T_{12} + t_0), T - t_0)\}_{i=1}^\infty$ to $\boldsymbol{q}_T$. Thus WLOG we assume that the sequence is chosen so that

$$\lim_{i\to\infty} \varphi(\boldsymbol{\theta}_i(T_{12} + t_0), T - t_0) = \boldsymbol{q}_T \neq \varphi(\pi_{\bar{\boldsymbol{b}}}(\tilde{\boldsymbol{\theta}}(t_0)), T - t_0).$$

We then show that there is a trajectory of the gradient flow that starts from $\pi_{\bar{\boldsymbol{b}}}(\tilde{\boldsymbol{\theta}}(t_0))$ and reaches $\boldsymbol{q}_T$ at time $T - t_0$, thereby causing a contradiction to our assumption that $\pi_{\bar{\boldsymbol{b}}}(\tilde{\boldsymbol{\theta}}(t_0))$ is non-branching.

For any pair of $L_0$-Lipschitz continuous functions $\boldsymbol{f}, \boldsymbol{g} : [t_0, T] \to \mathbb{R}^D$, define the $L^\infty$-distance to be $\|\boldsymbol{f} - \boldsymbol{g}\|_\infty := \sup_{t\in[t_0,T]} \|\boldsymbol{f}(t) - \boldsymbol{g}(t)\|_2$. Note that the space of $L_0$-Lipschitz functions with bounded function values is compact under $L^\infty$-distance, and therefore any sequence of functions in this space has a converging subsequence whose limit is also $L_0$-Lipschitz.

Let $C := \sup_i\{\|\boldsymbol{\theta}_i(T_{12} + t_0)\|_2\}$, then as $\{\boldsymbol{\theta}_i(T_{12} + t_0)\}$ is converging to $\pi_{\bar{\boldsymbol{b}}}(\tilde{\boldsymbol{\theta}}(t_0)) \neq \infty$, $C < \infty$. By Lemma E.11 we know each trajectory $\boldsymbol{\theta}_i(T_{12} + t)$ is $(Ce^{T-t_0})$-Lipschitz for $t \in [t_0, T]$. This means we can find a subsequence $1 \leq i_1 < i_2 < i_3 < \cdots$ that the trajectory $\{\boldsymbol{\theta}_{i_j}(T_{12} + t)\}$ $L^\infty$-converges on $[t_0, T]$ as $j \to \infty$. Then the pointwise limit $\boldsymbol{q}(t) := \lim_{j\to\infty}\boldsymbol{\theta}_{i_j}(T_{12} + t)$ exists for every $t \in [t_0, T]$. $\boldsymbol{q}(t_0) = \pi_{\bar{\boldsymbol{b}}}(\tilde{\boldsymbol{\theta}}(t_0))$, $\boldsymbol{q}(T) = \boldsymbol{q}_T$.

Finally we show that $\boldsymbol{q}(t)$ is indeed a valid gradient flow trajectory. Notice that $\boldsymbol{q}(t)$ is $(Ce^{T-t_0})$-Lipschitz, then by Rademacher theorem for $\boldsymbol{q}(t)$ is differentiable for a.e. $t \in [t_0, T]$. We are left to show $\boldsymbol{q}'(t) \in \partial^\circ \mathcal{L}(\boldsymbol{q}(t))$ whenever $\boldsymbol{q}$ is differentiable at $t$.

For any $\epsilon > 0$ that $[t, t + \epsilon] \subseteq [t_0, T]$, we investigate the behaviour of $\boldsymbol{q}(t)$ in the $\epsilon$-neighborhood of $t$. Let $\Omega_j$ be the set of $\tau \in [t_0, T]$ so that $\frac{\mathrm{d}}{\mathrm{d}\tau}\boldsymbol{\theta}_{i_k}(T_{12} + \tau) \in -\partial^\circ \mathcal{L}(\boldsymbol{\theta}_{i_k}(T_{12} + \tau))$. By definition of differential inclusion, $\Omega_j$ has full measure in $[t_0, T]$. Define $B_{j,\epsilon}$ be the following closed convex hull:

$$B_{j,\epsilon} = \overline{\text{conv}}\left\{\frac{\mathrm{d}}{\mathrm{d}\tau}\boldsymbol{\theta}_{i_k}(T_{12} + \tau) : k \geq j, \tau \in [t, t + \epsilon] \cap \Omega_j\right\}.$$

It is easy to see that $B_{j,\epsilon}$ is monotonic with respect to $j$. Then we know that for any $j$,

$$\frac{\boldsymbol{\theta}_{i_j}(T_{12} + t + \epsilon) - \boldsymbol{\theta}_{i_j}(T_{12} + t)}{\epsilon} = \frac{1}{\epsilon}\int_t^{t+\epsilon}\frac{\mathrm{d}}{\mathrm{d}\tau}\boldsymbol{\theta}_{i_j}(T_{12} + \tau)\mathrm{d}\tau \in B_{j,\epsilon},$$

Then taking the limits $j \to \infty$, as all $B_{j,\epsilon}$ are closed, we know $\frac{q(t+\epsilon)-q(t)}{\epsilon} \in \lim_{j \to \infty} B_{j,\epsilon}$.

Now let $C_{j,\epsilon}, C_\epsilon$ be the following closed convex hull of subgradients:

$$C_{j,\epsilon} = \overline{\mathrm{conv}} \left( \bigcup_{\substack{k \geq j \\ \tau \in [t, t+\epsilon]}} \partial^\circ \mathcal{L}(\boldsymbol{\theta}_{i_k}(T_{12} + \tau)) \right), \quad C_\epsilon = \overline{\mathrm{conv}} \left( \bigcup_{\tau \in [t, t+\epsilon]} \partial^\circ \mathcal{L}(\boldsymbol{q}(T_{12} + \tau)) \right).$$

Then we know $B_{j,\epsilon} \subseteq -C_{j,\epsilon}$ for all $j \geq 1$ and $\epsilon > 0$. Notice that $C_{j,\epsilon}$ and $C_\epsilon$ are also monotonic with respect to $j$ and $\epsilon$ respectively so we can take the respective limit. As for $\tau \in [t, t+\epsilon]$, $\lim_{j \to \infty} \boldsymbol{\theta}_{i_j}(T_{12} + \tau) = \boldsymbol{q}(T_{12} + \tau)$, by the upper-semicontinuity of $\partial^\circ \mathcal{L}$, $\lim_{j \to \infty} C_{j,\epsilon} \subseteq C_\epsilon$. Then $\frac{q(t+\epsilon)-q(t)}{\epsilon} \in \lim_{j \to \infty} B_{j,\epsilon} \subseteq \lim_{j \to \infty} C_{j,\epsilon} \subseteq C_\epsilon$.

When $t \in [t_0, T)$ and $\boldsymbol{q}(t)$ is differential at $t$, we can take the limit $\epsilon \to 0$, and by the upper-semicontinuity of $\partial^\circ \mathcal{L}$ again, we have

$$\boldsymbol{q}'(t) \in \lim_{\epsilon \to 0} C_\epsilon \subseteq \overline{\mathrm{conv}}(-\partial^\circ \mathcal{L}(\boldsymbol{q}(T_{12} + t))) = -\partial^\circ \mathcal{L}(\boldsymbol{q}(T_{12} + t))$$

as $\partial^\circ \mathcal{L}(\boldsymbol{q}(T_{12}+t))$ is closed convex for any $t$. Therefore $\boldsymbol{q}(t)$ is indeed a gradient flow trajectory. $\square$

*Proof for Lemma 5.4.* We can prove Lemma 5.4 by combining Lemmas E.7 to E.9 together. For $-\infty < t \leq t_0$, by Lemma E.8, $\left\| \tilde{\boldsymbol{\theta}}(t) - e^{\lambda_0 t} \hat{\boldsymbol{\theta}} \right\|_{\mathrm{M}} \leq \frac{4m\|\hat{\boldsymbol{\theta}}\|_{\mathrm{M}}^3}{\lambda_0} e^{3\lambda_0 t}$. With $\hat{\boldsymbol{\theta}} := (\bar{b}_+, \bar{b}_+ \bar{\boldsymbol{\mu}}, \bar{b}_-, \bar{b}_- \bar{\boldsymbol{\mu}})$, by choosing a threshold $t_s < t_0$ small enough, we can have for any $t \leq t_s$,

- $\tilde{a}_1(t) \geq e^{\lambda_0 t} \bar{b}_+ - \frac{4m\|\hat{\boldsymbol{\theta}}\|_{\mathrm{M}}^3}{\lambda_0} e^{3\lambda_0 t} > 0$;

- $\tilde{a}_2(t) \leq e^{\lambda_0 t} \bar{b}_- + \frac{4m\|\hat{\boldsymbol{\theta}}\|_{\mathrm{M}}^3}{\lambda_0} e^{3\lambda_0 t} < 0$;

- $\langle \tilde{\boldsymbol{w}}_1(t), \boldsymbol{w}^* \rangle \geq e^{\lambda_0 t} \bar{b}_+ \langle \bar{\boldsymbol{\mu}}, \boldsymbol{w}^* \rangle - \frac{4m\|\hat{\boldsymbol{\theta}}\|_{\mathrm{M}}^3}{\lambda_0} e^{3\lambda_0 t} > \frac{8m\|\hat{\boldsymbol{\theta}}\|_{\mathrm{M}}^3}{\lambda_0} e^{3\lambda_0 t} > 0$;

- $\langle \tilde{\boldsymbol{w}}_2(t), \boldsymbol{w}^* \rangle \leq e^{\lambda_0 t} \bar{b}_- \langle \bar{\boldsymbol{\mu}}, \boldsymbol{w}^* \rangle + \frac{4m\|\hat{\boldsymbol{\theta}}\|_{\mathrm{M}}^3}{\lambda_0} e^{3\lambda_0 t} < -\frac{8m\|\hat{\boldsymbol{\theta}}\|_{\mathrm{M}}^3}{\lambda_0} e^{3\lambda_0 t} < 0$.

Then $\tilde{\boldsymbol{\theta}}(t) \neq 0$ for all $t \leq t_s$. For $t > t_s$, we know $\tilde{\boldsymbol{\theta}}(t) \neq 0$ by applying Theorem B.19. Finally by Lemmas E.8 and E.9 we know $\lim_{\sigma_{\mathrm{init}} \to 0} \boldsymbol{\theta}(T_{12} + t) = \pi_{\bar{\boldsymbol{b}}}(\tilde{\boldsymbol{\theta}}(t))$ for all $t$. $\square$

# F  Proofs for Phase III

## F.1  Two Neuron Case: Margin Maximization

In this subsection we prove Theorem 5.5 for the symmetric datasets. By Theorem B.19 and Theorem 3.1, we know that gradient flow must converge in a KKT-margin direction of width-2 two-layer Leaky ReLU network (Definition B.8). Thus we first give some characterizations for KKT-margin directions by proving Lemma F.1 and Lemma F.2.

**Lemma F.1.** *Given $\boldsymbol{u}_1, \boldsymbol{u}_2 \in \mathbb{R}^d$, if $y_i(\phi(\langle \boldsymbol{u}_1, \boldsymbol{x}_i \rangle) - \phi(-\langle \boldsymbol{u}_2, \boldsymbol{x}_i \rangle)) \geq 1$ for all $i \in [n]$, then*

$$y_i(h_i^{(1)} \langle \boldsymbol{u}_2, \boldsymbol{x}_i \rangle + h_i^{(2)} \langle \boldsymbol{u}_1, \boldsymbol{x}_i \rangle) \geq 1,$$

*for all Clarke's sub-differentials $h_i^{(1)} \in \phi^\circ(\langle \boldsymbol{u}_1, \boldsymbol{x}_i \rangle), h_i^{(2)} \in \phi^\circ(-\langle \boldsymbol{u}_2, \boldsymbol{x}_i \rangle)$.*

*Proof.* We prove by cases for any fixed $i \in [n]$. By Assumption 4.1 we have

$$y_i(\phi(\langle \boldsymbol{u}_1, \boldsymbol{x}_i \rangle) - \phi(-\langle \boldsymbol{u}_2, \boldsymbol{x}_i \rangle)) \geq 1, \qquad -y_i(\phi(-\langle \boldsymbol{u}_1, \boldsymbol{x}_i \rangle) - \phi(\langle \boldsymbol{u}_2, \boldsymbol{x}_i \rangle)) \geq 1.$$

**Case 1.** Suppose that $\langle \boldsymbol{u}_1, \boldsymbol{x}_i \rangle \neq 0$, $\langle \boldsymbol{u}_2, \boldsymbol{x}_i \rangle \neq 0$. Then we have $h_i^{(1)}, h_i^{(2)} \in \{\alpha_{\text{leaky}}, 1\}$. If $h_i^{(1)} = h_i^{(2)}$, then $h_i^{(1)}\langle \boldsymbol{u}_2, \boldsymbol{x}_i \rangle = -\phi(-\langle \boldsymbol{u}_2, \boldsymbol{x}_i \rangle)$, $h_i^{(2)}\langle \boldsymbol{u}_1, \boldsymbol{x}_i \rangle = \phi(\langle \boldsymbol{u}_1, \boldsymbol{x}_i \rangle)$, and thus we have

$$
\begin{aligned}
y_i(h_i^{(1)}\langle \boldsymbol{u}_2, \boldsymbol{x}_i \rangle + h_i^{(2)}\langle \boldsymbol{u}_1, \boldsymbol{x}_i \rangle) &= y_i(-\phi(-\langle \boldsymbol{u}_2, \boldsymbol{x}_i \rangle) + \phi(\langle \boldsymbol{u}_1, \boldsymbol{x}_i \rangle)) \\
&= y_i(\phi(\langle \boldsymbol{u}_1, \boldsymbol{x}_i \rangle) - \phi(-\langle \boldsymbol{u}_2, \boldsymbol{x}_i \rangle)) \geq 1.
\end{aligned}
$$

Otherwise, $h_i^{(1)} \neq h_i^{(2)}$, then we have $h_i^{(1)}\langle \boldsymbol{u}_2, \boldsymbol{x}_i \rangle = \phi(\langle \boldsymbol{u}_2, \boldsymbol{x}_i \rangle)$, $h_i^{(2)}\langle \boldsymbol{u}_1, \boldsymbol{x}_i \rangle = -\phi(-\langle \boldsymbol{u}_1, \boldsymbol{x}_i \rangle)$, and thus

$$
\begin{aligned}
y_i(h_i^{(1)}\langle \boldsymbol{u}_2, \boldsymbol{x}_i \rangle + h_i^{(2)}\langle \boldsymbol{u}_1, \boldsymbol{x}_i \rangle) &= y_i\left(\phi(\langle \boldsymbol{u}_2, \boldsymbol{x}_i \rangle) - \phi(-\langle \boldsymbol{u}_1, \boldsymbol{x}_i \rangle)\right) \\
&= -y_i\left(\phi(-\langle \boldsymbol{u}_1, \boldsymbol{x}_i \rangle) - \phi(\langle \boldsymbol{u}_2, \boldsymbol{x}_i \rangle)\right) \geq 1.
\end{aligned}
$$

**Case 2.** Suppose that $\langle \boldsymbol{u}_1, \boldsymbol{x}_i \rangle = 0$ or $\langle \boldsymbol{u}_2, \boldsymbol{x}_i \rangle = 0$. WLOG we assume that $\langle \boldsymbol{u}_1, \boldsymbol{x}_i \rangle = 0$ (the case of $\langle \boldsymbol{u}_2, \boldsymbol{x}_i \rangle = 0$ can be proved similarly). Then we have

$$
-y_i\phi(-\langle \boldsymbol{u}_2, \boldsymbol{x}_i \rangle)) \geq 1, \qquad y_i\phi(\langle \boldsymbol{u}_2, \boldsymbol{x}_i \rangle)) \geq 1.
$$

If $\langle \boldsymbol{u}_2, \boldsymbol{x}_i \rangle = 0$, then the feasibility cannot be satisfied. So we must have $\langle \boldsymbol{u}_2, \boldsymbol{x}_i \rangle \neq 0$ and $h_i^{(2)} \in \{\alpha_{\text{leaky}}, 1\}$. This implies that $y_i\langle \boldsymbol{u}_2, \boldsymbol{x}_i \rangle \geq \frac{1}{\alpha_{\text{leaky}}}$.

Since $\langle \boldsymbol{u}_1, \boldsymbol{x}_i \rangle = 0$, we have $h_i^{(1)} \in [\alpha_{\text{leaky}}, 1]$. Therefore,

$$
y_i(h_i^{(1)}\langle \boldsymbol{u}_2, \boldsymbol{x}_i \rangle + h_i^{(2)}\langle \boldsymbol{u}_1, \boldsymbol{x}_i \rangle) = y_i h_i^{(1)}\langle \boldsymbol{u}_2, \boldsymbol{x}_i \rangle \geq y_i\alpha_{\text{leaky}}\langle \boldsymbol{u}_2, \boldsymbol{x}_i \rangle \geq 1,
$$

which completes the proof. $\qquad\square$

**Lemma F.2.** *If $(\boldsymbol{w}_1, \boldsymbol{w}_2, a_1, a_2)$ is along a KKT-margin direction of width-2 two-layer Leaky ReLU network and $a_1 > 0, a_2 < 0$, then $\boldsymbol{w}_1 = -\boldsymbol{w}_2$, $a_1 = -a_2 = \|\boldsymbol{w}_1\|_2$.*

*Proof.* WLOG we assume that $q_{\min}(\boldsymbol{\theta}) = 1$. By Definition B.8 and Lemma B.9, there exist $\lambda_1, \ldots, \lambda_n \geq 0$ and $h_1^{(1)}, \ldots, h_n^{(1)} \in \mathbb{R}, h_1^{(2)}, \ldots, h_n^{(2)} \in \mathbb{R}$ such that $h_i^{(1)} \in \phi^\circ(\langle \boldsymbol{w}_1, \boldsymbol{x}_i \rangle)$, $h_i^{(2)} \in \phi^\circ(\langle \boldsymbol{w}_2, \boldsymbol{x}_i \rangle)$, and the following conditions hold:

1. $\boldsymbol{w}_1 = a_1 \sum_{i \in [n]} \lambda_i y_i h_i^{(1)} \boldsymbol{x}_i$, $\boldsymbol{w}_2 = a_2 \sum_{i \in [n]} \lambda_i y_i h_i^{(2)} \boldsymbol{x}_i$;

2. $a_1 = \|\boldsymbol{w}_1\|_2$, $a_2 = -\|\boldsymbol{w}_2\|_2$;

3. For all $i \in [n]$, if $q_i(\boldsymbol{\theta}) \neq 1$ then $\lambda_i = 0$.

Let $\boldsymbol{u}_1 = a_1 \boldsymbol{w}_1$ and $\boldsymbol{u}_2 = -a_2 \boldsymbol{w}_2$. Let $\bar{\boldsymbol{u}}_1 := \frac{\boldsymbol{u}_1}{\|\boldsymbol{u}_1\|_2}, \bar{\boldsymbol{u}}_2 := -\frac{\boldsymbol{u}_2}{\|\boldsymbol{u}_2\|_2}$. Then the following conditions hold for all $i \in [n]$:

$$
\bar{\boldsymbol{u}}_1 - \sum_{i=1}^n \lambda_i h_i^{(1)} y_i \boldsymbol{x}_i = 0, \tag{25}
$$

$$
\bar{\boldsymbol{u}}_2 - \sum_{i=1}^n \lambda_i h_i^{(2)} y_i \boldsymbol{x}_i = 0, \tag{26}
$$

$$
\lambda_i(1 - y_i(\phi(\langle \boldsymbol{u}_1, \boldsymbol{x}_i \rangle) - \phi(-\langle \boldsymbol{u}_2, \boldsymbol{x}_i \rangle))) = 0. \tag{27}
$$

By homogeneity, $h_i^{(1)} \cdot \langle \boldsymbol{u}_1, \boldsymbol{x}_i \rangle = \phi(\langle \boldsymbol{u}_1, \boldsymbol{x}_i \rangle)$, $h_i^{(2)} \cdot \langle \boldsymbol{u}_2, \boldsymbol{x}_i \rangle = -\phi(-\langle \boldsymbol{u}_2, \boldsymbol{x}_i \rangle)$. Left-multiplying $(\boldsymbol{u}_1)^\top$ or $(\boldsymbol{u}_2)^\top$ on both sides of (25), we have

$$
\|\boldsymbol{u}_1\|_2 - \sum_{i=1}^n \lambda_i y_i \phi(\langle \boldsymbol{u}_1, \boldsymbol{x}_i \rangle) = 0, \tag{28}
$$

$$
\langle \bar{\boldsymbol{u}}_1, \bar{\boldsymbol{u}}_2 \rangle \|\boldsymbol{u}_2\|_2 - \sum_{i=1}^n \lambda_i h_i^{(1)} y_i \langle \boldsymbol{u}_2, \boldsymbol{x}_i \rangle = 0. \tag{29}
$$

Similarly, we have

$$\|\boldsymbol{u}_2\|_2 + \sum_{i=1}^{n} \lambda_i y_i \phi(-\langle \boldsymbol{u}_2, \boldsymbol{x}_i \rangle) = 0, \tag{30}$$

$$\langle \bar{\boldsymbol{u}}_1, \bar{\boldsymbol{u}}_2 \rangle \|\boldsymbol{u}_1\|_2 - \sum_{i=1}^{n} \lambda_i h_i^{(2)} y_i \langle \boldsymbol{u}_1, \boldsymbol{x}_i \rangle = 0. \tag{31}$$

Combining (28) and (30), we have

$$\|\boldsymbol{u}_1\|_2 + \|\boldsymbol{u}_2\|_2 = \sum_{i=1}^{n} \lambda_i y_i \phi(\langle \boldsymbol{u}_1, \boldsymbol{x}_i \rangle) - \sum_{i=1}^{n} \lambda_i y_i \phi(-\langle \boldsymbol{u}_2, \boldsymbol{x}_i \rangle) \tag{32}$$

$$= \sum_{i=1}^{n} \lambda_i y_i \left( \phi(\langle \boldsymbol{u}_1, \boldsymbol{x}_i \rangle) - \phi(-\langle \boldsymbol{u}_2, \boldsymbol{x}_i \rangle) \right) \tag{33}$$

$$= \sum_{i=1}^{n} \lambda_i, \tag{34}$$

where the last equality is due to (27).

Combining (29) and (31), we have

$$\langle \bar{\boldsymbol{u}}_1, \bar{\boldsymbol{u}}_2 \rangle (\|\boldsymbol{u}_1\|_2 + \|\boldsymbol{u}_2\|_2) = \sum_{i=1}^{n} \lambda_i h_i^{(1)} y_i \langle \boldsymbol{u}_2, \boldsymbol{x}_i \rangle + \sum_{i=1}^{n} \lambda_i h_i^{(2)} y_i \langle \boldsymbol{u}_1, \boldsymbol{x}_i \rangle \tag{35}$$

$$= \sum_{i=1}^{n} \lambda_i y_i \left( h_i^{(1)} \langle \boldsymbol{u}_2, \boldsymbol{x}_i \rangle + h_i^{(2)} \langle \boldsymbol{u}_1, \boldsymbol{x}_i \rangle \right) \tag{36}$$

$$\geq \sum_{i=1}^{n} \lambda_i, \tag{37}$$

where the last inequality is due to Lemma F.1. Since we have deduced that $\|\boldsymbol{u}_1\|_2 + \|\boldsymbol{u}_2\|_2 = \sum_{i=1}^{n} \lambda_i$, we further have

$$\langle \bar{\boldsymbol{u}}_1, \bar{\boldsymbol{u}}_2 \rangle (\|\boldsymbol{u}_1\|_2 + \|\boldsymbol{u}_2\|_2) \geq \|\boldsymbol{u}_1\|_2 + \|\boldsymbol{u}_2\|_2.$$

Combining this with $\langle \bar{\boldsymbol{u}}_1, \bar{\boldsymbol{u}}_2 \rangle \leq \|\bar{\boldsymbol{u}}_1\|_2 \|\bar{\boldsymbol{u}}_2\|_2 \leq 1$, we have $1 \leq \langle \bar{\boldsymbol{u}}_1, \bar{\boldsymbol{u}}_2 \rangle \leq 1$. So all the inequalities become equalities, and thus $\bar{\boldsymbol{u}}_1 = \bar{\boldsymbol{u}}_2$. (36) also equals to (37), so

$$y_i \left( h_i^{(1)} \langle \boldsymbol{u}_2, \boldsymbol{x}_i \rangle + h_i^{(2)} \langle \boldsymbol{u}_1, \boldsymbol{x}_i \rangle \right) = 1, \tag{38}$$

whenever $\lambda_i \neq 0$.

By (27), we have $y_i \left( h_i^{(1)} \langle \boldsymbol{u}_1, \boldsymbol{x}_i \rangle + h_i^{(2)} \langle \boldsymbol{u}_2, \boldsymbol{x}_i \rangle \right) = 1$ whenever $\lambda_i \neq 0$. Combining this with (38), we have

$$y_i (h_i^{(1)} - h_i^{(2)}) \langle \boldsymbol{u}_1, \boldsymbol{x}_i \rangle = y_i (h_i^{(1)} - h_i^{(2)}) \langle \boldsymbol{u}_2, \boldsymbol{x}_i \rangle.$$

Then we prove that $\langle \boldsymbol{u}_1, \boldsymbol{x}_i \rangle = \langle \boldsymbol{u}_2, \boldsymbol{x}_i \rangle$ by discussing two cases:

1. If $\langle \boldsymbol{u}_1, \boldsymbol{x}_i \rangle = 0$, then $\langle \boldsymbol{u}_2, \boldsymbol{x}_i \rangle = 0$ since $\bar{\boldsymbol{u}}_1 = \bar{\boldsymbol{u}}_2$;

2. Otherwise, we have $(h_i^{(1)}, h_i^{(2)}) = (1, \alpha_{\text{leaky}})$ or $(\alpha_{\text{leaky}}, 1)$ by symmetry, so $h_i^{(1)} \neq h_i^{(2)}$ and thus $\langle \boldsymbol{u}_1, \boldsymbol{x}_i \rangle = \langle \boldsymbol{u}_2, \boldsymbol{x}_i \rangle$.

This means $\boldsymbol{u}_1$ and $\boldsymbol{u}_2$ have the same projection onto the linear space spanned by $\{\boldsymbol{x}_i : \lambda_i \neq 0\}$. By (25) and (26), $\boldsymbol{u}_1$ and $\boldsymbol{u}_2$ are in the span of $\{\boldsymbol{x}_i : i \in [n], \lambda_i \neq 0\}$. Therefore, $\boldsymbol{u}_1 = \boldsymbol{u}_2$ and we can easily deduce that $\boldsymbol{w}_1 = -\boldsymbol{w}_2, a_1 = -a_2 = \|\boldsymbol{w}_1\|_2$. □

**Lemma F.3.** *If $\boldsymbol{\theta} = (\boldsymbol{w}_1, \boldsymbol{w}_2, a_1, a_2)$ is along a KKT-margin direction of width-2 two-layer Leaky ReLU network and $\|\boldsymbol{\theta}\|_2 = 1$, $a_1 \geq 0$ and $a_2 \leq 0$, then one of the following three cases is true:*

1. $\boldsymbol{\theta} = \frac{1}{2}(\boldsymbol{w}^*, -\boldsymbol{w}^*, 1, -1)$;

2. $\boldsymbol{\theta} = \frac{1}{\sqrt{2}}(\boldsymbol{w}^*, \boldsymbol{0}, 1, 0)$;

3. $\boldsymbol{\theta} = \frac{1}{\sqrt{2}}(\boldsymbol{0}, -\boldsymbol{w}^*, 0, -1)$.

*Proof.* Suppose $a_1 > 0$ and $a_2 < 0$, then by Lemma F.2, we know $\boldsymbol{w}_1 = -\boldsymbol{w}_2$, $a_1 = -a_2 = \|\boldsymbol{w}_1\|_2$. Since $q_i(\boldsymbol{\theta}) > 0, \forall i$, we know $\langle \boldsymbol{w}_1, \boldsymbol{x}_i \rangle \neq 0, \forall i$, which implies $q_i(\boldsymbol{\theta})$ is differentiable at $\boldsymbol{\theta}$. Let $\boldsymbol{\theta}' = (\boldsymbol{w}_1, a_1)$ and $[\boldsymbol{\theta}'; -\boldsymbol{\theta}'] = (\boldsymbol{w}_1, -\boldsymbol{w}_1, a_1, -a_1)$, we know $\boldsymbol{\theta}'$ is along the KKT direction of the following optimization problem:

$$\min \quad f([\boldsymbol{\theta}'; -\boldsymbol{\theta}'])$$
$$\text{s.t.} \quad g_i([\boldsymbol{\theta}'; -\boldsymbol{\theta}']) \leq 0, \qquad \forall i \in [n],$$

where $f([\boldsymbol{\theta}'; -\boldsymbol{\theta}']) = \|[\boldsymbol{\theta}'; -\boldsymbol{\theta}']\|_2^2 = 2\|\boldsymbol{w}_1\|_2^2 + 2a_1^2$, and $g_i([\boldsymbol{\theta}'; -\boldsymbol{\theta}']) = 1 - q_i([\boldsymbol{\theta}'; -\boldsymbol{\theta}']) = y_i a_i(\phi(\langle \boldsymbol{w}_1, \boldsymbol{x}_i \rangle) - \phi(-\langle \boldsymbol{w}_1, \boldsymbol{x}_i \rangle)) = a_1(1 + \alpha_{\text{leaky}})\langle \boldsymbol{w}_1, y_i \boldsymbol{x}_i \rangle$. With a standard analysis, we know $\boldsymbol{w}_1$ be in the direction of the max-margin classifier of the original problem, $\boldsymbol{w}^*$.

Next we discuss the case where $a_2 = 0$ ($a_1 = 0$ follows the same analysis). When $a_2 = 0$, since $q_i(\boldsymbol{\theta}) > 0$ for all $i$, we know $a_1 y_i \langle \boldsymbol{x}_i, \boldsymbol{x}_1 \rangle > 0$ for all $i$. Thus $q_i(\boldsymbol{\theta}) > q_{i+\frac{n}{2}}(\boldsymbol{\theta}) = \alpha_{\text{leaky}} q_i(\boldsymbol{\theta})$, which means only the second half constraints might be active. This reduces the optimization problem to a standard linear-max-margin problem, and $\boldsymbol{w}_1$ will be aligned with $\boldsymbol{w}^*$. $\qquad\square$

*Proof for Theorem 5.5.* By Theorem B.19 and Theorem 3.1, we know $\lim_{t \to +\infty} \frac{\boldsymbol{\theta}(t)}{\|\boldsymbol{\theta}(t)\|_2}$ must be along a KKT-margin direction. By Lemma F.3, we know that there are only 3 KKT-margin directions:

$$\frac{1}{2}(\boldsymbol{w}^*, -\boldsymbol{w}^*, 1, -1), \quad \frac{1}{\sqrt{2}}(\boldsymbol{w}^*, \boldsymbol{0}, 1, 0), \quad \frac{1}{\sqrt{2}}(\boldsymbol{0}, -\boldsymbol{w}^*, 0, -1).$$

Thus it suffices to show $\lim_{t \to +\infty} \frac{\boldsymbol{\theta}(t)}{\|\boldsymbol{\theta}(t)\|_2} \neq \frac{1}{\sqrt{2}}(\boldsymbol{w}^*, \boldsymbol{0}, 1, 0)$. ($\lim_{t \to +\infty} \frac{\boldsymbol{\theta}(t)}{\|\boldsymbol{\theta}(t)\|_2} \neq \frac{1}{\sqrt{2}}(\boldsymbol{w}^*, \boldsymbol{0}, 1, 0)$ would hold for the same reason.)

For convenience, we define $i' := i + n/2$ if $1 \leq i \leq n/2$ and $i' := i - n/2$ if $n/2 < i \leq n$. By Assumption 4.1 we know that $\boldsymbol{x}_{i'} = -\boldsymbol{x}_i$ and $y_{i'} = -y_i$.

We first define the angle between $\boldsymbol{w}^*$ and $\boldsymbol{w}_1(t)$ as $\beta_1(t) := \arccos \frac{\langle \boldsymbol{w}^*, \boldsymbol{w}_1(t) \rangle}{\|\boldsymbol{w}_1(t)\|_2}$ and angle between $-\boldsymbol{w}^*$ and $\boldsymbol{w}_2(t)$ as $\beta_2(t) := \arccos \frac{\langle -\boldsymbol{w}^*, \boldsymbol{w}_2(t) \rangle}{\|\boldsymbol{w}_2(t)\|_2}$. Since $\langle \boldsymbol{w}^*, \boldsymbol{w}_1(0) \rangle > 0$ and $\langle -\boldsymbol{w}^*, \boldsymbol{w}_2(0) \rangle > 0$, by Lemma B.20 we know that $\beta_1(t), \beta_2(t) \in [0, \pi/2)$ for all $t \geq 0$.

We also define $\epsilon := \min_{i \in [n]} \left\{ \arcsin \frac{\langle y_i \boldsymbol{x}_i, \boldsymbol{w}^* \rangle}{\|\boldsymbol{x}_i\|_2} \right\}$, which can be understood as the angle between $\boldsymbol{x}_i$ and the decision boundary determined by the linear separator $\boldsymbol{w}^*$.

Below we will prove by contradiction. Suppose $\lim_{t \to +\infty} \frac{\boldsymbol{\theta}(t)}{\|\boldsymbol{\theta}(t)\|_2} = \frac{1}{\sqrt{2}}(\boldsymbol{w}^*, \boldsymbol{0}, 1, 0) =: \bar{\boldsymbol{\theta}}_\infty$ holds. Then $\beta_1(t) \to 0$ and $\frac{\|\boldsymbol{w}_2(t)\|_2}{\|\boldsymbol{w}_1(t)\|_2} \to 0$ as $t \to +\infty$. Thus there must exist $T_1 > 0$ such that $\beta_1(t) \leq \epsilon/2$.

Note that $f_{\bar{\boldsymbol{\theta}}_\infty}(\boldsymbol{x}_i) = \frac{1}{2}\phi(\langle \boldsymbol{x}_i, \boldsymbol{w}^* \rangle)$ for all $i \in [n]$. By symmetry, for $i \in [n/2]$ we have

$$q_i(\bar{\boldsymbol{\theta}}_\infty) - q_{i'}(\bar{\boldsymbol{\theta}}_\infty) = f_{\bar{\boldsymbol{\theta}}_\infty}(\boldsymbol{x}_i) - f_{\bar{\boldsymbol{\theta}}_\infty}(-\boldsymbol{x}_i) = \frac{1}{2}\langle \boldsymbol{x}_i, \boldsymbol{w}^* \rangle - \frac{\gamma^*}{2}\langle \boldsymbol{x}_i, \boldsymbol{w}^* \rangle = \frac{1 - \gamma^*}{2}\langle \boldsymbol{x}_i, \boldsymbol{w}^* \rangle > 0.$$

By Theorem B.19, we also know $\|\boldsymbol{\theta}(t)\|_2 \to \infty$, so $q_i(\boldsymbol{\theta}) - q_{i'}(\boldsymbol{\theta}) = \|\boldsymbol{\theta}(t)\|_2 (q_i(\bar{\boldsymbol{\theta}}_\infty) - q_{i'}(\bar{\boldsymbol{\theta}}_\infty)) \to +\infty$ for all $i \in [n/2]$. Let $g_i(\boldsymbol{\theta}) := -\ell'(q_i(\boldsymbol{\theta}))$. Then

$$\frac{g_i(\boldsymbol{\theta}(t))}{g_{i'}(\boldsymbol{\theta}(t))} \sim \frac{\exp(-g_i(\boldsymbol{\theta}(t)))}{\exp(-g_{i'}(\boldsymbol{\theta}(t)))} = e^{-(g_i(\boldsymbol{\theta}(t)) - g_{i'}(\boldsymbol{\theta}(t)))} \to 0.$$

Thus there must exist $T_2 > 0$ such that $\frac{g_i(\boldsymbol{\theta}(t))}{g_{i'}(\boldsymbol{\theta}(t))} \leq \max \left\{ \frac{\cos \epsilon - \alpha_{\text{leaky}}}{1 - \alpha_{\text{leaky}} \cos \epsilon}, 1 \right\}$ for all $i \in [n/2]$ and $t \geq T_2$.

We will use these to show that $\frac{\langle \boldsymbol{w}_2(t), -\boldsymbol{w}^* \rangle}{\langle \boldsymbol{w}_1(t), \boldsymbol{w}^* \rangle}$ is non-decreasing for $t \geq T := \max\{T_1, T_2\}$, which further implies $\frac{\|\boldsymbol{w}_2(t)\|_2}{\|\boldsymbol{w}_1(t)\|_2}$ is lower bounded by some constant. Thus it contradicts with the assumption of convergence.

By Corollary B.18, we know that $a_1(t) = \|\boldsymbol{w}_1(t)\|_2$ and $a_2(t) = -\|\boldsymbol{w}_2(t)\|_2$ for all $t \geq 0$. Then for all $i \in [n]$, we have

$$f_{\boldsymbol{\theta}}(\boldsymbol{x}_i) = a_1\phi(\boldsymbol{w}_1^\top \boldsymbol{x}_i) + a_2\phi(\boldsymbol{w}_2^\top \boldsymbol{x}_i) = \|\boldsymbol{w}_1\|_2\phi(\boldsymbol{w}_1^\top \boldsymbol{x}_i) - \|\boldsymbol{w}_2\|_2\phi(\boldsymbol{w}_2^\top \boldsymbol{x}_i).$$

By (10), if $\boldsymbol{w}_1(t), \boldsymbol{w}_2(t) \in \Omega_{\mathcal{S}}$ then we have

$$\frac{\mathrm{d}\boldsymbol{w}_1}{\mathrm{d}t} = \frac{\|\boldsymbol{w}_1\|_2}{n} \sum_{i\in[n]} g_i(\boldsymbol{\theta})\phi'(\boldsymbol{w}_1^\top \boldsymbol{x}_i)y_i\boldsymbol{x}_i, \qquad -\frac{\mathrm{d}\boldsymbol{w}_2}{\mathrm{d}t} = \frac{\|\boldsymbol{w}_2\|_2}{n} \sum_{i\in[n]} g_i(\boldsymbol{\theta})\phi'(\boldsymbol{w}_2^\top \boldsymbol{x}_i)y_i\boldsymbol{x}_i.$$

By symmetry, we can rewrite them as

$$\frac{\mathrm{d}\boldsymbol{w}_1(t)}{\mathrm{d}t} = \frac{\|\boldsymbol{w}_1(t)\|_2}{n} \sum_{i\in[n/2]} \sigma_i^{(1)}(t)\boldsymbol{x}_i, \qquad -\frac{\mathrm{d}\boldsymbol{w}_2(t)}{\mathrm{d}t} = \frac{\|\boldsymbol{w}_2(t)\|_2}{n} \sum_{i\in[n/2]} \sigma_i^{(2)}(t)\boldsymbol{x}_i. \qquad (39)$$

where $\sigma_i^{(k)}(t) := g_i(\boldsymbol{\theta}(t))\phi'(\boldsymbol{w}_k^\top(t)\boldsymbol{x}_i) + g_{i'}(\boldsymbol{\theta}(t))\phi'(-\boldsymbol{w}_k^\top(t)\boldsymbol{x}_i)$. Note that this only holds for $\boldsymbol{w}_k(t) \in \Omega_{\mathcal{S}}$. By taking limits through (8), we know that for a.e. $t \geq 0$, there exists $\sigma_i^{(k)}(t)$ such that (39) holds and

$$\sigma_i^{(k)}(t) \in \begin{cases} \{g_i(\boldsymbol{\theta}) + \alpha_{\text{leaky}}g_{i'}(\boldsymbol{\theta})\} & \text{if } \boldsymbol{w}_k^\top \boldsymbol{x}_i > 0; \\ \{\alpha_{\text{leaky}}g_i(\boldsymbol{\theta}) + g_{i'}(\boldsymbol{\theta})\} & \text{if } \boldsymbol{w}_k^\top \boldsymbol{x}_i < 0; \quad (40) \\ \{\lambda g_i(\boldsymbol{\theta}) + (1 + \alpha_{\text{leaky}} - \lambda)g_{i'}(\boldsymbol{\theta}) : \alpha_{\text{leaky}} \leq \lambda \leq 1\} & \text{if } \boldsymbol{w}_k^\top \boldsymbol{x}_i = 0. \end{cases}$$

By chain rule, for a.e. $t \geq 0$ we have:

$$\frac{\mathrm{d}}{\mathrm{d}t} \ln \frac{\langle \boldsymbol{w}_2, -\boldsymbol{w}^* \rangle}{\langle \boldsymbol{w}_1, \boldsymbol{w}^* \rangle} = \frac{\langle \frac{\mathrm{d}\boldsymbol{w}_2}{\mathrm{d}t}, -\boldsymbol{w}^* \rangle}{\langle \boldsymbol{w}_2, -\boldsymbol{w}^* \rangle} - \frac{\langle \frac{\mathrm{d}\boldsymbol{w}_1}{\mathrm{d}t}, \boldsymbol{w}^* \rangle}{\langle \boldsymbol{w}_1, \boldsymbol{w}^* \rangle}$$

$$= \frac{\|\boldsymbol{w}_2\|_2}{\langle \boldsymbol{w}_2, -\boldsymbol{w}^* \rangle} \cdot \frac{1}{n} \sum_{i\in[n/2]} \sigma_i^{(2)} \langle \boldsymbol{x}_i, \boldsymbol{w}^* \rangle - \frac{\|\boldsymbol{w}_1\|_2}{\langle \boldsymbol{w}_1, \boldsymbol{w}^* \rangle} \cdot \frac{1}{n} \sum_{i\in[n/2]} \sigma_i^{(1)} \langle \boldsymbol{x}_i, \boldsymbol{w}^* \rangle$$

$$= \frac{1}{n} \sum_{i\in[n/2]} \left( \frac{\sigma_i^{(2)}}{\cos\beta_2} - \frac{\sigma_i^{(1)}}{\cos\beta_1} \right) \langle \boldsymbol{x}_i, \boldsymbol{w}^* \rangle.$$

Now we are ready to prove $\frac{\mathrm{d}}{\mathrm{d}t} \ln \frac{\langle \boldsymbol{w}_2, -\boldsymbol{w}^* \rangle}{\langle \boldsymbol{w}_1, \boldsymbol{w}^* \rangle} \geq 0$ for $t \geq T$. For this, we only need to show that $\frac{\sigma_i^{(2)}}{\cos\beta_2} \geq \frac{\sigma_i^{(1)}}{\cos\beta_1}$ in two cases.

Case 1. When $\beta_1 < \beta_2$, it suffices to show $\sigma_i^{(1)} \leq \sigma_i^{(2)}$. By our choice of $T_1$, we have $\beta_1 \leq \epsilon/2$, which implies $\boldsymbol{w}_1^\top \boldsymbol{x}_i > 0$ and $\sigma_i^{(1)} = g_i(\boldsymbol{\theta}) + \alpha_{\text{leaky}}g_{i'}(\boldsymbol{\theta})$ for all $i \in [n/2]$. Note that $g_i(\boldsymbol{\theta})) \leq g_{i'}(\boldsymbol{\theta})$ according to our choice of $T_2$. Then for any $\lambda \in [\alpha_{\text{leaky}}, 1]$ we have

$$\sigma_i^{(1)} = g_i(\boldsymbol{\theta}) + \alpha_{\text{leaky}}g_{i'}(\boldsymbol{\theta}) \leq \lambda g_i(\boldsymbol{\theta}) + (1 + \alpha_{\text{leaky}} - \lambda)g_{i'}(\boldsymbol{\theta}).$$

By (40), we therefore have $\sigma_i^{(1)} \leq \sigma_i^{(2)}$.

Case 2. If $\beta_1 \geq \beta_2$, then by our choice of $T_1$ we have $\epsilon/2 \geq \beta_1 \geq \beta_2$. Then for all $i \in [n/2]$, $\boldsymbol{w}_2^\top \boldsymbol{x}_i \leq 0$. So we have

$$\frac{\sigma_i^{(1)}}{\sigma_i^{(2)}} = \frac{g_i(\boldsymbol{\theta}) + \alpha_{\text{leaky}}g_{i'}(\boldsymbol{\theta})}{\alpha_{\text{leaky}}g_i(\boldsymbol{\theta}) + g_{i'}(\boldsymbol{\theta})} = \frac{\frac{g_i(\boldsymbol{\theta})}{g_{i'}(\boldsymbol{\theta})} + \alpha_{\text{leaky}}}{\alpha_{\text{leaky}}\frac{g_i(\boldsymbol{\theta})}{g_{i'}(\boldsymbol{\theta})} + 1} \leq \cos\epsilon \leq \cos\beta_1(t) \leq \frac{\cos\beta_1(t)}{\cos\beta_2(t)}.$$

Thus $\frac{\sigma_i^{(2)}}{\cos\beta_2} \geq \frac{\sigma_i^{(1)}}{\cos\beta_1}$.

Now we have shown that $\frac{\langle \boldsymbol{w}_2(t), -\boldsymbol{w}^* \rangle}{\langle \boldsymbol{w}_1(t), \boldsymbol{w}^* \rangle} \geq \frac{\langle \boldsymbol{w}_2(T), -\boldsymbol{w}^* \rangle}{\langle \boldsymbol{w}_1(T), \boldsymbol{w}^* \rangle} =: r_0$, where $r_0$ is a constant (ratio at time $T$). So for $t \geq T$,

$$\frac{\|\boldsymbol{w}_2(t)\|_2}{\|\boldsymbol{w}_1(t)\|_2} = \frac{\langle \boldsymbol{w}_2(t), -\boldsymbol{w}^* \rangle \cos\beta_1(t)}{\langle \boldsymbol{w}_1(t), \boldsymbol{w}^* \rangle \cos\beta_2(t)} \geq \frac{\langle \boldsymbol{w}_2(t), -\boldsymbol{w}^* \rangle \cos\epsilon}{\langle \boldsymbol{w}_1(t), \boldsymbol{w}^* \rangle} \geq r_0 \cos\epsilon, \qquad (41)$$

is lower bounded, which contradicts with $\lim_{t\to+\infty} \frac{\boldsymbol{\theta}(t)}{\|\boldsymbol{\theta}(t)\|_2} = \frac{1}{\sqrt{2}}(\boldsymbol{w}^*, \boldsymbol{0}, 1, 0)$. $\qquad \square$

## F.2 Directional Convergence of $L$-homogeneous Neural Nets

In this section we consider general $L$-homogeneous neural nets with logistic loss following the settings introduced in Section 3.1. We define $\alpha(\boldsymbol{\theta})$ and $\tilde{\gamma}(\boldsymbol{\theta})$ to be smoothed margin and its normalized version following Lyu and Li (2020).

$$\alpha(\boldsymbol{\theta}) = \ell^{-1}(n\mathcal{L}(\boldsymbol{\theta})), \qquad \tilde{\gamma}(\boldsymbol{\theta}) = \frac{\alpha(\boldsymbol{\theta})}{\|\boldsymbol{\theta}\|_2^L}.$$

Define $\zeta(t) := \int_0^t \left\| \frac{\mathrm{d}}{\mathrm{d}\tau} \frac{\boldsymbol{\theta}(\tau)}{\|\boldsymbol{\theta}(\tau)\|_2} \right\|_2 \mathrm{d}\tau$ to be the length of the trajectory swept by $\boldsymbol{\theta}/\|\boldsymbol{\theta}\|_2$ from time $0$ to $t$. Define $\beta(t)$ to be the cosine of the angle between $\boldsymbol{\theta}(t)$ and $\frac{\mathrm{d}\boldsymbol{\theta}(t)}{\mathrm{d}t}$.

$$\beta(t) := \frac{\left\langle \frac{\mathrm{d}\boldsymbol{\theta}(t)}{\mathrm{d}t}, \boldsymbol{\theta}(t) \right\rangle}{\left\| \frac{\mathrm{d}\boldsymbol{\theta}(t)}{\mathrm{d}t} \right\|_2 \cdot \|\boldsymbol{\theta}(t)\|_2}, \qquad \text{for a.e. } t \geq 0.$$

### F.2.1 Lemmas from Previous Works

We leverage the following two lemmas from Ji and Telgarsky (2020a) on desingularizing function. Formally, we say that $\Psi : [0, \nu)$ is a desingularizing function if $\Psi$ is continuous on $[0, \nu)$ with $\Psi(0) = 0$ and continuously differentiable on $(0, \nu)$ with $\Psi' > 0$.

**Lemma F.4** (Lemma 3.6, Ji and Telgarsky 2020a). *Given a locally Lipschitz definable function $f$ with an open domain $D \subseteq \{\boldsymbol{\theta} : \|\boldsymbol{\theta}\|_2 > 1\}$, for any $c, \eta > 0$, there exists $\nu > 0$ and a definable desingularizing function $\Psi$ on $[0, \nu)$ such that*

$$\Psi'(f(\boldsymbol{\theta})) \cdot \|\boldsymbol{\theta}\|_2 \left\| \bar{\partial}^\circ f(\boldsymbol{\theta}) \right\|_2 \geq 1,$$

*whenever $f(\boldsymbol{\theta}) \in (0, \nu)$ and $\left\| \bar{\partial}_\perp^\circ f(\boldsymbol{\theta}) \right\|_2 \geq c\|\boldsymbol{\theta}\|_2^\eta \left\| \bar{\partial}_{\mathrm{r}}^\circ f(\boldsymbol{\theta}) \right\|_2$.*

**Lemma F.5** (Corollary of Lemma 3.7, Ji and Telgarsky 2020a). *Given a locally Lipschitz definable function $f$ with an open domain $D \subseteq \{\boldsymbol{\theta} : \|\boldsymbol{\theta}\|_2 > 1\}$, for any $\lambda > 0$, there exists $\nu > 0$ and a definable desingularizing function $\Psi$ on $[0, \nu)$ such that*

$$\Psi'(f(\boldsymbol{\theta})) \cdot \|\boldsymbol{\theta}\|_2^{1+\lambda} \left\| \bar{\partial}^\circ f(\boldsymbol{\theta}) \right\|_2 \geq 1,$$

*whenever $f(\boldsymbol{\theta}) \in (0, \nu)$.*

For $\tilde{\gamma}(\boldsymbol{\theta})$, we have the following decomposition lemma from Ji and Telgarsky (2020a).

**Lemma F.6** (Lemma 3.4, Ji and Telgarsky 2020a). *If $\mathcal{L}(\boldsymbol{\theta}(t)) < \ell(0)/n$ at time $t = t_0$, it holds for a.e. $t \geq t_0$ that*

$$\frac{\mathrm{d}\tilde{\gamma}(\boldsymbol{\theta}(t))}{\mathrm{d}t} = \left\| \bar{\partial}_{\mathrm{r}}^\circ \tilde{\gamma}(\boldsymbol{\theta}(t)) \right\|_2 \left\| \bar{\partial}_{\mathrm{r}}^\circ \mathcal{L}(\boldsymbol{\theta}(t)) \right\|_2 + \left\| \bar{\partial}_\perp^\circ \tilde{\gamma}(\boldsymbol{\theta}(t)) \right\|_2 \left\| \bar{\partial}_\perp^\circ \mathcal{L}(\boldsymbol{\theta}(t)) \right\|_2.$$

For $a \in \mathbb{R} \cup \{+\infty, -\infty\}$, we say that $v$ is an asymptotic Clarke critical value of a locally Lipschitz function $f : \mathbb{R}^D \to \mathbb{R}$ if there exists a sequence of $(\boldsymbol{\theta}_j, \boldsymbol{g}_j)$, where $\boldsymbol{\theta}_j \in \mathbb{R}^D$ and $\boldsymbol{g}_j \in \partial^\circ f(\boldsymbol{\theta}_j)$, such that $\lim_{j \to +\infty} f(\boldsymbol{\theta}_j) = v$ and $\lim_{j \to +\infty}(1 + \|\boldsymbol{\theta}_j\|_2)\|\boldsymbol{g}_j\|_2 = 0$.

**Lemma F.7** (Corollary of Lemma B.10, Ji and Telgarsky 2020a). *$\tilde{\gamma}(\boldsymbol{\theta})$ only has finitely many asymptotic Clarke critical values.*

For $\beta(\boldsymbol{\theta})$, we have the following lemma from Lyu and Li (2020).

**Lemma F.8** (Lemma C.12, Lyu and Li 2020). *If $\mathcal{L}(\boldsymbol{\theta}(t)) < \ell(0)/n$ at time $t = t_0$, then there exists a sequence $t_1, t_2, \ldots$ such that $t_j \to +\infty$ and $\beta(t_j) \to 1$ as $j \to +\infty$.*

### F.2.2 Characterizing Margin Maximization with Asymptotic Clarke Critical Value

Before proving Theorem 5.6, we first prove the following theorem that characterizes margin maximization using asymptotic Clarke critical value.

**Theorem F.9.** *For homogeneous nets, if $\mathcal{L}(\boldsymbol{\theta}(0)) < \ell(0)/n$, then $\frac{\boldsymbol{\theta}(t)}{\|\boldsymbol{\theta}(t)\|_2}$ converges to some direction $\bar{\boldsymbol{\theta}}$ and $\gamma(\bar{\boldsymbol{\theta}})$ is an asymptotic Clarke critical value of $\tilde{\gamma}$.*

*Proof.* Note that Theorem 3.1 already implies that $\frac{\boldsymbol{\theta}(t)}{\|\boldsymbol{\theta}(t)\|_2}$ converges to some direction $\bar{\boldsymbol{\theta}}$. We only need to show that $\gamma(\bar{\boldsymbol{\theta}})$ is an asymptotic Clarke critical value of $\tilde{\gamma}$.

By Lemma F.8 and definition of $\beta$, there exists a sequence of $(\boldsymbol{\theta}_j, \boldsymbol{h}_j)$, where $\boldsymbol{h}_j \in -\partial^\circ \mathcal{L}(\boldsymbol{\theta}_j)$, such that $\tilde{\gamma}(\boldsymbol{\theta}_j) \to \gamma(\bar{\boldsymbol{\theta}})$, $\|\boldsymbol{\theta}_j\|_2 \to +\infty$, $\frac{\boldsymbol{\theta}_j}{\|\boldsymbol{\theta}_j\|_2} \to \bar{\boldsymbol{\theta}}$, $\frac{\langle \boldsymbol{h}_j, \boldsymbol{\theta}_j \rangle}{\|\boldsymbol{h}_j\|_2 \|\boldsymbol{\theta}_j\|_2} \to 1$ as $j \to +\infty$. By chain rule, we know that

$$\partial^\circ \tilde{\gamma}(\boldsymbol{\theta}) = \frac{\partial^\circ \alpha(\boldsymbol{\theta})}{\|\boldsymbol{\theta}\|_2^L} + \frac{L\alpha(\boldsymbol{\theta})}{\|\boldsymbol{\theta}\|_2^{L+2}}\boldsymbol{\theta} = \frac{n(\ell^{-1})'(n\mathcal{L}(\boldsymbol{\theta}))\partial^\circ \mathcal{L}(\boldsymbol{\theta})}{\|\boldsymbol{\theta}\|_2^L} + \frac{L\alpha(\boldsymbol{\theta})}{\|\boldsymbol{\theta}\|_2^{L+2}}\boldsymbol{\theta}$$

$$= \frac{n\partial^\circ \mathcal{L}(\boldsymbol{\theta})}{\|\boldsymbol{\theta}\|_2^L \ell'(\alpha(\boldsymbol{\theta}))} + \frac{L\alpha(\boldsymbol{\theta})}{\|\boldsymbol{\theta}\|_2^{L+2}}\boldsymbol{\theta}.$$

This means $\boldsymbol{g}_j := \frac{L\alpha(\boldsymbol{\theta}_j)}{\|\boldsymbol{\theta}_j\|_2^{L+2}}\boldsymbol{\theta}_j - \frac{n\boldsymbol{h}_j}{\|\boldsymbol{\theta}_j\|_2^L \ell'(\alpha(\boldsymbol{\theta}_j))} \in \partial^\circ \tilde{\gamma}(\boldsymbol{\theta})$. By definition of asymptotic Clarke critical value, it suffices to show that $\|\boldsymbol{\theta}_j\|_2 \cdot \|\boldsymbol{g}_j\|_2 \to 0$ as $j \to +\infty$.

By Lemma C.5 in (Ji and Telgarsky, 2020a), $\left| \frac{n\langle -\boldsymbol{h}_j, \boldsymbol{\theta}_j \rangle}{L \cdot \ell'(\alpha(\boldsymbol{\theta}_j))} - \alpha(\boldsymbol{\theta}) \right| \le 2\ln n + 1$. So

$$\lim_{j \to +\infty} \left| \frac{n\langle -\boldsymbol{h}_j, \boldsymbol{\theta}_j \rangle}{L \cdot \|\boldsymbol{\theta}_j\|_2^L \ell'(\alpha(\boldsymbol{\theta}_j))} - \tilde{\gamma}(\boldsymbol{\theta}_j) \right| = \lim_{j \to +\infty} \frac{1}{\|\boldsymbol{\theta}_j\|_2^L} \left| \frac{n\langle -\boldsymbol{h}_j, \boldsymbol{\theta}_j \rangle}{L \cdot \ell'(\alpha(\boldsymbol{\theta}_j))} - \alpha(\boldsymbol{\theta}_j) \right| = 0,$$

which implies that $\lim_{j \to +\infty} \frac{n\langle -\boldsymbol{h}_j, \boldsymbol{\theta}_j \rangle}{L \cdot \|\boldsymbol{\theta}_j\|_2^L \ell'(\alpha(\boldsymbol{\theta}_j))} = \lim_{j \to +\infty} \tilde{\gamma}(\boldsymbol{\theta}_j) = \gamma(\bar{\boldsymbol{\theta}})$. Now for the radial component of $\boldsymbol{g}_j$ we have

$$\|\boldsymbol{\theta}_j\|_2 \cdot \left| \left\langle \boldsymbol{g}_j, \frac{\boldsymbol{\theta}_j}{\|\boldsymbol{\theta}_j\|_2} \right\rangle \right| = \frac{n\langle \boldsymbol{h}_j, \boldsymbol{\theta}_j \rangle}{\|\boldsymbol{\theta}_j\|_2^L \ell'(\alpha(\boldsymbol{\theta}_j))} + \frac{L\alpha(\boldsymbol{\theta}_j)}{\|\boldsymbol{\theta}_j\|_2^L} \to -L\gamma(\bar{\boldsymbol{\theta}}) + L\gamma(\bar{\boldsymbol{\theta}}) = 0.$$

And for the tangential component we have

$$\|\boldsymbol{\theta}_j\|_2 \cdot \left\| \left( \boldsymbol{I} - \frac{\boldsymbol{\theta}_j \boldsymbol{\theta}_j^\top}{\|\boldsymbol{\theta}_j\|_2^2} \right) \boldsymbol{g}_j \right\|_2 = \frac{n}{\|\boldsymbol{\theta}_j\|_2^{L-1} \ell'(\alpha(\boldsymbol{\theta}_j))} \left\| \left( \boldsymbol{I} - \frac{\boldsymbol{\theta}_j \boldsymbol{\theta}_j^\top}{\|\boldsymbol{\theta}_j\|_2^2} \right) \boldsymbol{h}_j \right\|_2$$

$$= \frac{n\|\boldsymbol{h}_j\|_2}{\|\boldsymbol{\theta}_j\|_2^{L-1} \ell'(\alpha(\boldsymbol{\theta}_j))} \sqrt{1 - \frac{\langle \boldsymbol{\theta}_j, \boldsymbol{h}_j \rangle^2}{\|\boldsymbol{\theta}_j\|_2^2 \|\boldsymbol{h}_j\|_2^2}}$$

$$= \frac{n\langle -\boldsymbol{h}_j, \boldsymbol{\theta}_j \rangle}{\|\boldsymbol{\theta}_j\|_2^L \ell'(\alpha(\boldsymbol{\theta}_j))} \frac{\|\boldsymbol{h}_j\|_2 \|\boldsymbol{\theta}_j\|_2}{\langle -\boldsymbol{h}_j, \boldsymbol{\theta}_j \rangle} \sqrt{1 - \frac{\langle \boldsymbol{\theta}_j, \boldsymbol{h}_j \rangle^2}{\|\boldsymbol{\theta}_j\|_2^2 \|\boldsymbol{h}_j\|_2^2}}$$

$$\to L\gamma(\bar{\boldsymbol{\theta}}) \cdot 1 \cdot 0 = 0.$$

Combining these proves that $\|\boldsymbol{\theta}_j\|_2 \cdot \|\boldsymbol{g}_j\|_2 \to 0$. $\qquad\qquad\square$

### F.2.3 Proof for Theorem 5.6

Given Lemmas F.4 and F.5 from Ji and Telgarsky (2020a), we have the following inequality around any direction.

**Lemma F.10.** *Given any parameter direction $\bar{\boldsymbol{\theta}}^* \in \mathbb{S}^{D-1}$, for any $\kappa \in (L/2, L)$, there exists $\nu > 0$ and a definable desingularizing function $\Psi$ on $[0, \nu)$ such that the following holds.*

1. *For any $\boldsymbol{\theta}$, if $\gamma(\bar{\boldsymbol{\theta}}^*) - \tilde{\gamma}(\boldsymbol{\theta}) \in (0, \nu)$ and*

$$\left\| \bar{\partial}_\perp^\circ \tilde{\gamma}(\boldsymbol{\theta}) \right\|_2 \ge \frac{\gamma(\bar{\boldsymbol{\theta}}^*)}{4\ln n + 2} \|\boldsymbol{\theta}\|_2^{L-\kappa} \left\| \bar{\partial}_r^\circ \tilde{\gamma}(\boldsymbol{\theta}) \right\|_2, \tag{42}$$

   *then*

$$\Psi'(\gamma(\bar{\boldsymbol{\theta}}^*) - \tilde{\gamma}(\boldsymbol{\theta})) \cdot \|\boldsymbol{\theta}\|_2 \left\| \bar{\partial}^\circ \tilde{\gamma}(\boldsymbol{\theta}) \right\|_2 \ge 1. \tag{43}$$

2. *For any $\boldsymbol{\theta}$, if $\gamma(\bar{\boldsymbol{\theta}}^*) - \tilde{\gamma}(\boldsymbol{\theta}) \in (0, \nu)$,*

$$\Psi'(\gamma(\bar{\boldsymbol{\theta}}^*) - \tilde{\gamma}(\boldsymbol{\theta})) \cdot \|\boldsymbol{\theta}\|_2^{2\kappa-L+1} \left\| \bar{\partial}^\circ \tilde{\gamma}(\boldsymbol{\theta}) \right\|_2 \ge 1. \tag{44}$$

*Proof.* Applying Lemma F.4 with $f(\boldsymbol{\theta}) = \gamma(\bar{\boldsymbol{\theta}}^*) - \tilde{\gamma}(\boldsymbol{\theta}), c = \frac{\gamma(\bar{\boldsymbol{\theta}}^*)}{4\ln n + 2}, \eta = L - \kappa$, we know that there exists $\nu_1 > 0$ and a definable desingularizing function $\Psi_1$ on $[0, \nu_1)$ such that Item 1 holds for $\Psi_1$, i.e.,

$$\Psi_1'(\gamma^* - \tilde{\gamma}(\boldsymbol{\theta})) \cdot \|\boldsymbol{\theta}\|_2 \left\|\bar{\partial}^\circ \tilde{\gamma}(\boldsymbol{\theta})\right\|_2 \geq 1,$$

whenever (42) holds.

Applying Lemma F.5 with $f(\boldsymbol{\theta}) = \gamma(\bar{\boldsymbol{\theta}}^*) - \tilde{\gamma}(\boldsymbol{\theta}), \lambda = 2\kappa - L$, we know that there exists $\nu_2 > 0$ and a definable desingularizing function $\Psi_2$ on $[0, \nu_2)$ such that Item 2 holds for $\Psi_2$, i.e.,

$$\Psi_2'(\gamma(\bar{\boldsymbol{\theta}}^*) - \tilde{\gamma}(\boldsymbol{\theta})) \cdot \|\boldsymbol{\theta}\|_2^{2\kappa - L + 1} \left\|\bar{\partial}^\circ \tilde{\gamma}(\boldsymbol{\theta})\right\|_2 \geq 1.$$

Since $\Psi_1'(x) - \Psi_2'(x)$ is definable, there exists a sufficiently small constant $\nu > 0$ such that either $\Psi_1'(x) - \Psi_2'(x) \geq 0$ holds for all $x \in [0, \nu)$, or $\Psi_1'(x) - \Psi_2'(x) \leq 0$ holds for all $x \in [0, \nu)$. This means either $\Psi_1'(x) \geq \Psi_2'(x)$ for all $x \in [0, \nu)$ or $\Psi_2'(x) \geq \Psi_1'(x)$ for all $x \in [0, \nu)$. Let $\Psi(x) = \Psi_1(x)$ in the former case and $\Psi(x) = \Psi_2(x)$ in the latter case. Then $\Psi'(x) \geq \Psi_1'(x)$ and $\Psi'(x) \geq \Psi_2'(x)$, and thus both Items 1 and 2 hold. $\square$

Now we prove the following lemma, which will directly lead to Theorem 5.6. The core idea of the proof is essentially the same as that for Lemma 3.3 in Ji and Telgarsky (2020a). The key difference here is that the desingularizing function $\Psi$ in their lemma has dependence on the initial point, while our lemma does not have such dependence.

**Lemma F.11.** *Consider any $L$-homogeneous neural networks with definable output $f_{\boldsymbol{\theta}}(\boldsymbol{x}_i)$ and logistic loss. Given a local-max-margin direction $\bar{\boldsymbol{\theta}}^* \in \mathbb{S}^{D-1}$, there is a desingularizing function on $[0, \nu)$ and two constants $\epsilon_0 > 0, \rho_0 \geq 1$ such that for any $\boldsymbol{\theta}_0$ with norm $\|\boldsymbol{\theta}_0\|_2 \geq \rho_0$ and direction $\left\|\frac{\boldsymbol{\theta}_0}{\|\boldsymbol{\theta}_0\|_2} - \bar{\boldsymbol{\theta}}^*\right\|_2 \leq \epsilon_0$, the gradient flow $\boldsymbol{\theta}(t)$ starting with $\boldsymbol{\theta}_0$ satisfies*

$$\frac{\mathrm{d}\zeta(t)}{\mathrm{d}t} \leq -c\frac{\mathrm{d}\Psi(\gamma(\bar{\boldsymbol{\theta}}^*) - \tilde{\gamma}(\boldsymbol{\theta}(t)))}{\mathrm{d}t}, \qquad \text{for a.e. } t \in [0, T),$$

*where $T := \inf\{t \geq 0 : \tilde{\gamma}(\boldsymbol{\theta}(t)) \geq \gamma(\bar{\boldsymbol{\theta}}^*)\} \in \mathbb{R} \cup \{+\infty\}$.*

*Proof.* Fix an arbitrary $\kappa \in (L/2, L)$. Let $\Psi$ be the desingularizing function on $[0, \nu)$ obtained from Lemma F.10. WLOG, we can make $\nu < \gamma(\bar{\boldsymbol{\theta}}^*)/2$.

Let $\tilde{\gamma}_{\mathrm{inf}}(\rho, \epsilon)$ be the following lower bound for the initial smoothed margin $\tilde{\gamma}(\boldsymbol{\theta}_0)$:

$$\tilde{\gamma}_{\mathrm{inf}}(\rho, \epsilon) := \inf\left\{\tilde{\gamma}(\boldsymbol{\theta}) : \|\boldsymbol{\theta}\|_2 \geq \rho, \left\|\frac{\boldsymbol{\theta}}{\|\boldsymbol{\theta}\|_2} - \bar{\boldsymbol{\theta}}^*\right\|_2 \leq \epsilon\right\}. \tag{45}$$

We set $\rho_0$ to be sufficiently large and $\epsilon_0$ to be sufficiently small so that $\tilde{\gamma}_{\mathrm{inf}}(\rho_0, \epsilon_0) > \frac{1}{2}\gamma(\bar{\boldsymbol{\theta}}^*)$ and $\rho_0^{L-\kappa} \geq (4\ln n + 2)/\gamma(\bar{\boldsymbol{\theta}}^*)$. By chain rule, it suffices to prove

$$\frac{\mathrm{d}\tilde{\gamma}(\boldsymbol{\theta}(t))}{\mathrm{d}t} \geq \frac{1}{c\Psi'(\gamma(\bar{\boldsymbol{\theta}}^*) - \tilde{\gamma}(\boldsymbol{\theta}(t)))}\frac{\mathrm{d}\zeta(t)}{\mathrm{d}t}, \qquad \text{for a.e. } t \in [0, T), \tag{46}$$

where $c = \max\left\{2, \frac{\gamma(\bar{\boldsymbol{\theta}}^*)}{2\ln n + 1}\right\}$.

We consider two cases, where assume (42) is true in Case 1 and (42) is not true in Case 2. According to our choice of $\rho_0$ and the monotonicity of $\|\boldsymbol{\theta}(t)\|_2$, we have $\|\boldsymbol{\theta}(t)\|_2^{L-\kappa} \geq \rho_0^{L-\kappa} \geq \frac{4\ln n + 2}{\gamma(\bar{\boldsymbol{\theta}}^*)}$, and thus $\frac{\gamma(\bar{\boldsymbol{\theta}}^*)}{4\ln n + 2}\|\boldsymbol{\theta}(t)\|_2^{L-\kappa} \geq 1$. This means

$$\left\|\bar{\partial}_{\mathrm{r}}^\circ \tilde{\gamma}(\boldsymbol{\theta}(t))\right\|_2 + \left\|\bar{\partial}_{\perp}^\circ \tilde{\gamma}(\boldsymbol{\theta}(t))\right\|_2 \leq 2\left\|\bar{\partial}_{\mathrm{r}}^\circ \tilde{\gamma}(\boldsymbol{\theta}(t))\right\|_2 \tag{47}$$

in Case 1, and

$$\left\|\bar{\partial}_{\mathrm{r}}^\circ \tilde{\gamma}(\boldsymbol{\theta}(t))\right\|_2 + \left\|\bar{\partial}_{\perp}^\circ \tilde{\gamma}(\boldsymbol{\theta}(t))\right\|_2 < \frac{\gamma(\bar{\boldsymbol{\theta}}^*)}{2\ln n + 1}\|\boldsymbol{\theta}(t)\|_2^{L-\kappa}\left\|\bar{\partial}_{\mathrm{r}}^\circ \tilde{\gamma}(\boldsymbol{\theta}(t))\right\|_2 \tag{48}$$

in Case 2.

**Case 1.** For any $t \geq 0$, if $\frac{\mathrm{d}\boldsymbol{\theta}(t)}{\mathrm{d}t} = -\bar{\partial}^{\circ}\mathcal{L}(\boldsymbol{\theta}(t))$ and (42) hold for $\boldsymbol{\theta}(t)$. By Lemma F.6, we have the following lower bound for $\frac{\mathrm{d}\tilde{\gamma}(\boldsymbol{\theta}(t))}{\mathrm{d}t}$.

$$\frac{\mathrm{d}\tilde{\gamma}(\boldsymbol{\theta}(t))}{\mathrm{d}t} \geq \left\|\bar{\partial}^{\circ}_{\perp}\tilde{\gamma}(\boldsymbol{\theta}(t))\right\|_2 \left\|\bar{\partial}^{\circ}_{\perp}\mathcal{L}(\boldsymbol{\theta}(t))\right\|_2. \tag{49}$$

By triangle inequality and (47),

$$\|\bar{\partial}^{\circ}\tilde{\gamma}(\boldsymbol{\theta}(t))\|_2 \leq \|\bar{\partial}^{\circ}_{\mathrm{r}}\tilde{\gamma}(\boldsymbol{\theta}(t))\|_2 + \|\bar{\partial}^{\circ}_{\perp}\tilde{\gamma}(\boldsymbol{\theta}(t))\|_2 \leq 2\left\|\bar{\partial}^{\circ}_{\perp}\tilde{\gamma}(\boldsymbol{\theta}(t))\right\|_2.$$

So $\left\|\bar{\partial}^{\circ}_{\perp}\tilde{\gamma}(\boldsymbol{\theta}(t))\right\|_2 \geq \frac{1}{2}\|\bar{\partial}^{\circ}\tilde{\gamma}(\boldsymbol{\theta}(t))\|_2$. Combining this with (49) and noting that $\frac{\mathrm{d}\zeta(t)}{\mathrm{d}t} = \frac{1}{\|\boldsymbol{\theta}(t)\|_2}\left\|\bar{\partial}^{\circ}_{\perp}\mathcal{L}(\boldsymbol{\theta}(t))\right\|_2$, we have

$$\frac{\mathrm{d}\tilde{\gamma}(\boldsymbol{\theta}(t))}{\mathrm{d}t} \geq \frac{1}{2}\|\bar{\partial}^{\circ}\tilde{\gamma}(\boldsymbol{\theta}(t))\|_2 \cdot \left(\|\boldsymbol{\theta}(t)\|_2 \cdot \frac{\mathrm{d}\zeta(t)}{\mathrm{d}t}\right).$$

Applying (43) gives

$$\frac{\mathrm{d}\tilde{\gamma}(\boldsymbol{\theta}(t))}{\mathrm{d}t} \geq \frac{1}{2\Psi'(\gamma(\bar{\boldsymbol{\theta}}^*) - \tilde{\gamma}(\boldsymbol{\theta}(t)))}\frac{\mathrm{d}\zeta(t)}{\mathrm{d}t} \geq \frac{1}{c\Psi'(\gamma(\bar{\boldsymbol{\theta}}^*) - \tilde{\gamma}(\boldsymbol{\theta}(t)))}\frac{\mathrm{d}\zeta(t)}{\mathrm{d}t}.$$

**Case 2.** For any $t \geq 0$, if $\frac{\mathrm{d}\boldsymbol{\theta}(t)}{\mathrm{d}t} = -\bar{\partial}^{\circ}\mathcal{L}(\boldsymbol{\theta}(t))$ and (42) does not hold for $\boldsymbol{\theta}(t)$, i.e.,

$$\left\|\bar{\partial}^{\circ}_{\perp}\tilde{\gamma}(\boldsymbol{\theta}(t))\right\|_2 < \frac{\gamma(\bar{\boldsymbol{\theta}}^*)}{4\ln n + 2}\|\boldsymbol{\theta}(t)\|_2^{L-\kappa}\left\|\bar{\partial}^{\circ}_{\mathrm{r}}\tilde{\gamma}(\boldsymbol{\theta}(t))\right\|_2, \tag{50}$$

By Lemma F.6, we have the following lower bound for $\frac{\mathrm{d}\tilde{\gamma}(\boldsymbol{\theta}(t))}{\mathrm{d}t}$.

$$\frac{\mathrm{d}\tilde{\gamma}(\boldsymbol{\theta}(t))}{\mathrm{d}t} \geq \left\|\bar{\partial}^{\circ}_{\mathrm{r}}\tilde{\gamma}(\boldsymbol{\theta}(t))\right\|_2 \left\|\bar{\partial}^{\circ}_{\mathrm{r}}\mathcal{L}(\boldsymbol{\theta}(t))\right\|_2. \tag{51}$$

We lower bound $\left\|\bar{\partial}^{\circ}_{\mathrm{r}}\tilde{\gamma}(\boldsymbol{\theta}(t))\right\|_2$ and $\left\|\bar{\partial}^{\circ}_{\mathrm{r}}\mathcal{L}(\boldsymbol{\theta}(t))\right\|_2$ respectively in order to apply KL inequality (44).

**Bounding $\left\|\bar{\partial}^{\circ}_{\mathrm{r}}\tilde{\gamma}(\boldsymbol{\theta}(t))\right\|_2$ in Case 2.** By triangle inequality and (48),

$$\|\bar{\partial}^{\circ}\tilde{\gamma}(\boldsymbol{\theta}(t))\|_2 \leq \|\bar{\partial}^{\circ}_{\mathrm{r}}\tilde{\gamma}(\boldsymbol{\theta}(t))\|_2 + \|\bar{\partial}^{\circ}_{\perp}\tilde{\gamma}(\boldsymbol{\theta}(t))\|_2$$
$$< \frac{\gamma(\bar{\boldsymbol{\theta}}^*)}{2\ln n + 1}\|\boldsymbol{\theta}(t)\|_2^{L-\kappa}\left\|\bar{\partial}^{\circ}_{\mathrm{r}}\tilde{\gamma}(\boldsymbol{\theta}(t))\right\|_2,$$

which can be restated as

$$\left\|\bar{\partial}^{\circ}_{\mathrm{r}}\tilde{\gamma}(\boldsymbol{\theta}(t))\right\|_2 \geq \frac{2\ln n + 1}{\gamma(\bar{\boldsymbol{\theta}}^*)}\|\boldsymbol{\theta}(t)\|_2^{\kappa-L}\left\|\bar{\partial}^{\circ}\tilde{\gamma}(\boldsymbol{\theta}(t))\right\|_2. \tag{52}$$

**Bounding $\left\|\bar{\partial}^{\circ}_{\mathrm{r}}\mathcal{L}(\boldsymbol{\theta}(t))\right\|_2$ in Case 2.** By Lemma C.3 in Ji and Telgarsky (2020a),

$$\left\|\bar{\partial}^{\circ}_{\mathrm{r}}\tilde{\gamma}(\boldsymbol{\theta}(t))\right\|_2 \leq \frac{L \cdot (2\ln n + 1)}{\|\boldsymbol{\theta}(t)\|_2^{L+1}}.$$

Combining this with (50),

$$\left\|\bar{\partial}^{\circ}_{\perp}\tilde{\gamma}(\boldsymbol{\theta}(t))\right\|_2 < \frac{\gamma(\bar{\boldsymbol{\theta}}^*)}{4\ln n + 2}\|\boldsymbol{\theta}(t)\|_2^{L-\kappa} \cdot \frac{L \cdot (2\ln n + 1)}{\|\boldsymbol{\theta}(t)\|_2^{L+1}} = \frac{\gamma(\bar{\boldsymbol{\theta}}^*)}{2}L\|\boldsymbol{\theta}(t)\|_2^{-(1+\kappa)},$$

which can be rewritten as

$$\frac{L\gamma(\bar{\boldsymbol{\theta}}^*)}{2} > \left\|\bar{\partial}^{\circ}_{\perp}\tilde{\gamma}(\boldsymbol{\theta}(t))\right\|_2 \|\boldsymbol{\theta}(t)\|_2^{1+\kappa}. \tag{53}$$

By the chain rule and Lemma C.5 in Ji and Telgarsky (2020a),

$$\left\|\bar{\partial}^{\circ}_{\mathrm{r}}\alpha(\boldsymbol{\theta}(t))\right\|_2 = \frac{L \cdot \langle\boldsymbol{\theta}(t), \bar{\partial}^{\circ}\alpha(\boldsymbol{\theta}(t))\rangle}{\|\boldsymbol{\theta}(t)\|_2} \geq \frac{L\alpha(\boldsymbol{\theta}(t))}{\|\boldsymbol{\theta}(t)\|_2} = L\tilde{\gamma}(\boldsymbol{\theta}(t))\|\boldsymbol{\theta}(t)\|_2^{L-1}. \tag{54}$$

By the monotonicity of $\tilde{\gamma}(\boldsymbol{\theta}(t))$ during training, $\tilde{\gamma}(\boldsymbol{\theta}(t)) \geq \tilde{\gamma}(\boldsymbol{\theta}(0))$. Also note that

$$\tilde{\gamma}(\boldsymbol{\theta}(0)) \geq \tilde{\gamma}_0 > \frac{1}{2}\gamma(\bar{\boldsymbol{\theta}}^*),$$

where the first inequality is by definition of $\tilde{\gamma}_0$, and the second inequality is due to our choice of $\rho_0, \epsilon_0$. So we can replace $\tilde{\gamma}(\boldsymbol{\theta}(t))$ with $\frac{1}{2}\gamma(\bar{\boldsymbol{\theta}}^*)$ in the RHS of (54) and obtain

$$\left\|\bar{\partial}_{\mathrm{r}}^{\circ}\alpha(\boldsymbol{\theta}(t))\right\|_2 \geq \frac{L\gamma(\bar{\boldsymbol{\theta}}^*)}{2}\|\boldsymbol{\theta}(t)\|_2^{L-1}.$$

Combining this with (53) and noting that $\bar{\partial}_{\perp}^{\circ}\alpha(\boldsymbol{\theta}(t)) = \|\boldsymbol{\theta}(t)\|_2^L \, \bar{\partial}_{\perp}^{\circ}\tilde{\gamma}(\boldsymbol{\theta}(t))$, we have

$$\left\|\bar{\partial}_{\mathrm{r}}^{\circ}\alpha(\boldsymbol{\theta}(t))\right\|_2 \geq \|\boldsymbol{\theta}(t)\|_2^{L+\kappa} \left\|\bar{\partial}_{\perp}^{\circ}\tilde{\gamma}(\boldsymbol{\theta}(t))\right\|_2 \geq \|\boldsymbol{\theta}(t)\|_2^{\kappa} \left\|\bar{\partial}_{\perp}^{\circ}\alpha(\boldsymbol{\theta}(t))\right\|_2. \tag{55}$$

Recall that $\alpha(\boldsymbol{\theta}) = \ell^{-1}(\mathcal{L}(\boldsymbol{\theta}))$. By chain rule, $\bar{\partial}^{\circ}\alpha(\boldsymbol{\theta})$ is equal to the subgradient $\bar{\partial}^{\circ}\mathcal{L}(\boldsymbol{\theta})$ rescaled by some factor. Thus (55) implies

$$\left\|\bar{\partial}_{\mathrm{r}}^{\circ}\mathcal{L}(\boldsymbol{\theta}(t))\right\|_2 \geq \|\boldsymbol{\theta}(t)\|_2^{\kappa} \left\|\bar{\partial}_{\perp}^{\circ}\mathcal{L}(\boldsymbol{\theta}(t))\right\|_2. \tag{56}$$

**Applying (44) for Case 2.**   Putting (51), (52) and (56) together gives

$$\begin{aligned}
\frac{\mathrm{d}\tilde{\gamma}(\boldsymbol{\theta}(t))}{\mathrm{d}t} &\geq \left(\frac{2\ln n + 1}{\gamma(\bar{\boldsymbol{\theta}}^*)}\|\boldsymbol{\theta}(t)\|_2^{\kappa-L} \left\|\bar{\partial}^{\circ}\tilde{\gamma}(\boldsymbol{\theta}(t))\right\|_2\right) \cdot \left(\|\boldsymbol{\theta}(t)\|_2^{\kappa} \left\|\bar{\partial}_{\perp}^{\circ}\mathcal{L}(\boldsymbol{\theta}(t))\right\|_2\right) \\
&\geq \frac{2\ln n + 1}{\gamma(\bar{\boldsymbol{\theta}}^*)}\|\boldsymbol{\theta}(t)\|_2^{2\kappa-L} \left\|\bar{\partial}^{\circ}\tilde{\gamma}(\boldsymbol{\theta}(t))\right\|_2 \cdot \left\|\bar{\partial}_{\perp}^{\circ}\mathcal{L}(\boldsymbol{\theta}(t))\right\|_2 \\
&= \frac{2\ln n + 1}{\gamma(\bar{\boldsymbol{\theta}}^*)}\|\boldsymbol{\theta}(t)\|_2^{2\kappa-L+1} \left\|\bar{\partial}^{\circ}\tilde{\gamma}(\boldsymbol{\theta}(t))\right\|_2 \cdot \frac{\mathrm{d}\zeta(t)}{\mathrm{d}t},
\end{aligned}$$

where the last equality is due to $\frac{\mathrm{d}\zeta(t)}{\mathrm{d}t} = \frac{1}{\|\boldsymbol{\theta}(t)\|_2} \left\|\bar{\partial}_{\perp}^{\circ}\mathcal{L}(\boldsymbol{\theta}(t))\right\|_2$. Applying (44) gives

$$\frac{\mathrm{d}\tilde{\gamma}(\boldsymbol{\theta}(t))}{\mathrm{d}t} \geq \frac{2\ln n + 1}{\gamma(\bar{\boldsymbol{\theta}}^*)} \cdot \frac{1}{\Psi'(\gamma(\bar{\boldsymbol{\theta}}^*) - \tilde{\gamma}(\boldsymbol{\theta}(t)))} \frac{\mathrm{d}\zeta(t)}{\mathrm{d}t} \geq \frac{1}{c\Psi'(\gamma(\bar{\boldsymbol{\theta}}^*) - \tilde{\gamma}(\boldsymbol{\theta}(t)))} \frac{\mathrm{d}\zeta(t)}{\mathrm{d}t}.$$

**Final Proof Step.**   For a.e. $t \geq 0$, $\boldsymbol{\theta}(t)$ lies in either Case 1 or Case 2, so (46) holds, and we can rewrite it as

$$c\Psi'(\gamma(\bar{\boldsymbol{\theta}}^*) - \tilde{\gamma}(\boldsymbol{\theta}(t)))\frac{\mathrm{d}\tilde{\gamma}(\boldsymbol{\theta}(t))}{\mathrm{d}t} \geq \frac{\mathrm{d}\zeta(t)}{\mathrm{d}t}, \qquad \text{for a.e. } t \in [0, T).$$

By chain rule, the LHS is equal to $\frac{\mathrm{d}}{\mathrm{d}t}\left(c\Psi(\gamma(\bar{\boldsymbol{\theta}}^*) - \tilde{\gamma}(\boldsymbol{\theta}(t)))\right)$, which completes the proof.   $\square$

*Proof for Theorem 5.6.*   By Lemma F.11, we can choose $\epsilon_0, \rho_0$ such that

$$\frac{\mathrm{d}\zeta(t)}{\mathrm{d}t} \leq -c\frac{\mathrm{d}\Psi(\gamma(\bar{\boldsymbol{\theta}}^*) - \tilde{\gamma}(\boldsymbol{\theta}(t)))}{\mathrm{d}t}, \qquad \text{for a.e. } t \in [0, T),$$

where $T := \inf\{t \geq 0 : \tilde{\gamma}(\boldsymbol{\theta}(t)) \geq \gamma(\bar{\boldsymbol{\theta}}^*)\} \in \mathbb{R} \cup \{+\infty\}$. Then for all $t \in (0, T)$,

$$\zeta(t) \leq c\Psi(\gamma(\bar{\boldsymbol{\theta}}^*) - \tilde{\gamma}(\boldsymbol{\theta}_0)) \leq \delta(\epsilon_0, \rho_0) := c\Psi(\gamma(\bar{\boldsymbol{\theta}}^*) - \tilde{\gamma}_{\inf}(\rho_0, \epsilon_0)), \tag{57}$$

where $\tilde{\gamma}_{\inf}$ is defined in (45). We can choose $\epsilon_0$ small enough and $\rho_0$ large enough so that $\delta(\epsilon_0, \rho_0) > 0$ is as small as we want.

If $T = +\infty$, then (57) implies that $\frac{\boldsymbol{\theta}(t)}{\|\boldsymbol{\theta}(t)\|_2}$ converges to some $\bar{\boldsymbol{\theta}}$ as $t \to +\infty$, and $\|\bar{\boldsymbol{\theta}} - \bar{\boldsymbol{\theta}}^*\|_2 \leq \delta$ if $\delta(\epsilon_0, \rho_0) \leq \delta$.

If $T$ is finite, then by triangle inequality we have $\left\|\frac{\boldsymbol{\theta}(T)}{\|\boldsymbol{\theta}(T)\|_2} - \bar{\boldsymbol{\theta}}^*\right\|_2 \leq \epsilon_0 + \delta(\epsilon_0, \rho_0)$. Since $\bar{\boldsymbol{\theta}}^*$ is a local-max-margin direction, when $\epsilon_0$ and $\delta(\epsilon_0, \rho_0)$ are sufficiently small, $\tilde{\gamma}(\boldsymbol{\theta}) \leq \gamma(\boldsymbol{\theta}) \leq \gamma(\bar{\boldsymbol{\theta}}^*)$ holds for any $\boldsymbol{\theta}$ satisfying $\left\|\frac{\boldsymbol{\theta}}{\|\boldsymbol{\theta}\|_2} - \bar{\boldsymbol{\theta}}^*\right\|_2 \leq 2(\epsilon_0 + \delta(\epsilon_0, \rho_0))$. The definition of $T$ then implies that $\tilde{\gamma}(\boldsymbol{\theta}(T)) = \gamma(\boldsymbol{\theta}(T)) = \gamma(\bar{\boldsymbol{\theta}}^*)$. By Lemma B.1 from Lyu and Li (2020), $\tilde{\gamma}(\boldsymbol{\theta}(t))$ is non-decreasing over time, and if it stops increasing at some value, then the time derivative of $\frac{\boldsymbol{\theta}(t)}{\|\boldsymbol{\theta}(t)\|_2}$ must be zero.

Thus we have $\tilde{\gamma}(\boldsymbol{\theta}(t)) = \gamma(\bar{\boldsymbol{\theta}}^*)$ and $\frac{\mathrm{d}}{\mathrm{d}t} \frac{\boldsymbol{\theta}(t)}{\|\boldsymbol{\theta}(t)\|_2} = 0$ for all $t \geq T$, which implies that $\frac{\boldsymbol{\theta}(t)}{\|\boldsymbol{\theta}(t)\|_2}$ converges to $\bar{\boldsymbol{\theta}} := \frac{\boldsymbol{\theta}(T)}{\|\boldsymbol{\theta}(T)\|_2}$ as $t \to +\infty$. This again proves that $\|\bar{\boldsymbol{\theta}} - \bar{\boldsymbol{\theta}}^*\|_2 \leq \delta$.

Now we only need to show that $\gamma(\bar{\boldsymbol{\theta}}) = \gamma(\bar{\boldsymbol{\theta}}^*)$. In the case where $T$ is finite, we have $\gamma(\bar{\boldsymbol{\theta}}) = \gamma\left(\frac{\boldsymbol{\theta}(T)}{\|\boldsymbol{\theta}(T)\|_2}\right) = \gamma(\bar{\boldsymbol{\theta}}^*)$. In the case where $T = +\infty$, $\gamma(\bar{\boldsymbol{\theta}})$ is a asymptotic Clarke critical value of $\tilde{\gamma}$ by Theorem F.9. Since there are only finitely many asymptotic Clarke critical values (Lemma F.7), we can make $\delta(\epsilon_0, \rho_0)$ to be small enough so that the only asymptotic Clarke critical value that can be achieved near $\bar{\boldsymbol{\theta}}^*$ is $\gamma(\bar{\boldsymbol{\theta}}^*)$ itself. $\qquad\square$

### F.3 Proof for Theorem 4.3

*Proof.* By Lemma 5.4, $\lim_{\sigma_{\mathrm{init}} \to 0} \boldsymbol{\theta}(T_{12} + t) = \pi_{\bar{\boldsymbol{b}}}(\tilde{\boldsymbol{\theta}}(t))$. Using Lemma E.8 and noting that $\langle \boldsymbol{\mu}, \boldsymbol{w}^* \rangle = \frac{1}{n} \sum_{i \in [n]} \langle y_i \boldsymbol{x}_i, \boldsymbol{w}^* \rangle > 0$, we know that there exists $t \leq t_0$ such that $\tilde{\boldsymbol{\theta}}(t) = (\tilde{\boldsymbol{w}}_1(t), \tilde{\boldsymbol{w}}_2(t), \tilde{a}_1(t), \tilde{a}_2(t))$ satisfies $\tilde{a}_1(t) = \|\tilde{\boldsymbol{w}}_1(t)\|_2$, $\tilde{a}_2(t) = -\|\tilde{\boldsymbol{w}}_2(t)\|_2$, $\langle \tilde{\boldsymbol{w}}_1(t), \boldsymbol{w}^* \rangle > 0$ and $\langle \tilde{\boldsymbol{w}}_2(t), \boldsymbol{w}^* \rangle < 0$. Then by Theorem 5.5,

$$\lim_{t \to +\infty} \frac{\tilde{\boldsymbol{\theta}}(t)}{\|\tilde{\boldsymbol{\theta}}(t)\|_2} = \frac{1}{2}(\boldsymbol{w}^*, -\boldsymbol{w}^*, 1, -1) =: \tilde{\boldsymbol{\theta}}_\infty,$$

which also implies that

$$\lim_{t \to +\infty} \lim_{\sigma_{\mathrm{init}} \to 0} \frac{\boldsymbol{\theta}(T_{12} + t)}{\|\boldsymbol{\theta}(T_{12} + t)\|_2} = \lim_{t \to +\infty} \frac{\pi_{\bar{\boldsymbol{b}}}(\tilde{\boldsymbol{\theta}}(t))}{\|\pi_{\bar{\boldsymbol{b}}}(\tilde{\boldsymbol{\theta}}(t))\|_2} = \pi_{\bar{\boldsymbol{b}}}\left(\lim_{t \to +\infty} \frac{\tilde{\boldsymbol{\theta}}(t)}{\|\tilde{\boldsymbol{\theta}}(t)\|_2}\right) = \pi_{\bar{\boldsymbol{b}}}(\tilde{\boldsymbol{\theta}}_\infty).$$

$$\lim_{t \to +\infty} \lim_{\sigma_{\mathrm{init}} \to 0} \|\boldsymbol{\theta}(T_{12} + t)\|_2 = \lim_{t \to +\infty} \|\pi_{\bar{\boldsymbol{b}}}(\tilde{\boldsymbol{\theta}}(t))\|_2 = +\infty.$$

This means that for any $\epsilon > 0$ and $\rho > 0$, we can choose a time $t_1 \in \mathbb{R}$ such that $\left\|\frac{\boldsymbol{\theta}(T_{12} + t_1)}{\|\boldsymbol{\theta}(T_{12} + t_1)\|_2} - \pi_{\bar{\boldsymbol{b}}}(\tilde{\boldsymbol{\theta}}_\infty)\right\|_2 \leq \epsilon$ and $\|\boldsymbol{\theta}(T_{12} + t_1)\|_2 \geq \rho$ for any $\sigma_{\mathrm{init}}$ small enough. By Theorem 4.2, $\pi_{\bar{\boldsymbol{b}}}(\tilde{\boldsymbol{\theta}}_\infty)$ is a global-max-margin direction. Then Theorem 5.6 shows that there exists $\sigma_{\mathrm{init}}^{\max}$ such that for all $\sigma_{\mathrm{init}} < \sigma_{\mathrm{init}}^{\max}$, $\frac{\boldsymbol{\theta}(t)}{\|\boldsymbol{\theta}(t)\|_2} \to \bar{\boldsymbol{\theta}}$, where $\gamma(\bar{\boldsymbol{\theta}}) = \gamma(\pi_{\bar{\boldsymbol{b}}}(\tilde{\boldsymbol{\theta}}_\infty))$ and $\|\bar{\boldsymbol{\theta}} - \pi_{\bar{\boldsymbol{b}}}(\tilde{\boldsymbol{\theta}}_\infty)\|_2 \leq \delta$. Therefore, $\bar{\boldsymbol{\theta}}$ is a global-max-margin direction and $f^\infty(\boldsymbol{x}) = \frac{1 + \alpha_{\mathrm{leaky}}}{4} \langle \boldsymbol{w}^*, \boldsymbol{x} \rangle$ by Theorem 4.2. $\quad\square$

## G  Trajectory-based Analysis for Non-symmetric Case

The proofs for the non-symmetric case follow similar manners from phase I to phase III. The high-level idea is to show the following in the 3 phases:

1. In Phase I, every weight vector $\boldsymbol{w}_k$ in the first layer moves towards the direction of either $\boldsymbol{\mu}^+$ or $-\boldsymbol{\mu}^-$. At the end of Phase I the weight vectors towards $-\boldsymbol{\mu}^-$ have much smaller norms than those towards $\boldsymbol{\mu}^+$, thereby becoming negligible.

2. In Phase II, we show that the dynamics of $\boldsymbol{\theta}(t)$ is close to a one-neuron dynamic (after embedding) for a long time.

3. In Phase III, we show that the one-neuron classifier converges to the max-margin solution among one-neuron neural nets (while the embedded classifier may have suboptimal margin among $m$-neuron neural nets), and the gradient flow $\boldsymbol{\theta}(t)$ on the $m$-neuron neural net gets stuck at a KKT-direction near this embedded classifier.

### G.1  Additional Notations

In this section we highlight the additional notations that allow us to adapt the results from previous sections. For $\delta \geq 0$, define $\mathcal{C}^\delta$ to be the convex cone containing all the unit weight vectors that have $\delta$ margin over the dataset $\{(\boldsymbol{x}_i, y_i)\}_{i \in [n]}$.

$$\mathcal{C}^\delta := \left\{\lambda \boldsymbol{w} : \langle \boldsymbol{w}, y_i \boldsymbol{x}_i \rangle \geq \delta, \boldsymbol{w} \in \mathbb{S}^{d-1}, \lambda > 0, \forall i \in [n]\right\},$$
$$\mathcal{C} := \mathcal{C}^0 := \left\{\boldsymbol{w} \in \mathbb{R}^d : \boldsymbol{w} \neq \boldsymbol{0}, \langle \boldsymbol{w}, y_i \boldsymbol{x}_i \rangle \geq 0, \forall i \in [n]\right\}.$$

For $0 < \epsilon < 1$, we define

$$\mathcal{H}^\epsilon := \left\{ \frac{1}{2n} \sum_{i \in [n]} (1 + \epsilon_i) \alpha_i y_i \boldsymbol{x}_i : \alpha_i \in [\alpha_{\text{leaky}}, 1], \epsilon_i \in [-\epsilon, \epsilon], \forall i \in [n] \right\},$$

$$\mathcal{H} := \mathcal{H}^0 := \left\{ \frac{1}{2n} \sum_{i \in [n]} \alpha_i y_i \boldsymbol{x}_i : \alpha_i \in [\alpha_{\text{leaky}}, 1], \forall i \in [n] \right\}.$$

By Lemma B.14 we know $-\frac{\partial^\circ \mathcal{L}(\boldsymbol{\theta})}{\partial \boldsymbol{w}_k} \subseteq a_k \mathcal{H}^\epsilon$ if $\|\boldsymbol{\theta}\|_{\text{M}} \leq \sqrt{\frac{\epsilon}{2m}}$. Further we define

$$\mathcal{K}^\epsilon := \bigcup_{\lambda > 0} \lambda \mathcal{H}^\epsilon \quad \text{and} \quad \mathcal{K} := \mathcal{K}^0 = \bigcup_{\lambda > 0} \lambda \mathcal{H}^0$$

Then $-\frac{\partial^\circ \mathcal{L}(\boldsymbol{\theta})}{\partial \boldsymbol{w}_k} \subseteq \text{sgn}(a_k) \mathcal{K}^\epsilon$. For a set $S$, we will use $\mathring{S}$ to denote the interior of $S$.

Recall in Appendix A, for every $\boldsymbol{x}_i$, we define $\boldsymbol{x}_i^+ := \boldsymbol{x}_i$ if $y_i = 1$ and $\boldsymbol{x}_i^+ := \alpha_{\text{leaky}} \boldsymbol{x}_i$ if $y_i = -1$. Similarly, we define $\boldsymbol{x}_i^- := \alpha_{\text{leaky}} \boldsymbol{x}_i$ if $y_i = 1$ and $\boldsymbol{x}_i^+ := \boldsymbol{x}_i$ if $y_i = -1$. Then we define $\boldsymbol{\mu}^+$ to be the mean vector of $y_i \boldsymbol{x}_i^+$, and $\boldsymbol{\mu}^-$ to be the mean vector of $y_i \boldsymbol{x}_i^-$, that is,

$$\boldsymbol{\mu}^+ := \frac{1}{n} \sum_{i \in [n]} y_i \boldsymbol{x}_i^+, \qquad \boldsymbol{\mu}^- := \frac{1}{n} \sum_{i \in [n]} y_i \boldsymbol{x}_i^-.$$

We use $\bar{\boldsymbol{\mu}}^+ := \frac{\boldsymbol{\mu}^+}{\|\boldsymbol{\mu}^+\|_2}, \bar{\boldsymbol{\mu}}^- := \frac{\boldsymbol{\mu}^-}{\|\boldsymbol{\mu}^-\|_2}$ to denote $\boldsymbol{\mu}^+, \boldsymbol{\mu}^-$ after normalization. Similar to $\mathcal{K}^\epsilon$, we define $\mathcal{M}_+^\epsilon$ and $\mathcal{M}_-^\epsilon$ as the perturbed versions of $\boldsymbol{\mu}^+$ and $\boldsymbol{\mu}^-$ in the sense that $\mathcal{M}_+ := \{\lambda \boldsymbol{\mu}^+ : \lambda > 0\}$ and $\mathcal{M}_- := \{\lambda \boldsymbol{\mu}^- : \lambda > 0\}$.

$$\mathcal{M}_+^\epsilon = \left\{ \frac{\lambda}{2n} \sum_{i \in [n]} (1 + \epsilon_i) y_i \boldsymbol{x}_i^+ : \epsilon_i \in [-\epsilon, \epsilon], \lambda > 0 \right\},$$

$$\mathcal{M}_-^\epsilon = \left\{ \frac{\lambda}{2n} \sum_{i \in [n]} (1 + \epsilon_i) y_i \boldsymbol{x}_i^- : \epsilon_i \in [-\epsilon, \epsilon], \lambda > 0 \right\}.$$

## G.2 More about Our Assumptions

The following lemma shows that Assumption A.1 is a weaker assumption than Assumption A.2.

**Lemma G.1.** *Assumption A.1 implies Assumption A.2.*

*Proof.* Let $\boldsymbol{w}^\diamond$ be the principal direction defined in Assumption A.1. We can decompose $\boldsymbol{\mu} = \boldsymbol{\mu}_\perp + \boldsymbol{\mu}_\parallel$, where $\boldsymbol{\mu}_\parallel$ is the along the direction of $\boldsymbol{w}^\diamond$ and $\boldsymbol{\mu}_\perp$ is orthogonal to $\boldsymbol{w}^\diamond$. Assumption A.1 implies that for all $i, j \in [n]$,

$$-\langle y_i \boldsymbol{x}_i, y_j \boldsymbol{x}_j \rangle = -\langle y_i \boldsymbol{x}_i, \boldsymbol{w}^\diamond \rangle \cdot \langle y_j \boldsymbol{x}_j, \boldsymbol{w}^\diamond \rangle - \langle \boldsymbol{P}^\diamond(y_i \boldsymbol{x}_i), \boldsymbol{P}^\diamond(y_j \boldsymbol{x}_j) \rangle$$
$$\leq \|\boldsymbol{P}^\diamond \boldsymbol{x}_i\|_2 \|\boldsymbol{P}^\diamond \boldsymbol{x}_j\|_2.$$

Then for all $i \in [n]$, we have

$$\frac{1}{n} \sum_{j \in [n]} \max\{-\langle y_i \boldsymbol{x}_i, y_j \boldsymbol{x}_j \rangle, 0\} \leq \frac{1}{n} \sum_{j \in [n]} \|\boldsymbol{P}^\diamond \boldsymbol{x}_i\|_2 \|\boldsymbol{P}^\diamond \boldsymbol{x}_j\|_2$$
$$\leq \|\boldsymbol{P}^\diamond \boldsymbol{x}_i\|_2 \cdot \alpha_{\text{leaky}} \langle \boldsymbol{\mu}, \boldsymbol{w}^\diamond \rangle \frac{\gamma^\diamond}{\max_{j \in [n]} \|\boldsymbol{P}^\diamond \boldsymbol{x}_j\|_2}$$
$$\leq \alpha_{\text{leaky}} \langle \boldsymbol{\mu}, \boldsymbol{w}^\diamond \rangle \gamma^\diamond.$$

On the other hand, recall that $\gamma^\diamond := \min_{i\in[n]} y_i\langle \boldsymbol{w}^\diamond, \boldsymbol{x}_i\rangle$, then we have

$$\langle \boldsymbol{\mu}, y_i\boldsymbol{x}_i\rangle = \langle \boldsymbol{\mu}, \boldsymbol{w}^\diamond\rangle \langle y_i\boldsymbol{x}_i, \boldsymbol{w}^\diamond\rangle + \langle \boldsymbol{P}^\diamond\boldsymbol{\mu}, \boldsymbol{P}^\diamond(y_i\boldsymbol{x}_i)\rangle$$

$$\geq \langle \boldsymbol{\mu}, \boldsymbol{w}^\diamond\rangle \langle y_i\boldsymbol{x}_i, \boldsymbol{w}^\diamond\rangle - \frac{1}{n}\sum_{j\in[n]}\|\boldsymbol{P}^\diamond\boldsymbol{x}_j\|_2\|\boldsymbol{P}^\diamond\boldsymbol{x}_i\|_2$$

$$\geq \langle \boldsymbol{\mu}, \boldsymbol{w}^\diamond\rangle\gamma^\diamond - \|\boldsymbol{P}^\diamond\boldsymbol{x}_i\|_2\cdot\alpha_{\text{leaky}}\langle \boldsymbol{\mu}, \boldsymbol{w}^\diamond\rangle\frac{\gamma^\diamond}{\max_{j\in[n]}\|\boldsymbol{P}^\diamond\boldsymbol{x}_j\|_2}$$

$$\geq (1-\alpha_{\text{leaky}})\langle \boldsymbol{\mu}, \boldsymbol{w}^\diamond\rangle\gamma^\diamond.$$

Combining these proves that $\langle \boldsymbol{\mu}, y_i\boldsymbol{x}_i\rangle \geq \frac{1-\alpha_{\text{leaky}}}{n\cdot\alpha_{\text{leaky}}}\sum_{j\in[n]}\max\{-\langle y_i\boldsymbol{x}_i, y_j\boldsymbol{x}_j\rangle, 0\}$. $\qquad\square$

Lemma G.2 gives the main property we will use from Assumption A.2, i.e. $\mathcal{K}\subseteq\mathring{\mathcal{C}}$.

**Lemma G.2.** *For linearly separable dataset $\{(\boldsymbol{x}_i, y_i)\}_{i\in[n]}$ and $\alpha_{\text{leaky}}\in(0,1]$, Assumption A.2 is equivalent to $\mathcal{K}\subseteq\mathring{\mathcal{C}}$.*

*Proof.* By definition, we know

$$\mathcal{C} = \{\boldsymbol{w}\in\mathbb{R}^d : \boldsymbol{w}\neq\boldsymbol{0}, \langle \boldsymbol{w}, y_i\boldsymbol{x}_i\rangle \geq 0, \forall i\in[n]\},$$
$$\mathring{\mathcal{C}} = \{\boldsymbol{w}\in\mathbb{R}^d : \langle \boldsymbol{w}, y_i\boldsymbol{x}_i\rangle > 0, \forall i\in[n]\}.$$

and $\mathcal{K} = \left\{\lambda\sum_{i\in[n]}\alpha_i y_i\boldsymbol{x}_i : \alpha_i\in[\alpha_{\text{leaky}}, 1], \lambda > 0\right\}$. For any $i\in[n]$, we have

$$\langle \boldsymbol{\mu}, y_i\boldsymbol{x}_i\rangle > \frac{1-\alpha_{\text{leaky}}}{n\cdot\alpha_{\text{leaky}}}\sum_{j\in[n]}\max\{-\langle y_i\boldsymbol{x}_i, y_j\boldsymbol{x}_j\rangle, 0\}$$

$$\Longleftrightarrow \frac{1}{n}\sum_{j\in[n]}\left(\frac{1-\alpha_{\text{leaky}}}{\alpha_{\text{leaky}}}\min\{\langle y_i\boldsymbol{x}_i, y_j\boldsymbol{x}_j\rangle, 0\} + \langle y_i\boldsymbol{x}_i, y_j\boldsymbol{x}_j\rangle\right) > 0$$

$$\Longleftrightarrow \sum_{j\in[n]}\underbrace{((1-\alpha_{\text{leaky}})\min\{\langle y_i\boldsymbol{x}_i, y_j\boldsymbol{x}_j\rangle, 0\} + \alpha_{\text{leaky}}\langle y_i\boldsymbol{x}_i, y_j\boldsymbol{x}_j\rangle)}_{\Delta_{ij}} > 0$$

Note that $\Delta_{ij} = \min_{\alpha_j\in[\alpha_{\text{leaky}}, 1]}\langle y_i\boldsymbol{x}_i, \alpha_j y_j\boldsymbol{x}_j\rangle$. So

$$\sum_{j\in[n]}\Delta_{ij} = \sum_{j\in[n]}\min_{\alpha_j\in[\alpha_{\text{leaky}}, 1]}\langle y_i\boldsymbol{x}_i, \alpha_j y_j\boldsymbol{x}_j\rangle = \min_{\boldsymbol{\alpha}\in[\alpha_{\text{leaky}}, 1]^n}\left\langle y_i\boldsymbol{x}_i, \sum_{j\in[n]}\alpha_j y_j\boldsymbol{x}_j\right\rangle.$$

Therefore we have the following equivalence:

$$\text{Assumption A.2} \iff \forall i\in[n] : \min_{\boldsymbol{\alpha}\in[\alpha_{\text{leaky}}, 1]^n}\left\langle y_i\boldsymbol{x}_i, \sum_{j\in[n]}\alpha_j y_j\boldsymbol{x}_j\right\rangle > 0 \qquad (58)$$

$$\iff \forall \boldsymbol{w}\in\mathcal{K}, \forall i\in[n] : \langle y_i\boldsymbol{x}_i, \boldsymbol{w}\rangle > 0 \qquad (59)$$

$$\iff \mathcal{K}\subseteq\mathring{\mathcal{C}}. \qquad (60)$$

which completes the proof. $\qquad\square$

Lemma G.2 shows that every direction in $\mathcal{K}$ has non-zero margin. Below we let the $\delta$ be the minimum of the margin of unit-norm linear separators in $\mathcal{K}$:

$$\delta := \min_{\boldsymbol{w}\in\mathcal{K}\cap\mathbb{S}^{d-1}}\min_{i\in[n]}\langle y_i\boldsymbol{x}_i, \boldsymbol{w}\rangle.$$

By (58) we have $\delta > 0$, and thus $\mathcal{K}\subseteq\mathcal{C}^\delta$.

### G.3 Phase I

The overall result we will prove for phase I in the non-symmetric case is Lemma G.5. Compared to the symmetric case, even $G$ function is not linear anymore. Recall $G$ is defined as below:

$$G(\boldsymbol{w}) := \frac{-\ell'(0)}{n} \sum_{i \in [n]} y_i \phi(\boldsymbol{w}^\top \boldsymbol{x}_i) = \frac{1}{2n} \sum_{i \in [n]} y_i \phi(\boldsymbol{w}^\top \boldsymbol{x}_i).$$

It holds that $\forall \boldsymbol{w} \in \mathbb{R}^d$, $\partial^\circ G(\boldsymbol{w}) \subseteq \mathcal{K}$. Moreover, we have $\boldsymbol{w} \in \mathring{\mathcal{C}} \implies \partial^\circ G(\boldsymbol{w}) = \{\boldsymbol{\mu}^+\}$ and $\boldsymbol{w} \in -\mathring{\mathcal{C}} \implies \partial^\circ G(\boldsymbol{w}) = \{\boldsymbol{\mu}^-\}$. Thanks to Assumption A.2, we can show each neuron $\boldsymbol{w}_k(t)$ will eventually converge to areas with fixed sign pattern $\pm \mathcal{C}^{\delta/3}$ and thus $G$ will become linear. Lemma G.3 states this idea more formally. Its proof is a simpliciation to the realistic case, Lemma G.4, and thus omitted. We will not use Lemma G.3 in the future.

**Lemma G.3.** *For any dataset $\{(\boldsymbol{x}_i, y_i)\}_{i \in [n]}$ satisfying Assumption A.2, suppose $\boldsymbol{w}(0) \neq \lambda \boldsymbol{\mu}^-$, $\forall \lambda \geq 0$, and it holds that*

$$\frac{\mathrm{d}\boldsymbol{w}}{\mathrm{d}t} \in \|\boldsymbol{w}\|_2 \cdot \partial^\circ G(\boldsymbol{w}), \tag{61}$$

*then there exists $T_0 > 0$, such that $\boldsymbol{w}(T_0) \in \mathcal{C}^{\delta/2}$.*

However, in the realistic setting, each $\boldsymbol{w}_k$ is not following gradient flow of $G$ exactly — there are tiny correlations between different $\boldsymbol{w}_k$. And we will control those correlations by setting initialization very small. This yields Lemma G.4.

**Lemma G.4.** *Under Assumption A.2, if $\bar{\boldsymbol{\theta}} = (\bar{\boldsymbol{w}}_1, \ldots, \bar{\boldsymbol{w}}_m, \bar{a}_1, \ldots, \bar{a}_m)$ satisfies the following three conditions:*

1. *For all $k \in [m]$, $|\bar{a}_k| = \|\bar{\boldsymbol{w}}_k\|_2 \neq 0$;*

2. *If $\bar{a}_k > 0$, then $\bar{\boldsymbol{w}}_k \neq \lambda \boldsymbol{\mu}^-$ for any $\lambda > 0$;*

3. *If $\bar{a}_k < 0$, then $\bar{\boldsymbol{w}}_k \neq -\lambda \boldsymbol{\mu}^+$ for any $\lambda > 0$;*

*then there exist $T_0, \sigma_{\mathrm{init}}^{\max} > 0$, such that for any $\sigma_{\mathrm{init}} < \sigma_{\mathrm{init}}^{\max}$, the gradient flow $\boldsymbol{\theta}(t) = (\boldsymbol{w}_1(t), \ldots, \boldsymbol{w}_m(t), a_1(t), \ldots, a_m(t)) = \varphi(\sigma_{\mathrm{init}} \bar{\boldsymbol{\theta}}, t)$ satisfies the following at time $T_0$,*

$$\boldsymbol{w}_k(T_0) \in \begin{cases} \mathcal{C}^{\delta/3}, & \text{if } \bar{a}_k > 0, \\ -\mathcal{C}^{\delta/3}, & \text{if } \bar{a}_k < 0. \end{cases} \tag{62}$$

*Moreover, there are constants $A, B > 0$ such that $A\sigma_{\mathrm{init}} \leq \|\boldsymbol{w}_k(T_0)\|_2 \leq B\sigma_{\mathrm{init}}$.*

It is easy to see that the three conditions in Lemma G.4 hold with probability 1 over the random draw of $\bar{\boldsymbol{\theta}}_0 \sim \mathcal{D}_{\mathrm{init}}(1)$. Then after time $T_0$, all the neurons $\boldsymbol{w}_k$ are either in $\mathcal{C}^{\delta/3}$ or $-\mathcal{C}^{\delta/3}$, and will not leave it until $T_{\sigma_{\mathrm{init}}}^\epsilon$, which implies the sign patterns $\mathrm{sgn}(\langle \boldsymbol{x}_i, \boldsymbol{w}_k(t) \rangle) = s_k y_i$ is fixed for $t \in [T_0, T_{\sigma_{\mathrm{init}}}^\epsilon]$. Thus similar to the symmetric case, $\boldsymbol{\theta}(t)$ evolves approximately under power iteration and yields the following lemma.

**Lemma G.5.** *Suppose that Assumptions A.2 and A.3 hold. Let $T_1(\sigma_{\mathrm{init}}, r) := \frac{1}{\lambda_0^+} \ln \frac{r}{\sqrt{m}\sigma_{\mathrm{init}}}$. With probability 1 over the random draw of $\bar{\boldsymbol{\theta}}_0 = (\bar{\boldsymbol{w}}_1, \ldots, \bar{\boldsymbol{w}}_m, \bar{a}_1, \ldots, \bar{a}_m) \sim \mathcal{D}_{\mathrm{init}}(1)$, the prerequisites of Lemma G.4 are satisfied. In this case, there exists a vector $\bar{\boldsymbol{b}}(\sigma_{\mathrm{init}}) \in \mathbb{R}^m$ for any $\sigma_{\mathrm{init}} > 0$ such that the following statements hold:*

1. *There exist constants $C_1 > 0, C_2 > 0, T_0 \geq 0, r_{\max} > 0$ such that for $r \in (0, r_{\max}), \sigma_{\mathrm{init}} \in (0, C_1 r^3)$, any neuron $(\boldsymbol{w}_k, a_k)$ at time $T_0 + T_1(\sigma_{\mathrm{init}}, r)$ can be decomposed into*

$$\text{If } \bar{a}_k > 0: \quad \boldsymbol{w}_k(T_0 + T_1(\sigma_{\mathrm{init}}, r)) = r\bar{b}_k(\sigma_{\mathrm{init}})\bar{\boldsymbol{\mu}}^+ + \Delta \boldsymbol{w}_k,$$

$$a_k(T_0 + T_1(\sigma_{\mathrm{init}}, r)) = r\bar{b}_k(\sigma_{\mathrm{init}}) + \Delta a_k,$$

$$\text{If } \bar{a}_k < 0: \quad \boldsymbol{w}_k(T_0 + T_1(\sigma_{\mathrm{init}}, r)) = r^{1-\kappa}\bar{b}_k(\sigma_{\mathrm{init}})\bar{\boldsymbol{\mu}}^- + \Delta \boldsymbol{w}_k,$$

$$a_k(T_0 + T_1(\sigma_{\mathrm{init}}, r)) = r^{1-\kappa}\bar{b}_k(\sigma_{\mathrm{init}}) + \Delta a_k,$$

*where the error term $\Delta \boldsymbol{\theta} := (\Delta \boldsymbol{w}_1, \ldots, \Delta \boldsymbol{w}_m, \Delta a_1, \ldots, \Delta a_m)$ is upper bounded by $\|\Delta \boldsymbol{\theta}\|_{\mathrm{M}} \leq C_2 r^3$ and $\kappa$ is the gap $1 - \frac{\|\boldsymbol{\mu}^-\|_2}{\|\boldsymbol{\mu}^+\|_2} > 0$.*

2. *There exist constants $\bar{A}, \bar{B} > 0$ such that $|\bar{b}_k(\sigma_{\text{init}})| \in [\bar{A}, \bar{B}]$ whenever $\bar{a}_k > 0$ and $|\bar{b}_k(\sigma_{\text{init}})| \in [\sigma_{\text{init}}^\kappa \bar{A}, \sigma_{\text{init}}^\kappa \bar{B}]$ whenever $\bar{a}_k < 0$.*

As $\sigma_{\text{init}} \to 0$, $|\bar{b}_k(\sigma_{\text{init}})| \to 0$ for neurons with $\bar{a}_k < 0$, while $|\bar{b}_k(\sigma_{\text{init}})| \in [\bar{A}, \bar{B}]$ remains for neurons with $\bar{a}_k > 0$. This means when the initialization scale is small, only the neurons with $\bar{a}_k > 0$ remain effective and the others become negligible. Those effective neurons move their weight vectors towards the direction of $\bar{\mu}^+$, until the error term $\Delta\boldsymbol{\theta}$ becomes large.

### G.3.1 Proof of Lemma G.4

*Proof of Lemma G.4.* Let $s_k := \text{sgn}(\bar{a}_k)$. By Corollary B.18, $a_k(t) = s_k\|\boldsymbol{w}(t)\|_2$ for all $t \geq 0$. Define $T^\epsilon_{\sigma_{\text{init}}} := \inf\left\{t \geq 0 : \|\boldsymbol{\theta}(t)\|_M \geq \sqrt{\frac{\epsilon}{m}}\right\}$. By Lemma B.14, we have $\forall t \leq T^\epsilon_{\sigma_{\text{init}}}$, $-\frac{\partial^\circ \mathcal{L}(\boldsymbol{\theta}(t))}{\partial \boldsymbol{w}_k} \subseteq a_k(t)\mathcal{H}^\epsilon \subseteq s_k \mathcal{K}^\epsilon$. Since $\mathcal{K} \subseteq \mathcal{C}^\delta$, there exists $\epsilon_1 > 0$, such that for all $\epsilon < \epsilon_1$, $\mathcal{K}^\epsilon \subseteq \mathcal{C}^{2\delta/3}$. The high-level idea of the proof is that suppose $-\frac{\partial^\circ \mathcal{L}(\boldsymbol{\theta}(t))}{\partial \boldsymbol{w}_k} \subseteq s_k \mathcal{C}^{2\delta/3}$ holds for sufficiently long time $T_0$, $\boldsymbol{w}_k(t)$ will eventually end up in a cone $s_k \mathcal{C}^{\delta/3}$ slightly wider than $s_k \mathcal{C}^{2\delta/3}$, as long as the total distance traveled is sufficiently long. On the other hand, we can make $\sigma_{\text{init}}^{\max}$ sufficiently small, such that $T^\epsilon_{\sigma_{\text{init}}} \geq T_0$ for all $\sigma_{\text{init}} < \sigma_{\text{init}}^{\max}$.

By Lemma B.16 and Lipschitzness of $\ell$,

$$\left|\frac{1}{2}\frac{\mathrm{d}\|\boldsymbol{w}_k\|_2^2}{\mathrm{d}t}\right| = \left|\frac{1}{n}\sum_{i=1}^{n} \ell'(q_i(\boldsymbol{\theta}))y_i a_k \phi(\boldsymbol{w}_k^\top \boldsymbol{x}_i)\right| \leq \frac{1}{n}\sum_{i=1}^{n} |a_k| \cdot \|\boldsymbol{w}_k\|_2 \leq \|\boldsymbol{w}_k\|_2^2.$$

Then we have

$$\forall t \leq T^\epsilon_{\sigma_{\text{init}}}, \quad \|\boldsymbol{w}_k(t)\|_2 \in [\|\boldsymbol{w}_k(0)\|_2 e^{-t}, \|\boldsymbol{w}_k(0)\|_2 e^t]. \tag{63}$$

Thus for any $T_0 \geq 0$, if $\sigma_{\text{init}} \leq e^{-T_0}\frac{\sqrt{\frac{\epsilon}{m}}}{\|\bar{\boldsymbol{\theta}}\|_M}$, we have $T_0 \leq T^\epsilon_{\sigma_{\text{init}}}$.

In order to lower bound the total travel distance for each $\boldsymbol{w}_k(t)$, it turns out that it suffices to lower bound the $\inf_{t \in [0, T^\epsilon_{\sigma_{\text{init}}}]} \|\boldsymbol{w}_k(t)\|_2$ by $\bar{D}\sigma_{\text{init}}$, where $\bar{D} > 0$ is some constant. We will first show that we can guarantee the existence of such constant $\bar{D}$ by picking sufficiently small $\epsilon$. Then we will formally prove the original claim of Lemma G.4.

**Existence of $\bar{D}$.** By definitions of $\mathcal{M}_+$ and $\mathcal{M}_-$, it holds that $\forall k \in [m]$,

$$\bar{\boldsymbol{w}}_k \notin \begin{cases} \mathcal{M}_+, & \text{if } \bar{a}_k < 0; \\ -\mathcal{M}_-, & \text{if } \bar{a}_k > 0. \end{cases}$$

In other words

$$\bar{d} := \min\left\{\min_{k:\bar{a}_k<0} \text{dist}(\bar{\boldsymbol{w}}_k - \mathcal{M}_+, \boldsymbol{0}), \min_{k:\bar{a}_k>0} \text{dist}(\bar{\boldsymbol{w}}_k + \mathcal{M}_-, \boldsymbol{0})\right\} > 0.$$

By the continuity of the distance function, there exists $\epsilon_2 > 0$ such that $\forall \epsilon \in (0, \epsilon_2)$, it holds that

$$\min\left\{\min_{k:\bar{a}_k<0} \text{dist}(\bar{\boldsymbol{w}}_k - \mathcal{M}_+^\epsilon, \boldsymbol{0}), \min_{k:\bar{a}_k>0} \text{dist}(\bar{\boldsymbol{w}}_k + \mathcal{M}_-^\epsilon, \boldsymbol{0})\right\} \geq \frac{\bar{d}}{2}.$$

Now we take $\epsilon = \min\{\epsilon_1, \epsilon_2\}$. We will first show the existence of such $\bar{D}$ for $k \in [m]$ with $\bar{a}_k > 0$. And the same argument holds for $k$ with negative $\bar{a}_k$. Let $t_k := \sup\{t \leq T^\epsilon_{\sigma_{\text{init}}} : \boldsymbol{w}_k(t) \in -\mathcal{C}^{\delta/3}\}$. We note that $\boldsymbol{w}_k(t) \in -\mathcal{C}^{\delta/3}$ for all $t \leq t_k$. Otherwise, $\boldsymbol{w}_k(t') \notin -\mathcal{C}^{\delta/3}$ for some $t' < t_k$. On the one hand, we have $\boldsymbol{w}_k(t_k) \in \boldsymbol{w}_k(t') + \mathcal{K}^\epsilon \subseteq \boldsymbol{w}_k(t') + \mathcal{C}^{2\delta/3}$; on the other hand, we also know that $\boldsymbol{w}_k(t_k) \in -\mathcal{C}^{\delta/3}$ by continuity of the trajectory of $\boldsymbol{w}_k(t)$. This implies $-\mathcal{C}^{\delta/3} \cap (\mathcal{C}^{2\delta/3} + \boldsymbol{w}_k(t')) \neq \varnothing$, and thus $\boldsymbol{w}_k(t') \in -\mathcal{C}^{\delta/3} - \mathcal{C}^{2\delta/3} \subseteq -\mathcal{C}^{\delta/3} - \mathcal{C}^{\delta/3} \subseteq -\mathcal{C}^{\delta/3}$. Contradiction.

Now we have $\boldsymbol{w}_k(t) \in -\mathcal{C}^{\delta/3}$ for all $t \leq t_k$, and this implies that $\langle \boldsymbol{w}_k(t), \boldsymbol{x}_i\rangle < 0$ for all $i \in [n]$. Then $\frac{\mathrm{d}\boldsymbol{w}_k(t)}{\mathrm{d}t} = -\nabla_{\boldsymbol{w}_k}\mathcal{L}(\boldsymbol{\theta}(t)) \in \mathcal{M}_-^\epsilon$. Therefore we have $\inf_{t \in [0, t_k]} \|\boldsymbol{w}_k(t)\|_2 \geq \text{dist}(\boldsymbol{w}_k(0) + \mathcal{M}_-^\epsilon, \boldsymbol{0}) = \sigma_{\text{init}}\text{dist}(\bar{\boldsymbol{w}}_k + \mathcal{M}_-^\epsilon, \boldsymbol{0}) \geq \frac{\bar{d}\sigma_{\text{init}}}{2}$.

Below we show the norm lower bound for any $t$ such that $t \in [t_k, T^\epsilon_{\sigma_{\text{init}}}]$. Let $\bar{d}'$ be the minimum distance between any point in $-\mathcal{C}^{2\delta/3}$ and any point on unit sphere but not in $-\mathcal{C}^{\delta/2}$, that is,

$$\bar{d}' := \text{dist}(\mathbb{S}^{d-1} \setminus (-\mathcal{C}^{\delta/2}), -\mathcal{C}^{2\delta/3})$$

We claim that $\bar{d}' > 0$. Otherwise there is a sequence of $\{w_j\}$ with unit norm and $w_j \notin -\mathcal{C}^{\delta/2}$ satisfying that $\lim_{n \to \infty} \text{dist}(w_k, -\mathcal{C}^{2\delta/3}) = 0$. Let $\bar{w}$ be a limit point, then $\bar{w} \in -\mathcal{C}^{2\delta/3}$ since $-\mathcal{C}^{2\delta/3}$ is closed. Since $-\mathcal{C}^{2\delta/3} \subseteq -\mathring{\mathcal{C}}^{\delta/2}$, we further have $\bar{w} \in -\mathring{\mathcal{C}}^{\delta/2}$, which contradicts with the definition of limit point.

By the continuity of $w_k(t)$, we know $w_k(t_k) \notin -\mathcal{C}^{\delta/2}$. Thus for any $t \in [t_k, T^\epsilon_{\sigma_{\text{init}}}]$, we have $w_k(t) \in w_k(t_k) + \mathcal{C}^{2\delta/3}$ and $\inf_{t \in [t_k, T^\epsilon_{\sigma_{\text{init}}}]} \|w_k(t)\|_2 \geq \text{dist}(\mathbf{0}, w_k(t_k) + \mathcal{C}^{2\delta/3}) = \text{dist}(-\mathcal{C}^{2\delta/3}, w_k(t_k)) = \|w_k(t_k)\|_2 \text{dist}(-\mathcal{C}^{2\delta/3}, \frac{w_k(t_k)}{\|w_k(t_k)\|_2}) \geq \|w_k(t_k)\|_2 \cdot \bar{d}' \geq \frac{\bar{d}\bar{d}'\sigma_{\text{init}}}{2}$. We can apply the same argument for those $k$ with $\bar{a}_k < 0$, and finally we can conclude that $\|w_k(t)\|_2 \geq \bar{D}\sigma_{\text{init}}$ for all $t \in [0, T^\epsilon_{\sigma_{\text{init}}}]$ and $k \in [m]$, where $\bar{D} := \max\{1, \bar{d}'\}\frac{\bar{d}}{2}$.

**Convergence to $\mathcal{C}^{\delta/3}$.** For $c \geq 0$ and $i \in [n]$ define $\Gamma_i^c(w) := \langle w, y_i x_i \rangle - c \|w\|_2$. For all $k \in [m]$ and $t \leq T^\epsilon_{\sigma_{\text{init}}}$, it holds that

$$\frac{\mathrm{d}\Gamma_i^{\delta/3}(s_k w_k)}{\mathrm{d}t} = \left\langle \frac{\mathrm{d}w_k}{\mathrm{d}t}, s_k y_i x_i \right\rangle - (\delta/3) \left\langle \frac{\mathrm{d}w_k}{\mathrm{d}t}, \frac{w_k}{\|w_k\|_2} \right\rangle$$

$$\geq (2\delta/3) \left\| \frac{\mathrm{d}w_k}{\mathrm{d}t} \right\|_2 - (\delta/3) \left\| \frac{\mathrm{d}w_k}{\mathrm{d}t} \right\|_2 = (\delta/3) \left\| \frac{\mathrm{d}w_k}{\mathrm{d}t} \right\|_2,$$

where the inequality is because $\frac{\mathrm{d}w_k}{\mathrm{d}t} \subseteq a_k \mathcal{H}^\epsilon \subseteq a_k \mathcal{C}^{2\delta/3}$ and $\langle \frac{\mathrm{d}w_k}{\mathrm{d}t}, \frac{w_k}{\|w_k\|_2} \rangle \leq \|\frac{\mathrm{d}w_k}{\mathrm{d}t}\|_2$.

Let $h_{\min} := \inf_{w \in \mathcal{H}^\epsilon} \|w\|_2 = \min_{w \in \mathcal{H}^\epsilon} \|w\|_2 > 0$. Note that $|a_k(t)| = \|w_k(t)\|_2 \geq \bar{D}\sigma_{\text{init}}$. Using $\frac{\mathrm{d}w_k}{\mathrm{d}t} \subseteq a_k \mathcal{H}^\epsilon$ again we have

$$\frac{\mathrm{d}\Gamma_i^{\delta/3}(s_k w_k(t))}{\mathrm{d}t} \geq \frac{\delta h_{\min} \bar{D}\sigma_{\text{init}}}{3}.$$

Thus if we pick

$$T_0 := \max\left\{ \frac{3}{\delta h_{\min} \bar{D}} \max_{i \in [n], k \in [m]} \{-\Gamma_i^{\delta/3}(s_k \bar{w}_k)\}, 0 \right\}$$

and set $\sigma_{\text{init}}^{\max} \leq e^{-T_0} \frac{\sqrt{\frac{\epsilon}{m}}}{\|\bar{\theta}\|_{\text{M}}}$ then it holds that $T_0 \leq T^\epsilon_{\sigma_{\text{init}}}$ for all $\sigma_{\text{init}} \leq \sigma_{\text{init}}^{\max}$ and that

$$\Gamma_i^{\delta/3}(s_k w_k(T_0)) \geq \frac{\delta h_{\min} \bar{D}\sigma_{\text{init}} T_0}{3} + \Gamma_i^{\delta/3}(s_k w_k(0))$$

$$\geq \sigma_{\text{init}} \left( \frac{\delta h_{\min} \bar{D} T_0}{3} + \Gamma_i^{\delta/3}(s_k \bar{w}_k(0)) \right)$$

$$\geq 0,$$

which implies (62).

Finally, by (63), it suffices to pick $A = e^{-T_0} \min_{k \in [m]} \|\bar{w}_k\|_2$ and $B = e^{T_0} \max_{k \in [m]} \|\bar{w}_k\|_2$. $\qquad \square$

### G.3.2 Proof of Lemma G.5

Note that $G(w) = \langle w, \mu^+ \rangle$ for $w \in \mathcal{C}^{\delta/3}$ and $G(w) = \langle w, \mu^- \rangle$ for $w \in -\mathcal{C}^{\delta/3}$. Similar to the first-phase analysis to the symmetric case, we use $\tilde{\varphi}(\tilde{\theta}_0, t)$ to denote the trajectory of gradient flow on $\tilde{\mathcal{L}}$:

$$\tilde{\mathcal{L}}(\theta) := \ell(0) + \sum_{k \in [m]} a_k G(w_k).$$

Throughout this subsection, we will set $T_0$ and $\epsilon$ as defined in the proof of Lemma G.4, and therefore by Lemma G.4, we know there is $\sigma_{\text{init}}^{\max} > 0$, s.t. $a_k(T_0) w_k(T_0) \in \mathcal{C}^{\delta/3}$ for all $\sigma_{\text{init}} \leq \sigma_{\text{init}}^{\max}$. This

means the dynamics of $\tilde{\boldsymbol{\theta}}(t) = (\tilde{\boldsymbol{w}}_1(t), \ldots, \tilde{\boldsymbol{w}}_m(t), \tilde{a}_1(t), \ldots, \tilde{a}_m(t)) = \tilde{\varphi}(\boldsymbol{\theta}(T_0), t - T_0)$ can be described by linear ODE for $T_0 \le t \le T_{\sigma_{\text{init}}}^{\epsilon}$.

$$\text{If } \bar{a}_k > 0: \qquad \frac{\mathrm{d}\tilde{\boldsymbol{w}}_k}{\mathrm{d}t} = \tilde{a}_k \boldsymbol{\mu}^+, \qquad \frac{\mathrm{d}\tilde{a}_k}{\mathrm{d}t} = \langle \tilde{\boldsymbol{w}}_k, \boldsymbol{\mu}^+ \rangle;$$

$$\text{If } \bar{a}_k < 0: \qquad \frac{\mathrm{d}\tilde{\boldsymbol{w}}_k}{\mathrm{d}t} = \tilde{a}_k \boldsymbol{\mu}^-, \qquad \frac{\mathrm{d}\tilde{a}_k}{\mathrm{d}t} = \langle \tilde{\boldsymbol{w}}_k, \boldsymbol{\mu}^- \rangle.$$

Let $\boldsymbol{M}_+ := \begin{bmatrix} \boldsymbol{0} & \boldsymbol{\mu}^+ \\ (\boldsymbol{\mu}^+)^\top & 0 \end{bmatrix}$ and $\boldsymbol{M}_- := \begin{bmatrix} \boldsymbol{0} & \boldsymbol{\mu}^- \\ (\boldsymbol{\mu}^-)^\top & 0 \end{bmatrix}$. The largest eigenvalues for $\boldsymbol{M}_+$ and $\boldsymbol{M}_-$ are $\lambda_0^+ := \|\boldsymbol{\mu}^+\|_2$ and $\lambda_0^- := \|\boldsymbol{\mu}^-\|_2$ respectively. Then the above linear ODE can be solved as

$$\text{If } \bar{a}_k > 0: \qquad \begin{bmatrix} \tilde{\boldsymbol{w}}_k(T_0 + t) \\ \tilde{a}_k(T_0 + t) \end{bmatrix} = \exp(t\boldsymbol{M}_+) \begin{bmatrix} \boldsymbol{w}_k(T_0) \\ a_k(T_0) \end{bmatrix}; \tag{64}$$

$$\text{If } \bar{a}_k < 0: \qquad \begin{bmatrix} \tilde{\boldsymbol{w}}_k(T_0 + t) \\ \tilde{a}_k(T_0 + t) \end{bmatrix} = \exp(t\boldsymbol{M}_-) \begin{bmatrix} \boldsymbol{w}_k(T_0) \\ a_k(T_0) \end{bmatrix}. \tag{65}$$

**Lemma G.6.** *Let* $\tilde{\boldsymbol{\theta}}(t) = \tilde{\varphi}(\boldsymbol{\theta}(T_0), t - T_0)$. *Then for all* $T_0 \le t \le T_{\sigma_{\text{init}}}^{\epsilon}$, *it holds that*

$$\|\tilde{\boldsymbol{\theta}}(t)\|_{\mathrm{M}} \le \exp((t - T_0)\lambda_0^+)\|\tilde{\boldsymbol{\theta}}(T_0)\|_{\mathrm{M}}.$$

*Proof.* By Assumption A.3, we have $\lambda_0^+ > \lambda_0^-$. By definition and Cauchy-Schwartz inequality,

$$\left\|\frac{\mathrm{d}\tilde{\boldsymbol{w}}_k}{\mathrm{d}t}\right\|_2 \le \lambda_0^+ |\tilde{a}_k|, \qquad \left|\frac{\mathrm{d}\tilde{a}_k}{\mathrm{d}t}\right| \le \lambda_0^+ \|\tilde{\boldsymbol{w}}_k\|_2.$$

So we have $\|\tilde{\boldsymbol{\theta}}(t)\|_{\mathrm{M}} \le \|\boldsymbol{\theta}(T_0)\|_{\mathrm{M}} + \int_{T_0}^{T_{\sigma_{\text{init}}}^{\epsilon}} \lambda_0^+ \|\tilde{\boldsymbol{\theta}}(\tau)\|_{\mathrm{M}} \mathrm{d}\tau$. Then we can finish the proof by Grönwall's inequality (12). $\square$

**Lemma G.7.** *For* $\boldsymbol{\theta}(T_0)$ *with* $|a_k(T_0)| = \|\boldsymbol{w}_k(T_0)\|_2$ *and* $a_k(T_0)\boldsymbol{w}_k(T_0) \in \mathcal{C}^{\delta/3}$, *we have*

$$\|\boldsymbol{\theta}(t) - \tilde{\varphi}(\boldsymbol{\theta}(T_0), t - T_0)\|_{\mathrm{M}} \le \frac{4m\|\boldsymbol{\theta}(T_0)\|_{\mathrm{M}}^3}{\lambda_0^+} \exp(3\lambda_0(t - T_0)),$$

*for all* $T_0 \le t \le \frac{1}{\lambda_0^+} \ln \frac{\sqrt{\min\{\epsilon, \lambda_0^+\}}}{\sqrt{4m}\|\boldsymbol{\theta}(T_0)\|_{\mathrm{M}}}$.

*Proof.* Let $\tilde{\boldsymbol{\theta}}(t) = \tilde{\varphi}(\boldsymbol{\theta}(T_0), t - T_0)$. Let

$$t_0 := \min\{T_{\sigma_{\text{init}}}^{\epsilon}, \inf\{t \ge T_0 : \|\boldsymbol{\theta}(t)\|_{\mathrm{M}} \ge 2\|\boldsymbol{\theta}(T_0)\|_{\mathrm{M}} \exp(\lambda_0^+(t - T_0))\}.$$

and it holds that $\forall T_0 \le t \le t_0$, all neurons of $\tilde{\boldsymbol{\theta}}(t), \boldsymbol{\theta}(t)$ are either in $\mathcal{C}^{\delta/3}$ or $-\mathcal{C}^{\delta/3}$, thus $\tilde{\boldsymbol{\theta}}(t), \boldsymbol{\theta}(t)$ are in the same differentiable region of $\tilde{\mathcal{L}}$. By Corollary B.13, the following holds for a.e. $t \ge 0$,

$$\left\|\frac{\mathrm{d}\boldsymbol{\theta}}{\mathrm{d}t} - \frac{\mathrm{d}\tilde{\boldsymbol{\theta}}}{\mathrm{d}t}\right\|_{\mathrm{M}} \le \sup\left\{\|\boldsymbol{\delta} - \nabla\tilde{\mathcal{L}}(\boldsymbol{\theta})\|_{\mathrm{M}} : \boldsymbol{\delta} \in \partial^\circ \mathcal{L}(\boldsymbol{\theta})\right\} + \|\nabla\tilde{\mathcal{L}}(\boldsymbol{\theta}) - \nabla\tilde{\mathcal{L}}(\tilde{\boldsymbol{\theta}})\|_{\mathrm{M}}$$

$$\le m\|\boldsymbol{\theta}(t)\|_{\mathrm{M}}^3 + \lambda_0^+ \|\boldsymbol{\theta} - \tilde{\boldsymbol{\theta}}\|_{\mathrm{M}}.$$

Then we can argue as the proof for Lemma D.2 to show that

$$\|\boldsymbol{\theta}(t) - \tilde{\boldsymbol{\theta}}(t)\|_{\mathrm{M}} \le \frac{4m\|\boldsymbol{\theta}(T_0)\|_{\mathrm{M}}^3}{\lambda_0} \exp(3\lambda_0^+(t - T_0))$$

for all $t \in [T_0, t_0]$. If $t_0 < T_0 + \frac{1}{2\lambda_0^+} \ln \frac{\min\{\lambda_0^+, \epsilon\}}{4m\|\boldsymbol{\theta}(T_0)\|_{\mathrm{M}}^2}$, then for all $T_0 \le t \le t_0$, we have

$$\|\boldsymbol{\theta}(t)\|_{\mathrm{M}} \le \|\boldsymbol{\theta}(T_0)\|_{\mathrm{M}} \sqrt{\frac{\min\{\lambda_0^+, \epsilon\}}{4m\|\boldsymbol{\theta}(T_0)\|_{\mathrm{M}}^2}} < \sqrt{\frac{\epsilon}{m}},$$

which implies that $t_0 < T^\epsilon_{\sigma_{\text{init}}}$ by definition of $T^\epsilon_{\sigma_{\text{init}}}$. Moreover,

$$\|\boldsymbol{\theta}(t)\|_{\mathrm{M}} \leq \|\tilde{\boldsymbol{\theta}}(t)\|_{\mathrm{M}} + \frac{4m\|\boldsymbol{\theta}(T_0)\|_{\mathrm{M}}^3}{\lambda_0^+} \exp(3\lambda_0^+(t - T_0))$$

$$\leq \|\tilde{\boldsymbol{\theta}}(t)\|_{\mathrm{M}} + \frac{4m\|\boldsymbol{\theta}(T_0)\|_{\mathrm{M}}^2}{\lambda_0^+} \exp(2\lambda_0^+(t_0 - T_0)) \cdot \|\boldsymbol{\theta}(T_0)\|_{\mathrm{M}} \exp(\lambda_0^+(t - T_0))$$

$$< \|\tilde{\boldsymbol{\theta}}(t)\|_{\mathrm{M}} + \|\boldsymbol{\theta}(T_0)\|_{\mathrm{M}} \exp(\lambda_0^+(t - T_0)).$$

By Lemma G.6, $\|\tilde{\boldsymbol{\theta}}(t)\|_{\mathrm{M}} \leq \|\boldsymbol{\theta}(T_0)\|_{\mathrm{M}} \exp(\lambda_0^+(t - T_0))$. So $\|\boldsymbol{\theta}(t)\|_{\mathrm{M}} < 2\|\boldsymbol{\theta}(T_0)\|_{\mathrm{M}} \exp(\lambda_0^+(t - T_0))$ for all $T_0 \leq t \leq t_0$, which contradicts to the definition of $t_0$. Therefore, $t_0 \geq \frac{1}{2\lambda_0^+} \ln \frac{\min\{\epsilon, \lambda_0^+\}}{4m\|\boldsymbol{\theta}(T_0)\|_{\mathrm{M}}^2} = \frac{1}{\lambda_0^+} \ln \frac{\sqrt{\min\{\epsilon, \lambda_0^+\}}}{\sqrt{4m}\|\boldsymbol{\theta}(T_0)\|_{\mathrm{M}}}$. $\square$

*Proof for Lemma G.5.* Let $r_{\max} := \frac{\sqrt{\min\{\lambda_0^+, \epsilon\}}}{2}$ and $C_1 := \sigma_{\text{init}} r_{\max}^{-3}$. We only need to prove the statements for all $\sigma_{\text{init}} < \sigma_{\text{init}}^{\max} = C_1 r_{\max}^3$.

We fix a pair of $\sigma_{\text{init}} < \sigma_{\text{init}}^{\max}$ and $r < r_{\max}$ satisfying $\sigma_{\text{init}} < C_1 r^3$. For convenience, we use $\bar{\boldsymbol{b}}, T_1$ to denote $\bar{\boldsymbol{b}}(\sigma_{\text{init}}), T_1(\sigma_{\text{init}}, r)$ for short.

Let $\boldsymbol{\theta}(t) = \varphi(\sigma_{\text{init}}\bar{\boldsymbol{\theta}}_0, t)$. It is easy to see that the prerequisites of Lemma G.4 are satisfied with probability 1. Below we only focus on the case where the prerequisites of Lemma G.4 are satisfied. Let $T_0, \sigma_{\text{init}}^{\max}, A, B$ be the constants from Lemma G.4. Let $\tilde{\boldsymbol{\theta}}(t) = \tilde{\varphi}(\boldsymbol{\theta}(T_0), t - T_0)$.

For $\bar{a}_k > 0$, we define

$$\bar{b}_k := \frac{\langle \boldsymbol{w}_k(T_0), \bar{\boldsymbol{\mu}}^+ \rangle + a_k(T_0)}{2\sqrt{m}\sigma_{\text{init}}},$$

and for $\bar{a}_k < 0$, we define

$$\bar{b}_k := \frac{\langle \boldsymbol{w}_k(T_0), \bar{\boldsymbol{\mu}}^- \rangle + a_k(T_0)}{2(\sqrt{m}\sigma_{\text{init}})^{1-\kappa}}.$$

$r \leq r_{\max}$ and $\sigma_{\text{init}} < \sigma_{\text{init}}^{\max}$.

**Proof for Item 1.** By Lemma G.7, we have

$$\|\boldsymbol{\theta}(T_0 + T_1) - \tilde{\boldsymbol{\theta}}(T_0 + T_1)\|_{\mathrm{M}} \leq \frac{4m\|\boldsymbol{\theta}(T_0)\|_{\mathrm{M}}^3}{\lambda_0^+} \exp(3\lambda_0^+ T_1) = \frac{4\|\boldsymbol{\theta}(T_0)\|_{\mathrm{M}}^3}{\lambda_0^+ \sqrt{m}\sigma_{\text{init}}^3} r^3. \quad (66)$$

Now we turn to characterize $\tilde{\boldsymbol{\theta}}(T_0 + T_1)$. Note that $\bar{\boldsymbol{\mu}}_2^+ := \frac{1}{\sqrt{2}}[\bar{\boldsymbol{\mu}}^+, 1]^\top$ and $\bar{\boldsymbol{\mu}}_2^- := \frac{1}{\sqrt{2}}[\bar{\boldsymbol{\mu}}^-, 1]^\top$ are the top eigenvectors of $\boldsymbol{M}_+$ and $\boldsymbol{M}_-$ respectively. Let $\kappa := 1 - \frac{\|\boldsymbol{\mu}^-\|_2}{\|\boldsymbol{\mu}^+\|_2}$. Recall that $T_1 := \frac{1}{\lambda_0^+} \ln \frac{r}{\sqrt{m}\sigma_{\text{init}}}$. Then for $\bar{a}_k > 0$, we have

$$\exp(T_1\lambda_0^+)\bar{\boldsymbol{\mu}}_2^+(\bar{\boldsymbol{\mu}}_2^+)^\top \begin{bmatrix} \boldsymbol{w}_k(T_0) \\ a_k(T_0) \end{bmatrix} = \left(\frac{r}{\sqrt{m}\sigma_{\text{init}}}\right) \bar{\boldsymbol{\mu}}_2^+(\bar{\boldsymbol{\mu}}_2^+)^\top \begin{bmatrix} \boldsymbol{w}_k(T_0) \\ a_k(T_0) \end{bmatrix} = r\bar{b}_k \begin{bmatrix} \bar{\boldsymbol{\mu}}^+ \\ 1 \end{bmatrix},$$

where the last equality is by definition of $\bar{b}_k$. Similarly for $\bar{a}_k > 0$, we have

$$\exp(T_1\lambda_0^-)\bar{\boldsymbol{\mu}}_2^-(\bar{\boldsymbol{\mu}}_2^-)^\top \begin{bmatrix} \boldsymbol{w}_k(T_0) \\ a_k(T_0) \end{bmatrix} = \left(\frac{r}{\sqrt{m}\sigma_{\text{init}}}\right)^{1-\kappa} \bar{\boldsymbol{\mu}}_2^-(\bar{\boldsymbol{\mu}}_2^-)^\top \begin{bmatrix} \boldsymbol{w}_k(T_0) \\ a_k(T_0) \end{bmatrix} = r^{1-\kappa}\bar{b}_k \begin{bmatrix} \bar{\boldsymbol{\mu}}^- \\ 1 \end{bmatrix}.$$

Combining these with (64) and (65), then for $\bar{a}_k > 0$ we have

$$\left\| \begin{bmatrix} \tilde{\boldsymbol{w}}_k(T_0 + t) \\ \tilde{a}_k(T_0 + t) \end{bmatrix} - r\bar{b}_k \begin{bmatrix} \bar{\boldsymbol{\mu}}^+ \\ 1 \end{bmatrix} \right\|_2 \leq \left\| \left(\exp(T_1\boldsymbol{M}_+) - \exp(T_1\lambda_0^+)\bar{\boldsymbol{\mu}}_2^+(\bar{\boldsymbol{\mu}}_2^+)^\top\right) \begin{bmatrix} \boldsymbol{w}_k(T_0) \\ a_k(T_0) \end{bmatrix} \right\|_2$$

$$\leq \left\| \begin{bmatrix} \boldsymbol{w}_k(T_0) \\ a_k(T_0) \end{bmatrix} \right\|_2$$

$$\leq \sqrt{2}\|\boldsymbol{\theta}(T_0)\|_{\mathrm{M}}.$$

and for $\bar{a}_k < 0$ we have

$$\left\| \begin{bmatrix} \tilde{\boldsymbol{w}}_k(T_0 + t) \\ \tilde{a}_k(T_0 + t) \end{bmatrix} - r^{1-\kappa} \bar{b}_k \begin{bmatrix} \bar{\boldsymbol{\mu}}^- \\ 1 \end{bmatrix} \right\|_2 \leq \left\| \left( \exp(T_1 \boldsymbol{M}_-) - \exp(T_1 \lambda_0^-) \bar{\boldsymbol{\mu}}_2^- (\bar{\boldsymbol{\mu}}_2^-)^\top \right) \begin{bmatrix} \boldsymbol{w}_k(T_0) \\ a_k(T_0) \end{bmatrix} \right\|_2$$

$$\leq \left\| \begin{bmatrix} \boldsymbol{w}_k(T_0) \\ a_k(T_0) \end{bmatrix} \right\|_2$$

$$\leq \sqrt{2} \|\boldsymbol{\theta}(T_0)\|_{\mathrm{M}}.$$

Then by definition of $\Delta\boldsymbol{\theta}$ and (66), we have

$$\|\Delta\boldsymbol{\theta}\|_{\mathrm{M}} \leq \frac{4\|\boldsymbol{\theta}(T_0)\|_{\mathrm{M}}^3}{\lambda_0^+ \sqrt{m} \sigma_{\mathrm{init}}^3} r^3 + \sqrt{2} \|\boldsymbol{\theta}(T_0)\|_{\mathrm{M}}.$$

Applying the upper bound $\|\boldsymbol{w}_k(T_0)\|_2 \leq B\sigma_{\mathrm{init}}$ from Lemma G.4, we then have

$$\|\Delta\boldsymbol{\theta}\|_{\mathrm{M}} \leq \frac{4B^3 \sigma_{\mathrm{init}}^3}{\lambda_0^+ \sqrt{m} \sigma_{\mathrm{init}}^3} r^3 + \sqrt{2} B\sigma_{\mathrm{init}} = \frac{4B^3}{\lambda_0^+ \sqrt{m}} r^3 + \sqrt{2} B\sigma_{\mathrm{init}},$$

Finally, recalling that $\sigma_{\mathrm{init}} \leq C_1 r^3$, we can conclude that $\|\Delta\boldsymbol{\theta}\|_{\mathrm{M}} \leq C_2 r^3$, where $C_2 := \frac{4B^3}{\lambda_0^+ \sqrt{m}} + \sqrt{2} BC_1$.

**Item 2.** Now it only remains to lower and upper bound $|\bar{b}_k|$. By Lemma G.4, $a_k(T_0) \boldsymbol{w}_k(T_0) \in \mathcal{C}^{\delta/3}$. Then $\mathrm{sgn}(a_k(T_0))\langle \boldsymbol{w}_k(T_0), \bar{\boldsymbol{\mu}}^+ \rangle \geq 0$ and thus

if $\bar{a}_k > 0$ : $\langle \boldsymbol{w}_k(T_0), \bar{\boldsymbol{\mu}}^+ \rangle + a_k(T_0) \in \left[ \|\boldsymbol{w}_k(T_0)\|_2, 2 \cdot \|\boldsymbol{w}_k(T_0)\|_2 \right] \subseteq [A\sigma_{\mathrm{init}}, 2B\sigma_{\mathrm{init}}];$

if $\bar{a}_k < 0$ : $\langle \boldsymbol{w}_k(T_0), \bar{\boldsymbol{\mu}}^- \rangle + a_k(T_0) \in \left[ -2 \cdot \|\boldsymbol{w}_k(T_0)\|_2, -\|\boldsymbol{w}_k(T_0)\|_2 \right] \subseteq [-2B\sigma_{\mathrm{init}}, -A\sigma_{\mathrm{init}}].$

Then for every $\bar{b}_k$,

$$\text{if } \bar{a}_k > 0 : \qquad |\bar{b}_k| = \frac{|\langle \boldsymbol{w}_k(T_0), \bar{\boldsymbol{\mu}}^+ \rangle + a_k(T_0)|}{2\sqrt{m}\sigma_{\mathrm{init}}} \in \left[ \frac{A}{2\sqrt{m}}, \frac{B}{\sqrt{m}} \right];$$

$$\text{if } \bar{a}_k < 0 : \qquad |\bar{b}_k| = \frac{|\langle \boldsymbol{w}_k(T_0), \bar{\boldsymbol{\mu}}^- \rangle + a_k(T_0)|}{2(\sqrt{m}\sigma_{\mathrm{init}})^{1-\kappa}} \in \left[ \frac{\sigma_{\mathrm{init}}^\kappa A}{2m^{(1-\kappa)/2}}, \frac{\sigma_{\mathrm{init}}^\kappa B}{m^{(1-\kappa)/2}} \right].$$

Letting $\bar{A} := \frac{A}{2m^{(1-\kappa)/2}}$ and $\bar{B} := \frac{B}{m^{(1-\kappa)/2}}$ completes the proof. $\qquad\square$

### G.4 Phase II

As shown in our analysis for Phase I, if the intialization scale is small, the weight vectors of neurons with $\bar{a}_k > 0$ move towards the direction of $\bar{\boldsymbol{\mu}}^+$, and all the other neurons are negligible. Now we show that the dynamic of $\boldsymbol{\theta}(t)$ is close to that of a one-neuron dynamic in a similar manner as we do for the symmetric case.

First we slightly extend the definition of embedding. For $\hat{\boldsymbol{\theta}} = (\hat{\boldsymbol{w}}_1, \hat{\boldsymbol{w}}_2, \hat{a}_1, \hat{a}_2)$ and an embedding vector $\boldsymbol{b} \in \mathbb{R}^m$, we say that $\boldsymbol{b}$ is compatible with $\hat{\boldsymbol{\theta}}$ if the following holds:

1. If $b_+ = 0$, then $\|\hat{\boldsymbol{w}}_1\|_2 = |\hat{a}_1| = 0$;
2. If $b_- = 0$, then $\|\hat{\boldsymbol{w}}_2\|_2 = |\hat{a}_2| = 0$.

When $\boldsymbol{b}$ is compatible with $\hat{\boldsymbol{\theta}}$, we define the (exact) embedding from two-neuron into $m$-neuron neural nets as $\pi_{\boldsymbol{b}}(\hat{\boldsymbol{\theta}}) := (\boldsymbol{w}_1, \ldots, \boldsymbol{w}_m, a_1, \ldots, a_m)$, where

$$a_k = \begin{cases} \frac{b_k}{b_+} \hat{a}_1, & \text{if } b_k > 0 \\ \frac{b_k}{b_-} \hat{a}_2, & \text{if } b_k < 0, \\ \boldsymbol{0}, & \text{if } b_k = 0 \end{cases} \qquad \boldsymbol{w}_k = \begin{cases} \frac{b_k}{b_+} \hat{\boldsymbol{w}}_1, & \text{if } b_k > 0 \\ \frac{b_k}{b_-} \hat{\boldsymbol{w}}_2, & \text{if } b_k < 0 . \\ \boldsymbol{0}, & \text{if } b_k = 0 \end{cases}$$

One can easily show that Lemma 5.3 continue to hold when $\boldsymbol{b}$ is compatible with $\hat{\boldsymbol{\theta}}$.

**Lemma G.8.** *Let $\bar{b}(\sigma_{\text{init}})$ be the same vector as in the statement of Lemma G.5. Let $T_{12}(\sigma_{\text{init}}) := T_0 + \frac{1}{\lambda_0^+} \ln \frac{1}{\sqrt{m}\sigma_{\text{init}}}$ and $T_2(r) := \frac{1}{\lambda_0^+} \ln \frac{1}{r}$. For width $m \geq 1$, the following statements hold with probability $1 - 2^{-m}$ over the random draw of $\bar{\theta}_0 = (\bar{w}_1, \ldots, \bar{w}_m, \bar{a}_1, \ldots, \bar{a}_m) \sim \mathcal{D}_{\text{init}}(1)$. Let $\sigma_1, \sigma_2, \ldots$ be any sequence of initialization scales so that $\sigma_j$ converges to $0$ as $j \to +\infty$ and the limit $\hat{b} := \lim_{j \to +\infty} \bar{b}(\sigma_j)$ exists.*

1. *$\hat{b}_+ > 0$ and $\hat{b}_- = 0$;*

2. *For the two-neuron dynamics starting with rescaled initialization in the direction of $\hat{\theta} := (\hat{b}_+ \bar{\mu}^+, \mathbf{0}, \hat{b}_+, 0)$, the following limit exists for all $t \geq 0$,*

$$\tilde{\theta}(t) := \lim_{r \to 0} \varphi\left(r\hat{\theta}, T_2(r) + t\right) \neq \mathbf{0}; \tag{67}$$

3. *For the $m$-neuron dynamics of $\theta_j(t)$ with initialization scale $\sigma_{\text{init}} = \sigma_j$, the following holds for all $t \geq 0$,*

$$\lim_{j \to \infty} \theta_j\left(T_{12}(\sigma_j) + t\right) = \pi_{\hat{b}}(\tilde{\theta}(t)). \tag{68}$$

*Proof.* The proof is similar to Lemma 5.4 for the symmetric case. Apply Theorem E.4 and then the lemma is straightforward. $\qquad\square$

### G.5 Phase III

In Phase III, we show that the dynamic of $\theta(t)$ converges to the same classifier as the one-neuron dynamic.

Let $\mathcal{S}^+ := \arg\min_{i \in [n]} \left\{ y_i \langle w^+, x_i^+ \rangle \right\} \subseteq [n]$. Let $\Delta^{h-1} = \left\{ p \in \mathbb{R}^h : \sum_{i \in [h]} p_i = 1, p_i \geq 1 \right\}$ be the probability simplex. Let $\Lambda^+ := \left\{ \lambda \in \Delta^{n-1} : \lambda_i = 0, \forall i \notin \mathcal{S}^+ \right\}$.

The theorem below characterizes the solution found by the one-neuron dynamic.

**Theorem G.9.** *Under Assumption 3.2, for $m = 1$, if initially $a_1 = \|w_1\|_2$, $\langle w_1, w^* \rangle > 0$, then $\theta(t)$ directionally converges to the following global-max-margin direction,*

$$\lim_{t \to +\infty} \frac{\theta(t)}{\|\theta(t)\|_2} = \frac{1}{\sqrt{2}}(w^+, 1).$$

*Proof.* By Theorem B.19, $\mathcal{L}(\theta(t)) \to 0$. Then by Theorem 3.1, $\frac{\theta(t)}{\|\theta(t)\|_2}$ converges along a KKT-margin direction. Combining this with Lemma B.17, we know that this direction must has the form $\frac{1}{\sqrt{2}}(\bar{w}, 1)$ for some $\bar{w} \in \mathbb{S}^{d-1}$.

By Definition B.8, $y_i \cdot \frac{1}{2}\phi(\langle \bar{w}, x_i \rangle) > 0$ and $\bar{w}$ can be expressed by a convex combination of $y_i \phi'(\langle \bar{w}, x_i \rangle) x_i$ among $i \in \arg\min\{\frac{1}{2}\phi(\langle \bar{w}, x_i \rangle)\}$. Equivalently. we know that $y_i \langle \bar{w}, x_i^+ \rangle$ and $\bar{w}$ can be expressed by a convex combination of $y_i x_i^+$ among $i \in \mathcal{S}^+$. Then the only possibility is $\bar{w} = w^+$. $\qquad\square$

Now we turn to analyze the trajectory of $\theta(t)$ on $m$-neuron neural net. First we prove the following lemma, then we prove Theorem G.11 for local-max-margin directions.

**Lemma G.10.** *Let $\Theta_- := \{\theta = (w_1, \ldots, w_m, a_1, \ldots, a_m) : m \geq 1, a_k \leq 0\}$. Then we have the following characterization for the global maximum of the normalized margin on the dataset $\{(x_i, y_i) : i \in \mathcal{S}^+\}$:*

$$\sup_{\theta \in \Theta_-} \left\{ \frac{\min_{i \in \mathcal{S}^+} q_i(\theta)}{\|\theta\|_2^2} \right\} = \inf_{\lambda \in \Lambda^+} \sup_{u \in \mathbb{S}^{d-1}} \left\{ -\frac{1}{2} \sum_{i \in [n]} \lambda_i y_i \phi(\langle u, x_i \rangle) \right\}$$

*Proof.* The proof is inspired by Chizat and Bach (2020, Proposition 12). By Lemma B.9, the maximum normalized margin is attained when $|a_k| = \|w_k\|_2$ for all $k \in [m]$. Note that we can

rewrite each neuron output $a_k\phi(\langle \boldsymbol{w}_k, \boldsymbol{x}_i\rangle)$ as $-a_k^2\phi(\langle \boldsymbol{w}_k/\|\boldsymbol{w}_k\|_2, \boldsymbol{x}_i\rangle)$ for any such solution, and it is easy to see $\sum_{k\in[m]} a_k^2 = \frac{1}{2}\|\boldsymbol{\theta}\|_2^2$. Let $\mathcal{V}$ be the set of probability distributions supported on finitely many points of $\mathbb{S}^{d-1}$. Then

$$\sup_{\boldsymbol{\theta}\in\Theta_-}\left\{\frac{\min_{i\in\mathcal{S}^+} q_i(\boldsymbol{\theta})}{\|\boldsymbol{\theta}\|_2^2}\right\} = \sup_{\substack{m\geq 1, \boldsymbol{p}\in\Delta^{m-1}\\ \boldsymbol{u}_1,\ldots,\boldsymbol{u}_m\in\mathbb{S}^{d-1}}} \min_{i\in\mathcal{S}^+}\left\{-\frac{1}{2}y_i\sum_{k\in[m]}p_k\phi(\langle \boldsymbol{u}_k, \boldsymbol{x}_i\rangle)\right\}$$

$$= \sup_{\nu\in\mathcal{V}}\min_{i\in\mathcal{S}^+}\mathbb{E}_{\boldsymbol{u}\sim\nu}\left[-\frac{1}{2}y_i\phi(\langle \boldsymbol{u}, \boldsymbol{x}_i\rangle)\right].$$

By minimax theorem, we can swap the order between sup and min in the following way:

$$\sup_{\nu\in\mathcal{V}}\min_{i\in\mathcal{S}^+}\mathbb{E}_{\boldsymbol{u}\sim\nu}\left[-\frac{1}{2}y_i\phi(\langle \boldsymbol{u}, \boldsymbol{x}_i\rangle)\right] = \sup_{\nu\in\mathcal{V}}\inf_{\boldsymbol{\lambda}\in\Lambda^+}\mathbb{E}_{\boldsymbol{u}\sim\nu}\left[-\frac{1}{2}\sum_{i\in\mathcal{S}^+}\lambda_i y_i\phi(\langle \boldsymbol{u}, \boldsymbol{x}_i\rangle)\right]$$

$$= \inf_{\boldsymbol{\lambda}\in\Lambda^+}\sup_{\nu\in\mathcal{V}}\mathbb{E}_{\boldsymbol{u}\sim\nu}\left[-\frac{1}{2}\sum_{i\in\mathcal{S}^+}\lambda_i y_i\phi(\langle \boldsymbol{u}, \boldsymbol{x}_i\rangle)\right]$$

$$= \inf_{\boldsymbol{\lambda}\in\Lambda^+}\sup_{\boldsymbol{u}\in\mathbb{S}^{d-1}}\left\{-\frac{1}{2}\sum_{i\in\mathcal{S}^+}\lambda_i y_i\phi(\langle \boldsymbol{u}, \boldsymbol{x}_i\rangle)\right\},$$

which proves the claim. $\qquad\square$

**Theorem G.11.** *Let* $\hat{\boldsymbol{\theta}} := (\frac{1}{\sqrt{2}}\boldsymbol{w}^+, \frac{1}{\sqrt{2}}, \boldsymbol{0}, 0)$ *and* $P$ *be a non-empty subset of* $[m]$. *Let* $\bar{\mathcal{Q}}$ *be the following subset of* $\mathbb{S}^{D-1}$:

$$\bar{\mathcal{Q}} := \{\boldsymbol{\theta} = (\boldsymbol{w}_1,\ldots,\boldsymbol{w}_m, a_1,\ldots, a_m)\in\mathbb{S}^{D-1} : a_k\geq 0 \text{ for all } k\in P \text{ and } a_k\leq 0 \text{ otherwise}\}.$$

*For any embedding vect* $\boldsymbol{b}$ *be an embedding vector satisfying the following:*

- $\boldsymbol{b}$ *is compatible with* $\hat{\boldsymbol{\theta}}$;
- $b_k > 0$ *for all* $k\in P$;
- $b_k = 0$ *for all* $k\notin P$;

*the following statements are true under Assumption A.5,*

1. $\pi_{\boldsymbol{b}}(\hat{\boldsymbol{\theta}})$ *is a local maximizer of* $\gamma(\boldsymbol{\theta})$ *among* $\boldsymbol{\theta}\in\bar{\mathcal{Q}}$;

2. *If* $\boldsymbol{\theta}\in\bar{\mathcal{Q}}$ *has the same normalized margin as* $\pi_{\boldsymbol{b}}(\hat{\boldsymbol{\theta}})$ *and* $\boldsymbol{\theta}$ *is sufficiently close to* $\pi_{\boldsymbol{b}}(\hat{\boldsymbol{\theta}})$, *then* $f_{\boldsymbol{\theta}}(\boldsymbol{x}) = f_{\hat{\boldsymbol{\theta}}}(\boldsymbol{x})$ *for all* $\boldsymbol{x}\in\mathbb{R}^d$.

*Proof.* It is easy to see that $\pi_{\boldsymbol{b}}(\hat{\boldsymbol{\theta}})$ is a KKT-margin direction with $\gamma(\pi_{\boldsymbol{b}}(\hat{\boldsymbol{\theta}})) = \gamma(\hat{\boldsymbol{\theta}}) = \frac{1}{2}\gamma^+$. Also, $\arg\min_{i\in[n]}\{q_i(\pi_{\boldsymbol{b}}(\hat{\boldsymbol{\theta}}))\} = \arg\min_{i\in[n]}\{q_i(\hat{\boldsymbol{\theta}})\} = \mathcal{S}^+$. Let $\epsilon > 0$ be a small constant such that the following holds whenever $\|\boldsymbol{\theta} - \pi_{\boldsymbol{b}}(\hat{\boldsymbol{\theta}})\|_{\mathrm{M}} < \epsilon$:

1. $\mathrm{sgn}(\langle \boldsymbol{w}_k, \boldsymbol{x}_i\rangle) = \mathrm{sgn}(\langle \boldsymbol{w}^+, \boldsymbol{x}_i\rangle)$ for all $i\in[n]$ and for all $k\in P$;

2. $\arg\min_{i\in[n]}\{q_i(\boldsymbol{\theta})\}\subseteq\mathcal{S}^+$.

Let $\boldsymbol{\theta}\in\bar{\mathcal{Q}}$ be any parameter satisfying $\|\boldsymbol{\theta} - \pi_{\boldsymbol{b}}(\hat{\boldsymbol{\theta}})\|_{\mathrm{M}} < \epsilon$. We can decompose $\boldsymbol{\theta}$ into $\boldsymbol{\theta}^+ + \boldsymbol{\theta}^-$, where $\boldsymbol{\theta}^+ = (\boldsymbol{w}_1^+,\ldots,\boldsymbol{w}_m^+, a_1^+,\ldots, a_m^+)$, $\boldsymbol{\theta}^- = (\boldsymbol{w}_1^-,\ldots,\boldsymbol{w}_m^-, a_1^-,\ldots, a_m^-)$, and

$$\boldsymbol{w}_k^+ = \mathbb{1}_{[k\in P]}\boldsymbol{w}_k, a_k^+ = \mathbb{1}_{[k\in P]}a_k, \qquad \boldsymbol{w}_k^- = \mathbb{1}_{[k\notin P]}\boldsymbol{w}_k, a_k^- = \mathbb{1}_{[k\notin P]}a_k.$$

Let $r_+ = \|\boldsymbol{\theta}^+\|_2$ and $r_- = \|\boldsymbol{\theta}^-\|_2$. Define $\bar{\boldsymbol{\theta}}^+$ and $\bar{\boldsymbol{\theta}}^-$ to be two unit-norm parameters so that $\boldsymbol{\theta}^+ = r_+\bar{\boldsymbol{\theta}}^+$, $\boldsymbol{\theta}^- = r_-\bar{\boldsymbol{\theta}}^-$. Then we have

$$\gamma(\boldsymbol{\theta}) = \min_{i\in\mathcal{S}^+}\{q_i(\boldsymbol{\theta})\} = \min_{i\in\mathcal{S}^+}\{q_i(\boldsymbol{\theta}^+) + q_i(\boldsymbol{\theta}^-)\} = \min_{i\in\mathcal{S}^+}\{r_+^2 q_i(\bar{\boldsymbol{\theta}}^+) + r_-^2 q_i(\bar{\boldsymbol{\theta}}^-)\}$$

Note that $r_+^2 + r_-^2 = 1$. By minimax theorem (similar to Lemma G.10),

$$\min_{i \in \mathcal{S}^+} \left\{ r_+^2 q_i(\bar{\boldsymbol{\theta}}^+) + r_-^2 q_i(\bar{\boldsymbol{\theta}}^-) \right\} \leq \min_{\boldsymbol{\lambda} \in \Lambda^+} \max \left\{ \sum_{i \in \mathcal{S}^+} \lambda_i q_i(\bar{\boldsymbol{\theta}}^+), \sum_{i \in \mathcal{S}^+} \lambda_i q_i(\bar{\boldsymbol{\theta}}^-) \right\}.$$

By definition of $\boldsymbol{w}^+$ and KKT conditions, we can find $\boldsymbol{\lambda}^* \in \Lambda^+$ so that $\sum_{i \in \mathcal{S}^+} \lambda_i^* \boldsymbol{x}^+ = \gamma^+ \boldsymbol{w}^+$. Letting $\boldsymbol{\lambda} = \boldsymbol{\lambda}^*$ for the above inequality, we can obtain

$$\gamma(\boldsymbol{\theta}) \leq \max \left\{ \sum_{i \in \mathcal{S}^+} \lambda_i^* q_i(\bar{\boldsymbol{\theta}}^+), \sum_{i \in \mathcal{S}^+} \lambda_i^* q_i(\bar{\boldsymbol{\theta}}^-) \right\}.$$

We only need to prove that both $\sum_{i \in \mathcal{S}^+} \lambda_i^* q_i(\bar{\boldsymbol{\theta}}^+)$ and $\sum_{i \in \mathcal{S}^+} \lambda_i^* q_i(\bar{\boldsymbol{\theta}}^-)$ are no more than $\frac{1}{2}\gamma^+$. Note that combining Assumption A.5 and Lemma G.10 directly implies that $\sum_{i \in \mathcal{S}^+} \lambda_i^* q_i(\bar{\boldsymbol{\theta}}^-) < \frac{1}{2}\gamma^+$. Now we focus on $\sum_{i \in \mathcal{S}^+} \lambda_i^* q_i(\bar{\boldsymbol{\theta}}^+)$.

According to our choice of $\epsilon$, we have $a_k \phi(\langle \boldsymbol{w}_k, \boldsymbol{x}_i \rangle) = \langle a_k \boldsymbol{w}_k, \boldsymbol{x}_i^+ \rangle$. For $\sum_{i \in \mathcal{S}^+} \lambda_i^* q_i(\bar{\boldsymbol{\theta}}^+)$, we have

$$\sum_{i \in \mathcal{S}^+} \lambda_i^* q_i(\bar{\boldsymbol{\theta}}^+) = \sum_{i \in \mathcal{S}^+} \lambda_i^* y_i \sum_{k \in P} \langle a_k \boldsymbol{w}_k, \boldsymbol{x}_i^+ \rangle = \sum_{k \in P} a_k \left\langle \boldsymbol{w}_k, \sum_{i \in \mathcal{S}^+} \lambda_i^* y_i \boldsymbol{x}_i^+ \right\rangle$$

$$= \sum_{k \in P} a_k \left\langle \boldsymbol{w}_k, \gamma^+ \boldsymbol{w}^+ \right\rangle.$$

By Cauchy-Schwartz inequality,

$$\sum_{k \in P} a_k \left\langle \boldsymbol{w}_k, \gamma^+ \boldsymbol{w}^+ \right\rangle \leq \sqrt{\sum_{k \in P} a_k^2} \cdot \sqrt{\sum_{k \in P} \langle \boldsymbol{w}_k, \gamma^+ \boldsymbol{w}^+ \rangle^2} \leq \frac{1}{\sqrt{2}} \cdot \frac{1}{\sqrt{2}} \gamma^+ = \frac{1}{2}\gamma^+.$$

This proves that $\sum_{i \in \mathcal{S}^+} \lambda_i^* q_i(\bar{\boldsymbol{\theta}}^+) \leq \frac{1}{2}\gamma^+$, and thus $\gamma(\boldsymbol{\theta}) \leq \frac{1}{2}\gamma^+ = \gamma(\hat{\boldsymbol{\theta}})$. Therefore Item 1 is true.

For Item 2, we only need to note that the equality in $\gamma(\boldsymbol{\theta}) \leq \frac{1}{2}\gamma^+$ only holds if $r_- = 0$ and $\boldsymbol{w}_k = a_k \boldsymbol{w}^+$ for all $k \in P$, so $f_{\boldsymbol{\theta}}$ represents the same function as $f_{\hat{\boldsymbol{\theta}}}$. $\qquad\square$

For proving Theorem A.7, we only need to show this:

**Theorem G.12.** *For any sequence of $\sigma_1, \sigma_2, \ldots$ converging to $0$, there is a subsequence $\sigma_{p_1}, \sigma_{p_2}, \ldots$ and a constant $\sigma_{\mathrm{init}}^{\max}$ such that Theorem A.7 holds for $\sigma_{\mathrm{init}} = \sigma_{p_i}$ as long as $\sigma_{p_i} < \sigma_{\mathrm{init}}^{\max}$.*

*Proof for Theorem A.7.* Assume to the contrary that Theorem A.7 does not hold. Then there exists $\boldsymbol{x} \in \mathbb{R}^d$ and a sequence of initialization scales $\sigma_1, \sigma_2, \ldots$ converging to $0$ such that $f^\infty(\boldsymbol{x}) \neq \frac{1}{2}\phi(\langle \boldsymbol{w}^+, \boldsymbol{x} \rangle)$ for any $\sigma_j$. However, by Theorem G.12, we can find a subsequence $\sigma_{p_1}, \sigma_{p_2}, \ldots$ and a constant $\sigma_{\mathrm{init}}^{\max}$ such that $\frac{1}{2}\phi(\langle \boldsymbol{w}^+, \boldsymbol{x} \rangle)$ holds for $\sigma_{p_i}$ as long as $\sigma_{p_i} < \sigma_{\mathrm{init}}^{\max}$, contradiction. $\qquad\square$

*Proof for Theorem G.12.* With probability 1 over the random draw of $\bar{\boldsymbol{\theta}}_0 \sim \mathcal{D}_{\mathrm{init}}(1)$, by Lemma G.5, the prerequisites of Lemma G.4 hold and we can find a subsequence of initialization scales $\sigma_{p_1}, \sigma_{p_2}, \ldots$ so that the limit $\hat{\boldsymbol{b}} := \lim_{j \to +\infty} \bar{\boldsymbol{b}}(\sigma_{p_j})$ exists.

Let $\boldsymbol{\theta}_j(t) = \varphi(\sigma_{p_j}\bar{\boldsymbol{\theta}}_0, t)$. By Lemma G.8, with probability $1 - 2^{-m}$, $\lim_{j \to \infty} \boldsymbol{\theta}_j(T_{12}(\sigma_{p_j}) + t) = \pi_{\hat{\boldsymbol{b}}}(\tilde{\boldsymbol{\theta}}(t))$. By Theorem G.9, $\lim_{t \to +\infty} \frac{\tilde{\boldsymbol{\theta}}(t)}{\|\tilde{\boldsymbol{\theta}}(t)\|_2} = \frac{1}{\sqrt{2}}(\boldsymbol{w}^+, \boldsymbol{0}, 1, 0) =: \tilde{\boldsymbol{\theta}}_\infty$. Then we can argue in a similar way as Theorem 4.3 to show that for any $\epsilon > 0$ and $\rho > 0$, we can choose a time $t_1 \in \mathbb{R}$ such that $\left\| \frac{\boldsymbol{\theta}_j(T_{12}(\sigma_{p_j}) + t_1)}{\|\boldsymbol{\theta}_j(T_{12}(\sigma_{p_j} + t_1))\|_2} - \pi_{\hat{\boldsymbol{b}}}(\tilde{\boldsymbol{\theta}}_\infty) \right\|_2$ and $\|\boldsymbol{\theta}_j(T_{12}(\sigma_{p_j}) + t_1)\|_2 \geq \rho$ for $\sigma_{p_j}$ small enough.

By Corollary B.18, the trajectory of gradient flow starting with $\sigma_{p_j}\bar{\boldsymbol{\theta}}_0$ lies in the set $\mathcal{Q} := \{\boldsymbol{\theta} : a_k\bar{a}_k \geq 0 \text{ for all } k \in [m]\}$ for all $j \geq 1$, that is, every $a_k$ has the same sign as its initial value during training. By a variant of Theorem 5.6, there exists $\sigma_{\mathrm{init}}^{\max}$ such that for all $\sigma_{p_j} < \sigma_{\mathrm{init}}^{\max}$, $\frac{\boldsymbol{\theta}(t)}{\|\boldsymbol{\theta}(t)\|_2} \to \bar{\boldsymbol{\theta}} \in \mathcal{Q}$, where $\gamma(\bar{\boldsymbol{\theta}}) = \gamma(\pi_{\bar{\boldsymbol{b}}}(\tilde{\boldsymbol{\theta}}_\infty))$ and $\|\bar{\boldsymbol{\theta}} - \pi_{\bar{\boldsymbol{b}}}(\tilde{\boldsymbol{\theta}}_\infty)\|_2 \leq \delta$. Applying Theorem G.11 proves that $f^\infty(\boldsymbol{x}) = \frac{1}{2}\phi(\langle \boldsymbol{w}^+, \boldsymbol{x} \rangle)$ for $\sigma_{p_j} < \sigma_{\mathrm{init}}^{\max}$. $\qquad\square$

# H Proofs for the Orthogonally Separable Case

In this section, we revisit the orthogonally separable setting considered by Phuong and Lampert (2021). Suprisingly, in this setting, all KKT points which contains at least one positive neuron and negative neuron are indeed global-max-margin directions and unique in function space. This means it is possible to prove the global optimality of margin in Phuong and Lampert (2021)'s setting even without a trajectory-based analysis.

**Definition H.1** (Orthogonally Separable Data, Phuong and Lampert 2021). *A binary classification dataset $\{(\boldsymbol{x}_1, y_1), \ldots, (\boldsymbol{x}_n, y_n)\}$ is called orthogonally separable if for all $i, j \in [n]$, if $\boldsymbol{x}_i^\top \boldsymbol{x}_j > 0$ whenever $y_i = y_j$ and $\boldsymbol{x}_i^\top \boldsymbol{x}_j \leq 0$ whenever $y_i = -y_j$.*

Let $\boldsymbol{\theta} = (\boldsymbol{w}_1, \ldots, \boldsymbol{w}_m, a_1, \ldots, a_m) \in \mathbb{R}^D$ and $f_{\boldsymbol{\theta}}(\boldsymbol{x}) := \sum_{i=1}^m a_i \phi(\langle \boldsymbol{x}, \boldsymbol{w}_i \rangle)$ where $\phi$ is ReLU, i.e., $\phi(x) = \max\{0, x\}$. The following theorem shows that for orthogonally separable data, all KKT-margin directions are global-max-margin directions.

**Theorem H.2.** *Suppose the dataset is orthogonally separable, for all KKT-margin directions $\boldsymbol{\theta} \in \mathbb{S}^D$, their corresponding functions $f_{\boldsymbol{\theta}}$ are the same and thus they are all global-max-margin directions.*

The Theorem H.2 is a simple corollary of the following lemma Lemma H.3.

**Lemma H.3.** *If $\boldsymbol{\theta}$ satisfies the KKT conditions of (P), then for $a_k \neq 0$, $|a_k| = \|\boldsymbol{w}_k\|_2$ and $(\sum_{j:a_j a_k > 0} a_j^2) \frac{\boldsymbol{w}_k}{a_k}$ is the global minimizer of the following optimization problem (Q):*

$$\min_{\boldsymbol{w}} \quad \frac{1}{2} \|\boldsymbol{w}\|_2^2 \quad \text{s.t.} \ \langle \boldsymbol{w}, \boldsymbol{x}_i \rangle \geq 1, \quad \text{for all } i \in [n] \text{ with } y_i = \operatorname{sgn}(a_k). \tag{Q}$$

*In other words, all the non-zero $a_k, \boldsymbol{w}_k$ can be split into 2 groups according to the sign of $a_k$, where in each group, $\frac{\boldsymbol{w}_k}{a_k}$ is the same.*

*Proof of Theorem H.2.* By Lemma H.3, we know for any $\boldsymbol{\theta}$ satisfying the KKT condition of (P),

$$\boldsymbol{w}_k = \left( \sum_{j:a_j a_k > 0} a_j^2 \right)^{-1} a_k \boldsymbol{w}^{\operatorname{sgn}(a_k)}, \tag{69}$$

where $\boldsymbol{w}^{\operatorname{sgn}(a_k)}$ ($\boldsymbol{w}^+$ or $\boldsymbol{w}^-$) are the unique global minimzer of the constrained convex optimization of (Q).

Thus $\|\boldsymbol{\theta}\|_2^2 = \sum_{i \in [m]} (|a_i|^2 + \|\boldsymbol{x}_i\|_2^2) = 2 \sum_{i \in [m]} |a_i|^2 = \|\boldsymbol{w}^-\|_2 + \|\boldsymbol{w}^+\|_2$ is the same for all $\boldsymbol{\theta}$ satisfying the condition in the theorem statement. Here the last equality uses (69) and $|a_k| = \|\boldsymbol{w}_k\|_2$.

Next we check the uniqueness of $f_{\boldsymbol{\theta}}$. For any $\boldsymbol{x}$, we have

$$f_{\boldsymbol{\theta}}(\boldsymbol{x}) = \sum_{k \in [m]} a_k \phi(\langle \boldsymbol{x}, \boldsymbol{w}_k \rangle) = \phi\left( \left\langle \boldsymbol{x}, \sum_{k:a_k > 0} a_k \boldsymbol{w}_k \right\rangle \right) + \phi\left( \left\langle \boldsymbol{x}, \sum_{k:a_k < 0} a_k \boldsymbol{w}_k \right\rangle \right)$$
$$= \phi(\langle \boldsymbol{x}, \boldsymbol{w}^+ \rangle) + \phi(\langle \boldsymbol{x}, \boldsymbol{w}^- \rangle),$$

which completes the proof. $\square$

*Proof of Lemma H.3.* By KKT conditions (Definition B.8), there exist $\lambda_1, \ldots \lambda_n \geq 0$, such that for each $k \in [m]$, there are $h_1^{(k)}, \ldots, h_n^{(k)} \in \mathbb{R}$ such that for all $i \in [n]$, $h_i^{(k)} \in \phi^\circ(\langle \boldsymbol{w}_k, \boldsymbol{x}_i \rangle)$, and the following conditions hold:

$$\boldsymbol{w}_k = a_k \sum_{i \in [n]} \lambda_i h_i^{(k)} y_i \boldsymbol{x}_i, \qquad a_k = \sum_{i \in [n]} \lambda_i y_i \phi(\boldsymbol{w}_k^\top \boldsymbol{x}_i),$$

and $\lambda_i = 0$ whenever $y_i f_{\boldsymbol{\theta}}(\boldsymbol{x}_i) > 1$. By Lemma B.9, $\|\boldsymbol{w}_k\|_2 = |a_k|$.

We claim that for all $i \in [n]$ so that $\lambda_i h_i^{(k)} > 0$, it holds that $y_i = \operatorname{sgn}(a_k)$ and $\langle \boldsymbol{w}_k, \boldsymbol{x}_i \rangle > 0$. Let $i \in [n]$ be any index so that $\lambda_i h_i^{(k)} > 0$. Then $h_i^{(k)} > 0$. By KKT conditions,

$$\langle \boldsymbol{w}_k, \boldsymbol{x}_i \rangle = \left\langle a_k \sum_{j \in [n]} \lambda_j h_j^{(k)} y_j \boldsymbol{x}_j, \boldsymbol{x}_i \right\rangle = a_k y_i \sum_{j \in [n]} \lambda_j h_j^{(k)} \langle y_j \boldsymbol{x}_j, y_i \boldsymbol{x}_i \rangle. \tag{70}$$

Since $\phi(x) = \max\{x, 0\}$, it holds that $\langle \boldsymbol{w}_k, \boldsymbol{x}_i \rangle \geq 0$; otherwise $h_i^{(k)} \in \phi^\circ(\langle \boldsymbol{w}_k, \boldsymbol{x}_i \rangle) = \{0\}$, which contradicts to $h_i^{(k)} > 0$. Then (70) implies that the product of $a_k y_i$ and $\sum_{j \in [n]} \lambda_j h_j^{(k)} \langle y_j \boldsymbol{x}_j, y_i \boldsymbol{x}_i \rangle$ is non-negative. By orthogonal separability, $\langle y_j \boldsymbol{x}_j, y_i \boldsymbol{x}_i \rangle \geq 0$ and thus $\sum_{j \in [n]} \lambda_j h_j^{(k)} \langle y_j \boldsymbol{x}_j, y_i \boldsymbol{x}_i \rangle \geq \lambda_i h_i^{(k)} \|y_i \boldsymbol{x}_i\|_2^2 > 0$. Then we can conclude that $a_k y_i \geq 0$ and thus $y_i = \mathrm{sgn}(a_k)$. Since $y_i = \mathrm{sgn}(a_k)$ and $a_k \neq 0$, indeed we have $a_k y_i > 0$. Now using (70) again, we obtain $\langle \boldsymbol{w}_k, \boldsymbol{x}_i \rangle \geq a_k y_i \cdot \lambda_i h_i^{(k)} \|y_i \boldsymbol{x}_i\|_2^2 > 0$ if $\lambda_i h_i^{(k)} > 0$.

Furthermore, for any $a_k \neq 0$, since $\|\boldsymbol{w}_k\|_2 = |a_k| > 0$, there is at least one index $j_* \in [n]$ such that $\lambda_{j_*} h_{j_*}^{(k)} > 0$ (otherwise $\boldsymbol{w}_k = \boldsymbol{0}$ by KKT conditions). For all $i \in [n]$, again by (70), it holds that

$$\mathrm{sgn}(a_k) y_i \langle \boldsymbol{w}_k, \boldsymbol{x}_i \rangle = |a_k| \sum_{j \in [n]} \lambda_j h_j^{(k)} \langle y_j \boldsymbol{x}_j, y_i \boldsymbol{x}_i \rangle \geq |a_k| \lambda_{j_*} h_{j_*}^{(k)} \langle y_{j_*} \boldsymbol{x}_{j_*}, y_i \boldsymbol{x}_i \rangle > 0,$$

where the last inequality is from the assumption of orthogonally separability. This further implies $h_i^{(k)} = \mathbb{1}_{[y_i = \mathrm{sgn}(a_k)]}$ and thus $\boldsymbol{w}_k = a_k \sum_{i=1}^{n} \mathbb{1}_{[y_i = \mathrm{sgn}(a_k)]} \lambda_i y_i \boldsymbol{x}_i$ for all $k \in [m]$.

Therefore we can split the neurons with non-zero $a_k$ into two parts: $K^+ = \{k \in [m] : a_k > 0\}$, $K^- = \{k \in [m] : a_k < 0\}$. Every $k \in K^+$ satisfies the following:

$$a_k = \|\boldsymbol{w}_k\|_2, \tag{71}$$

$$\boldsymbol{w}_k = a_k \sum_{i=1}^{n} \mathbb{1}_{[y_i = 1]} \lambda_i \boldsymbol{x}_i. \tag{72}$$

This implies $\forall k \in K^+$, $\frac{\boldsymbol{w}_k}{a_k} = \frac{\boldsymbol{w}_k}{\|\boldsymbol{w}_k\|_2} = \sum_{i=1}^{n} \mathbb{1}_{[y_i = 1]} \lambda_i \boldsymbol{x}_i$. Define $\bar{\boldsymbol{w}} := \sum_{k \in K^+} a_k \boldsymbol{w}_k$, then

$$\bar{\boldsymbol{w}} = \left( \sum_{k \in K^+} a_k^2 \right) \sum_{i=1}^{n} \mathbb{1}_{[y_i = 1]} \lambda_i \boldsymbol{x}_i.$$

Recall that $\lambda_i = 0$ whenever $y_i f_{\boldsymbol{\theta}}(\boldsymbol{x}_i) > 1$. When $y_i = 1$, $f_{\boldsymbol{\theta}}(\boldsymbol{x}_i)$ can be rewritten as

$$f_{\boldsymbol{\theta}}(\boldsymbol{x}_i) = \sum_{k \in [m]} a_k \mathbb{1}_{[\mathrm{sgn}(a_k) = 1]} \langle \boldsymbol{w}_k, \boldsymbol{x}_i \rangle = \langle \boldsymbol{x}_i, \bar{\boldsymbol{w}} \rangle.$$

So we can verify that $\bar{\boldsymbol{w}}$ satisfies the KKT conditions of the following constrained convex optimization problem:

$$\min_{\boldsymbol{w}} \|\boldsymbol{w}\|_2^2 \tag{73}$$

$$\text{s.t. } \langle \boldsymbol{w}, \boldsymbol{x}_i \rangle \geq 1, \quad \text{for all } i \in [n] \text{ with } y_i = 1. \tag{74}$$

By convexity, $\bar{\boldsymbol{w}}$ is the unique minimizer of the above problem. The negative part $K^-$ can be analyzed in the same way. $\qquad \square$

# I  Additional Discussions

## I.1  Illustrations for Figure 1

In this section we further illustrate the the relationship between KKT-margin and max-margin directions, as the examples have showed in Figure 1.

### I.1.1  Left: Symmetric Data

**Example.**  For some symmetric data, there are KKT-margin directions with non-linear decision boundary (and thus by Theorem 4.2 are not global-max-margin directions).

Let $\lambda_i$ be the dual variable for $(x_i, y_i)$, then the KKT conditions (Definition B.8 and Lemma B.9) ask

1. for all $k \in [m]$, $\boldsymbol{w}_k \in \sum_{i \in [n]} \lambda_i y_i a_k \phi^\circ(\boldsymbol{w}_k^\top \boldsymbol{x}_i) \boldsymbol{x}_i$;
2. for all $k \in [m]$, $|a_k| = \|\boldsymbol{w}_k\|_2$;

3. for all $i \in [n]$, if $q_i(\boldsymbol{\theta}) \neq q_{\min}(\boldsymbol{\theta})$ then $\lambda_i = 0$ (recall that $q_i(\boldsymbol{\theta}) = y_i f_{\boldsymbol{\theta}}(\boldsymbol{x}_i)$).

For $\alpha_{\text{leaky}} = 0$, the example is simpler. Consider the following case: the data points are $\boldsymbol{x}_1 = (1, -1)$, $\boldsymbol{x}_2 = (1, 0)$, $\boldsymbol{x}_3 = (1, 1)$ with label 1 and the symmetric counterpart $\boldsymbol{x}_4 = (-1, 1)$, $\boldsymbol{x}_5 = (-1, 0)$, $\boldsymbol{x}_6 = (-1, -1)$ with label $-1$. As we have proved, the global-max-margin solution is a linear function and in this case $\boldsymbol{w}^* = (1, 0)$. On the other hand, for hidden neurons $m \geq 3$, one KKT-margin direction is as follows:

$$\begin{cases} a_1 = 2^{-1/4} & \boldsymbol{w}_1 = \frac{1}{2^{3/4}}(1, 1) \\ a_2 = 2^{-1/4} & \boldsymbol{w}_2 = \frac{1}{2^{3/4}}(1, -1) \\ a_3 = -1 & \boldsymbol{w}_3 = (-1, 0) \\ a_k = 0 & \boldsymbol{w}_k = \boldsymbol{0} \qquad \text{for all } k > 3. \end{cases}$$

In this case, all the data points $\boldsymbol{x}_i$ share the same output margin $q_i(\boldsymbol{\theta})$, so they are all support vectors. A possible choice of dual variables is $\boldsymbol{\lambda} = (\frac{1}{\sqrt{2}}, 0, \frac{1}{\sqrt{2}}, 0, 1, 0)$. It is easy to verify that this KKT-margin direction does not have linear decision boundary and is thus not global-max-margin.

For $\alpha_{\text{leaky}} > 0$, we can adapt the above case to construct a KKT point. Let $\beta$ be a solution to the equation

$$(2\sin^2 \beta + \cos \beta)\alpha_{\text{leaky}}^2 - (1 + \cos \beta)\alpha_{\text{leaky}} + \cos 2\beta = 0.$$

Let the data be $\boldsymbol{x}_1 = (1, \cot \beta), \boldsymbol{x}_2 = (1, 0), \boldsymbol{x}_3 = (1, -\cot \beta)$ with label 1 and the corresponding opposites $\boldsymbol{x}_4 = (-1, -\cot \beta), \boldsymbol{x}_5 = (-1, 0), \boldsymbol{x}_6 = (-1, \cot \beta)$ with label $-1$. Then we can have a KKT point with $\boldsymbol{\lambda} = (\frac{\sin^2 \beta}{\cos \beta(1 - \alpha_{\text{leaky}})}, 0, \frac{\sin^2 \beta}{\cos \beta(1 - \alpha_{\text{leaky}})}, 0, 1 - \frac{2\alpha_{\text{leaky}} \sin^2 \beta}{(1 - \alpha_{\text{leaky}}) \cos \beta}, 0)$ and

$$\begin{cases} a_1 = (2(1 + \alpha_{\text{leaky}}) \cos \beta)^{-1/2} & \boldsymbol{w}_1 = a_1 \sin \beta(\cot \beta, 1) \\ a_2 = a_1 & \boldsymbol{w}_2 = a_2 \sin \beta(\cot \beta, -1) \\ a_3 = -(1 + \alpha_{\text{leaky}})^{-1/2} & \boldsymbol{w}_3 = -a_3(-1, 0) \\ a_k = 0 & \boldsymbol{w}_k = \boldsymbol{0} \qquad \text{for all } k > 3. \end{cases}$$

When $\cos 2\beta = 0$ we already have solution $\alpha_{\text{leaky}} = 0$, and it is easy to verify that for any $\alpha_{\text{leaky}} \in [0, 1)$ there is a solution $\beta$ that satisfy the KKT conditions. Thus in the leaky ReLU case we are considering in the previous chapters, there are also KKT-margin directions that have non-linear decision boundaries and therefore have sub-optimal margin.

### I.1.2  Middle and Right: Non-symmetric Data

In Figure 1 we further show two examples of non-symmetric data that gradient flow from small initialization converges to a linear-boundary classifier that has a suboptimal margin.

The idea of the middle plot dataset comes from Shah et al. (2020). In the middle subplot, we exhibit a data example that is linear separable in the first dimension $x$ but not linear separable in the second dimension $y$. The data is distributed on $(A_\epsilon, 1)$ and $(A_\epsilon, -1)$ with label 1 and on $(-A_{\epsilon'}, 0)$ with label $-1$ (here $A_c = [c, \infty)$ is an interval in one dimension). We add identical entries $c$ to all the data in the third dimension $z$ so in the $x - y$ plane with $z = c$ the two-layer ReLU network can represent decision patterns with bias.

To apply Theorem A.7 on this dataset, we need to make $c$ smaller than $\epsilon$ and $\epsilon/\epsilon' \ll \alpha_{\text{leaky}}$, so that the points at $(\epsilon, 1)$ and $(\epsilon, -1)$ becomes the support vectors for the one-neuron function. Also we can control the principal direction by taking more data points from the positive class, and then gradient flow will converge to the one-neuron max-margin solution as predicted by Theorem A.7. This solution cannot be global max-margin when $\epsilon \ll 1$, as a two-neuron network can express a function where these two support vectors have much larger distances to the decision boundary (and possess larger output margins).

In the right plot, we add three hints to a linear separable dataset so that gradient flow converges to the solution with a linear decision boundary and suboptimal margin. The result follows from Theorem 6.2.

### I.1.3  Experimental Results

We run gradient descent with small learning rate and 0.001 times the He intialization (He et al., 2015) on the two-layer LeakyReLU network for the examples in Figure 1. The contours of the neural net

outputs are displayed in Figure 2. In the three settings the neural nets actually converge to linear classifiers.

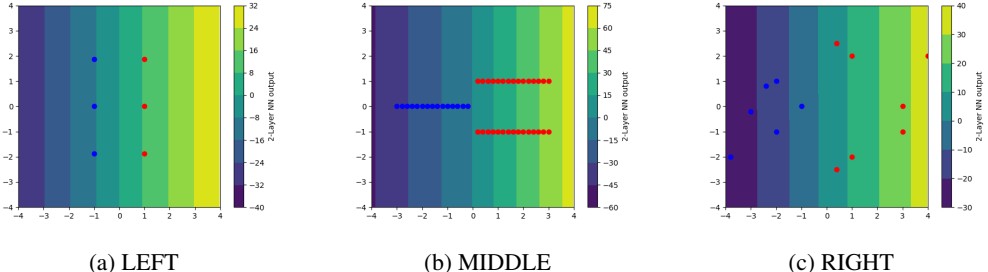

(a) LEFT           (b) MIDDLE           (c) RIGHT

Figure 2: Two-layer Leaky ReLU neural nets converge to functions with linear decision boundary for the examples in Figure 1. The output contours are displayed in colors, and lighter colors mean higher outputs.

## I.2   On the Non-branching Starting Point Assumptions

In the proofs of the main theorems we make assumptions regarding the starting point of gradient flow trajectories being non-branching (Assumption 4.6 for the symmetric case and Assumption A.6 for the non-symmetric case). The assumptions address a technical difficulty due to the potential non-uniqueness of gradient flow trajectories on general non-smooth loss functions. The motivations for these assumptions are explained below.

### I.2.1   The non-uniqueness of gradient flow trajectories

Gradient flow trajectories are unique on smooth loss functions by the classic theory of ordinary differential equations. In this case, for trajectory defined by $\frac{\mathrm{d}\boldsymbol{\theta}}{\mathrm{d}t} = -\nabla \mathcal{L}(\boldsymbol{\theta})$, at any point $\boldsymbol{\theta}_0$, if both $\nabla \mathcal{L}(\boldsymbol{\theta}_0)$ and $\nabla^2 \mathcal{L}(\boldsymbol{\theta}_0)$ are continuous, then the trajectory is unique as long as it exists.

For the non-smooth case with differential inclusion $\frac{\mathrm{d}\boldsymbol{\theta}}{\mathrm{d}t} \in -\partial^\circ \mathcal{L}(\boldsymbol{\theta})$, when $\mathcal{L}$ is continuous and convex, the Clarke subdifferentials agree with the subdifferentials for convex functions, and gradient flow trajectory is also unique (for instance see Bolte et al. 2010). However, on loss functions that are non-smooth and non-convex, gradient flow may not be unique and the trajectory may branch at non-differentiable points (see Figure 3). When a non-differentiable point is atop a "ridge", a gradient flow reaching it may go down different slopes next. Then any starting points wherefrom gradient flow can reach such on-the-ridge points are not non-branching starting points as the trajectory is not unique. For instance, with $\mathcal{L}(\boldsymbol{\theta}) = -|\langle \boldsymbol{\theta}, \boldsymbol{w} \rangle|$, then the trajectory with $\boldsymbol{\theta}(t) = 0$ for $t < t_s$ and $\boldsymbol{\theta}(t) = \pm(t - t_s)\boldsymbol{w}$ for $t \geq t_s$ is a valid gradient flow trajectory for any $t_s \geq 0$. On the other hand, when the point is either at the bottom of a "valley" or at a "refraction edge", the trajectory would not split. Figure 3 sketches in red the possible gradient flow trajectories in different circumstances.

In the case of two-layer Leaky ReLU network dynamics, there are settings where Assumption 4.6 or Assumption A.6 holds. When data points are orthogonally separable (Definition H.1), all starting points are non-branching. In this case, the output of each Leaky ReLU neuron will change monotonically. By the chain rule, for any neuron $k \in [m]$, on any data sample $i \in [n]$,

$$\left\langle \frac{\mathrm{d}\boldsymbol{w}_k}{\mathrm{d}t}, x_i \right\rangle \in -\frac{a_k}{n} \sum_{j \in [n]} \ell'(q_j(\boldsymbol{\theta})) y_j \phi^\circ(\langle \boldsymbol{w}_k, \boldsymbol{x}_i \rangle) \langle x_i, x_j \rangle .$$

Then as $y_i y_j \langle \boldsymbol{x}_i, \boldsymbol{x}_j \rangle \geq 0$ by the orthogonally separability, the sign of RHS is controlled by $\mathrm{sgn}(a_k y_i)$. With Theorem B.19, we know each $a_k$ does not change its sign along the gradient flow trajectory, and therefore $\langle \boldsymbol{w}_k, x_i \rangle$ changes monotonically. Then following the arguements of the classic theory of ordinary differential equation, by applying Grönwall's inequality to both intervals $\{t : \langle \boldsymbol{w}_k(t), x_i \rangle > 0\}$ and $\{t : \langle \boldsymbol{w}_k(t), x_i \rangle \leq 0\}$ we know the trajectory is unique. In this setting all the non-differentiable landscapes resemble the "refraction edges".

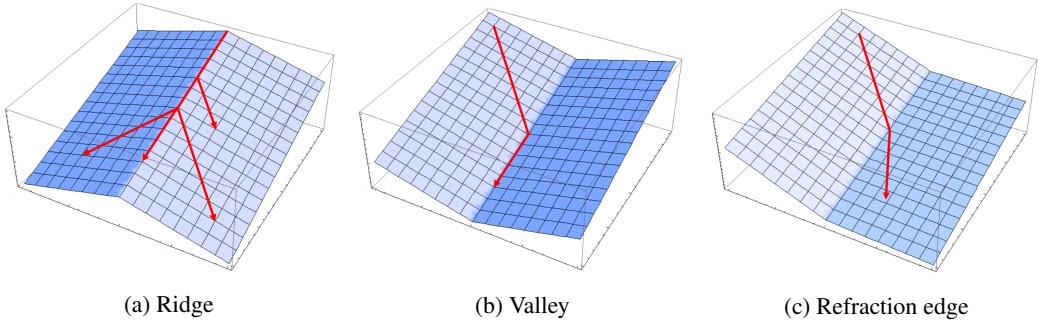

| | (a) Ridge | (b) Valley | (c) Refraction edge |

Figure 3: Gradient flow trajectories behave differently in different landscapes. The trajectory may be non-unique only after arriving at a point on the "ridge".

In the general cases, it is a future research direction to find other analyses that can replace the non-branching starting point assumptions, and doing so may deepen our understanding in the trajectory behaviors in non-smooth settings.

## J    Additional Experiments

We conducted several additional experiments on synthetic datasets. The goal is to show that 2-layer Leaky ReLU networks actually converges to the max-margin linear classifiers in different settings with moderately small initialization. The results are summarized in Table 1 and Figure 4.

| Dataset size | SVM test error | 2-Layer neural net test error |
|---|---|---|
| 10 | 30.2% | 30.5 % |
| 20 | 19.7% | 18.9% |
| 30 | 17.6% | 15.6% |
| 40 | 8.0% | 7.1% |
| 50 | 6.4% | 5.9% |
| 60 | 6.3% | 5.1% |
| 70 | 7.6% | 6.5% |
| 80 | 3.9% | 3.1% |
| 90 | 6.1% | 5.2% |
| 100 | 2.9% | 2.9% |

Table 1: Test errors for SVM max-margin linear classifiers and 2-Layer ReLU neural networks are nearly the same across different data size.

**Data.**    $n = 10, 20, \cdots, 100$ data points are randomly sampled from the standard gaussian distribution $\mathcal{N}(0, \boldsymbol{I})$ in the space of dimension $d = 50$, and are classified with a linear classifier through zero. Then the points are translated mildly away from the classifier to make a small nonzero margin that assists learning.

**Model and Training.**    We used the two-layer leaky ReLU network with hidden layer width $m = 100$ and with bias terms. In out setting the bias term is equivalent to adding an extra dimension of value 0.1 to all the data points. We trained our model with the gradient descent method from 0.001 times the He initialization (He et al., 2015) and initial learning rate 0.01. The learning rate is raised after interpolation to boost margin increase.

We compare the neural network output with the max-margin linear classfier produced by the support vector machine (SVM) on hinge loss. In Table 1, the test errors are calculated from 10000 test points from the same distribution. In Figure 4, we drawn the decision boundaries for both the SVM max-margin linear classifier and the neural network restricted to a plane passing 0. The results show that the neural network classifier converges to the max-margin linear classfier in our setting.

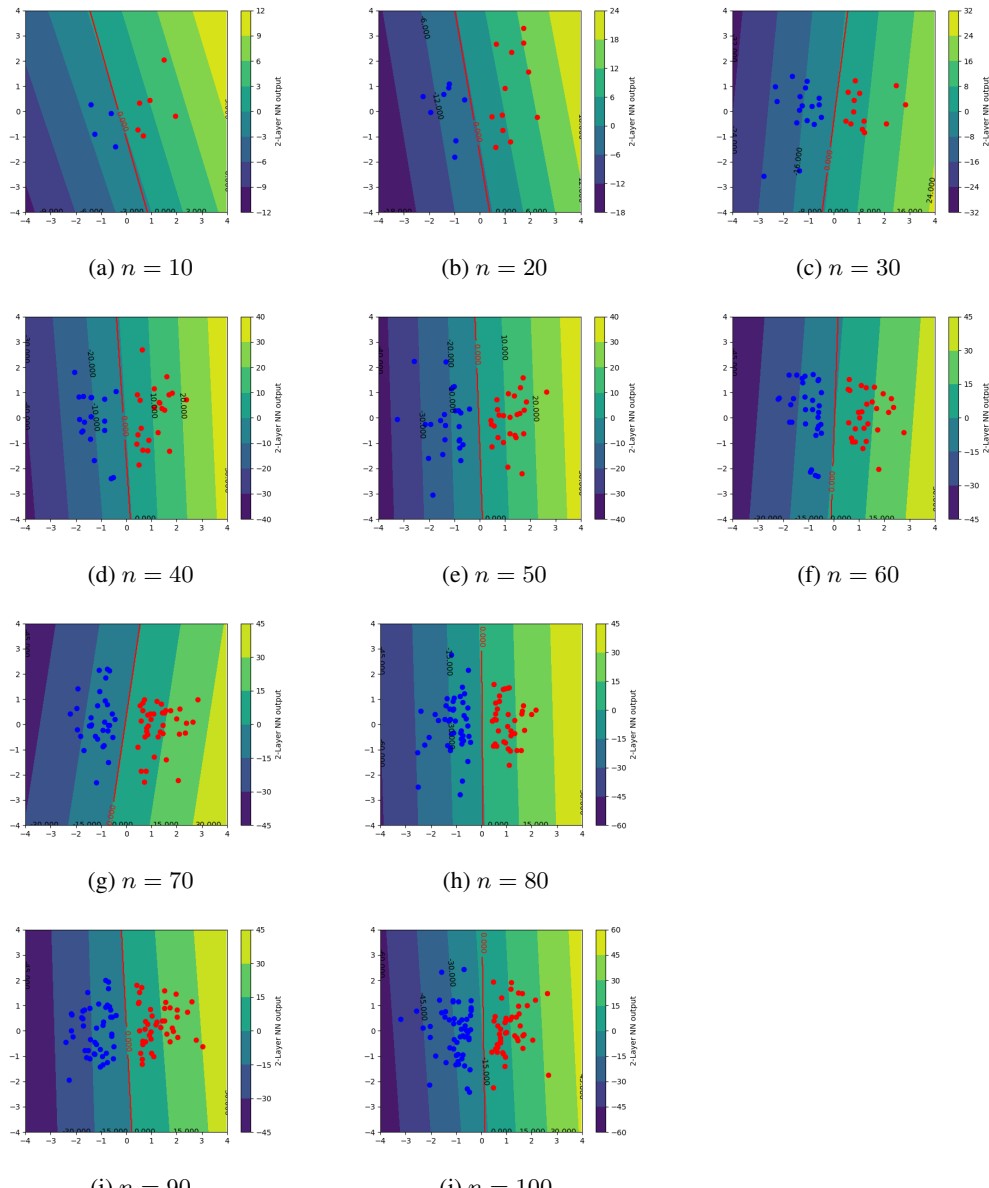

Figure 4: Two-layer Leaky ReLU neural net converges in direction to the SVM max-margin linear classifier. **Red and Blue Dots:** two classes of data points. **Red Lines:** the decision boundaries of the SVM max-margin linear classifiers. **Background:** the contours of two-layer leaky-ReLU neural network outputs. Lighter colors mean higher outputs. The underlying true separator is the vertical line through zero.