# OpenReview forum: "Gradient Descent on Two-layer Nets: Margin Maximization and Simplicity Bias"
_NeurIPS.cc/2021/Conference — NeurIPS 2021 Poster_

### Official Review · Reviewer_6TUB · 2021-06-26

**Rating:** 6
**Confidence:** 3

**Summary:**

This paper proves that gradient descent may converge to the “max-margin” linear solution that attains the global optimum of the loss on finite two-layer Leaky ReLU nets trained on linearly separable and symmetric data.

The contribution of this paper is mainly reflected in the strict assumption of data distribution (linearly separable and symmetric data). It is proved that GD has good convergence property in relatively simple nonconvex optimization problems (finite two-layer Leaky ReLU nets), and tends to get relatively “simple” solutions (the model complexity of “max-margin” linear solution is low, and it is resistant to overfitting risk).


**Ethical Concerns:**

NAN

**Limitations And Societal Impact:**

In this paper, some non-symmetric results are given in the appendix.

Theoretical research has no social impact.

**Main Review:**

Originality: This paper studies the properties of the convergence solution of GD under the specific data distribution and simplified network model, which is not a new problem. The originality of this paper is mainly reflected in the new assumption of data distribution (linearly separable and symmetric data), which leads to stronger conclusions than previous work (from KKT margin direction to global Max margin direction).

Quality & Clarity: This paper is theoretical research, and the process of proof is rigorous. In writing, the main results and the secondary lemmas are properly arranged and expressed clearly. Although there are some types in the appendix, it does not affect the understanding.

In the appendix, line 551 and line 560  have the missing references.

Significance: It is difficult to extend the assumptions to the deep neural network structure. Its significance only lies in helping us understand some simple structures in DNNs. Considering that the assumptions of the model and data distribution in this paper are not difficult to be realized through experiments, I hope that the authors can verify their theoretical results through experimental design by synthesizing the dataset.

If the author can provide some experimental results to verify the theoretical conclusion, I would be willing to raise the score to 7-8.


**Time Spent Reviewing:**

5 hours

---

> ### Author Response · Authors · 2021-08-11
> **Response to Reviewer 6TUB**
>
> Thanks for your attention on the experiment part. We have conducted experiments on two-layer LeakyReLU networks and show that GD with small initialization finds the linear classifier for all examples plotted in Figure 1.
>
> For high-dimensional experiments, we compare the test errors of neural networks and SVMs on linearly separable and symmetric datasets, drawn from two 50-dim half-Gaussians separated by a margin. The test errors are similar across different sample sizes.
>
> | Dataset size | SVM test error | 2-Layer neural net test error|
> |------------------|----------------------------------|-------------------------------------------------------|
> | 10 | 26.4% | 26.0% |
> | 20 | 18.0% | 17.1% |
> | 30 | 17.0% | 15.6% |
> | 40 | 11.9% | 11.3% |
> | 50 | 10.2% | 10.0% |
> | 60 | 9.6% | 6.5% |
> | 70 | 2.8% | 2.1% |
> | 80 | 5.5% | 5.2% |
> | 90 | 7.6% | 7.2% |
> | 100 |  6.4% | 4.6% |
>
> We also study the importance of the initialization scale. In the symmetric case, experiments show that GD finds the max-margin linear classifier even for larger initialization (e.g., He init), but when the dataset is no longer symmetric, the final outcome may be non-linear with larger initialization.
>
> Changing the depth from 2 to 10, the results are always the same as above. This means these empirical findings can be safely extended to deep neural networks, and a more general theory is waiting to be discovered.

---

### Official Review · Reviewer_zJ7E · 2021-07-05

**Rating:** 6
**Confidence:** 4

**Summary:**

The paper studies the implicit bias for finite two layers Leaky ReLU networks. For linearly separable symmetric data, the authors show that the global max-margin direction corresponds to a linear classifier. In addition, under additional regularity conditions and for a sufficiently small initialization scale, the authors show gradient flow achieves the global max-margin. For non-symmetric yet linearly separable data, the authors show that gradient flow can be led to converge to a linear classifier with suboptimal margin by adding 3 data points.

**Ethical Concerns:**

None.

**Limitations And Societal Impact:**

The authors mentioned and justified most of the limitations of their results. However, some of the limitations should be more thoroughly discussed, as mentioned in the main review.

**Main Review:**

The paper studies the implicit bias for finite two layers Leaky ReLU networks and logistic loss.

The paper main contributions are:
- For linearly separable *symmetric* datasets, the authors show that the global max-margin separator is linear.
- Moreover, the authors show that in this setting and under additional regularity conditions and sufficiently small initialization scale, gradient flow converges to a global max-margin *linear* classifier.
In particular, the authors note that previous works only showed convergence to a KKT-margin direction which in general may be non-linear.
- These results connect current implicit bias max-margin results with simplicity bias works.
- However, for non-symmetric yet linearly separable data, the authors show that gradient flow can be led to converge to a linear classifier with suboptimal margin by adding 3 data points.

My main questions\concerns:
- How small should the initialization scale be for the result stated in theorem 4.3 to hold?
The authors mention that small initialization is needed to avoid the NTK regime and yet, to get a better sense of the results, it’s important to understand if they only hold for infinitesimal initialization or does they also hold for intermediate initialization scales (where the training accuracy might also have some effect, e.g. [1]).
Some experimental results on a synthetic dataset that demonstrate the paper’s main findings might help clarify this point.
- Symmetric dataset: The authors justified the symmetric dataset assumption by mentioning that this assumption can be easily satisfied with data augmentation. They also give a negative result for what happens when the dataset is not symmetric. However, since the mentioned data augmentation is not common practice, I worry that the symmetry assumption is too restrictive and weakens the paper’s positive results.

Clarity/writing:
In general, the paper is well written, and I did not see any critical issues. Two suggestions that I think can improve the clarity:
- The o-minimal structure should be briefly explained\given some intuition.
- Some explanation should be given for why assumption A.5 is necessary.

Minor comments:
- Line 31: nature-->natural
- Equation between lines 159 and 160: ‘.’-->’,’
- Line 344: corollary general theorem--> corollary of the general theorem

Citations:

[1] E. Moroshko, S. Gunasekar, B. Woodworth, J. D. Lee, N. Srebro, D. Soudry, "Implicit Bias in Deep Linear Classification: Initialization Scale vs Training Accuracy", NeurIPS 2020.

---

Update: Thank you for your response and for explaining the assumption.


**Time Spent Reviewing:**

12

---

> ### Author Response · Authors · 2021-08-11
> **Response to Reviewer zJ7E**
>
> Thanks for your careful review! We have responded to your concern on small initialization and symmetric dataset assumption in [our general comment](https://openreview.net/forum?id=Aa5oPXc_1IV&noteId=mx9FMql_Mw). We will improve the clarity in the next version.
> * We will add the definition of o-minimal structure in the next version. It is a very broad function class that includes fully-connected, convolutional, recurrent neural networks with ReLU, LeakyReLU, ELU, sigmoid activation, and square, hinge, logistic loss functions, but an exception is that the activation or loss cannot be periodic (e.g., sin and cos). The functions in this class share nice properties in math.
> * Similar to Assumption 4.6, Assumption A.5 is used to ensure that the trajectory of gradient flow is unique. This is mainly due to the technical difficulty of dealing with non-smooth functions, and we will further clarify this point in the next version.

---

### Official Review · Reviewer_Eg4Q · 2021-07-10

**Rating:** 7
**Confidence:** 4

**Summary:**

This paper considers margin maximization and simplicity bias properties of gradient flow on a two-layer leaky-ReLU network with linearly separable data. If the data is also symmetric (i.e., whenever (x,y) is in the dataset, (-x,-y) is also in the dataset), this paper proves that the max-margin network basically represents the max-margin linear classifier. Moreover, while prior results ensure convergence to KKT points which may not give the maximum margin, this paper shows that under a few technical conditions gradient flow can indeed maximize the margin. On the other hand, if the data is not symmetric, this paper provides an example where gradient flow cannot globally maximize the margin.

**Limitations And Societal Impact:**

As mentioned in Section 7, the linear separability assumption is restrictive. However, in my opinion the symmetry condition is more restrictive: under this condition, if the data is not linearly separable, then it cannot be separated by a network either (this can be proved using the proof of Theorem 4.2). Moreover, one common sufficient condition for a positive definite NTK is that no feature vectors are parallel, which do not hold under the symmetry condition. The symmetry condition is also used in phase I of the gradient flow analysis. I think these points should be discussed more clearly in the body.

**Main Review:**

I think this paper provides a few interesting results and techniques:

(1) Theorem 4.2 uses the symmetry condition to ensure the max-margin network is basically the max-margin linear classifier; I think this is interesting, and the proof of Theorem 4.2 is also nice.

(2) Section 5 gives a multi-phase analysis: in the first phase, even though random initialization is used, as long as the initialization is small enough, this paper shows gradient flow will approximately move the weight vectors on the first layer to two opposite directions. Then in the second phase, the gradient flow dynamic is approximated by a dynamic of two neurons, and finally in the third phase, this paper proves margin maximization. I think this multi-phase analysis is novel and interesting.

(3) The negative result given by Theorem 6.2 is also interesting, confirming that margin maximization may not hold in general.

**Time Spent Reviewing:**

8

---

> ### Author Response · Authors · 2021-08-11
> **Response to Reviewer Eg4Q**
>
> Thanks for the positive review and the appreciation of our results and techniques! We have responded to your concern on symmetric dataset assumption in [our general comment](https://openreview.net/forum?id=Aa5oPXc_1IV&noteId=mx9FMql_Mw), and we will discuss this more clearly in the main body of our paper.

---

### Official Review · Reviewer_ZQSg · 2021-07-16

**Rating:** 6
**Confidence:** 4

**Summary:**

This paper studies the gradient descent for two-layer leaky-ReLU neural networks on binary classification problems with the logistic loss and shows the simplicity bias where the neural network converges to the linear function that maximizes the margin under appropriate assumptions.

**Ethics Review Area:**

["I don’t know"]

**Limitations And Societal Impact:**

The authors adequately addressed the limitations and potential negative social impact

**Main Review:**

[Contributions]

The line of research that characterizes the implicit bias of the optimization method is important to explain the generalization capability without explicit regularization, especially for the nonconvex models such as neural networks. In this sense, the paper makes certain contributions summarized below:

- Under linear separable and symmetric assumptions, the study shows that the two-layer leaky-ReLU network with the parameter maximizing the normalized margin takes the linear function.

- The study shows that the gradient flow of the two-layer leaky ReLU converges to the max-margin linear classifier in direction.

Advantages over related studies are that (i) the study shows the convergence to the max-margin linear function rather than KKT-margin direction which yields a somewhat complicated function, and (ii) the study does not require an assumption called Neural Agreement Regime (NAR) which is used to conclude the similar result in [Sarussi et al. (2021)].

[Weaknesses]

I think this study is interesting, but I'm not sure when Assumption 4.6 is satisfied. Since one advantage is the validity of the required assumption, it is better to elaborate on the description of this assumption. Could you explain the situation where this assumption is satisfied?
I would like to increase the score if the authors well address this concern.

[Questions]

- There is no explanation about the dependency on the constant $c_{ainit}$ which is introduced for the initialization of $a$. I would like to know how this constant will affect the results. The following paper shows that the training dynamics fall into the kernel regime when the initialization scale of $a$ is small depending on the network width, although the problem setting and the way of initialization is different from this submission.

E Weinan, Chao Ma, and Lei Wu. A comparative analysis of optimization and generalization properties of two-layer neural network and random feature models under gradient descent dynamics, 2020.

- I would like to know how important small-scale initialization is. I acknowledge its importance in the context of feature learning. I want to know the importance in the context of the margin. Why does it fail to converge to a linear function when the initialization scale is large?

[Minor comments]

There are some typos in Appendix. For instance, see lines 551, 556, and 561. Moreover in Lemma B.5, $\ell$ should be replaced with $\ell'$

-----
After reading the rebuttal:
The authors have addressed my concern about Assumption 4.6. They well explained the motivation of this assumption and provided an example. Although a given example is rather limited, it is useful in finding a new research direction. I would like to increase the score to 6. I hope some comments on this assumption will be added in the revision.

**Time Spent Reviewing:**

6

---

> ### Author Response · Authors · 2021-08-11
> **Response to Reviewer ZQSg**
>
> Thanks for the valuable feedback. Our response to your question on small initialization is given in [our general comment](https://openreview.net/forum?id=Aa5oPXc_1IV&noteId=mx9FMql_Mw). Now we respond to your concerns on Assumption 4.6 and $c_{ainit}$.
>
> **The validity of Assumption 4.6.** Thanks for the suggestions and we will elaborate more on Assumption 4.6 in the next version.  The assumption addresses a difficulty due to the potential non-uniqueness of gradient flow trajectories on general non-smooth loss functions but we indicate why it is not too divorced from real training.
> * Gradient flow trajectories are unique on smooth loss functions by the classic theory of ordinary differential equations.
> * For continuous and convex loss functions (which may be non-smooth), gradient flow trajectories are also unique [1].
> * On loss functions that are non-smooth and non-convex, gradient flow may not be unique and the trajectory may branch at non-differentiable points. When a non-differentiable point is atop a ridge,  a gradient flow reaching it may go down different slopes next. Then two trajectories from the same starting point may split at such points on the ridges, and thus making the solution non-unique. For instance, consider the differential inclusion $dx/dt\in\partial^{\circ} f(x)$ with $f(x)=-|x|$ and $x(0)=0$, then both $x=0$, $x(s)=0$ ($s\leq c$) and $x(s)=\pm (s-c)$ ($s>c$) for any $c\geq 0$ are valid gradient flow trajectories. In this example, the trajectory may split at any time after reaching 0. On the other hand, when a non-differentiable point is in a valley, it would not cause trajectory splitting, and any trajectory from it will go along the valley next. In the above example when $f(x)$ is the leaky ReLU function, the flow would not split at 0 as the Clarke subdifferential of f at 0 only includes positive values. This holds also for $f(x)=|x|$ as the trajectory would remain 0 after reaching 0.
> * In the case of two-layer Leaky ReLU network dynamics, there are settings that this assumption holds. When data points are orthogonal or orthogonally separable (i.e. all the pairwise inner products of data samples within the same class are non-negative and those over different classes are non-positive), all starting points are non-branching. In this case, the output of each Leaky ReLU neuron will change monotonically, and the way signs (of Leaky ReLU neurons) change is unique at any non-differentiable point. Therefore the trajectory is unique.
>
> It is a future research direction to find other analyses that can replace assumption 4.6, and doing so may deepen our understanding in the trajectory behaviors in non-smooth settings. We will add this explanation to the final version.
>
> **The importance of $c_{ainit}$ and the relationship to (E et al., 2020).** We first clarify that our main results hold for any constant $c_{ainit}$, and we didn’t attempt to compute the exact dependency of $\sigma_{init}$ on $c_{ainit}$. For the result in (E et al., 2020) as mentioned by the reviewer, e.g., their Theorem 3.3, it not only requires the scale of $a$ to be sufficiently small depending on the network width, but also needs the network width sufficiently large depending on the smallest eigenvalue of the feature kernel, or $\sigma_{init}$ in our case. Their result doesn’t apply in our setting, since we only make claims for sufficiently small $\sigma_{init}$ after fixing the network width and $c_{ainit}$.
>
> [1] Bolte, Jérôme, et al. "Characterizations of Łojasiewicz inequalities: subgradient flows, talweg, convexity." Transactions of the American Mathematical Society 2010. https://www.ams.org/journals/tran/2010-362-06/S0002-9947-09-05048-X/S0002-9947-09-05048-X.pdf

---

### Decision · Program_Chairs · 2021-09-27

**Decision:**

Accept (Poster)

**Comment:**

This paper refines the implicit bias story for two-layer networks, analyzing settings where it is guaranteed to prefer certain simple linear solutions.  While the reviewers had some concerns, overall reviews and discussion were positive.  I urge the authors to carefully address reviewer comments during their final revisions.